



**Global Carbon Budget 2021**
Pierre Friedlingstein[1,2], Matthew W. Jones[3], Michael O'Sullivan[1], Robbie M. Andrew[4],
Dorothee, C. E. Bakker[5], Judith Hauck[6], Corinne Le Quéré[3], Glen P. Peters[4], Wouter
Peters[7,8], Julia Pongratz[9,10], Stephen Sitch[11], Josep G. Canadell[12], Philippe Ciais[13], Rob B.
Jackson[14], Simone R. Alin[15], Peter Anthoni[16], Nicholas R. Bates[17], Meike Becker[18,19], Nicolas
Bellouin[20], Laurent Bopp[2], Thi Tuyet Trang Chau[13], Frédéric Chevallier[13], Louise P. Chini[21],
Margot Cronin[22], Kim I. Currie[23], Bertrand Decharme[24], Laique M. Djeutchouang[25,26], Xinyu
Dou[27], Wiley Evans[28], Richard A. Feely[15], Liang Feng[29], Thomas Gasser[30], Dennis Gilfillan[31],
Thanos Gkritzalis[32], Giacomo Grassi[33], Luke Gregor[34], Nicolas Gruber[34], Özgür Gürses[6], Ian
Harris[35], Richard A. Houghton[36], George C. Hurtt[21], Yosuke Iida[37], Tatiana Ilyina[10], Ingrid T.
Luijkx[7], Atul Jain[38], Steve D. Jones[18,19], Etsushi Kato[39], Daniel Kennedy[40], Kees Klein
Goldewijk[41], Jürgen Knauer[12,42], Jan Ivar Korsbakken[4], Arne Körtzinger[43], Peter
Landschützer[10], Siv K. Lauvset[19, 44], Nathalie Lefèvre[45], Sebastian Lienert[46], Junjie Liu[47],
Gregg Marland[48,49], Patrick C. McGuire[50], Joe R. Melton[51], David R. Munro[52,53], Julia E.M.S.
Nabel[10,54], Shin-Ichiro Nakaoka[55], Yosuke Niwa[55,56], Tsuneo Ono[57], Denis Pierrot[58], Benjamin
Poulter[59], Gregor Rehder[60], Laure Resplandy[61], Eddy Robertson[62], Christian Rödenbeck[54],
Thais M. Rosan[11], Jörg Schwinger[19, 44], Clemens Schwingshackl[9], Roland Séférian[24],
Adrienne J. Sutton[15], Colm Sweeney[53], Toste Tanhua[43], Pieter P. Tans[63], Hanqin Tian[64],
Bronte Tilbrook[65,66], Francesco Tubiello[67], Guido van der Werf[68], Nicolas Vuichard[13], Chisato
Wada[55], Rik Wanninkhof[58], Andrew J. Watson[11], David Willis[3], Andrew J. Wiltshire[62],
Wenping Yuan[69], Chao Yue[13], Xu Yue[70], Sönke Zaehle[54], Jiye Zeng[55]
[1] College of Engineering, Mathematics and Physical Sciences, University of Exeter, Exeter EX4 4QF,
UK
[2] Laboratoire de Météorologie Dynamique / Institut Pierre-Simon Laplace, CNRS, Ecole Normale
Supérieure / Université PSL, Sorbonne Université, Ecole Polytechnique, Paris, France
[3] Tyndall Centre for Climate Change Research, School of Environmental Sciences, University of East
Anglia, Norwich Research Park, Norwich NR4 7TJ, UK
[4] CICERO Center for International Climate Research, Oslo 0349, Norway
[5] School of Environmental Sciences, University of East Anglia, Norwich Research Park, Norwich NR4
7TJ, UK
[6] Alfred-Wegener-Institut, Helmholtz-Zentrum für Polar- und Meeresforschung, Am Handelshafen
12, 27570 Bremerhaven
[7] Wageningen University, Environmental Sciences Group, P.O. Box 47, 6700AA, Wageningen, The
Netherlands
[8] University of Groningen, Centre for Isotope Research, Groningen, The Netherlands
[9] Ludwig-Maximilians-Universität München, Luisenstr. 37, 80333 München, Germany
[10] Max Planck Institute for Meteorology, Bundesstr. 53, 20146 Hamburg, Germany
[11] College of Life and Environmental Sciences, University of Exeter, Exeter EX4 4RJ, UK
[12] CSIRO Oceans and Atmosphere, Canberra, ACT 2101, Australia
[13] Laboratoire des Sciences du Climat et de l'Environnement, LSCE/IPSL, CEA-CNRS-UVSQ, Université
Paris-Saclay, F-91198 Gif-sur-Yvette, France
[14] Department of Earth System Science, Woods Institute for the Environment, and Precourt Institute
for Energy, Stanford University, Stanford, CA 94305–2210, United States of America
[15] National Oceanic & Atmospheric Administration, Pacific Marine Environmental Laboratory
(NOAA/PMEL), 7600 Sand Point Way NE, Seattle, WA 98115, USA



Karlsruhe Institute of Technology, Institute of Meteorology and Climate Research/Atmospheric Environmental Research, 82467 Garmisch-Partenkirchen, Germany
Bermuda Institute of Ocean Sciences (BIOS), 17 Biological Lane, Ferry Reach, St. Georges, GEO1, Bermuda
Geophysical Institute, University of Bergen, Bergen, Norway
Bjerknes Centre for Climate Research, Bergen, Norway
Department of Meteorology, University of Reading, Reading, UK
Department of Geographical Sciences, University of Maryland, College Park, Maryland 20742, USA
Marine Institute Ireland, Galway, Rinville, Ireland
NIWA, Union Place West, Dunedin, New Zealand
CNRM, Université de Toulouse, Météo-France, CNRS, Toulouse, France
Department of Oceanography, University of Cape Town, Cape Town, 7701, South Africa
SOCCO, Council for Scientific and Industrial Research, Cape Town, 7700, South Africa
Department of Earth System Science, Tsinghua University, Beijing, China
Hakai Institute, Heriot Bay, BC, Canada
National Centre for Earth Observation, University of Edinburgh, UK
International Institute for Applied Systems Analysis (IIASA), Schlossplatz 1 A-2361 Laxenburg, Austria
North Carolina School for Science and Mathematics, Durham, North Carolina, USA
Flanders Marine Institute (VLIZ), InnovOceanSite, Wandelaarkaai 7, 8400 Ostend, Belgium
European Commission, Joint Research Centre, 21027 Ispra (VA), Italy
Environmental Physics Group, ETH Zürich, Institute of Biogeochemistry and Pollutant Dynamics and Center for Climate Systems Modeling (C2SM), 8092 Zurich, Switzerland
NCAS-Climate, Climatic Research Unit, School of Environmental Sciences, University of East Anglia, Norwich Research Park, Norwich, NR4 7TJ, UK
Woodwell Climate Research Center, Falmouth, MA 02540, USA
Atmosphere and Ocean Department, Japan Meteorological Agency, Minato-Ku, Tokyo 105-8431, Japan
Department of Atmospheric Sciences, University of Illinois, Urbana, IL 61821, USA
Institute of Applied Energy (IAE), Minato-ku, Tokyo 105-0003, Japan
National Center for Atmospheric Research, Climate and Global Dynamics, Terrestrial Sciences Section, Boulder, CO 80305, USA
Utrecht University, Faculty of Geosciences, Department IMEW, Copernicus Institute of Sustainable Development, Heidelberglaan 2, P.O. Box 80115, 3508 TC, Utrecht, the Netherlands
Hawkesbury Institute for the Environment, Western Sydney University, Penrith, New South Wales, Australia
GEOMAR Helmholtz Centre for Ocean Research Kiel, Düsternbrooker Weg 20, 24105 Kiel, Germany
NORCE Norwegian Research Centre, Jahnebakken 5, 5007 Bergen, Norway
LOCEAN/IPSL laboratory, Sorbonne Université, CNRS/IRD/MNHN, Paris, France
Climate and Environmental Physics, Physics Institute and Oeschger Centre for Climate Change Research, University of Bern, Bern, Switzerland
Jet Propulsion Laboratory, California Institute of Technology, Pasadena, CA, USA.
Research Institute for Environment, Energy, and Economics, Appalachian State University, Boone, North Carolina, USA
Department of Geological and Environmental Sciences, Appalachian State University, Boone, North Carolina, USA



Department of Meteorology, Department of Geography & Environmental Science, National Centre for Atmospheric Science, University of Reading, Reading, UK
Climate Research Division, Environment and Climate Change Canada, Victoria, BC, Canada
Cooperative Institute for Research in Environmental Sciences, University of Colorado, Boulder, CO, 80305, USA
National Oceanic & Atmospheric Administration/Global Monitoring Laboratory (NOAA/GML), Boulder, CO, 80305, USA
Max Planck Institute for Biogeochemistry, Jena, Germany
Earth System Division, National Institute for Environmental Studies, 16-2 Onogawa, Tsukuba, Ibaraki, 305-8506 Japan
Meteorological Research Institute, 1-1 Nagamine, Tsukuba, Ibaraki, 305-0052 Japan
Japan Fisheries Research and Education Agency, 2-12-4 Fukuura, Kanazawa-Ku, Yokohama 236-8648, Japan
National Oceanic & Atmospheric Administration/Atlantic Oceanographic & Meteorological Laboratory (NOAA/AOML), Miami, FL 33149, USA
NASA Goddard Space Flight Center, Biospheric Sciences Laboratory, Greenbelt, Maryland 20771, USA
Leibniz Institute for Baltic Sea Research Warnemuende (IOW), Seestrasse 15; 18119 Rostock, Germany
Princeton University, Department of Geosciences and Princeton Environmental Institute, Princeton, NJ, USA
Met Office Hadley Centre, FitzRoy Road, Exeter EX1 3PB, UK
National Oceanic and Atmospheric Administration, Earth System Research Laboratory (NOAA ESRL),Boulder, CO 80305, USA
School of Forestry and Wildlife Sciences, Auburn University, 602 Ducan Drive, Auburn, AL 36849, USA
CSIRO Oceans and Atmosphere, PO Box 1538, Hobart Tasmania 7001, Australia
Australian Antarctic Partnership Program, University of Tasmania, Hobart, Australia
Statistics Division, Food and Agriculture Organization of the United Nations, Via Terme di Caracalla, Rome 00153, Italy
Faculty of Earth and Life Sciences, VU University, Amsterdam, The Netherlands
School of Atmospheric Sciences, Sun Yat-sen University, Zhuhai, Guangdong 510245, China
School of Environmental Science and Engineering, Nanjing University of Information Science and Technology (NUIST)

**Correspondence**: Pierre Friedlingstein (p.friedlingstein@exeter.ac.uk)

**Abstract**

Accurate assessment of anthropogenic carbon dioxide ($CO_2$) emissions and their redistribution among the atmosphere, ocean, and terrestrial biosphere in a changing climate is critical to better understand the global carbon cycle, support the development of climate policies, and project future climate change. Here we describe and synthesize data sets and methodology to quantify the five major components of the global carbon budget and their uncertainties. Fossil $CO_2$ emissions ($E_{FOS}$) are based on energy statistics and



cement production data, while emissions from land-use change ($E_{LUC}$), mainly deforestation,
are based on land-use and land-use change data and bookkeeping models. Atmospheric $CO_2$
concentration is measured directly, and its growth rate ($G_{ATM}$) is computed from the annual
changes in concentration. The ocean $CO_2$ sink ($S_{OCEAN}$) is estimated with global ocean
biogeochemistry models and observation-based data-products. The terrestrial $CO_2$ sink
($S_{LAND}$) is estimated with dynamic global vegetation models. The resulting carbon budget
imbalance ($B_{IM}$), the difference between the estimated total emissions and the estimated
changes in the atmosphere, ocean, and terrestrial biosphere, is a measure of imperfect data
and understanding of the contemporary carbon cycle. All uncertainties are reported as ±1σ.
For the first time, an approach is shown to reconcile the difference in our $E_{LUC}$ estimate with
the one from national greenhouse gases inventories, supporting the assessment of
collective countries' climate progress.
For the year 2020, $E_{FOS}$ declined by 5.4% relative to 2019, with fossil emissions at 9.5 ± 0.5
GtC yr$^{-1}$ (9.3 ± 0.5 GtC yr$^{-1}$ when the cement carbonation sink is included), $E_{LUC}$ was 0.9 ± 0.7
GtC yr$^{-1}$, for a total anthropogenic $CO_2$ emission of 10.2 ± 0.8 GtC yr$^{-1}$ (37.4 ± 2.9 Gt$CO_2$).
Also, for 2020, $G_{ATM}$ was 5.0 ± 0.2 GtC yr$^{-1}$ (2.4 ± 0.1 ppm yr$^{-1}$), $S_{OCEAN}$ was 3.0 ± 0.4 GtC yr$^{-1}$
and $S_{LAND}$ was 2.9 ± 1 GtC yr$^{-1}$, with a $B_{IM}$ of -0.8 GtC yr$^{-1}$. The global atmospheric $CO_2$
concentration averaged over 2020 reached 412.45 ± 0.1 ppm. Preliminary data for 2021,
suggest a rebound in $E_{FOS}$ relative to 2020 of +4.9% (4.1% to 5.7%) globally.
Overall, the mean and trend in the components of the global carbon budget are consistently
estimated over the period 1959-2020, but discrepancies of up to 1 GtC yr$^{-1}$ persist for the
representation of annual to semi-decadal variability in $CO_2$ fluxes. Comparison of estimates
from multiple approaches and observations shows: (1) a persistent large uncertainty in the
estimate of land-use changes emissions, (2) a low agreement between the different
methods on the magnitude of the land $CO_2$ flux in the northern extra-tropics, and (3) a
discrepancy between the different methods on the strength of the ocean sink over the last
decade. This living data update documents changes in the methods and data sets used in
this new global carbon budget and the progress in understanding of the global carbon cycle
compared with previous publications of this data set (Friedlingstein et al., 2020;
Friedlingstein et al., 2019; Le Quéré et al., 2018b, 2018a, 2016, 2015b, 2015a, 2014, 2013).
The data presented in this work are available at https://doi.org/10.18160/gcp-2021
(Friedlingstein et al., 2021).



## 1 Executive Summary

**Global fossil CO$_2$ emissions (excluding cement carbonation) in 2021 are returning towards their 2019 levels after decreasing [5.4%] in 2020.** The 2020 decrease was 0.52 GtC yr$^{-1}$ (1.9 GtCO$_2$ yr$^{-1}$), bringing 2020 emissions to 9.5 ± 0.5 GtC yr$^{-1}$ (34.8 ± 1.8 GtCO$_2$ yr$^{-1}$), comparable to the emissions level of 2012. Preliminary estimates based on data available in October 2021 and a projection for the rest of the year suggest fossil CO$_2$ emissions will rebound 4.9% in 2021 (4.1% to 5.7%), bringing emissions at 9.9 GtC yr$^{-1}$ (36.4 GtCO$_2$ yr$^{-1}$), back to about the same level as in 2019 (10.0 ± 0.5 GtC yr$^{-1}$, 36.7 ± 1.8 GtCO$_2$ yr$^{-1}$). Emissions from coal and gas in 2021 are expected to rebound above 2019 levels, while emissions from oil are still below their 2019 level. Emissions in China are expected to be 5.5% higher in 2021 than in 2019, reaching 3.0 GtC (11.1 GtCO$_2$) and also higher in India with a 4.4% increase in 2021 relative to 2019, reaching 0.75 GtC (2.7 GtCO$_2$). In contrast, projected 2021 emissions in the United States (1.4 GtC, 5.1 GtCO$_2$), European Union (0.8 GtC, 2.8 GtCO$_2$), and the rest of the world (4.0 GtC, 14.8 GtCO$_2$, in aggregate) remain respectively 3.7%, 4.2%, and 4.2% below their 2019 levels. These patterns reflect the stringency of the COVID-19 confinement levels and the background trends in emissions in these countries.

**Fossil CO$_2$ emissions significantly decreased in 23 countries during the decade 2010-2019.** Altogether, these 23 countries contribute to about 2.5 GtC yr$^{-1}$ fossil fuel CO$_2$ emissions over the last decade, only about one quarter of world CO$_2$ fossil emissions.

**Global CO$_2$ emissions from land-use, land-use change, and forestry (LUC) converge based on revised data of land-use change and show a small decrease over the past two decades.** Near constant gross emissions estimated at 3.8 ± 0.6 GtC yr$^{-1}$ in the 2011-2020 decade are only partly offset by growing carbon removals on managed land of 2.7 ± 0.4 GtC yr$^{-1}$, resulting in the net emissions in managed land of 1.1 ± 0.7 GtC yr$^{-1}$ (4.1 ± 2.6 GtCO2 yr$^{-1}$). These net emissions decreased by 0.2 GtC in 2020 compared to 2019 levels, with large uncertainty. Preliminary estimates for emissions in 2021 suggest a 0.1 GtC decrease for 2021, giving net emissions of 0.8 GtC yr$^{-1}$ (2.9 GtCO2 yr$^{-1}$). The convergence of different emission estimates does not reflect the high uncertainty in land-use change datasets, which likely underestimate interannual variability and the (rising) importance of degradation, highlighting the need for accurate land-use data. For the first time, we link the global carbon



budget models' estimates to the official country reporting of national greenhouse gases
inventories. While the global carbon budget distinguishes anthropogenic from natural
drivers of land carbon fluxes, country reporting is area based and attributes part of the
natural terrestrial sink on managed land to the land-use sector. Accounting for this
redistribution, the two approaches are shown to be consistent with each other.
**The remaining carbon budget for a 50% likelihood to limit global warming to 1.5°C, 1.7°C**
**and 2°C has shrunk to 120 GtC (420 GtCO$_2$), 210 GtC (770 GtCO$_2$) and 350 GtC (1270 GtCO$_2$)**
**respectively, equivalent to 11, 20 and 32 years from the beginning of 2022, assuming 2021**
**emissions levels.** Total anthropogenic emissions were 10.4 GtC yr$^{-1}$ (38.0 GtCO$_2$ yr-1) in
2020, with a preliminary estimate of 10.7 GtC yr$^{-1}$ (39.4 GtCO2 yr$^{-1}$) for 2021. The remaining
carbon budget to keep global temperatures below the climate targets of the Paris
Agreement has shrunk by 21 GtC (77 GtCO$_2$) relative to the remaining carbon budget
estimate assessed in the IPCC AR6 Working Group 1 assessment. Reaching net zero CO$_2$
emissions by 2050 entails cutting total anthropogenic CO$_2$ emissions by about 0.4 GtC (1.4
GtCO$_2$) each year on average, comparable to the decrease during 2020, highlighting the
scale of the action needed.
**The concentration of CO$_2$ in the atmosphere is set to reach 414.7 ppm in 2021, 49% above**
**pre-industrial levels.** The atmospheric CO$_2$ growth was 5.1 ± 0.02 GtC yr$^{-1}$ during the decade
2011-2020 (47% of total CO$_2$ emissions) with a preliminary 2021 growth rate estimate of
around 4.2 GtC yr$^{-1}$. The 2020 decrease in total CO$_2$ emissions of about 0.7 GtC propagated
to a reduction of the atmospheric CO$_2$ growth rate of 0.38GtC (0.18 ppm).
**The ocean CO$_2$ sink resumed a more rapid growth in the past decade after low or no**
**growth during the 1991-2002 period.** However, the growth of the ocean CO$_2$ sink in the
past decade has an uncertainty of a factor of three, with estimates based on data products
and estimates based on models showing an ocean sink increase of 0.9 GtC yr$^{-1}$ and 0.3 GtC
yr$^{-1}$ since 2010, respectively. The discrepancy in the trend originates from all latitudes but is
largest in the Southern Ocean. The ocean CO$_2$ sink was 2.8 ± 0.4 GtC yr$^{-1}$ during the decade
2011-2020 (26% of total CO$_2$ emissions), with a preliminary 2021 estimate of around 2.9 GtC
yr$^{-1}$.



**The land $CO_2$ sink continued to increase during the 2011-2020 period primarily in response**
**to increased atmospheric $CO_2$, albeit with large interannual variability.**   The land $CO_2$ sink
was 3.1 ± 0.6 GtC yr$^{-1}$ during the 2011-2020 decade (29% of total $CO_2$ emissions), 0.5 GtC yr$^{-1}$
larger than during the previous decade (2000-2009), with a preliminary 2021 estimate of
around 3.3 GtC yr$^{-1}$. Year to year variability in the land sink is about 1 GtC yr$^{-1}$, making small
annual changes in anthropogenic emissions hard to detect in global atmospheric $CO_2$
concentration.



## 1 Introduction

The concentration of carbon dioxide ($CO_2$) in the atmosphere has increased from approximately 277 parts per million (ppm) in 1750 (Joos and Spahni, 2008), the beginning of the Industrial Era, to 412.4 ± 0.1 ppm in 2020 (Dlugokencky and Tans, 2021); Fig. 1). The atmospheric $CO_2$ increase above pre-industrial levels was, initially, primarily caused by the release of carbon to the atmosphere from deforestation and other land-use change activities (Canadell et al., 2021). While emissions from fossil fuels started before the Industrial Era, they became the dominant source of anthropogenic emissions to the atmosphere from around 1950 and their relative share has continued to increase until present. Anthropogenic emissions occur on top of an active natural carbon cycle that circulates carbon between the reservoirs of the atmosphere, ocean, and terrestrial biosphere on time scales from sub-daily to millennia, while exchanges with geologic reservoirs occur at longer timescales (Archer et al., 2009).

The global carbon budget (GCB) presented here refers to the mean, variations, and trends in the perturbation of $CO_2$ in the environment, referenced to the beginning of the Industrial Era (defined here as 1750). This paper describes the components of the global carbon cycle over the historical period with a stronger focus on the recent period (since 1958, onset of atmospheric $CO_2$ measurements), the last decade (2011-2020), the last year (2020) and the current year (2021). We quantify the input of $CO_2$ to the atmosphere by emissions from human activities, the growth rate of atmospheric $CO_2$ concentration, and the resulting changes in the storage of carbon in the land and ocean reservoirs in response to increasing atmospheric $CO_2$ levels, climate change and variability, and other anthropogenic and natural changes (Fig. 2). An understanding of this perturbation budget over time and the underlying variability and trends of the natural carbon cycle is necessary to understand the response of natural sinks to changes in climate, $CO_2$ and land-use change drivers, and to quantify the permissible emissions for a given climate stabilization target.

The components of the $CO_2$ budget that are reported annually in this paper include separate and independent estimates for the $CO_2$ emissions from (1) fossil fuel combustion and oxidation from all energy and industrial processes; also including cement production and carbonation ($E_{FOS}$; GtC yr$^{-1}$) and (2) the emissions resulting from deliberate human activities on land, including those leading to land-use change ($E_{LUC}$; GtC yr$^{-1}$); and their partitioning



among (3) the growth rate of atmospheric $CO_2$ concentration ($G_{ATM}$; GtC yr$^{-1}$), and the
uptake of $CO_2$ (the 'CO$_2$ sinks') in (4) the ocean ($S_{OCEAN}$; GtC yr$^{-1}$) and (5) on land ($S_{LAND}$; GtC
yr$^{-1}$). The $CO_2$ sinks as defined here conceptually include the response of the land (including
inland waters and estuaries) and ocean (including coasts and territorial seas) to elevated
$CO_2$ and changes in climate and other environmental conditions, although in practice not all
processes are fully accounted for (see Section 2.7). Global emissions and their partitioning
among the atmosphere, ocean and land are in reality in balance. Due to the combination of
imperfect spatial and/or temporal data coverage, errors in each estimate, and smaller terms
not included in our budget estimate (discussed in Section 2.7), the independent estimates
(1) to (5) above do not necessarily add up to zero. We therefore (a) additionally assess a set
of global atmospheric inverse model results that by design close the global carbon balance
(see Section 2.6), and (b) estimate a budget imbalance ($B_{IM}$), which is a measure of the
mismatch between the estimated emissions and the estimated changes in the atmosphere,
land and ocean, as follows:
$$B_{IM} = E_{FOS} + E_{LUC} - (G_{ATM} + S_{OCEAN} + S_{LAND}) \tag{1}$$
$G_{ATM}$ is usually reported in ppm yr$^{-1}$, which we convert to units of carbon mass per year, GtC
yr$^{-1}$, using 1 ppm = 2.124 GtC (Ballantyne et al., 2012; Table 1). All quantities are presented
in units of gigatonnes of carbon (GtC, $10^{15}$ gC), which is the same as petagrams of carbon
(PgC; Table 1). Units of gigatonnes of $CO_2$ (or billion tonnes of $CO_2$) used in policy are equal
to 3.664 multiplied by the value in units of GtC.
We also include a quantification of $E_{FOS}$ by country, computed with both territorial and
consumption-based accounting (see Section 2), and discuss missing terms from sources
other than the combustion of fossil fuels (see Section 2.7).
The global $CO_2$ budget has been assessed by the Intergovernmental Panel on Climate
Change (IPCC) in all assessment reports (Prentice et al., 2001; Schimel et al., 1995; Watson
et al., 1990; Denman et al., 2007; Ciais et al., 2013; Canadell et al., 2021), and by others (e.g.
Ballantyne et al., 2012). The Global Carbon Project (GCP, www.globalcarbonproject.org, last
access: 15 October 2021) has coordinated this cooperative community effort for the annual
publication of global carbon budgets for the year 2005 (Raupach et al., 2007; including fossil
emissions only), year 2006 (Canadell et al., 2007), year 2007 (GCP, 2008), year 2008 (Le
Quéré et al., 2009), year 2009 (Friedlingstein et al., 2010), year 2010 (Peters et al., 2012b),



year 2012 (Le Quéré et al., 2013; Peters et al., 2013), year 2013 (Le Quéré et al., 2014), year
2014 (Le Quéré et al., 2015a; Friedlingstein et al., 2014), year 2015 (Jackson et al., 2016; Le
Quéré et al., 2015b), year 2016 (Le Quéré et al., 2016), year 2017 (Le Quéré et al., 2018a;
Peters et al., 2017), year 2018 (Le Quéré et al., 2018b; Jackson et al., 2018)  year 2019
(Friedlingstein et al., 2019; Jackson et al., 2019; Peters et al., 2020) and more recently the
year 2020 (Friedlingstein et al.,  2020; Le Quéré et al., 2021) . Each of these papers updated
previous estimates with the latest available information for the entire time series.
We adopt a range of ±1 standard deviation (σ) to report the uncertainties in our estimates,
representing a likelihood of 68% that the true value will be within the provided range if the
errors have a Gaussian distribution, and no bias is assumed. This choice reflects the difficulty
of characterising the uncertainty in the $CO_2$ fluxes between the atmosphere and the ocean
and land reservoirs individually, particularly on an annual basis, as well as the difficulty of
updating the $CO_2$ emissions from land-use change. A likelihood of 68% provides an
indication of our current capability to quantify each term and its uncertainty given the
available information. The uncertainties reported here combine statistical analysis of the
underlying data, assessments of uncertainties in the generation of the data sets, and expert
judgement of the likelihood of results lying outside this range. The limitations of current
information are discussed in the paper and have been examined in detail elsewhere
(Ballantyne et al., 2015; Zscheischler et al., 2017). We also use a qualitative assessment of
confidence level to characterise the annual estimates from each term based on the type,
amount, quality, and consistency of the evidence as defined by the IPCC (Stocker et al.,

22   2013).

This paper provides a detailed description of the data sets and methodology used to
compute the global carbon budget estimates for the industrial period, from 1750 to 2020,
and in more detail for the period since 1959. It also provides decadal averages starting in
1960 including the most recent decade (2011-2020), results for the year 2020, and a
projection for the year 2021. Finally, it provides cumulative emissions from fossil fuels and
land-use change since the year 1750, the pre-industrial period; and since the year 1850, the
reference year for historical simulations in IPCC AR6 (Eyring et al., 2016). This paper is
updated every year using the format of 'living data' to keep a record of budget versions and
the changes in new data, revision of data, and changes in methodology that lead to changes



in estimates of the carbon budget. Additional materials associated with the release of each
new version will be posted at the Global Carbon Project (GCP) website
(http://www.globalcarbonproject.org/carbonbudget, last access: 15 October 2021), with
fossil fuel emissions also available through the Global Carbon Atlas
(http://www.globalcarbonatlas.org, last access: 15 October 2021). With this approach, we
aim to provide the highest transparency and traceability in the reporting of $CO_2$, the key
driver of climate change.
**2  Methods**
Multiple organizations and research groups around the world generated the original
measurements and data used to complete the global carbon budget. The effort presented
here is thus mainly one of synthesis, where results from individual groups are collated,
analysed, and evaluated for consistency. We facilitate access to original data with the
understanding that primary data sets will be referenced in future work (see Table 2 for how
to cite the data sets). Descriptions of the measurements, models, and methodologies follow
below, and detailed descriptions of each component are provided elsewhere.
This is the 16th version of the global carbon budget and the tenth revised version in the
format of a living data update in Earth System Science Data. It builds on the latest published
global carbon budget of Friedlingstein et al. (2020). The main changes are: the inclusion of
(1) data to year 2020 and a projection for the global carbon budget for year 2021; (2) a Kaya
analysis to identify the driving factors behind the recent trends in fossil fuel emissions
(changes in population, GDP per person, energy use per GDP, and $CO_2$ emissions per unit
energy), (3) an estimate of the ocean sink from models and data-products combined, (4) an
assessment of the relative contributions of increased atmospheric $CO_2$ and climate change
in driving the land and ocean sinks, and  (5) an assessment of the current trends in
anthropogenic emissions and implications for the remaining carbon budget for specific
climate targets. The main methodological differences between recent annual carbon
budgets (2016-2020) are summarised in Table 3 and previous changes since 2006 are
provided in Table A7.



**2.1 Fossil $CO_2$ emissions ($E_{FOS}$)**
**2.1.1 Historical period 1850-2020**
The estimates of global and national fossil $CO_2$ emissions ($E_{FOS}$) include the oxidation of fossil
fuels through both combustion (e.g., transport, heating) and chemical oxidation (e.g. carbon
anode decomposition in aluminium refining) activities, and the decomposition of carbonates
in industrial processes (e.g. the production of cement). We also include $CO_2$ uptake from the
cement carbonation process. Several emissions sources are not estimated or not fully
covered: coverage of emissions from lime production are not global, and decomposition of
carbonates in glass and ceramic production are included only for UNFCCC Annex 1 countries
for lack of activity data. These omissions are considered to be minor. Short-cycle carbon
emissions - for example from combustion of biomass - are not included.
Our estimates of fossil $CO_2$ emissions are derived using the standard approach of activity
data and emission factors, relying on data collection by many other parties. Our goal is to
produce the best estimate of this flux, and we therefore use a prioritisation framework to
combine data from different sources that have used different methods, while being careful
to avoid double counting and undercounting of emissions sources. The CDIAC-FF emissions
dataset, derived largely from UN energy data, forms the foundation, and we extend
emissions to year Y-1 using energy growth rates reported by BP. We then proceed to replace
estimates using data from what we consider to be superior sources, for example Annex 1
countries' official submissions to the UNFCCC. All data points are potentially subject to
revision, not just the latest year. For full details see Andrew and Peters (2021).
Other estimates of global fossil $CO_2$ emissions exist, and these are compared by Andrew
(2020a). The most common reason for differences in estimates of global fossil $CO_2$ emissions
is a difference in which emissions sources are included in the datasets. Datasets such as
those published by BP, the US Energy Information Administration, and the International
Energy Agency's '$CO_2$ emissions from fuel combustion' are all generally limited to emissions
from combustion of fossil fuels. In contrast, datasets such as PRIMAP-hist, CEDS, EDGAR,
and GCP's dataset aim to include all sources of fossil $CO_2$ emissions. See Andrew (2020a) for
detailed comparisons and discussion.
Cement absorbs $CO_2$ from the atmosphere over its lifetime, a process known as 'cement
carbonation'. We estimate this $CO_2$ sink as the average of two studies in the literature (Cao



et al., 2020; Guo et al., 2021). Both studies use the same model, developed by Xi et al.
(2016), with different parameterisations and input data. Since carbonation is a function of
both current and previous cement production, we extend these estimates by one year to
2020 by using the growth rate derived from the smoothed cement emissions (10-year
smoothing) fitted to the carbonation data.
We use the Kaya Identity for a simple decomposition of $CO_2$ emissions into the key drivers
(Raupach et al., 2007). While there are variations (Peters et al 2017), we focus here on a
decomposition of $CO_2$ emissions into population, GDP per person, energy use per GDP, and
$CO_2$ emissions per energy. Multiplying these individual components together returns the
$CO_2$ emissions. Using the decomposition, it is possible to attribute the change in $CO_2$
emissions to the change in each of the drivers. This method gives a first order understanding
of what causes $CO_2$ emissions to change each year.
**2.1.2   2021 projection**
We provide a projection of global $CO_2$ emissions in 2021 by combining separate projections
for China, USA, EU, India, and all other countries combined. The methods are different for
each of these. For China we combine monthly fossil fuel production data from the National
Bureau of Statistics, import/export data from the Customs Administration, and monthly coal
consumption estimates from SX Coal (2021), giving us partial data for the growth rates to
date of natural gas, petroleum, and cement, and of the consumption itself for raw coal. We
then use a regression model to project full-year emissions based on historical observations.
For the USA our projection is taken directly from the Energy Information Administration's
(EIA) Short-Term Energy Outlook (EIA, 2021), combined with the year-to-date growth rate of
cement production. For the EU we use monthly energy data from Eurostat to derive
estimates of monthly $CO_2$ emissions through July, with coal emissions extended first through
September using a statistical relationship with reported electricity generation from coal and
other factors, then through December assuming normal seasonal patterns. EU emissions
from natural gas - a strongly seasonal cycle - are extended through December using bias-
adjusted Holt-Winters exponential smoothing (Chatfield, 1978). EU emissions from oil are
derived using the EIA's projection of oil consumption for Europe. EU cement emissions are
based on available year-to-date data from two of the largest producers, Germany and
Poland. India's projected emissions are derived from estimates through August (September





for coal) using the methods of Andrew (2020b) and extrapolated assuming normal seasonal
patterns. Emissions for the rest of the world are derived using projected growth in economic
production from the IMF (2021) combined with extrapolated changes in emissions intensity
of economic production. More details on the $E_{FOS}$ methodology and its 2021 projection can
be found in Appendix C.1.
**2.2    $CO_2$ emissions from land-use, land-use change and forestry ($E_{LUC}$)**
The net $CO_2$ flux from land-use, land-use change and forestry ($E_{LUC}$, called land-use change
emissions in the rest of the text) includes $CO_2$ fluxes from deforestation, afforestation,
logging and forest degradation (including harvest activity), shifting cultivation (cycle of
cutting forest for agriculture, then abandoning), and regrowth of forests following wood
harvest or abandonment of agriculture. Emissions from peat burning and drainage are
added from external datasets.
Three bookkeeping approaches (updated estimates each of BLUE (Hansis et al., 2015),
OSCAR (Gasser et al., 2020), and H&N2017 (Houghton and Nassikas, 2017)) were used to
quantify gross sources and sinks and the resulting net $E_{LUC}$. Uncertainty estimates were
derived from the DGVMs ensemble for the time period prior to 1960, using for the recent
decades an uncertainty range of ±0.7 GtC yr-1, which is a semi-quantitative measure for
annual and decadal emissions and reflects our best value judgment that there is at least 68%
chance (±1σ) that the true land-use change emission lies within the given range, for the
range of processes considered here. This uncertainty range had been increased from 0.5 GtC
yr-1 after new bookkeeping models were included that indicated a larger spread than
assumed before (Le Quéré et al., 2018). Projections for 2021 are based on fire activity from
tropical deforestation and degradation as well as emissions from peat fires and drainage.
Our $E_{LUC}$ estimates follow the definition of global carbon cycle models of $CO_2$ fluxes related
to land-use and land management and differ from IPCC definitions adopted in national GHG
inventories (NGHGI) for reporting under the UNFCCC, which additionally generally include,
through adoption of the IPCC so-called managed land proxy approach, the terrestrial fluxes
occurring on land defined by countries as managed. This partly includes fluxes due to
environmental change (e.g. atmospheric $CO_2$ increase), which are part of $S_{LAND}$ in our
definition. This causes the global emission estimates to be smaller for NGHGI than for the



global carbon budget definition (Grassi et al., 2018). The same is the case for FAO estimates
of carbon fluxes on forest land, which include, compared to $S_{LAND}$, both anthropogenic and
natural sources on managed land (Tubiello et al., 2021). Using the approach outlined in
Grassi et al. (2021), here we map as additional information the two definitions to each
other, to provide a comparison of the anthropogenic carbon budget to the official country
reporting to the climate convention. More details on the $E_{LUC}$ methodology can be found in
Appendix C.2.

## 2.3  Growth rate in atmospheric $CO_2$ concentration ($G_{ATM}$)

### 2.3.1  Historical period

The rate of growth of the atmospheric $CO_2$ concentration is provided for years 1959-2020 by
the US National Oceanic and Atmospheric Administration Earth System Research Laboratory
(NOAA/ESRL; Dlugokencky and Tans, 2021), which is updated from Ballantyne et al. (2012)
and includes recent revisions to the calibration scale of atmospheric $CO_2$ measurements
(Hall et al., 2021). For the 1959-1979 period, the global growth rate is based on
measurements of atmospheric $CO_2$ concentration averaged from the Mauna Loa and South
Pole stations, as observed by the $CO_2$ Program at Scripps Institution of Oceanography
(Keeling et al., 1976). For the 1980-2020 time period, the global growth rate is based on the
average of multiple stations selected from the marine boundary layer sites with well-mixed
background air (Ballantyne et al., 2012), after fitting each station with a smoothed curve as
a function of time, and averaging by latitude band (Masarie and Tans, 1995). The annual
growth rate is estimated by Dlugokencky and Tans (2021) from atmospheric $CO_2$
concentration by taking the average of the most recent December-January months
corrected for the average seasonal cycle and subtracting this same average one year earlier.
The growth rate in units of ppm $yr^{-1}$ is converted to units of GtC $yr^{-1}$ by multiplying by a
factor of 2.124 GtC per ppm, assuming instantaneous mixing of $CO_2$ throughout the
atmosphere (Ballantyne et al., 2012).
Starting in 2020, NOAA/ESRL now provides estimates of atmospheric $CO_2$ concentrations
with respect to a new calibration scale, referred to as WMO-CO2-X2019, in line with the
recommendation of the World Meteorological Organization (WMO) Global Atmosphere
Watch (GAW) community (Hall et al., 2021). The WMO-CO2-X2019 scale improves upon the





earlier WMO-CO2-X2007 scale by including a broader set of standards, which contain $CO_2$ in
a wider range of concentrations that span the range 250-800 ppm (versus 250–520 ppm for
WMO-CO2-X2007). In addition, NOAA/ESRL made two minor corrections to the analytical
procedure used to quantify $CO_2$ concentrations, fixing an error in the second virial
coefficient of $CO_2$ and accounting for loss of a small amount of $CO_2$ to materials in the
manometer during the measurement process.  The difference in concentrations measured
using WMO-CO2-X2019 versus WMO-CO2-X2007 is ~+0.18 ppm at 400 ppm and the
observational record of atmospheric $CO_2$ concentrations have been revised accordingly. The
revisions have been applied retrospectively in all cases where the calibrations were
performed by NOAA/ESRL, thus affecting measurements made by members of the WMO-
GAW programme and other regionally coordinated programmes (e.g., Integrated Carbon
Observing System, ICOS). Changes to the $CO_2$ concentrations measured across these
networks propagate to the global mean $CO_2$ concentrations. Comparing the estimates of
$G_{ATM}$ made by Dlugokencky and Tans (2020), used in the Global Carbon Budget 2020
(Friedlingstein et al., 2020), with updated estimates from Dlugokencky and Tans (2021),
used here, we find that $G_{ATM}$ reduced on average by -0.06 GtC $yr^{-1}$ during 2010-2019 and by -
0.01 GtC $yr^{-1}$ during 1959-2019 (well within the uncertainty ranges reported below). Hence
the change in analytical procedures made by NOAA/ESRL has a negligible impact on the
atmospheric growth rate $G_{ATM}$.
The uncertainty around the atmospheric growth rate is due to four main factors. First, the
long-term reproducibility of reference gas standards (around 0.03 ppm for 1σ from the
1980s; Dlugokencky and Tans, 2021). Second, small unexplained systematic analytical errors
that may have a duration of several months to two years come and go. They have been
simulated by randomizing both the duration and the magnitude (determined from the
existing evidence) in a Monte Carlo procedure. Third, the network composition of the
marine boundary layer with some sites coming or going, gaps in the time series at each site,
etc (Dlugokencky and Tans, 2021). The latter uncertainty was estimated by NOAA/ESRL with
a Monte Carlo method by constructing 100 "alternative" networks (Masarie and Tans, 1995;
NOAA/ESRL, 2019). The second and third uncertainties, summed in quadrature, add up to
0.085 ppm on average (Dlugokencky and Tans, 2021). Fourth, the uncertainty associated
with using the average $CO_2$ concentration from a surface network to approximate the true





atmospheric average $CO_2$ concentration (mass-weighted, in 3 dimensions) as needed to
assess the total atmospheric $CO_2$ burden. In reality, $CO_2$ variations measured at the stations
will not exactly track changes in total atmospheric burden, with offsets in magnitude and
phasing due to vertical and horizontal mixing. This effect must be very small on decadal and
longer time scales, when the atmosphere can be considered well mixed. Preliminary
estimates suggest this effect would increase the annual uncertainty, but a full analysis is not
yet available. We therefore maintain an uncertainty around the annual growth rate based
on the multiple stations data set ranges between 0.11 and 0.72 GtC yr$^{-1}$, with a mean of 0.61
GtC yr$^{-1}$ for 1959-1979 and 0.17 GtC yr$^{-1}$ for 1980-2020, when a larger set of stations were
available as provided by Dlugokencky and Tans (2021) but recognise further exploration of
this uncertainty is required. At this time, we estimate the uncertainty of the decadal
averaged growth rate after 1980 at 0.02 GtC yr$^{-1}$ based on the calibration and the annual
growth rate uncertainty but stretched over a 10-year interval. For years prior to 1980, we
estimate the decadal averaged uncertainty to be 0.07 GtC yr$^{-1}$ based on a factor
proportional to the annual uncertainty prior and after 1980 (0.02 * [0.61/0.17] GtC yr$^{-1}$).
We assign a high confidence to the annual estimates of $G_{ATM}$ because they are based on
direct measurements from multiple and consistent instruments and stations distributed
around the world (Ballantyne et al., 2012; Hall et al., 2021).
To estimate the total carbon accumulated in the atmosphere since 1750 or 1850, we use an
atmospheric $CO_2$ concentration of 277 ± 3 ppm or 286 ± 3 ppm, respectively, based on a
cubic spline fit to ice core data (Joos and Spahni, 2008). For the construction of the
cumulative budget shown in Figure 3, we use the fitted estimates of $CO_2$ concentration from
Joos and Spahni (2008) to estimate the annual atmospheric growth rate using the
conversion factors shown in Table 1.  The uncertainty of ±3 ppm (converted to ±1σ) is taken
directly from the IPCC's AR5 assessment (Ciais et al., 2013). Typical uncertainties in the
growth rate in atmospheric $CO_2$ concentration from ice core data are equivalent to ±0.1-
0.15 GtC yr$^{-1}$ as evaluated from the Law Dome data (Etheridge et al., 1996) for individual 20-
year intervals over the period from 1850 to 1960 (Bruno and Joos, 1997).




**2.3.2   2021 projection**
We provide an assessment of $G_{ATM}$ for 2021 based on the monthly calculated global
atmospheric $CO_2$ concentration (GLO) through August (Dlugokencky and Tans, 2021), and
bias-adjusted Holt–Winters exponential smoothing with additive seasonality (Chatfield,
1978) to project to January 2022. Additional analysis suggests that the first half of the year
(the boreal winter-spring-summer transition) shows more interannual variability than the
second half of the year (the boreal summer-autumn-winter transition), so that the exact
projection method applied to the second half of the year has a relatively smaller impact on
the projection of the full year.  Uncertainty is estimated from past variability using the
standard deviation of the last 5 years' monthly growth rates.
**2.4   Ocean $CO_2$ sink**
The reported estimate of the global ocean anthropogenic $CO_2$ sink $S_{OCEAN}$ is derived as the
average of two estimates. The first estimate is derived as the mean over an ensemble of
eight global ocean biogeochemistry models (GOBMs, Table 4 and Table A2). The second
estimate is obtained as the mean over an ensemble of seven observation-based data-
products (Table 4 and Table A3). The GOBMs simulate both the natural and anthropogenic
$CO_2$ cycles in the ocean. They constrain the anthropogenic air-sea $CO_2$ flux (the dominant
component of $S_{OCEAN}$) by the transport of carbon into the ocean interior, which is also the
controlling factor of present-day ocean carbon uptake in the real world. They cover the full
globe and all seasons and were recently evaluated against surface ocean carbon
observations, suggesting they are suitable to estimate the annual ocean carbon sink (Hauck
et al., 2020). The data-products are tightly linked to observations of $fCO_2$ (fugacity of $CO_2$,
which equals $pCO_2$ corrected for the non-ideal behaviour of the gas; Pfeil et al., 2013), which
carry imprints of temporal and spatial variability, but are also sensitive to uncertainties in
gas-exchange parameterizations and data-sparsity. Their asset is the assessment of
interannual and spatial variability (Hauck et al., 2020).  We further use two diagnostic ocean
models to estimate $S_{OCEAN}$ over the industrial era (1781-1958).
The global $fCO_2$-based flux estimates were adjusted to remove the pre-industrial ocean
source of $CO_2$ to the atmosphere of 0.61 GtC yr$^{-1}$ from river input to the ocean (the average
of 0.45 ± 0.18 GtC yr$^{-1}$ by Jacobson et al. (2007) and 0.78 ± 0.41 GtC yr$^{-1}$ by Resplandy et al.,



2018), to satisfy our definition of $S_{OCEAN}$ (Hauck et al., 2020). The river flux adjustment was
distributed over the latitudinal bands using the regional distribution of Aumont et al. (2001;
North: 0.16 GtC yr$^{-1}$, Tropics: 0.15 GtC yr$^{-1}$, South: 0.30 GtC yr$^{-1}$), acknowledging that the
boundaries of Aumont et al (2001; namely 20°S and 20°N) are not consistent with the
boundaries otherwise used in the GCB (30°S and 30°N). A recent modelling study (Lacroix et
al., 2020) suggests that more of the riverine outgassing is located in the tropics than in the
Southern Ocean; and hence this regional distribution is associated with a major uncertainty.
Anthropogenic perturbations of river carbon and nutrient transport to the ocean are not
considered (see section 2.7).
We derive $S_{OCEAN}$ from GOBMs by using a simulation (sim A) with historical forcing of climate
and atmospheric $CO_2$, accounting for model biases and drift from a control simulation (sim
B) with constant atmospheric $CO_2$ and normal year climate forcing. A third simulation (sim
C) with historical atmospheric $CO_2$ increase and normal year climate forcing is used to
attribute the ocean sink to $CO_2$ (sim C minus sim B) and climate (sim A minus sim C) effects.
Data-products are adjusted to represent the full ocean area by a simple scaling approach
when coverage is below 98%. GOBMs and data-products fall within the observational
constraints over the 1990s (2.2 ± 0.7 GtC yr$^{-1}$ , Ciais et al., 2013) after applying adjustments .
We assign an uncertainty of ± 0.4 GtC yr$^{-1}$ to the ocean sink based on a combination of
random (ensemble standard deviation) and systematic uncertainties (GOBMs bias in
anthropogenic carbon accumulation, previously reported uncertainties in $fCO_2$-based data-
products; see section C.3.3). We assess a medium confidence level to the annual ocean $CO_2$
sink and its uncertainty because it is based on multiple lines of evidence, it is consistent with
ocean interior carbon estimates (Gruber et al., 2019, see section 3.5.5) and the results are
consistent in that the interannual variability in the GOBMs and data-based estimates are all
generally small compared to the variability in the growth rate of atmospheric $CO_2$
concentration. We refrain from assigning a high confidence because of the systematic
deviation between the GOBM and data-product trends since around 2002. More details on
the $S_{OCEAN}$ methodology can be found in Appendix C.3.
The ocean $CO_2$ sink forecast for the year 2021 is based on the annual historical and
estimated 2021 atmospheric $CO_2$ concentration (Dlugokencky and Tans 2021), historical and
estimated 2021 annual global fossil fuel emissions from this year's carbon budget, and the
spring (March, April, May) Oceanic Niño Index (ONI) index (NCEP, 2021). Using a non-linear




regression approach, i.e., a feed-forward neural network, atmospheric $CO_2$, the ONI index
and the fossil fuel emissions are used as training data to best match the annual ocean $CO_2$
sink (i.e. combined $S_{OCEAN}$ estimate from GOBMs and data products) from 1959 through
2020 from this year's carbon budget. Using this relationship, the 2021 $S_{OCEAN}$ can then be
estimated from the projected 2021 input data using the non-linear relationship established
during the network training. To avoid overfitting, the neural network was trained with a
variable number of hidden neurons (varying between 2-5) and 20% of the randomly
selected training data were withheld for independent internal testing. Based on the best
output performance (tested using the 20% withheld input data), the best performing
number of neurons was selected. In a second step, we trained the network 10 times using
the best number of neurons identified in step 1 and different sets of randomly selected
training data. The mean of the 10 trainings is considered our best forecast, whereas the
standard deviation of the 10 ensembles provides a first order estimate of the forecast
uncertainty. This uncertainty is then combined with the $S_{OCEAN}$ uncertainty (0.4 GtC $yr^{-1}$) to
estimate the overall uncertainty of the 2021 prediction.
**2.5    Terrestrial $CO_2$ sink**
The terrestrial land sink ($S_{LAND}$) is thought to be due to the combined effects of fertilisation
by rising atmospheric $CO_2$ and N inputs on plant growth, as well as the effects of climate
change such as the lengthening of the growing season in northern temperate and boreal
areas. $S_{LAND}$ does not include land sinks directly resulting from land-use and land-use change
(e.g., regrowth of vegetation) as these are part of the land-use flux ($E_{LUC}$), although system
boundaries make it difficult to attribute exactly $CO_2$ fluxes on land between $S_{LAND}$ and $E_{LUC}$
(Erb et al., 2013).
$S_{LAND}$ is estimated from the multi-model mean of 17 DGVMs (Table A1). As described in
Appendix C.4, DGVMs simulations include all climate variability and $CO_2$ effects over land,
with 12 DGVMs also including the effect of N inputs. The DGVMs estimate of $S_{LAND}$ does not
include the export of carbon to aquatic systems or its historical perturbation, which is
discussed in Appendix D3. See Appendix C.4 for DGVMs evaluation and uncertainty
assessment for $S_{LAND}$, using the International Land Model Benchmarking system (ILAMB;
Collier et al., 2018). More details on the $S_{LAND}$ methodology can be found in Appendix C.4.
Like the ocean forecast, the land $CO_2$ sink ($S_{LAND}$) forecast is based on the annual historical
and estimated 2021 atmospheric $CO_2$ concentration (Dlugokencky and Tans 2021), historical
and estimated 2021 annual global fossil fuel emissions from this year's carbon budget, and
the summer (June, July, August) ONI index (NCEP, 2021). All training data are again used to
best match $S_{LAND}$ from 1959 through 2020 from this year's carbon budget using a feed-
forward neural network. To avoid overfitting, the neural network was trained with a variable
number of hidden neurons (varying between 2-15), larger than for $S_{OCEAN}$ prediction due to
the stronger land carbon interannual variability. As done for $S_{OCEAN}$, a pre-training selects the
optimal number of hidden neurons based on 20% withheld input data, and in a second step,
an ensemble of 10 forecasts is produced to provide the mean forecast plus uncertainty. This
uncertainty is then combined with the $S_{LAND}$ uncertainty for 2020 (1.0 GtC yr$^{-1}$) to estimate
the overall uncertainty of the 2021 prediction.
**2.6    The atmospheric perspective**
The world-wide network of in-situ atmospheric measurements and satellite derived
atmospheric $CO_2$ column (xCO$_2$) observations put a strong constraint on changes in the
atmospheric abundance of $CO_2$. This is true globally (hence our large confidence in $G_{ATM}$),
but also regionally in regions with sufficient observational density found mostly in the extra-
tropics. This allows atmospheric inversion methods to constrain the magnitude and location
of the combined total surface $CO_2$ fluxes from all sources, including fossil and land-use
change emissions and land and ocean $CO_2$ fluxes. The inversions assume $E_{FOS}$ to be well
known, and they solve for the spatial and temporal distribution of land and ocean fluxes
from the residual gradients of $CO_2$ between stations that are not explained by fossil fuel
emissions. By design, such systems thus close the carbon balance ($B_{IM}$ = 0) and thus provide
an additional perspective on the independent estimates of the ocean and land fluxes.
This year's release includes six inversion systems that are described in Table A4. Each system
is rooted in Bayesian inversion principles but uses slightly different methodologies. These
differences concern the selection of atmospheric $CO_2$ data and the choice of a-priori fluxes
to refine with these datas. They also differ in spatial and temporal resolution, assumed
correlation structures, and mathematical approach of the models (see references in Table
A4 for details). Importantly, the systems use a variety of transport models, which was
demonstrated to be a driving factor behind differences in atmospheric inversion-based flux



estimates, and specifically their distribution across latitudinal bands (Gaubert et al., 2019;
Schuh et al., 2019). Multiple inversion systems (UoE, CTE, and CAMS) were previously tested
with satellite $xCO_2$ retrievals from GOSAT or OCO-2 measurements, but their results at the
larger scales (as discussed in this work) did not deviate substantially from their in-situ
counterparts and are therefore not separately included. One inversion this year (CMS-Flux)
used ACOS-GOSAT v9 retrievals between July 2009 and Dec 2014 and OCO-2 b10 retrievals
between Jan 2015 to Dec 2015, in addition to the in-situ observational $CO_2$ mole fraction
records.
The original products delivered by the inverse modelers were modified to facilitate the
comparison to the other elements of the budget, specifically on 3 accounts: (1) global total
fossil fuel emissions, (2) riverine $CO_2$ transport, and (3) cement carbonation $CO_2$ uptake.
Details are given below. We note that with these adjustments the inverse results no longer
represent the net atmosphere-surface exchange over land/ocean areas as sensed by
atmospheric observations. Instead for land they become the net loss/uptake of $CO_2$ by
vegetation and soils that is not exported by fluvial systems, similar to the DGVMs estimates.
For oceans, they become the net uptake of anthropogenic $CO_2$, similar to the GOBMs
estimates.
The inversion systems prescribe global fossil fuel emissions based on the GCP's Gridded
Fossil Emissions Dataset version 2021.2 (GCP-GridFEDv2021.2; Jones et al., 2021b), which is
an update to 2019 of the first version of GCP-GridFED presented by Jones et al. (2021a).
GCP-GridFEDv2021.2 scales gridded estimates of $CO_2$ emissions from EDGARv4.3.2
(Janssens-Maenhout et al., 2019) within national territories to match national emissions
estimates provided by the GCB for the years 1959-2020, which were compiled following the
methodology described in Section 2.1 with all datasets available on August 14th 2021 (R.
Andrew, *pers. comm.*). Small differences between the systems due to for instance regridding
to the transport model resolution are corrected for in the latitudinal partitioning we
present, to ensure agreement with the estimate of $E_{FOS}$ in this budget. We also note that the
ocean fluxes used as prior by 5 out of 6 inversions are part of the suite of the ocean process
model or fCO2 data products suite listed in Section 2.4. Although these fluxes are further
adjusted by the atmospheric inversions, it makes the inversion estimates of the ocean fluxes
not completely independent of $S_{OCEAN}$ assessed here.



To facilitate comparisons to the independent $S_{OCEAN}$ and $S_{LAND}$, we used the same corrections
for transport and outgassing of carbon transported from land to ocean, as done for the
observation-based estimates of $S_{OCEAN}$ (see Appendix C.3). Furthermore, the inversions did
not include a cement carbonation sink (see section 2.1) and therefore this GCB component
is implicitly part of their total land sink estimate. In the numbers presented in this budget,
each year's global carbonation sink from cement was subtracted from each year's estimated
land sink in each inversion, distributed proportional to fossil fuel emissions per region
(North-Tropics-South).
The atmospheric inversions are evaluated using vertical profiles of atmospheric $CO_2$
concentrations (Fig. B4). More than 30 aircraft programs over the globe, either regular
programs or repeated surveys over at least 9 months, have been used to assess model
performance (with space-time observational coverage sparse in the SH and tropics, and
denser in NH mid-latitudes; Table A6). The six models are compared to the independent
aircraft $CO_2$ measurements between 2 and 7 km above sea level between 2001 and 2020.
Results are shown in Fig. B4 and discussed in Section 3.7.
With a relatively small ensemble (N=6) of systems that moreover share some a-priori fluxes
used with one another, or with the process-based models, it is difficult to justify using their
mean and standard deviation as a metric for uncertainty across the ensemble. We therefore
report their full range (min-max) without their mean. More details on the atmospheric
inversions methodology can be found in Appendix C.5.
**2.7    Processes not included in the global carbon budget**
The contribution of anthropogenic CO and CH4 to the global carbon budget is not fully
accounted for in Eq. (1) and is described in Appendix D1. The contributions of other
carbonates to $CO_2$ emissions is described in Appendix D2. The contribution of anthropogenic
changes in river fluxes is conceptually included in Eq. (1) in $S_{OCEAN}$ and in $S_{LAND}$, but it is not
represented in the process models used to quantify these fluxes. This effect is discussed in
Appendix D3. Similarly, the loss of additional sink capacity from reduced forest cover is
missing in the combination of approaches used here to estimate both land fluxes ($E_{LUC}$ and
$S_{LAND}$) and its potential effect is discussed and quantified in Appendix D4.



**3      Results**
For each component of the global carbon budget, we present results for three different time
periods: the full historical period, from 1850 to 2020, the six decades in which we have
atmospheric concentration records from Mauna Loa (1960-2020), a specific focus on last
year (2020), and the projection for the current year (2021). Subsequently, we assess the
combined constraints from the budget components (often referred to as a bottom-up
budget) against the top-down constraints from inverse modeling of atmospheric
observations. We do this for the global balance of the last decade, as well as for a regional
breakdown of land and ocean sinks by broad latitude bands.
**3.1      Fossil $CO_2$ Emissions**
**3.1.1      Historical period 1850-2020**
Cumulative fossil $CO_2$ emissions for 1850-2020 were 455 ± 25 GtC, including the cement
carbonation sink (Fig. 3, Table 8) .
In this period, 46% of fossil $CO_2$ emissions came from coal, 35% from oil, 14% from natural
gas, 3% from decomposition of carbonates, and 1% from flaring.
In 1850, the UK stood for 62% of global fossil $CO_2$ emissions. In 1891 the combined
cumulative emissions of the current members of the European Union reached and
subsequently surpassed the level of the UK. Since 1917 US cumulative emissions have been
the largest. Over the entire period 1850-2020, US cumulative emissions amount to 110GtC
(25% of world total) , the EU's to 80 GtC (18%), and China's to 60 GtC (14%).
There are three additional global datasets that include all sources of fossil $CO_2$ emissions:
CDIAC-FF (Gilfillan and Marland, 2021), CEDS version v_2021_04_21 (Hoesly et al., 2018);
O'Rourke et al., 2021) and PRIMAP-hist version 2.3.1 (Gütschow et al., 2016, 2021), although
these datasets are not independent. CDIAC-FF has the lowest cumulative emissions over
1750-2018 at 437 GtC, GCP has 443 GtC, CEDS 445 GtC, PRIMAP-hist TP 453 GtC, and
PRIMAP-hist CR 455 GtC. CDIAC-FF excludes emissions from lime production, while both
CDIAC-FF and GCP exclude emissions from international bunker fuels prior to 1950. CEDS
has higher emissions from international shipping in recent years, while PRIMAP-hist has
higher fugitive emissions than the other datasets. However, in general these four datasets
are in relative agreement as to total historical global emissions of fossil $CO_2$.



### 3.1.2 Recent period 1960-2020

Global fossil $CO_2$ emissions, $E_{FOS}$ (including the cement carbonation sink), have increased every decade from an average of 3.0 ± 0.2 GtC $yr^{-1}$ for the decade of the 1960s to an average of 9.5 ± 0.5 GtC $yr^{-1}$ during 2011-2020 (Table 6, Fig. 2 and Fig. 5). The growth rate in these emissions decreased between the 1960s and the 1990s, from 4.3% $yr^{-1}$ in the 1960s (1960-1969), 3.2% $yr^{-1}$ in the 1970s (1970-1979), 1.6% $yr^{-1}$ in the 1980s (1980-1989), to 0.9% $yr^{-1}$ in the 1990s (1990-1999). After this period, the growth rate began increasing again in the 2000s at an average growth rate of 3.0% $yr^{-1}$, decreasing to 0.6% $yr^{-1}$ for the last decade (2011-2020). China's emissions increased by +1.0% $yr^{-1}$ on average over the last 10 years dominating the global trend, followed by India's emissions increase by +3.9% $yr^{-1}$, while emissions decreased in EU27 by −1.9% $yr^{-1}$, and in the USA by −1.1% $yr^{-1}$. Fig.6 illustrates the spatial distribution of fossil fuel emissions for the 2011-2020 period.

$E_{FOS}$ includes the uptake of $CO_2$ by cement via carbonation which has increased with increasing stocks of cement products, from an average of 20 MtC $yr^{-1}$ (0.02 GtC $yr^{-1}$) in the 1960s to an average of 200 MtC $yr^{-1}$ (0.2 GtC $yr^{-1}$) during 2011-2020 (Fig. 5).

### 3.1.3 Final year 2020

The estimate of global fossil $CO_2$ emissions for 2020 is 5.4% lower than in 2019, declining 0.5 GtC to reach 9.5 ± 0.5 GtC (9.3 ± 0.5 GtC when including the cement carbonation sink) in 2020 (Fig. 5), distributed among coal (40%), oil (32%), natural gas (21%), cement (5%) and others (2%). Compared to the previous year, 2020 emissions from coal, oil and gas declined by 4.4%, 9.7% and 2.3% respectively, while emissions from cement increased by 0.8%. All growth rates presented are adjusted for the leap year, unless stated otherwise.

In 2020, the largest absolute contributions to global fossil $CO_2$ emissions were from China (31%), the USA (14%), the EU27 (7%), and India (7%). These four regions account for 59% of global $CO_2$ emissions, while the rest of the world contributed 41%, including international aviation and marine bunker fuels (2.9% of the total). Growth rates for these countries from 2019 to 2020 were +1.4% (China), -10.6% (USA), −10.9% (EU27), and -7.3% (India), with -7.0% for the rest of the world. The per-capita fossil $CO_2$ emissions in 2020 were 1.2 tC $person^{-1}$ $yr^{-1}$ for the globe, and were 3.9 (USA), 2.0 (China), 1.6 (EU27) and 0.5 (India) tC $person^{-1}$ $yr^{-1}$ for the four highest emitting countries (Fig. 5).



The decline in emissions of -5.4% in 2020 is close to the projected decline of -6.7%, which
was the median of four approaches, published in Friedlingstein et al. (2020). Of the four
approaches, the 'GCP' method was closest at -5.8%. That method was based on national
emissions projections for China, the USA, the EU27, and India using reported monthly
activity data when available and projections of gross domestic product corrected for trends
in fossil fuel intensity ($I_{FOS}$) for the rest of the world. Of the regions, the projection for the
EU27 was least accurate, and the reasons for this are discussed by Andrew (2021).
**3.1.4   Year 2021 Projection**
Globally, we estimate that global fossil $CO_2$ emissions will rebound 4.9% in 2021 (4.1% to
5.7%) to 9.9 GtC (36.4 $GtCO_2$), returning near their 2019 emission levels of 10.0 GtC (36.7
$GtCO_2$). Global increase in 2021 emissions per fuel types are +5.7% (range 4.5% to 6.8%) for
coal, +4.4% (range 3.0% to 5.8%) for oil, +4.3% (range 3.2% to 5.4%) for natural gas, and
+6.5% (range 4.8% to 8.3%) for cement.
For China, projected fossil emissions in 2021 are expected to increase by 4.0% (range 2.1%
to 5.8%) compared with 2020 emissions, bringing 2021 emissions for China around 3.0 GtC
$yr^{-1}$ (11.1 $GtCO_2$ $yr^{-1}$). Chinese emissions appear to have risen in both 2020 and 2021 despite
the economic disruptions of COVID-19. Increases in fuel specific projections for China are
+2.5% for coal, +6.0% for oil, +15.3% natural gas, and +6.4% for cement.
For the USA, the Energy Information Administration (EIA) emissions projection for 2021
combined with cement clinker data from USGS gives an increase of 7.6% (range 5.3% to
10.0%) compared to 2020, bringing USA 2021 emissions around 1.4 GtC $yr^{-1}$ (5.1 $GtCO_2$ $yr^{-1}$).
This is based on separate projections for coal +20.4%, oil +9.1%, natural gas -0.4%, and
cement +0.7%.
For the European Union, our projection for 2021 is for an increase of 7.6% (range 5.6% to
9.5%) over 2020, with 2021 emissions around 0.8 GtC $yr^{-1}$ (2.8 $GtCO_2$ $yr^{-1}$). This is based on
separate projections for coal of +15.4%, oil +4.3%, natural gas +7.6%, and cement -0.2%.
For India, our projection for 2021 is an increase of 12.6% (range of 10.7% to 13.6%) over
2020, with 2021 emissions around 0.7 GtC $yr^{-1}$ (2.7 $GtCO_2$ $yr^{-1}$). This is based on separate
projections for coal of +14.8%, oil +6.7%, natural gas +4.7%, and cement +21.4%.



For the rest of the world, the expected growth rate for 2021 is 2.9% (range 1.8% to 4.1%).
This is computed using the GDP projection for the world (excluding China, the USA, the EU,
and India) of 4.4% made by the IMF (2021) and a decrease in $I_{FOS}$ of -1.7%yr$^{-1}$, which is the
average over 2011-2020. The uncertainty range is based on the standard deviation of the
interannual variability in $I_{FOS}$ during 2011–2020 of 0.6%yr$^{-1}$ and our estimates of uncertainty
in the IMF's GDP forecast of 0.6%. The methodology allows independent projections for
coal, oil, natural gas, cement, and other components, which add to the total emissions in
the rest of the world. The fuel specific projected 2021 growth rates for the rest of the world
are: +3.0% (range 0.5% to 5.6%) for coal, +2.1% (-0.5% to +4.7%) for oil, +3.9% (2.4% to
5.5%) for natural gas, +4.6% (+2.5% to +6.7%) for cement.
Independently, the IEA has published two forecasts of global fossil energy $CO_2$ emissions
(i.e., a subset of fossil $CO_2$ emissions), first in April (4.8%; IEA, 2021a) and so revised in
October at 4% (IEA, 2021b). Carbon Monitor produces estimates of global emissions with
low temporal lag, and their estimates suggest that emissions in the first eight months of
2021 were 7.0% higher than in the same period in 2020 (Carbon Monitor, 2021).
**3.2    Emissions from Land Use Changes**
**3.2.1    Historical period 1850-2020**
Cumulative $CO_2$ emissions from land-use changes ($E_{LUC}$) for 1850-2020 were 200 ± 65 GtC
(Table 8; Fig. 3; Fig. 13). The cumulative emissions from $E_{LUC}$ are particularly uncertain, with
large spread among individual estimates of 140 GtC (updated H&N2017), 270 GtC (BLUE),
and 195 GtC (OSCAR) for the three bookkeeping models and a similar wide estimate of 190 ±
60 GtC for the DGVMs (all cumulative numbers are rounded to the nearest 5GtC). These
estimates are broadly consistent with indirect constraints from vegetation biomass
observations, giving a cumulative source of 155 ± 50 GtC over the 1901-2012 period  (Li et
al., 2017). However, given the large spread a best estimate is difficult to ascertain.
**3.2.2    Recent period 1960-2020**
In contrast to growing fossil emissions, $CO_2$ emissions from land-use, land-use change and
forestry have remained relatively constant, at around 1.3 ± 0.7 GtC yr$^{-1}$ over the 1970-1999
period, and even show a slight decrease over the last 20 years (Table 6) but with large
spread across estimates (Table 5, Fig. 7). Emissions are relatively constant in the DGVMs



ensemble of models since the 1970s, with similar mean values until the 1990s as the
bookkeeping mean and large model spread (Table 5, Fig. 7). The DGVMs average grows
larger than the bookkeeping average in the recent decades and shows no sign of decreasing
emissions, which is, however, expected as DGVM-based estimates include the loss of
additional sink capacity, which grows with time, while the bookkeeping estimates do not
(Appendix D4).
$E_{LUC}$ is a net term of various gross fluxes, which comprise emissions and removals. Gross
emissions are on average 2-4 times larger than the net $E_{LUC}$ emissions, and remained largely
constant over the last 60 years, with a moderate increase from an average of 3.4 ± 0.9 GtC
$yr^{-1}$ for the decade of the 1960s to an average of 3.8 ± 0.6 GtC $yr^{-1}$ during 2011-2020 (Fig.7,
Table 5), showing the relevance of land management such as harvesting or rotational
agriculture. Increases in gross removals, from 1.9 ± 0.4 GtC $yr^{-1}$ for the 1960s to 2.7 ± 0.4 GtC
$yr^{-1}$ for 2011-2020, were larger than the increase in gross emissions. Since the processes
behind gross removals, foremost forest regrowth and soil recovery, are all slow, while gross
emissions include a large instantaneous component, short-term changes in land-use
dynamics, such as a temporary decrease in deforestation, influences gross emissions
dynamics more than gross removals dynamics. It is these relative changes to each other that
explain the decrease in net $E_{LUC}$ emissions over the last two decades and the last few years.
Gross fluxes differ more across the three bookkeeping estimates than net fluxes, which is
expected due to different process representation; in particular, treatment of shifting
cultivation, which increases both gross emissions and removals, differs across models.
There is a decrease in net $CO_2$ emissions from land-use change over the last decade (Fig. 7,
Table 6), in contrast to earlier estimates of no clear trend across $E_{LUC}$ estimates
(Friedlingstein et al., 2020, Hong et al., 2021). The trend in the last decade is now about -4%
per year, compared to the +1.8% per year reported by Friedlingstein et al. (2020). This
decrease is principally attributable to changes in $E_{LUC}$ estimates from BLUE and OSCAR,
which relate to changes in the underlying land-use forcing, LUH2 (Chini et al. 2021, Hurtt et
al. 2020) based on HYDE3.3 (Klein Goldewijk et al., 2017a, b). HYDE3.3 now incorporates
updated estimates of agricultural areas by the FAO (see Appendix C.2.2) and uses multi-
annual land cover maps from satellite remote sensing (ESA CCI Land Cover) to constrain
contemporary land cover patterns. These changes lead to lower global $E_{LUC}$ estimates in the



last two decades compared to earlier versions of the global carbon budget due most notably
to lower emissions from cropland expansion, particularly in the tropical regions. Rosan et al.
(2021) showed that for Brazil, the new HYDE3.3 version is closer to independent, regional
estimates of land-use and land cover change (MapBiomas, 2021) with respect to spatial
patterns, but it shows less land-use and land cover changes than these independent
estimates, while HYDE3.2-based estimates had shown higher changes. The update in land-
use forcing leads to a decrease in estimated emissions in Brazil across several models after
the documented deforestation peak of 2003-2004 that preceded policies and monitoring
systems decreasing deforestation rates. However, estimated emissions based on the new
land-use forcing do not reflect the rise in Brazilian deforestation in the recent few years
(Silva Junior, 2021), and associated increasing emissions from deforestation would have
been missed here. The update in FAO agricultural areas in Brazil also implied that substantial
interannual variability reported to earlier FAO assessment and captured by the HYDE3.2
version since 2000 was removed. Due to the asymmetry of (fast) decay (like clearing by fire)
and (slower) regrowth, such reduced variability is expected to decrease annual emissions.
Also, the approach by Houghton and Nassikas (2017) smooths land use area changes before
calculating carbon fluxes by a 5-year running mean, hence the three emission estimates are
in better agreement than in previous GCB estimates. However, differences still exist, which
highlight the need for accurate knowledge of land-use transitions and their spatial and
temporal variability. A further caveat is that global land-use change data for model input
does not capture forest degradation, which often occurs on small scale or without forest
cover changes easily detectable from remote sensing and poses a growing threat to forest
area and carbon stocks that may surpass deforestation effects (e.g., Matricardi et al., 2020,
Qin et al., 2021).
Highest land-use emissions occur in the tropical regions of all three continents, including the
Arc of Deforestation in the Amazon basin (Fig. 6b). This is related to massive expansion of
cropland, particularly in the last few decades in Latin America, Southeast Asia, and sub-
Saharan Africa Emissions (Hong et al., 2021), to a substantial part for export (Pendrill et al.,
2019). Emission intensity is high in many tropical countries, particularly of Southeast Asia,
due to high rates of land conversion in regions of carbon-dense and often still pristine,
undegraded natural forests (Hong et al., 2021). Emissions are further increased by peat fires



in equatorial Asia (GFED4s, van der Werf et al., 2017). Uptake due to land-use change
occurs, particularly in Europe, partly related to expanding forest area as a consequence of
the forest transition in the 19th and 20th century and subsequent regrowth of forest (Fig. 6b)
(Mather 2001; McGrath et al., 2015).
National GHG inventory data (NGHGI) under the LULUCF sector or data submitted by
countries to FAOSTAT differ from the global models' definition of $E_{LUC}$ we adopt here in that
in the NGHGI reporting, the natural fluxes ($S_{LAND}$) are counted towards $E_{LUC}$ when they occur
on managed land (Grassi et al., 2018). In order to compare our results to the NGHGI
approach, we perform a re-mapping of our $E_{LUC}$ estimate by including the $S_{LAND}$ over
managed forest from the DGVMs simulations (following Grassi et al., 2021) to the
bookkeeping $E_{LUC}$ estimate (see Appendix C.2.3). For the 2010-2019 period, we estimate
that 1.5 GtC yr$^{-1}$ of $S_{LAND}$ occurred on managed forests and is then reallocated to $E_{LUC}$ here, as
done in the NGHGI method. Doing so, our mean estimate of $E_{LUC}$ is reduced from a source of
1.2 GtC to a sink of -0.4 GtC, very similar to the NGHGI estimate of -0.3 GtC (Table A.8).
Though estimates between GHGI, FAOSTAT, individual process-based models and the
mapped budget estimates still differ in value and need further analysis, the approach taken
here provides a possibility to relate the global models' and NGHGI approach to each other
routinely and thus link the anthropogenic carbon budget estimates of land $CO_2$ fluxes
directly to the Global Stocktake, as part of UNFCCC Paris Agreement.
**3.2.3   Final year 2020**
The global $CO_2$ emissions from land-use change are estimated as 0.9 ± 0.7 GtC in 2020, 0.2
GtC lower than 2019, which had featured particularly large peat and tropical
deforestation/degradation fires. The surge in deforestation fires in the Amazon, causing
about 30% higher emissions from deforestation and degradation fires in 2019 over the
previous decade, continued into 2020 (GFED4.1s, van der Werf et al., 2017). However, the
unusually dry conditions for a non-El Niño year that occurred in Indonesia in 2019 and led to
fire emissions from peat burning, deforestation and degradation in equatorial Asia to be
about twice as large as the average over the previous decade (GFED4.1s, van der Werf et al.,
2017) ceased in 2020. However, confidence in the annual change remains low.





Land-use change and related emissions may have been affected by the COVID-19 pandemic
(e.g. Poulter et al., 2021). Although emissions from tropical deforestation and degradation
fires have been decreasing from 2019 to 2020 on the global scale, they increased in Latin
America (GFED4s; van der Werf et al., 2017). During the period of the pandemic,
environmental protection policies and their implementation may have been weakened in
Brazil (Vale et al., 2021). In other countries, too, monitoring capacities and legal
enforcement of measures to reduce tropical deforestation have been reduced due to
budget restrictions of environmental agencies or impairments to ground-based monitoring
that prevents land grabs and tenure conflicts (Brancalion et al., 2020, Amador-Jiménez et
al., 2020). Effects of the pandemic on trends in fire activity or forest cover changes are hard
to separate from those of general political developments and environmental changes and
the long-term consequences of disruptions in agricultural and forestry economic activities
(e.g., Gruère and Brooks, 2020; Golar et al., 2020; Beckman and Countryman, 2021) remain
to be seen.
**3.2.4   Year 2021 Projection**
With wet conditions in Indonesia and a below-average fire season in South America our
preliminary estimate of $E_{LUC}$ for 2021 is substantially lower than the 2011-2020 average. By
the end of September 2021 emissions from tropical deforestation and degradation fires
were estimated to be 192 TgC, down from 347 TgC in 2019 and 288 in 2020 (315 TgC 1997-
2020 average). Peat fire emissions in Equatorial Asia were estimated to be 1 TgC, down from
117 TgC in 2019 and 2 TgC in 2020 (74 TgC 1997-2020 average) (GFED4.1s, van der Werf et
al., 2017). Based on the fire emissions until the end of September, we expect $E_{LUC}$ emissions
of around 0.8 GtC in 2021. Note that although our extrapolation is based on tropical
deforestation and degradation fires, degradation attributable to selective logging, edge-
effects or fragmentation will not be captured.
**3.3   Total anthropogenic emissions**
Cumulative anthropogenic $CO_2$ emissions for 1850-2020 totalled 660 ± 65 GtC (2420 ± 240
$GtCO_2$), of which almost 70% (455 GtC) occurred since 1960 and more than 30% (205 GtC)
since 2000 (Table 6 and 8). Total anthropogenic emissions more than doubled over the last



60 years, from 4.6 ± 0.7 GtC yr⁻¹ for the decade of the 1960s to an average of 10.6 ± 0.8 GtC
yr⁻¹ during 2011-2020.
The total anthropogenic $CO_2$ emissions from fossil plus land-use change amounted to 10.2 ±
0.8 GtC (37.2 ± 2.9 GtCO₂) in 2020, while for 2021, we project global total anthropogenic
$CO_2$ emissions from fossil and land use changes to be around 10.5 GtC (38.5 GtCO₂).
During the historical period 1850-2020, 30% of historical emissions were from land use
change and 70% from fossil emissions. However, fossil emissions have grown significantly
since 1960 while land use changes have not, and consequently the contributions of land use
change to total anthropogenic emissions were smaller during recent periods (17% during
the period 1960-2020 and 10% during 2011-2020).
**3.4    Atmospheric $CO_2$**
**3.4.1   Historical period 1850-2020**
Atmospheric $CO_2$ concentration was approximately 277 parts per million (ppm) in 1750
(Joos and Spahni, 2008), reaching 300ppm in the 1910s, 350ppm in the late 1980s, and
reaching 412.44 ± 0.1 ppm in 2020 (Dlugokencky and Tans, 2021); Fig. 1). The mass of
carbon in the atmosphere increased by 48% from 590 GtC in 1750 to 876 GtC in 2020.
Current $CO_2$ concentrations in the atmosphere are unprecedented in the last 2 million years
and the current rate of atmospheric $CO_2$ increase is at least 10 times faster than at any other
time during the last 800,000 years (Canadell et al., 2021).
**3.4.2   Recent period 1960-2020**
The growth rate in atmospheric $CO_2$ level increased from 1.7 ± 0.07 GtC yr⁻¹ in the 1960s to
5.1 ± 0.02 GtC yr⁻¹ during 2011-2020 with important decadal variations (Table 6, Fig. 3 and
Fig 4).
During the last decade (2011-2020), the growth rate in atmospheric $CO_2$ concentration
continued to increase, albeit with large interannual variability (Fig. 4).
The airborne fraction (AF), defined as the ratio of atmospheric $CO_2$ growth rate to total
anthropogenic emissions:
$AF = G_{ATM} / (E_{FOS} + E_{LUC})$                               (2)



provides a diagnostic of the relative strength of the land and ocean carbon sinks in removing
part of the anthropogenic $CO_2$ perturbation. The evolution of AF over the last 60 years
shows no significant trend, remaining nearly at around 45%, albeit showing a large
interannual variability driven by the year-to-year variability in $G_{ATM}$ (Fig. 8). The observed
stability of the airborne fraction over the 1960-2020 period indicates that the ocean and
land $CO_2$ sinks have been removing on average about 55% of the anthropogenic emissions
(see sections 3.5 and 3.6).
**3.4.3   Final year 2020**
The growth rate in atmospheric $CO_2$ concentration was 5.0 ± 0.2 GtC (2.37 ± 0.08 ppm) in
2020 (Fig. 4; Dlugokencky and Tans, 2021), very close to the 2011-2020 average. The 2020
decrease in $E_{FOS}$ and $E_{LUC}$ of about 0.7 GtC propagated to an atmospheric $CO_2$ growth rate
reduction of 0.38 GtC (0.18 ppm), given the significant interannual variability of the land
carbon sink.
**3.4.4   Year 2021 Projection**
The 2021 growth in atmospheric $CO_2$ concentration ($G_{ATM}$) is projected to be about 4.2 GtC
(1.98 ppm) based on GLO observations until the end of July 2021, bringing the atmospheric
$CO_2$ concentration to an expected level of 414.7 ppm averaged over the year, 49% over the
pre-industrial level.
**3.5     Ocean Sink**
**3.5.1   Historical period 1850-2020**
Cumulated since 1850, the ocean sink adds up to 170 ± 35 GtC, with two thirds of this
amount being taken up by the global ocean since 1960. Over the historical period, the ocean
sink increased in pace with the anthropogenic emissions exponential increase (Fig. 3b).
Since 1850, the ocean has removed 26% of total anthropogenic emissions.
**3.5.2   Recent period 1960-2020**
The ocean $CO_2$ sink increased from 1.1 ± 0.4 GtC $yr^{-1}$ in the 1960s to 2.8 ± 0.4 GtC $yr^{-1}$ during
2011-2020 (Table 6), with interannual variations of the order of a few tenths of GtC $yr^{-1}$ (Fig.
9). The ocean-borne fraction ($S_{OCEAN}/(E_{FOS}+E_{LUC})$) has been remarkably constant around 25%



on average (Fig. 8). Variations around this mean illustrate decadal variability of the ocean
carbon sink. So far, there is no indication of a decrease in the ocean-borne fraction from
1960 to 2020. The increase of the ocean sink is primarily driven by the increased
atmospheric $CO_2$ concentration, with the strongest $CO_2$ induced signal in the North Atlantic
and the Southern Ocean (Fig. 10a). The effect of climate change is much weaker, reducing
the ocean sink globally by $0.12 \pm 0.07$ GtC yr$^{-1}$ or 5% (2011-2020, range -0.8 to -7.4%), and
does not show clear spatial patterns across the GOBMs ensemble (Fig. 10b). This is the
combined effect of change and variability in all atmospheric forcing fields, previously
attributed to wind and temperature changes in one model (LeQuéré et al., 2010).
The global net air-sea $CO_2$ flux is a residual of large natural and anthropogenic $CO_2$ fluxes
into and out of the ocean with distinct regional and seasonal variations (Fig. 6 and B1).
Natural fluxes dominate on regional scales, but largely cancel out when integrated globally
(Gruber et al., 2009). Mid-latitudes in all basins and the high-latitude North Atlantic
dominate the ocean $CO_2$ uptake where low temperatures and high wind speeds facilitate
$CO_2$ uptake at the surface (Takahashi et al., 2009). In these regions, formation of mode,
intermediate and deep-water masses transport anthropogenic carbon into the ocean
interior, thus allowing for continued $CO_2$ uptake at the surface. Outgassing of natural $CO_2$
occurs mostly in the tropics, especially in the equatorial upwelling region, and to a lesser
extent in the North Pacific and polar Southern Ocean, mirroring a well-established
understanding of regional patterns of air-sea $CO_2$ exchange (e.g., Takahashi et al., 2009,
Gruber et al., 2009). These patterns are also noticeable in the Surface Ocean $CO_2$ Atlas
(SOCAT) dataset, where an ocean fCO$_2$ value above the atmospheric level indicates
outgassing (Fig. B1). This map further illustrates the data-sparsity in the Indian Ocean and
the southern hemisphere in general.
Interannual variability of the ocean carbon sink is driven by climate variability with a first-
order effect from a stronger ocean sink during large El Niño events (e.g., 1997-1998) (Fig. 9;
Rödenbeck et al., 2014, Hauck et al., 2020). The GOBMs show the same patterns of decadal
variability as the mean of the fCO$_2$-based data products, with a stagnation of the ocean sink
in the 1990s and a strengthening since the early 2000s (Fig. 9, Le Quéré et al., 2007;
Landschützer et al., 2015, 2016; DeVries et al., 2017; Hauck et al., 2020; McKinley et al.,
2020). Different explanations have been proposed for this decadal variability, ranging from





the ocean's response to changes in atmospheric wind and pressure systems (e.g., Le Quéré
et al., 2007, Keppler and Landschützer, 2019), including variations in upper ocean
overturning circulation (DeVries et al., 2017) to the eruption of Mount Pinatubo and its
effects on sea surface temperature and slowed atmospheric $CO_2$ growth rate in the 1990s
(McKinley et al., 2020). The main origin of the decadal variability is a matter of debate with a
number of studies initially pointing to the Southern Ocean (see review in Canadell et al.,
2021), but also contributions from the North Atlantic and North Pacific (Landschützer et al.,
2016, DeVries et al., 2019), or a global signal (McKinley et al., 2020) were proposed.
Although all individual GOBMs and data-products fall within the observational constraint,
the ensemble means of GOBMs, and data-products adjusted for the riverine flux diverge
over time with a mean offset increasing from 0.24 GtC yr$^{-1}$ in the 1990s to 0.66 GtC yr$^{-1}$ in
the decade 2011-2020 and reaching 1.1 GtC yr$^{-1}$ in 2020. The $S_{OCEAN}$ trend diverges with a
factor two difference since 2002 (GOBMs: 0.3 ± 0.1 GtC yr$^{-1}$ per decade, data-products: 0.7 ±
0.2 GtC yr$^{-1}$ per decade, best estimate: 0.5 GtC yr$^{-1}$ per decade) and with a factor of three
since 2010 (GOBMs: 0.3 ± 0.1 GtC yr$^{-1}$ per decade, data-products: 0.9 ± 0.3 GtC yr$^{-1}$ per
decade , best estimate: 0.6 GtC yr$^{-1}$ per decade). The GOBMs estimate is lower than in the
previous global carbon budget (Friedlingstein et al., 2020), because one high-sink model was
not available. The effect of two models (CNRM, MOM6-COBALT) revising their estimates
downwards was largely balanced by two models revising their estimate upwards (FESOM-
REcoM, PlankTOM).
The discrepancy between the two types of estimates stems mostly from a larger Southern
Ocean sink in the data-products prior to 2001, and from a larger $S_{OCEAN}$ trend in the northern
and southern extra-tropics since then (Fig. 12). Possible explanations for the discrepancy in
the Southern Ocean could be missing winter observations and data sparsity in general
(Bushinsky et al., 2019, Gloege et al., 2021), model biases (as indicated by the large model
spread in the South, Figure 12, and the larger model-data mismatch, Figure B2), or
uncertainties in the regional river flux adjustment (Hauck et al., 2020, Lacroix et al., 2020).
During 2010-2016, the ocean $CO_2$ sink appears to have intensified in line with the expected
increase from atmospheric $CO_2$ (McKinley et al., 2020). This effect is stronger in the $fCO_2$-
based data products (Fig. 9, GOBMs: +0.43 GtC yr$^{-1}$, data-products: +0.56 GtC yr$^{-1}$). The
reduction of -0.09 GtC yr$^{-1}$ (range: -0.30 to +0.12 GtC yr$^{-1}$) in the ocean $CO_2$ sink in 2017 is



consistent with the return to normal conditions after the El Niño in 2015/16, which caused
an enhanced sink in previous years. After 2017, the GOBMs ensemble mean suggests the
ocean sink levelling off at about 2.5 GtC yr$^{-1}$, whereas the data-products' estimate increases
by 0.3 GtC yr$^{-1}$ over the same period.
**3.5.3   Final year 2020**
The estimated ocean $CO_2$ sink was 3.0 ± 0.4 GtC in 2020. This is the average of GOBMs and
data-products, and is a small increase of 0.02 GtC compared to 2019, in line with the
competing effects from an expected sink strengthening from atmospheric $CO_2$ growth and
expected sink weakening from La Nina conditions. There is, however, a substantial
difference between GOBMs and $fCO_2$-based data-products in their mean 2020 $S_{OCEAN}$
estimate (GOBMs: 2.5 GtC, data-products: 3.5 GtC). While the GOBMs simulate a stagnation
of the sink from 2019 to 2020 (-0.02 ±0.11 GtCGtC), the data-products suggest an increase
by 0.06 GtC, although not significant at the 1σ level (±0.13 GtC). Four models and four data
products show an increase of $S_{OCEAN}$ (GOBMs up to +0.18 GtC, data-product up to +0.21
GtC), while four models and three data products show no change or a decrease of $S_{OCEAN}$
(GOBMs down to -0.12 GtC, data-products down to -0.13 GtC; Fig. 9). The data-products
have a larger uncertainty at the tails of the reconstructed time series (e.g., Watson et al.,
2020). Specifically, the data-products' estimate of the last year is regularly adjusted in the
following release owing to the tail effect and an incrementally increasing data availability
with 1-5 years lag (Figure 9 bottom).
**3.5.4   Year 2021 Projection**
Using a feed-forward neural network method (see section 2.4) we project an ocean sink of
2.9 GtC for 2021.  This is a reduction of the sink by 0.1 GtC relative to the 2020 value which
we attribute to La Niña conditions in January to May 2021 and projections of a re-
emergence of La Niña later in the year.
**3.5.5   Model Evaluation**
The evaluation of the ocean estimates (Fig. B2) shows an RMSE from annually detrended
data of 1.3 to 2.8 µatm for the seven $fCO_2$-based data products over the globe, relative to
the $fCO_2$ observations from the SOCAT v2021 dataset for the period 1990-2020. The GOBMs





RMSEs are larger and range from 3.3 to 5.9 µatm. The RMSEs are generally larger at high
latitudes compared to the tropics, for both the data products and the GOBMs. The data
products have RMSEs of 1.3 to 3.6 µatm in the tropics, 1.3 to 2.7 µatm in the north, and 2.2
to 6.1 µatm in the south. Note that the data products are based on the SOCAT v2021
database, hence the latter are not independent dataset for the evaluation of the data
products. The GOBMs RMSEs are more spread across regions, ranging from 2.7 to 4.3 µatm
in the tropics, 2.9 to 6.9 µatm in the North, and 6.4 to 9.8 µatm in the South. The higher
RMSEs occur in regions with stronger climate variability, such as the northern and southern
high latitudes (poleward of the subtropical gyres). The upper-range of the model RMSEs
have decreased somewhat relative to Friedlingstein et al. (2020), owing to one model with
upper-end RMSE not being represented this year, and the reduction of RMSE in one model
(MPIOM-HAMOCC6), presumably related to the inclusion of riverine carbon fluxes.
The additional simulation C allows to separate the steady-state anthropogenic carbon
component (sim C - sim B) and to compare the model flux and DIC inventory change directly
to the interior ocean estimate of Gruber et al (2019) without further assumptions. The
GOBMs ensemble average of steady-state anthropogenic carbon inventory change 1994-
2007 amounts to 2.1 GtC yr$^{-1}$, and is significantly lower than the 2.6 ± 0.3 GtC yr$^{-1}$ estimated
by Gruber et al (2019). Only the three models with the highest sink estimate fall within the
range reported by Gruber et al. (2019). This suggests that most of the models
underestimates anthropogenic carbon uptake by the ocean likely due to biases in ocean
carbon transport and mixing from the surface mixed layer to the ocean interior.
The reported $S_{OCEAN}$ estimate from GOBMs and data-products is 2.1 ± 0.4 GtC yr$^{-1}$ over the
period 1994 to 2007, which is in agreement with the ocean interior estimate of 2.2 ± 0.4 GtC
yr$^{-1}$ when accounting for the climate effect on the natural $CO_2$ flux of −0.4 ± 0.24 GtC yr$^{-1}$
(Gruber et al., 2019) to match the definition of $S_{OCEAN}$ used here (Hauck et al., 2020). This
comparison depends critically on the estimate of the climate effect on the natural $CO_2$ flux,
which is smaller from the GOBMs (section 3.5.2) than in Gruber et al. (2019).



### 3.6    Land Sink
### 3.6.1    Historical period 1850-2020
Cumulated since 1850, the terrestrial $CO_2$ sink amounts to 195 ± 45 GtC, 30% of total
anthropogenic emissions. Over the historical period, the sink increased in pace with the
anthropogenic emissions exponential increase (Fig. 3b).
### 3.6.2    Recent period 1960-2020
The terrestrial $CO_2$ sink increased from 1.2 ± 0.5 GtC yr$^{-1}$ in the 1960s to 3.1 ± 0.6 GtC yr$^{-1}$
during 2010-2019, with important interannual variations of up to 2 GtC yr$^{-1}$ generally
showing a decreased land sink during El Niño events (Fig. 7), responsible for the
corresponding enhanced growth rate in atmospheric $CO_2$ concentration. The larger land $CO_2$
sink during 2010-2019 compared to the 1960s is reproduced by all the DGVMs in response
to the combined atmospheric $CO_2$ increase and the changes in climate, and consistent with
constraints from the other budget terms (Table 5).
Over the period 1960 to present the increase in the global terrestrial $CO_2$ sink is largely
attributed to the $CO_2$ fertilization effect in the models (Prentice et al., 2001, Piao et al.,
2009), directly stimulating plant photosynthesis and increased plant water use in water
limited systems, with a small negative contribution of climate change (Fig. 10). There is a
range of evidence to support a positive terrestrial carbon sink in response to increasing
atmospheric $CO_2$, albeit with uncertain magnitude (Walker et al., 2021). As expected from
theory the greatest $CO_2$ effect is simulated in the tropical forest regions, associated with
warm temperatures and long growing seasons (Hickler et al., 2008) (Fig. 10a). However,
evidence from tropical intact forest plots indicate an overall decline in the land sink across
Amazonia (1985-2011), attributed to enhanced mortality offsetting productivity gains
(Brienen et al., 2005, Hubau et al., 2020). During 2011-2020 the land sink is positive in all
regions (Fig. 6) with the exception of central and eastern Brazil, Southwest USA and
northern Mexico, Southeast Europe and Central Asia, South Africa, and eastern Australia,
where the negative effects of climate variability and change (i.e. reduced rainfall)
counterbalance $CO_2$ effects. This is clearly visible on Figure 10 where the effects of $CO_2$ (Fig.
10a) and climate (Fig. 10b) as simulated by the DGVMs are isolated. The negative effect of
climate is the strongest in most of South America, Central America, Southwest US and



Central Europe (Fig. 10b). Globally, climate change reduces the land sink by 0.45 ± 0.39 GtC
yr$^{-1}$ (2011-2020).
In the past years several regions experienced record-setting fire events. While global burned
area has declined over the past decades mostly due to declining fire activity in savannas
(Andela et al., 2017), forest fire emissions are rising and have the potential to counter the
negative fire trend in savannas (Zheng et al., 2021). Noteworthy events include the 2019-
2020 Black Summer event in Australia (emissions of roughly 0.2 GtC; van der Velde et al.,
2021) and Siberia in 2021 where emissions approached 0.4 GtC or three times the 1997-
2020 average according to GFED4s. While other regions, including Western US and
Mediterranean Europe, also experienced intense fire seasons in 2021 their emissions are
substantially lower.
Despite these regional negative effects of climate change on $S_{LAND}$, the efficiency of land to
remove anthropogenic $CO_2$ emissions has remained broadly constant over the last six
decades, with a land-borne fraction ($S_{LAND}/(E_{FOS}+E_{LUC})$) of ~30% (Fig 8).
**3.6.3  Final year 2020**
The terrestrial $CO_2$ sink from the DGVMs ensemble was 2.9 ± 1.0 GtC in 2020, slightly below
the decadal average of 3.1 GtC yr$^{-1}$ (Fig. 4, Table 6). We note that the DGVMs estimate for
2020 is significantly larger than the 2.1 ± 0.9 GtC yr$^{-1}$ estimate from the residual sink from
the global budget ($E_{FOS}+E_{LUC}-G_{ATM}-S_{OCEAN}$) (Table 5).
**3.6.4  Year 2021 Projection**
Using a feed-forward neural network method (see section 2.5) we project a land sink of 3.3
GtC for 2021.  This is an increase of the land sink by 0.3 GtC relative to the 2020 value which
we attribute to La Niña conditions in 2021.
**3.6.5  Model Evaluation**
The evaluation of the DGVMs (Fig. B3) shows generally high skill scores across models for
runoff, and to a lesser extent for vegetation biomass, GPP, and ecosystem respiration (Fig.
B3, left panel). Skill score was lowest for leaf area index and net ecosystem exchange, with a
widest disparity among models for soil carbon. Further analysis of the results will be





provided separately, focusing on the strengths and weaknesses in the DGVMs ensemble and
its validity for use in the global carbon budget.
**3.7    Partitioning the carbon sinks**
**3.7.1    Global sinks and spread of estimates**
In the period 2011-2020, the bottom-up view of total global carbon sinks provided by the
GCB ($S_{OCEAN}$ + $S_{LAND}$– $E_{LUC}$) agrees closely with the top-down budget delivered by the
atmospheric inversions. Figure 11 shows both total sink estimates of the last decade split by
land and ocean, which match the difference between $G_{ATM}$ and $E_{FOS}$ to within 0.06–0.17 GtC
yr$^{-1}$ for inverse models, and to 0.3 GtC yr$^{-1}$ for the GCB mean. The latter represents the $B_{IM}$
discussed in Section 3.8, which by design is minimal for the inverse models.
The distributions based on the individual models and data products reveal substantial
spread but converge near the decadal means quoted in Tables 5 and 6. Sink estimates for
$S_{OCEAN}$ and from inverse models are mostly non-Gaussian, while the ensemble of DGVMs
appears more normally distributed justifying the use of a multi-model mean and standard
deviation for their errors in the budget. Noteworthy is that the tails of the distributions
provided by the land and ocean bottom-up estimates would not agree with the global
constraint provided by the fossil fuel emissions and the observed atmospheric $CO_2$ growth
rate ($E_{FOS}$ – $G_{ATM}$). This illustrates the power of the atmospheric joint constraint from $G_{ATM}$
and the global $CO_2$ observation network it derives from.
**3.7.2    Total atmosphere-to-land fluxes**
The total atmosphere-to-land fluxes ($S_{LAND}$ – $E_{LUC}$), calculated here as the difference between
$S_{LAND}$ from the DGVMs and $E_{LUC}$ from the bookkeeping models, amounts to a 1.9 ± 0.9 GtC yr$^{-}$
$^{1}$ sink during 2011-2020 (Table 5). Estimates of total atmosphere-to-land fluxes ($S_{LAND}$ – $E_{LUC}$)
from the DGVMs alone (1.6 ± 0.6 GtC yr$^{-1}$) are consistent with this estimate and also with
the global carbon budget constraint ($E_{FOS}$ – $G_{ATM}$ – $S_{OCEAN}$, 1.7 ± 0.8 GtC yr$^{-1}$ Table 5).
Consistent with the bookkeeping models estimates, the DGVM-based $E_{LUC}$ is substantially
lower than in Friedlingstein et al., (2020) due to the improved land cover forcing (see
section 3.2.2), increasing their total atmosphere-to-land fluxes and hence the consistency
with the budget constraint. For the last decade (2011-2020), the inversions estimate the net



atmosphere-to-land uptake to lie within a range of 1.3 to 2.0 GtC yr$^{-1}$, consistent with the
GCB and DGVMs estimates of $S_{LAND} - E_{LUC}$ (Figure 11, Figure 12 top row).
**3.7.3   Total atmosphere-to-ocean fluxes**
For the 2011-2020 period, the GOBMs (2.5 ± 0.6 GtC yr$^{-1}$) produce a lower estimate for the
ocean sink than the $fCO_2$-based data products (3.1 ± 0.5 GtC yr$^{-1}$), which shows up in Figure
11 as a separate peak in the distribution from the GOBMs (triangle symbols pointing right)
and from the $fCO_2$-based products (triangle symbols pointing left). Atmospheric inversions
(2.6 to 3.1 GtC yr$^{-1}$) also suggest higher ocean uptake in the recent decade (Figure 11, Figure
12 top row). In interpreting these differences, we caution that the riverine transport of
carbon taken up on land and outgassing from the ocean is a substantial (0.6 GtC yr$^{-1}$) and
uncertain term that separates the various methods. A recent estimate of decadal ocean
uptake from observed $O_2/N_2$ ratios (Tohjima et al., 2019) also points towards a larger ocean
sink, albeit with large uncertainty (2012-2016: 3.1 ± 1.5 GtC yr$^{-1}$).
**3.7.4   Regional breakdown and interannual variability**
Figure 12 also shows the latitudinal partitioning of the total atmosphere-to-surface fluxes
excluding fossil $CO_2$ emissions ($S_{OCEAN} + S_{LAND} - E_{LUC}$) according to the multi-model average
estimates from GOBMs and ocean $fCO_2$-based products ($S_{OCEAN}$) and DGVMs ($S_{LAND} - E_{LUC}$),
and from atmospheric inversions ($S_{OCEAN}$ and $S_{LAND} - E_{LUC}$).
**3.7.4.1   North**
Despite being one of the most densely observed and studied regions of our globe, annual
mean carbon sink estimates in the northern extra-tropics (north of 30°N) continue to differ
by about 0.5 GtC yr$^{-1}$. The atmospheric inversions suggest an atmosphere-to-surface sink
($S_{OCEAN} + S_{LAND} - E_{LUC}$) for 2011-2020 of 2.0 to 3.4 GtC yr$^{-1}$, which is higher than the process
models' estimate of 2.1 ± 0.5 GtC yr$^{-1}$ (Fig. 12). The GOBMs (1.1 ± 0.2 GtC yr$^{-1}$), $fCO_2$-based
data products (1.3 ± 0.1 GtC yr$^{-1}$), and inversion models (0.9 to 1.5 GtC yr$^{-1}$) produce
consistent estimates of the ocean sink. Thus, the difference mainly arises from the total land
flux ($S_{LAND} - E_{LUC}$) estimate, which is 1.0 ± 0.4 GtC yr$^{-1}$ in the DGVMs compared to 0.7 to 2.4
GtC yr$^{-1}$ in the atmospheric inversions (Figure 12, second row).
Discrepancies in the northern land fluxes conforms with persistent issues surrounding the
quantification of the drivers of the global net land $CO_2$ flux (Arneth et al., 2017; Huntzinger





et al., 2017) and the distribution of atmosphere-to-land fluxes between the tropics and high
northern latitudes (Baccini et al., 2017; Schimel et al., 2015; Stephens et al., 2007; Ciais et al.
2019; Gaubert et al,. 2019).
In the northern extratropics, the process models, inversions, and $fCO_2$-based data products
consistently suggest that most of the variability stems from the land (Fig. 12). Inversions
generally estimate similar interannual variations (IAV) over land to DGVMs (0.28 – 0.47 vs
0.20 – 0.73 GtC yr$^{-1}$, averaged over 1990-2020), and they have higher IAV in ocean fluxes
(0.03 – 0.19 GtC yr$^{-1}$)  relative to GOBMs  (0.03 – 0.05 GtC yr$^{-1}$, Fig. B2), and $fCO_2$-based data
products (0.03 – 0.09 GtC yr$^{-1}$).
**3.7.4.2  Tropics**
In the tropics (30°S-30°N), both the atmospheric inversions and process models estimate a
total carbon balance ($S_{OCEAN}$+$S_{LAND}$-$E_{LUC}$) that is close to neutral over the past decade. The
GOBMs (0.0 ± 0.3 GtC yr$^{-1}$), $fCO_2$-based data products (0.03 ± 0.2 GtC yr$^{-1}$), and inversion
models (-0.2 to 0.2 GtC yr$^{-1}$) all indicate an approximately neutral tropical ocean flux (see
Fig. B1 for spatial patterns). DGVMs indicate a net land sink ($S_{LAND}$-$E_{LUC}$) of 0.6 ± 0.3 GtC yr$^{-1}$,
whereas the inversion models indicate a net land flux between -0.7 and 0.9 GtC yr$^{-1}$, though
with high uncertainty (Figure 12, third row).
The tropical lands are the origin of most of the atmospheric $CO_2$ interannual variability
(Ahlström et al., 2015), consistently among the process models and inversions (Fig. 12). The
interannual variability in the tropics is similar among the ocean data products (0.07 – 0.15
GtC yr−1) and the models (0.07 – 0.15 GtC yr$^{-1}$, Fig. B2), which is the highest ocean sink
variability of all regions. The DGVMs and inversions indicate that atmosphere-to-land $CO_2$
fluxes are more variable than atmosphere-to-ocean $CO_2$ fluxes in the tropics, with
interannual variability of 0.4 to 1.2 and 0.6 to 1.1 GtC yr$^{-1}$ respectively.
**3.7.4.3  South**
In the southern extra-tropics (south of 30°S), the atmospheric inversions suggest a total
atmosphere-to-surface sink ($S_{OCEAN}$+$S_{LAND}$-$E_{LUC}$) for 2011-2020 of 1.6 to 1.9 GtC yr$^{-1}$, slightly
higher than the process models' estimate of 1.4 ± 0.3 GtC yr$^{-1}$ (Fig. 12). An approximately
neutral total land flux ($S_{LAND}$-$E_{LUC}$) for the southern extra-tropics is estimated by both the
DGVMs (0.02 ± 0.05 GtC yr$^{-1}$) and the inversion models (sink of -0.1 to 0.2 GtC yr$^{-1}$). This



means nearly all carbon uptake is due to oceanic sinks south of 30°S.  The southern ocean
flux in the $fCO_2$-based data products (1.7 ± 0.1 GtC yr$^{-1}$ ) and inversion estimates (1.4 to 1.8
GtCyr-1) is higher than in the GOBMs (1.4 ± 0.3 GtC yr$^{-1}$ ) (Figure 12, bottom row). This might
be explained by the data-products potentially underestimating the winter $CO_2$ outgassing
south of the Polar Front (Bushinsky et al., 2019), by model biases, or by the uncertainty in
the regional distribution of the river flux adjustment (Aumont et al., 2001, Lacroix et al.,
2020) applied to $fCO_2$-based data products and inverse models to isolate the anthropogenic
$S_{OCEAN}$ flux. $CO_2$ fluxes from this region are more sparsely sampled by all methods, especially
in wintertime (Fig. B1).
The interannual variability in the southern extra-tropics is low because of the dominance of
ocean area with low variability compared to land areas. The split between land ($S_{LAND}$-$E_{LUC}$)
and ocean ($S_{OCEAN}$) shows a substantial contribution to variability in the south coming from
the land, with no consistency between the DGVMs and the inversions or among inversions.
This is expected due to the difficulty of separating exactly the land and oceanic fluxes when
viewed from atmospheric observations alone. The $S_{OCEAN}$ interannual variability was found to
be higher in the $fCO_2$-based data products (0.09 to 0.14 GtC yr−1) compared to GOBMs (0.04
to 0.06 GtC yr−1) in 1990-2020 (Fig. B2). Model subsampling experiments recently
illustrated that observation-based products may overestimate decadal variability in the
Southern Ocean carbon sink by 30% due to data sparsity, based on one data product with
the highest decadal variability (Gloege et al., 2021).
**3.7.4.4 Tropical vs northern land uptake**
A continuing conundrum is the partitioning of the global atmosphere-land flux between the
northern hemisphere land, and the tropical land (Stephens et al., 2017; Pan et al., 2011;
Gaubert et al., 2019). It is of importance because each region has its own history of land-use
change, climate drivers, and impact of increasing atmospheric $CO_2$ and nitrogen deposition.
Quantifying the magnitude of each sink is a prerequisite to understanding how each
individual driver impacts the tropical and mid/high-latitude carbon balance.
We define the North-South (N-S) difference as net atmosphere-land flux north of 30N
minus the net atmosphere-land flux south of 30°N. For the inversions, the N-S difference
ranges from -0.1 GtC yr$^{-1}$ to 2.9 GtC yr$^{-1}$ across this year's inversion ensemble with an equal



preference across models for either a small Northern land sink and a tropical land sink
(small N-S difference), a medium Northern land sink and a neutral tropical land flux
(medium N-S difference), or a large Northern land sink and a tropical land source (large N-S
difference).
In the ensemble of DGVMs the N-S difference is 0.5 ± 0.5 GtC $yr^{-1}$, a much narrower range
than the one from inversions. Only three DGVMs have a N-S difference larger than 1.0 GtC
$yr^{-1}$. The larger agreement across DGVMs than across inversions is to be expected as there is
no correlation between Northern and Tropical land sinks in the DGVMs as opposed to the
inversions where the sum of the two regions being well-constrained leads to an anti-
correlation between these two regions. The much smaller spread in the N-S difference
between the DGVMs could help to scrutinize the inverse models further. For example, a
large northern land sink and a tropical land source in an inversion would suggest a large
sensitivity to $CO_2$ fertilization (the dominant factor driving the land sinks) for Northern
ecosystems, which would be not mirrored by tropical ecosystems. Such a combination could
be hard to reconcile with the process understanding gained from the DGVMs ensembles and
independent measurements (e.g., FACE experiments). Such investigations will be further
pursued in the upcoming assessment from REgional Carbon Cycle Assessment and Processes
(RECCAP2; Ciais et al., 2020).
**3.8   Closing the Global Carbon Cycle**
**3.8.1   Partitioning of Cumulative Emissions and Sink Fluxes**
The global carbon budget over the historical period (1850-2020) is shown in Fig. 3.
Emissions during the period 1850-2020 amounted to 660 ± 65 GtC and were partitioned
among the atmosphere (270 ± 5 GtC; 41%), ocean (170 ± 35 GtC; 26%), and the land (195 ±
45 GtC; 30%). The cumulative land sink is almost equal to the cumulative land-use emissions
(200 ± 65 GtC), making the global land nearly neutral over the whole 1850-2020 period.
The use of nearly independent estimates for the individual terms shows a cumulative
budget imbalance of 25 GtC (4%) during 1850-2020 (Fig. 3, Table 8), which, if correct,
suggests that emissions are slightly too high by the same proportion (4%) or that the
combined land and ocean sinks are slightly underestimated (by about 7%). The bulk of the
imbalance could originate from the estimation of large $E_{LUC}$ between the mid 1920s and the





mid 1960s which is unmatched by a growth in atmospheric $CO_2$ concentration as recorded in
ice cores (Fig. 3). However, the known loss of additional sink capacity of 30-40 GtC (over the
1850-2020 period) due to reduced forest cover has not been accounted for in our method
and would further exacerbate the budget imbalance (Section 2.7.4).
For the more recent 1960-2020 period where direct atmospheric $CO_2$ measurements are
available, 375 ± 20 GtC (82%) of the total emissions ($E_{FOS}$ + $E_{LUC}$) were caused by fossil $CO_2$
emissions, and 80 ± 45 GtC (18%) by land-use change (Table 8). The total emissions were
partitioned among the atmosphere (205 ± 5 GtC; 47%), ocean (115 ± 25 GtC; 25%), and the
land (135 ± 25 GtC; 30%), with a near zero unattributed budget imbalance. All components
except land-use change emissions have significantly grown since 1960, with important
interannual variability in the growth rate in atmospheric $CO_2$ concentration and in the land
$CO_2$ sink (Fig. 4), and some decadal variability in all terms (Table 6). Differences with
previous budget releases are documented in Fig. B5.
The global carbon budget averaged over the last decade (2011-2020) is shown in Fig. 2, Fig.
13 (right panel) and Table 6. For this time period, 90% of the total emissions ($E_{FOS}$ + $E_{LUC}$)
were from fossil $CO_2$ emissions ($E_{FOS}$), and 10% from land-use change ($E_{LUC}$). The total
emissions were partitioned among the atmosphere (47%), ocean (26%) and land (29%), with
a near-zero unattributed budget imbalance (~3%). For single years, the budget imbalance
can be larger (Figure 4). For 2020, the combination of our sources and sinks estimates leads
to a $B_{IM}$ of -0.8 GtC, suggesting an underestimation of the anthropogenic sources
(potentially $E_{LUC}$), and/or an overestimation of the combined land and ocean sinks
**3.8.2  Carbon Budget Imbalance**
The carbon budget imbalance ($B_{IM}$; Eq. 1, Fig.4) quantifies the mismatch between the
estimated total emissions and the estimated changes in the atmosphere, land, and ocean
reservoirs. The mean budget imbalance from 1960 to 2020 is very small (average of 0.03 GtC
$yr^{-1)}$) and shows no trend over the full time series. The process models (GOBMs and DGVMs)
and data-products have been selected to match observational constraints in the 1990s, but
no further constraints have been applied to their representation of trend and variability.
Therefore, the near-zero mean and trend in the budget imbalance is seen as evidence of a
coherent community understanding of the emissions and their partitioning on those time



scales (Fig. 4). However, the budget imbalance shows substantial variability of the order of
±1 GtC yr$^{-1}$, particularly over semi-decadal time scales, although most of the variability is
within the uncertainty of the estimates. The positive carbon imbalance during the 1960s,
and early 1990s, indicates that either the emissions were overestimated, or the sinks were
underestimated during these periods. The reverse is true for the 1970s, 1980s, and for the
2011-2020 period (Fig. 4, Table 6).
We cannot attribute the cause of the variability in the budget imbalance with our analysis,
we only note that the budget imbalance is unlikely to be explained by errors or biases in the
emissions alone because of its large semi-decadal variability component, a variability that is
untypical of emissions and has not changed in the past 60 years despite a near tripling in
emissions (Fig. 4). Errors in $S_{LAND}$ and $S_{OCEAN}$ are more likely to be the main cause for the
budget imbalance. For example, underestimation of the $S_{LAND}$ by DGVMs has been reported
following the eruption of Mount Pinatubo in 1991 possibly due to missing responses to
changes in diffuse radiation (Mercado et al., 2009). Although in GCB2021 we have for the
first time accounted for aerosol effects on solar radiation quantity and quality (diffuse vs
direct), most DGVMs only used the former as input (i.e., total solar radiation). Thus, the
ensemble mean may not capture the full effects of volcanic eruptions, i.e. associated with
high light scattering sulphate aerosols, on the land carbon sink (O'Sullivan et al., 2021).
DGVMs are suspected to overestimate the land sink in response to the wet decade of the
1970s (Sitch et al., 2008). Quasi-decadal variability in the ocean sink has also been reported,
with all methods agreeing on a smaller than expected ocean $CO_2$ sink in the 1990s and a
larger than expected sink in the 2000s (Fig. 9; Landschützer et al., 2016, DeVries et al., 2019,
Hauck et al., 2020, McKinley et al., 2020). Errors in sink estimates could also be driven by
errors in the climatic forcing data, particularly precipitation for $S_{LAND}$ and wind for $S_{OCEAN}$.
The budget imbalance ($B_{IM}$) was negative (-0.3 GtC yr$^{-1}$) on average over 2011-2020,
although the $B_{IM}$ uncertainty is large (1.1 GtC yr$^{-1}$ over the decade). Also, the $B_{IM}$ shows
substantial departure from zero on yearly time scales (Fig. 4), highlighting unresolved
variability of the carbon cycle, likely in the land sink ($S_{LAND}$), given its large year to year
variability (Fig. 4e and 7).
Both the budget imbalance ($B_{IM}$, Table 6) and the residual land sink from the global budget
($E_{FOS}+E_{LUC}-G_{ATM}-S_{OCEAN}$, Table 5) include an error term due to the inconsistencies that arises



from using $E_{LUC}$ from bookkeeping models, and $S_{LAND}$ from DGVMs, most notably the loss of
additional sink capacity (see section 2.7). Other differences include a better accounting of
land use changes practices and processes in bookkeeping models than in DGVMs, or the
bookkeeping models error of having present-day observed carbon densities fixed in the
past. That the budget imbalance shows no clear trend towards larger values over time is an
indication that these inconsistencies probably play a minor role compared to other errors in
$S_{LAND}$ or $S_{OCEAN}$.
Although the budget imbalance is near zero for the recent decades, it could be due to
compensation of errors. We cannot exclude an overestimation of $CO_2$ emissions, particularly
from land-use change, given their large uncertainty, as has been suggested elsewhere (Piao
et al., 2018), combined with an underestimate of the sinks. A larger $S_{LAND}$ would reconcile
model results with inversion estimates for fluxes in the total land during the past decade
(Fig. 12; Table 5). Likewise, a larger $S_{OCEAN}$ is also possible given the higher estimates from
the data-products (see section 3.1.2, Fig. 9 and Fig. 12) and the recently suggested upward
correction of the ocean carbon sink (Watson et al., 2020, Fig. 9). If $S_{OCEAN}$ were to be based
on data-products alone, with all data-products including the Watson et al. (2020)
adjustment, this would result in a 2011-2020 $S_{OCEAN}$ of nearly 4 GtC $yr^{-1}$, outside of the range
supported by the atmospheric inversions, with a negative $B_{IM}$ of more than 1 GtC $yr^{-1}$
indicating that a closure of the budget could only be achieved with either anthropogenic
emissions being larger and/or the net land sink being substantially smaller than estimated
here. More integrated use of observations in the Global Carbon Budget, either on their own
or for further constraining model results, should help resolve some of the budget imbalance
(Peters et al., 2017).
**4    Tracking progress towards mitigation targets**
Fossil $CO_2$ emissions growth peaked at 3% per year during the 2000s, driven by the rapid
growth in Chinese emissions. In the last decade, however, the growth rate for the preceding
10 years has slowly declined, reaching a low 0.4% per year from 2012-2021 (including the
2020 global decline and the expected 2021 emissions rebound). While this slowdown in
global fossil $CO_2$ emissions growth is welcome, it is far from what is needed to be consistent
with the temperature goals of the Paris Agreement.



1. Since the 1990s, the average growth rate of fossil $CO_2$ emissions has continuously declined
2. across the group of developed countries of the Organisation for Economic Co-operation and
3. Development (OECD), with emissions peaking in around 2005 and now declining at around
4. 1% yr$^{-1}$ (Le Quéré et al., 2021). In the decade 2010-2019, territorial fossil $CO_2$ emissions
5. decreased significantly (at the 95% confidence level) in 23 countries whose economies grew
6. significantly (also at the 95% confidence level): Barbados, Belgium, Croatia, Czech Republic,
7. Denmark, Finland, France, Germany, Israel, Japan, Luxembourg, North Macedonia, Malta,
8. Mexico, Netherlands, Slovakia, Slovenia, Solomon Islands, Sweden, Switzerland, Tuvalu,
9. United Kingdom and the USA (updated from Le Quéré et al., 2019). Altogether, these 23
10. countries contribute to 2.5 GtC yr$^{-1}$ over the last decade, about one quarter of world $CO_2$
11. fossil emissions. Consumption-based emissions are also falling significantly in 15 of these
12. countries (Belgium, Croatia, Czech Republic, Denmark, Finland, France, Germany, Israel,
13. Japan, Mexico, Netherlands, Slovenia, Sweden, United Kingdom, and the USA). Figure 14
14. shows that the emission declines in the USA and the EU27 are primarily driven by increased
15. decarbonisation ($CO_2$ emissions per unit energy) in the last decade compared to the
16. previous, with smaller contributions in the EU27 from slightly weaker economic growth and
17. slightly larger declines in energy per GDP. These countries have stable or declining energy
18. use and so decarbonisation policies replace existing fossil fuel infrastructure (Le Quéré et al.
19. 2019).

20. In contrast, fossil $CO_2$ emissions continue to grow in non-OECD countries, although the
21. growth rate has slowed from over 5% yr$^{-1}$ during the 2000s to around 2% yr$^{-1}$ in the last
22. decade. A large part of this slowdown in non-OECD countries is due to China, which has
23. seen emissions growth declining from nearly 10% yr$^{-1}$ in the 2000s to 2% yr$^{-1}$ in the last
24. decade. Excluding China, non-OECD emissions grew at 3% yr$^{-1}$ in the 2000s compared to 2%
25. yr$^{-1}$ in the last decade. Figure 14 shows that compared to the previous decade, China has
26. had weaker economic growth in the last decade and a larger decarbonisation rate, with
27. more rapid declines in energy per GDP which are now back to levels during the 1990s. India
28. and the rest of the world have strong economic growth that is not compensated by
29. decarbonisation or declines in energy per GDP, implying fossil $CO_2$ emissions continue to
30. grow. Despite the high deployment of renewables in some countries (e.g., India), fossil
31. energy sources continue to grow to meet growing energy demand (Le Quéré et al. 2019).



Globally, fossil $CO_2$ emissions growth is slowing, and this is primarily due to the emergence
of climate policy and emission declines in OECD countries (Eskander and Fankhauser 2020).
At the aggregated global level, decarbonisation shows a strong and growing signal in the last
decade, with smaller contributions from lower economic growth and declines in energy per
GDP. Despite the slowing growth in global fossil $CO_2$ emissions, emissions are still growing,
far from the reductions needed to meet the ambitious climate goals of the UNFCCC Paris
agreement.
We update the remaining carbon budget assessed by the IPCC AR6 (Canadell et al., 2021),
accounting for the 2020 and estimated 2021 emissions from fossil fuel combustion ($E_{FOS}$)
and land use changes ($E_{LUC}$). From January 2022, the remaining carbon (50% likelihood) for
limiting global warming to 1.5°C, 1.7°C and 2°C is estimated to amount to 120, 210, and 350
GtC (420, 770, 1270 GtCO₂). These numbers include an uncertainty based on model spread
(as in IPCC AR6), which is reflected through the percent likelihood of exceeding the given
temperature threshold. These remaining abouts correspond to respectively about 11, 20
and 32 years from beginning of 2020, at the 2021 level of total $CO_2$ emissions. Reaching net
zero $CO_2$ emissions by 2050 entails cutting total anthropogenic $CO_2$ emissions by about 0.4
GtC (1.4 GtCO₂) each year on average, comparable to the decrease during 2020.
**5    Discussion**
Each year when the global carbon budget is published, each flux component is updated for
all previous years to consider corrections that are the result of further scrutiny and
verification of the underlying data in the primary input data sets. Annual estimates may be
updated with improvements in data quality and timeliness (e.g., to eliminate the need for
extrapolation of forcing data such as land-use). Of all terms in the global budget, only the
fossil $CO_2$ emissions and the growth rate in atmospheric $CO_2$ concentration are based
primarily on empirical inputs supporting annual estimates in this carbon budget. The carbon
budget imbalance, yet an imperfect measure, provides a strong indication of the limitations
in observations in understanding and representing processes in models, and/or in the
integration of the carbon budget components.
The persistent unexplained variability in the carbon budget imbalance limits our ability to
verify reported emissions (Peters et al., 2017) and suggests we do not yet have a complete



understanding of the underlying carbon cycle dynamics on annual to decadal timescales.
Resolving most of this unexplained variability should be possible through different and
complementary approaches. First, as intended with our annual updates, the imbalance as an
error term is reduced by improvements of individual components of the global carbon
budget that follow from improving the underlying data and statistics and by improving the
models through the resolution of some of the key uncertainties detailed in Table 9. Second,
additional clues to the origin and processes responsible for the variability in the budget
imbalance could be obtained through a closer scrutiny of carbon variability in light of other
Earth system data (e.g., heat balance, water balance), and the use of a wider range of
biogeochemical observations to better understand the land-ocean partitioning of the carbon
imbalance (e.g. oxygen, carbon isotopes). Finally, additional information could also be
obtained through higher resolution and process knowledge at the regional level, and
through the introduction of inferred fluxes such as those based on satellite $CO_2$ retrievals.
The limit of the resolution of the carbon budget imbalance is yet unclear, but most certainly
not yet reached given the possibilities for improvements that lie ahead.
Estimates of global fossil $CO_2$ emissions from different datasets are in relatively good
agreement when the different system boundaries of these datasets are considered
(Andrew, 2020a). But while estimates of $E_{FOS}$ are derived from reported activity data
requiring much fewer complex transformations than some other components of the budget,
uncertainties remain, and one reason for the apparently low variation between datasets is
precisely the reliance on the same underlying reported energy data. The budget excludes
some sources of fossil $CO_2$ emissions, which available evidence suggests are relatively small
(<1%). We have added emissions from lime production in China and the US, but these are
still absent in most other non-Annex I countries, and before 1990 in other Annex I countries.
Further changes to $E_{FOS}$ this year are documented by Andrew and Peters (2021).
Estimates of $E_{LUC}$ suffer from a range of intertwined issues, including the poor quality of
historical land-cover and land-use change maps, the rudimentary representation of
management processes in most models, and the confusion in methodologies and boundary
conditions used across methods (e.g., Arneth et al., 2017; Pongratz et al., 2014, see also
Section 2.7.4 on the loss of sink capacity; Bastos et al., 2021). Uncertainties in current and
historical carbon stocks in soils and vegetation also add uncertainty in the $E_{LUC}$ estimates.



Unless a major effort to resolve these issues is made, little progress is expected in the
resolution of $E_{LUC}$. This is particularly concerning given the growing importance of $E_{LUC}$ for
climate mitigation strategies, and the large issues in the quantification of the cumulative
emissions over the historical period that arise from large uncertainties in $E_{LUC}$.
By adding the DGVMs estimates of $CO_2$ fluxes due to environmental change from countries'
managed forest areas (part of $S_{LAND}$ in this budget) to the budget $E_{LUC}$ estimate, we
successfully reconciled the large gap between our $E_{LUC}$ estimate and the land use flux from
NGHGIs using the approach described in Grassi et al. (2021). This latter estimate has been
used in the recent UNFCCC's Synthesis Report on Nationally Determined Contribution
(UNFCCC, 2021b) to enable the total national emission estimates to be comparable with
those of the IPCC. However, while Grassi et al. (2021) used only one DGVM, here 17 DGVMs
are used, thus providing a more robust value to be used as potential adjustment in the
policy context, e.g., to help assessing the collective countries' progress towards the goal of
the Paris Agreement and avoiding double-accounting for the sink in managed forests. In the
absence of this adjustment, collective progress would hence appear better than it is (Grassi
et al. 2021).
The comparison of GOBMs, data products and inversions highlights substantial discrepancy
in the Southern Ocean (Fig. 12, Hauck et al., 2020). The long-standing sparse data coverage
of $fCO_2$ observations in the Southern compared to the Northern Hemisphere (e.g., Takahashi
et al., 2009) continues to exist (Bakker et al., 2016, 2021, Fig. B1) and to lead to substantially
higher uncertainty in the $S_{OCEAN}$ estimate for the Southern Hemisphere (Watson et al., 2020,
Gloege et al., 2021). This discrepancy, which also hampers model improvement, points to
the need for increased high-quality $fCO_2$ observations especially in the Southern Ocean. At
the same time, model uncertainty is illustrated by the large spread of individual GOBM
estimates (indicated by shading in Fig. 12) and highlights the need for model improvement.
Further uncertainty stems from the regional distribution of the river flux adjustment term
being based on one model study yielding the largest riverine outgassing flux south of 20°S
(Aumont et al., 2001), with a recent study questioning this distribution (Lacroix et al., 2020).
The diverging trends in $S_{OCEAN}$ from different methods is a matter of concern, which is
unresolved. The assessment of the net land-atmosphere exchange from DGVMs and
atmospheric inversions also shows substantial discrepancy, particularly for the estimate of



the total land flux over the northern extra-tropic. This discrepancy highlights the difficulty to
quantify complex processes ($CO_2$ fertilisation, nitrogen deposition and fertilisers, climate
change and variability, land management, etc.) that collectively determine the net land $CO_2$
flux. Resolving the differences in the Northern Hemisphere land sink will require the
consideration and inclusion of larger volumes of observations.
We provide metrics for the evaluation of the ocean and land models and the atmospheric
inversions (Figs. B2 to B4). These metrics expand the use of observations in the global
carbon budget, helping 1) to support improvements in the ocean and land carbon models
that produce the sink estimates, and 2) to constrain the representation of key underlying
processes in the models and to allocate the regional partitioning of the $CO_2$ fluxes. However,
GOBMs skills have changed little since the introduction of the ocean model evaluation. An
additional simulation this year allows for direct comparison with interior ocean
anthropogenic carbon estimates and suggests that the models underestimate
anthropogenic carbon uptake and storage. This is an initial step towards the introduction of
a broader range of observations that we hope will support continued improvements in the
annual estimates of the global carbon budget.
We assessed before that a sustained decrease of –1% in global emissions could be detected
at the 66% likelihood level after a decade only (Peters et al., 2017). Similarly, a change in
behaviour of the land and/or ocean carbon sink would take as long to detect, and much
longer if it emerges more slowly. To continue reducing the carbon imbalance on annual to
decadal time scales, regionalising the carbon budget, and integrating multiple variables are
powerful ways to shorten the detection limit and ensure the research community can
rapidly identify issues of concern in the evolution of the global carbon cycle under the
current rapid and unprecedented changing environmental conditions.
**6     Conclusions**
The estimation of global $CO_2$ emissions and sinks is a major effort by the carbon cycle
research community that requires a careful compilation and synthesis of measurements,
statistical estimates, and model results. The delivery of an annual carbon budget serves two
purposes. First, there is a large demand for up-to-date information on the state of the
anthropogenic perturbation of the climate system and its underpinning causes. A broad



stakeholder community relies on the data sets associated with the annual carbon budget
including scientists, policy makers, businesses, journalists, and non-governmental
organizations engaged in adapting to and mitigating human-driven climate change. Second,
over the last decades we have seen unprecedented changes in the human and biophysical
environments (e.g., changes in the growth of fossil fuel emissions, impact of COVID-19
pandemic, Earth's warming, and strength of the carbon sinks), which call for frequent
assessments of the state of the planet, a better quantification of the causes of changes in
the contemporary global carbon cycle, and an improved capacity to anticipate its evolution
in the future. Building this scientific understanding to meet the extraordinary climate
mitigation challenge requires frequent, robust, transparent, and traceable data sets and
methods that can be scrutinized and replicated. This paper via 'living data' helps to keep
track of new budget updates.
**7      Data availability**
The data presented here are made available in the belief that their wide dissemination will
lead to greater understanding and new scientific insights of how the carbon cycle works,
how humans are altering it, and how we can mitigate the resulting human-driven climate
change. The free availability of these data does not constitute permission for publication of
the data. For research projects, if the data are essential to the work, or if an important
result or conclusion depends on the data, co-authorship may need to be considered for the
relevant data providers. Full contact details and information on how to cite the data shown
here are given at the top of each page in the accompanying database and summarised in
Table 2.
The accompanying database includes two Excel files organised in the following
spreadsheets:
File Global_Carbon_Budget_2021v1.0.xlsx includes the following:
1. Summary
2. The global carbon budget (1959-2020);
3. The historical global carbon budget (1750-2020);
4. Global $CO_2$ emissions from fossil fuels and cement production by fuel type, and the per-
capita emissions (1959-2020);



5. $CO_2$ emissions from land-use change from the individual methods and models (1959-

2        2020);

6. Ocean $CO_2$ sink from the individual ocean models and $fCO_2$-based products (1959-

4        2020);

7. Terrestrial $CO_2$ sink from the DGVMs (1959-2020).
File National_Carbon_Emissions_2021v1.0.xlsx includes the following:
1. Summary
2. Territorial country $CO_2$ emissions from fossil $CO_2$ emissions (1959-2020);
3. Consumption country $CO_2$ emissions from fossil $CO_2$ emissions and emissions transfer

11        from the international trade of goods and services (1990-2019) using CDIAC/UNFCCC

12        data as reference;

4. Emissions transfers (Consumption minus territorial emissions; 1990-2019);
5. Country definitions;
6. Details of disaggregated countries;
7. Details of aggregated countries.
Both spreadsheets are published by the Integrated Carbon Observation System (ICOS)
Carbon Portal and are available at https://doi.org/10.18160/gcp-2021 (Friedlingstein et al.,
2021). National emissions data are also available from the Global Carbon Atlas
(http://www.globalcarbonatlas.org/, last access: 21 October 2021) and from Our World in
Data (https://ourworldindata.org/co2-emissions, last access: 21 October 2021).
**Author contributions**
PF, MWJ, MOS, CLQ, RMA, DCEB, JH, GPP, WP, JP and SS designed the study, conducted the
analysis, and wrote the paper with input from JGC, PC and RBJ. RMA, GPP and JIK produced
the fossil fuel emissions and their uncertainties and analysed the emissions data. DG and
GM provided fossil fuel emission data. JP, TG, CS and RAH provided the bookkeeping land-
use change emissions. JH, LB, OG, NG, TI, LR, JS, RS and DW provided an update of the global



ocean biogeochemical models. SRA, TTTC, LD, LG, YI, PL, CR, AJW and JZ provided an update
of the ocean $fCO_2$ data products, with synthesis by JH. MB, NRB, KIC, MC, WE, RAF, SRA, TG,
AK, NL, SKL, DRM, ClS, CoS, SN, CW, TO, DP, GR, AJS, BT, TT, CW, and RW provided ocean
$fCO_2$ measurements for the year 2020, with synthesis by DCEB and SDJ. PA, BD, AKJ, DK, EK,
JK, SL, PCM, JRM, JEMSN, BP, HT, NV, AJW, WY, XY and SZ provided an update of the
Dynamic Global Vegetation Models, with synthesis by SS. WP, FC, LF, ITL, JL, YN and CR
provided an updated atmospheric inversion, developed the protocol and produced the
evaluation, with synthesis by WP. RMA provided predictions of the 2021 emissions and
atmospheric $CO_2$ growth rate. PL provided the predictions of the 2021 ocean and land sinks.
LPC, GCH, KKG, TMS and GRvdW provided forcing data for land-use change. GG, FT, and CY
provided data for the land-use change NGHGI mapping. PPT provided key atmospheric $CO_2$
data. MWJ produced the historical record of atmospheric $CO_2$ concentration and growth
rate, including the atmospheric $CO_2$ forcing. MOS and NB produced the aerosol diffuse
radiative forcing for the DGVMs. IH provided the climate forcing data for the DGVMs. ER
provided the evaluation of the DGVMs. MWJ provided the emissions prior for use in the
inversion models. XD provided seasonal emissions data for years 2019-2020 for the emission
prior. MWJ and MOS developed a new data management pipeline which automates many
aspects of the data collation, analysis, plotting and synthesis. PF, MWJ, and MOS revised all
figures, tables, text and/or numbers to ensure the update is clear from the 2020 edition and
in phase with the globalcarbonatlas.org.
**Competing interests.** The authors declare that they have no conflict of interest.



**Acknowledgements**
We thank all people and institutions who provided the data used in this global carbon budget
2021 and the Global Carbon Project members for their input throughout the development of
this publication. We thank Nigel Hawtin for producing Figure 2 and Figure 13. We thank Omar
Jamil and Freddy Wordingham for technical support. We thank Ed Dlugokencky for providing
atmospheric $CO_2$ measurements. We thank Vivek Arora, Ian G.C. Ashton, Erik Buitenhuis,
Fatemeh Cheginig, Christian Ethé, Marion Gehlen, Lonneke Goddijn-Murphy, T. Holding,
Fabrice Lacroix, Enhui Liao, Pedro M.S. Monteiro, Naiqing Pan, Tristan Quaife, Shijie Shu,
Jamie D. Shutler, Jade Skye, Anthony Walker, and David K. Woolf for their involvement in the
development, use and analysis of the models and data-products used here. We thank Markus
Ritschel, Carmen Rodriguez, Claire Lo Monaco, Nicolas Metzl, Vassilis Kitidis, Sören Gutekunst,
Anne Willstrand Wranne, Tobias Steinhoff, Jessica N. Cross, Natalie M. Monacci, Alice Benoit-
Cattin, Sólveig R. Ólafsdóttir, Joe Salisbury, Doug Vandemark and Christopher W. Hunt, who
contributed to the provision of surface ocean $CO_2$ observations for the year 2020 (see Table
A5). We also thank Benjamin Pfeil, Rocío Castaño-Primo, Camilla Landa and Maren Karlsen of
the Ocean Thematic Centre of the EU Integrated Carbon Observation System (ICOS) Research
Infrastructure, Kevin O'Brien and Eugene Burger of NOAA's Pacific Marine Environmental
Laboratory and Alex Kozyr of NOAA's National Centers for Environmental Information, for
their contribution to surface ocean $CO_2$ data and metadata management. We thank the
scientists, institutions, and funding agencies responsible for the collection and quality control
of the data in SOCAT as well as the International Ocean Carbon Coordination Project (IOCCP),
the Surface Ocean Lower Atmosphere Study (SOLAS) and the Integrated Marine Biosphere
Research (IMBeR) program for their support. We thank data providers ObsPack



GLOBALVIEWplus v6.1 and NRT v6.1.1 for atmospheric $CO_2$ observations. We thank the
individuals and institutions that provided the databases used for the models evaluations used
here. We thank Fortunat Joos, Samar Khatiwala and Timothy DeVries for providing historical
data. NV thanks the whole ORCHIDEE group. YN thanks CSIRO, EC, EMPA, FMI, IPEN, JMA,
LSCE, NCAR, NIES, NILU, NIWA, NOAA, SIO, and TU/NIPR for providing data for NISMON-CO2.
JL thanks the Jet Propulsion Laboratory, California Institute of Technology. This is PMEL
contribution 5317. SDJ thanks the data management team at the Bjerknes Climate Data
Centre. WE thanks the Tula Foundation for funding support. Australian ocean $CO_2$ data were
sourced from Australia's Integrated Marine Observing System (IMOS); IMOS is enabled by the
National Collaborative Research Infrastructure Strategy (NCRIS). MC thanks Anthony English,
Clynt Gregory and Gordon Furey (P&O Maritime Services) and Tobias Steinhoff for support.
NL thanks the crew of the Cap San Lorenzo and the US IMAGO of IRD Brest for technical
support. GR is grateful for the skillful technical support of M. Glockzin and B. Sadkowiak. MWJ
thanks Anthony J. De-Gol for his technical and conceptual assistance with the development
of GCP-GridFED. FAOSTAT is funded by FAO member states through their contributions to the
FAO Regular Programme, data contributions by national experts are greatly acknowledged.
The views expressed in this paper are the authors' only and do not necessarily reflect those
of FAO. Finally, we thank all funders who have supported the individual and joint
contributions to this work (see Table A9), as well as the reviewers of this manuscript and
previous versions, and the many researchers who have provided feedback.



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

Ramonet, Michel, Delmotte, Marc, Schmidt, Martina, Gheusi, Francois, Mihalopoulos, N.,
Morgui, J.A., Andrews, Arlyn, Dlugokencky, Ed, Lee, John, Sweeney, Colm, Thoning, Kirk,
Tans, Pieter, De Wekker, Stephan, Fischer, Marc L., Jaffe, Dan, McKain, Kathryn, Viner, Brian,
Miller, John B., Karion, Anna, Miller, Charles, Sloop, Christopher D., Saito, Kazuyuki, Aoki,
Shuji, Morimoto, Shinji, Goto, Daisuke, Steinbacher, Martin, Myhre, Cathrine Lund,
Hermanssen, Ove, Stephens, Britton, Keeling, Ralph, Afshar, Sara, Paplawsky, Bill, Cox,
Adam, Walker, Stephen, Schuldt, Kenneth, Mukai, Hitoshi, Machida, Toshinobu, Sasakawa,
Motoki, Nomura, Shohei, Ito, Akihiko, Iida, Yosuke, and Jones, Matthew W.: Long-term
global CO2 fluxes estimated by NICAM-based Inverse Simulation for Monitoring CO2
(NISMON-CO2) (ver.2020.1), https://doi.org/10.17595/20201127.001, 2020.
Obermeier, W. A., Nabel, J. E. M. S., Loughran, T., Hartung, K., Bastos, A., Havermann, F.,
Anthoni, P., Arneth, A., Goll, D. S., Lienert, S., Lombardozzi, D., Luyssaert, S., McGuire, P. C.,
Melton, J. R., Poulter, B., Sitch, S., Sullivan, M. O., Tian, H., Walker, A. P., Wiltshire, A. J.,
Zaehle, S., and Pongratz, J.: Modelled land use and land cover change emissions – a spatio-
temporal comparison of different approaches, 12, 635–670, https://doi.org/10.5194/esd-
21  12-635-2021, 2021.

O'Rourke, P. R., Smith, S. J., Mott, A., Ahsan, H., McDuffie, E. E., Crippa, M., Klimont, Z.,
McDonald, B., Wang, S., Nicholson, M. B., Feng, L., and Hoesly, R. M.: CEDS v_2021_04_21
Release Emission Data, https://doi.org/10.5281/zenodo.4741285, 2021.
Orr, J. C., Najjar, R. G., Aumont, O., Bopp, L., Bullister, J. L., Danabasoglu, G., Doney, S. C.,
Dunne, J. P., Dutay, J.-C., Graven, H., Griffies, S. M., John, J. G., Joos, F., Levin, I., Lindsay, K.,
Matear, R. J., McKinley, G. A., Mouchet, A., Oschlies, A., Romanou, A., Schlitzer, R.,
Tagliabue, A., Tanhua, T., and Yool, A.: Biogeochemical protocols and diagnostics for the
CMIP6 Ocean Model Intercomparison Project (OMIP), 10, 2169–2199,
https://doi.org/10.5194/gmd-10-2169-2017, 2017.





O'Sullivan, M., Zhang, Y., Bellouin, N., Harris, I., Mercado, L. M., Sitch, S., Ciais, P., and
Friedlingstein, P.: (under review) Aerosol-light interactions reduce the carbon budget
imbalance, Environmental Research Letters, 2021.
Palmer, P. I., Feng, L., Baker, D., Chevallier, F., Bösch, H., and Somkuti, P.: Net carbon
emissions from African biosphere dominate pan-tropical atmospheric CO2 signal, Nat
Commun, 10, 3344, https://doi.org/10.1038/s41467-019-11097-w, 2019.
Pan, Y., Birdsey, R. A., Fang, J., Houghton, R., Kauppi, P. E., Kurz, W. A., Phillips, O. L.,
Shvidenko, A., Lewis, S. L., Canadell, J. G., Ciais, P., Jackson, R. B., Pacala, S. W., McGuire, A.
D., Piao, S., Rautiainen, A., Sitch, S., and Hayes, D.: A Large and Persistent Carbon Sink in the
World's Forests, Science, 333, 988–993, https://doi.org/10.1126/science.1201609, 2011.
Patra, P. K., Takigawa, M., Watanabe, S., Chandra, N., Ishijima, K., and Yamashita, Y.:
Improved Chemical Tracer Simulation by MIROC4.0-based Atmospheric Chemistry-Transport
Model (MIROC4-ACTM), SOLA, 14, 91–96, https://doi.org/10.2151/sola.2018-016, 2018.
Pendrill, F., Persson, U. M., Godar, J., Kastner, T., Moran, D., Schmidt, S., and Wood, R.:
Agricultural and forestry trade drives large share of tropical deforestation emissions, Global
Environmental Change, 56, 1–10, https://doi.org/10.1016/j.gloenvcha.2019.03.002, 2019.
Peters, G. P., Andrew, R., and Lennox, J.: Constructing an environmentally-extended multi-
regional input–output table using the GTAP database, Economic Systems Research, 23, 131–
152, https://doi.org/10.1080/09535314.2011.563234, 2011a.
Peters, G. P., Minx, J. C., Weber, C. L., and Edenhofer, O.: Growth in emission transfers via
international trade from 1990 to 2008, Proceedings of the National Academy of Sciences,
108, 8903–8908, https://doi.org/10.1073/pnas.1006388108, 2011b.
Peters, G. P., Davis, S. J., and Andrew, R.: A synthesis of carbon in international trade,
Biogeosciences, 9, 3247–3276, https://doi.org/10.5194/bg-9-3247-2012, 2012a.
Peters, G. P., Marland, G., Le Quéré, C., Boden, T., Canadell, J. G., and Raupach, M. R.: Rapid
growth in CO2 emissions after the 2008–2009 global financial crisis, Nature Clim Change, 2,
2–4, https://doi.org/10.1038/nclimate1332, 2012b.



Peters, G. P., Andrew, R. M., Boden, T., Canadell, J. G., Ciais, P., Le Quéré, C., Marland, G.,
Raupach, M. R., and Wilson, C.: The challenge to keep global warming below 2 °C, Nature
Clim Change, 3, 4–6, https://doi.org/10.1038/nclimate1783, 2013.
Peters, G. P., Le Quéré, C., Andrew, R. M., Canadell, J. G., Friedlingstein, P., Ilyina, T.,
Jackson, R. B., Joos, F., Korsbakken, J. I., McKinley, G. A., Sitch, S., and Tans, P.: Towards real-
time verification of CO2 emissions, Nature Clim Change, 7, 848–850,
https://doi.org/10.1038/s41558-017-0013-9, 2017.
Peters, G. P., Andrew, R. M., Canadell, J. G., Friedlingstein, P., Jackson, R. B., Korsbakken, J.
I., Le Quéré, C., and Peregon, A.: Carbon dioxide emissions continue to grow amidst slowly
emerging climate policies, Nat. Clim. Chang., 10, 3–6, https://doi.org/10.1038/s41558-019-

11  0659-6, 2020.

Petrescu, A. M. R., Peters, G. P., Janssens-Maenhout, G., Ciais, P., Tubiello, F. N., Grassi, G.,
Nabuurs, G.-J., Leip, A., Carmona-Garcia, G., Winiwarter, W., Höglund-Isaksson, L., Günther,
D., Solazzo, E., Kiesow, A., Bastos, A., Pongratz, J., Nabel, J. E. M. S., Conchedda, G., Pilli, R.,
Andrew, R. M., Schelhaas, M.-J., and Dolman, A. J.: European anthropogenic AFOLU
greenhouse gas emissions: a review and benchmark data, Earth Syst. Sci. Data, 12, 961–
1001, https://doi.org/10.5194/essd-12-961-2020, 2020.
Pfeil, B., Olsen, A., Bakker, D. C. E., Hankin, S., Koyuk, H., Kozyr, A., Malczyk, J., Manke, A.,
Metzl, N., Sabine, C. L., Akl, J., Alin, S. R., Bates, N., Bellerby, R. G. J., Borges, A., Boutin, J.,
Brown, P. J., Cai, W.-J., Chavez, F. P., Chen, A., Cosca, C., Fassbender, A. J., Feely, R. A.,
González-Dávila, M., Goyet, C., Hales, B., Hardman-Mountford, N., Heinze, C., Hood, M.,
Hoppema, M., Hunt, C. W., Hydes, D., Ishii, M., Johannessen, T., Jones, S. D., Key, R. M.,
Körtzinger, A., Landschützer, P., Lauvset, S. K., Lefèvre, N., Lenton, A., Lourantou, A.,
Merlivat, L., Midorikawa, T., Mintrop, L., Miyazaki, C., Murata, A., Nakadate, A., Nakano, Y.,
Nakaoka, S., Nojiri, Y., Omar, A. M., Padin, X. A., Park, G.-H., Paterson, K., Perez, F. F.,
Pierrot, D., Poisson, A., Ríos, A. F., Santana-Casiano, J. M., Salisbury, J., Sarma, V. V. S. S.,
Schlitzer, R., Schneider, B., Schuster, U., Sieger, R., Skjelvan, I., Steinhoff, T., Suzuki, T.,
Takahashi, T., Tedesco, K., Telszewski, M., Thomas, H., Tilbrook, B., Tjiputra, J., Vandemark,
D., Veness, T., Wanninkhof, R., Watson, A. J., Weiss, R., Wong, C. S., and Yoshikawa-Inoue,
H.: A uniform, quality controlled Surface Ocean CO2 Atlas (SOCAT), Earth Syst. Sci. Data, 5,



125–143, https://doi.org/10.5194/essd-5-125-2013, 2013.
Piao, S., Ciais, P., Friedlingstein, P., de Noblet-Ducoudré, N., Cadule, P., Viovy, N., and Wang,
T.: Spatiotemporal patterns of terrestrial carbon cycle during the 20th century, 23,
https://doi.org/10.1029/2008GB003339, 2009.
Piao, S., Huang, M., Liu, Z., Wang, X., Ciais, P., Canadell, J. G., Wang, K., Bastos, A.,
Friedlingstein, P., Houghton, R. A., Le Quéré, C., Liu, Y., Myneni, R. B., Peng, S., Pongratz, J.,
Sitch, S., Yan, T., Wang, Y., Zhu, Z., Wu, D., and Wang, T.: Lower land-use emissions
responsible for increased net land carbon sink during the slow warming period, Nature
Geosci, 11, 739–743, https://doi.org/10.1038/s41561-018-0204-7, 2018.
Pongratz, J., Reick, C. H., Houghton, R. A., and House, J. I.: Terminology as a key uncertainty
in net land use and land cover change carbon flux estimates, Earth Syst. Dynam., 5, 177–
195, https://doi.org/10.5194/esd-5-177-2014, 2014.
Potapov, P., Hansen, M. C., Laestadius, L., Turubanova, S., Yaroshenko, A., Thies, C., Smith,
W., Zhuravleva, I., Komarova, A., Minnemeyer, S., and Esipova, E.: The last frontiers of
wilderness: Tracking loss of intact forest landscapes from 2000 to 2013, 3, e1600821,
https://doi.org/10.1126/sciadv.1600821, 2017.
Poulter, B., Frank, D. C., Hodson, E. L., and Zimmermann, N. E.: Impacts of land cover and
climate data selection on understanding terrestrial carbon dynamics and the CO2 airborne
fraction, Biogeosciences, 8, 2027–2036, https://doi.org/10.5194/bg-8-2027-2011, 2011.
Poulter, B., Freeborn, P. H., Jolly, W. M., and Varner, J. M.: COVID-19 lockdowns drive
decline in active fires in southeastern United States, PNAS, 118,
https://doi.org/10.1073/pnas.2105666118, 2021.
Prather, M.: Interactive comment on "Carbon dioxide and climate impulse response
functions for the computation of greenhouse gas metrics: a multi-model analysis" by F. Joos
et al., 6, 2012.
Prentice, I. C., Farquhar, G. D., Fasham, M. J. R., Goulden, M. L., Heimann, M., Jaramillo, V.
J., Kheshgi, H. S., Le Quéré, C., Scholes, R. J., and Wallace, D. W. R.: The Carbon Cycle and
Atmospheric Carbon Dioxide, in Climate Change 2001: The Scientific Basis. Contribution of





Working Group I to the Third Assessment Report of the Intergovernmental Panel on Climate
Change, edited by: Houghton, J. T., Ding, Y., Griggs, D. J., Noguer, M., van der Linden, P. J.,
Dai, X., Maskell, K., and Johnson, C. A., Cambridge University Press, Cambridge, United
Kingdom and New York, NY, USA, 183–237, 2001.
Price, J. T. and Warren, R.: Literature Review of the Potential of "Blue Carbon" Activities to
Reduce Emissions, available at: https://avoid-net-uk.cc.ic.ac.uk/wp-
content/uploads/delightful-downloads/2016/03/Literature-review-of-the-potential-of-blue-
carbon-activities-to-reduce-emissions-AVOID2-WPE2.pdf, last access: 25 October 2021,

9  2016.

Qin, Y., Xiao, X., Wigneron, J.-P., Ciais, P., Brandt, M., Fan, L., Li, X., Crowell, S., Wu, X.,
Doughty, R., Zhang, Y., Liu, F., Sitch, S., and Moore, B.: Carbon loss from forest degradation
exceeds that from deforestation in the Brazilian Amazon, Nat. Clim. Chang., 11, 442–448,
https://doi.org/10.1038/s41558-021-01026-5, 2021.
Qiu, C., Ciais, P., Zhu, D., Guenet, B., Peng, S., Petrescu, A. M. R., Lauerwald, R., Makowski,
D., Gallego-Sala, A. V., Charman, D. J., and Brewer, S. C.: Large historical carbon emissions
from cultivated northern peatlands, 7, eabf1332, https://doi.org/10.1126/sciadv.abf1332,

17  2021.

Raupach, M. R., Marland, G., Ciais, P., Le Quere, C., Canadell, J. G., Klepper, G., and Field, C.
B.: Global and regional drivers of accelerating CO2 emissions, Proceedings of the National
Academy of Sciences, 104, 10288–10293, https://doi.org/10.1073/pnas.0700609104, 2007.
Regnier, P., Friedlingstein, P., Ciais, P., Mackenzie, F. T., Gruber, N., Janssens, I. A., Laruelle,
G. G., Lauerwald, R., Luyssaert, S., Andersson, A. J., Arndt, S., Arnosti, C., Borges, A. V., Dale,
A. W., Gallego-Sala, A., Goddéris, Y., Goossens, N., Hartmann, J., Heinze, C., Ilyina, T., Joos,
F., LaRowe, D. E., Leifeld, J., Meysman, F. J. R., Munhoven, G., Raymond, P. A., Spahni, R.,
Suntharalingam, P., and Thullner, M.: Anthropogenic perturbation of the carbon fluxes from
land to ocean, Nature Geosci, 6, 597–607, https://doi.org/10.1038/ngeo1830, 2013.
Reick, C. H., Gayler, V., Goll, D., Hagemann, S., Heidkamp, M., Nabel, J. E. M. S., Raddatz, T.,
Roeckner, E., Schnur, R., and Wilkenskjeld, S.: JSBACH 3 - The land component of the MPI
Earth System Model: documentation of version 3.2, https://doi.org/10.17617/2.3279802,





1    2021.

Remaud, M., Chevallier, F., Cozic, A., Lin, X., and Bousquet, P.: On the impact of recent
developments of the LMDz atmospheric general circulation model on the simulation of CO2
transport, 11, 4489, https://doi.org/10.5194/gmd-11-4489-2018, 2018.
Resplandy, L., Keeling, R. F., Rödenbeck, C., Stephens, B. B., Khatiwala, S., Rodgers, K. B.,
Long, M. C., Bopp, L., and Tans, P. P.: Revision of global carbon fluxes based on a
reassessment of oceanic and riverine carbon transport, Nature Geosci, 11, 504–509,
https://doi.org/10.1038/s41561-018-0151-3, 2018.
Rhein, M., Rintoul, S. R., Aoki, S., Campos, E., Chambers, D., Feely, R. A., Gulev, S., Johnson,
G. C., Josey, S. A., Kostianoy, A., Mauritzen, C., Roemmich, D., and Talley, L. D.:
Observations: Ocean, edited by: Stocker, T. F., Qin, D., Plattner, G.-K., Tignor, M., Allen, S. K.,
Boschung, J., Nauels, A., Xia, Y., Bex, V., and Midgley, P. M., Cambridge University Press,
255–316, 2013.
Rodenbeck, C., Houweling, S., Gloor, M., and Heimann, M.: CO2 flux history 1982–2001
inferred from atmospheric data using a global inversion of atmospheric transport, 46, 2003.
Rödenbeck, C., Keeling, R. F., Bakker, D. C. E., Metzl, N., Olsen, A., Sabine, C., and Heimann,
M.: Global surface-ocean pCO2 and sea–air CO2 flux variability from an observation-driven
ocean mixed-layer scheme, 9, 193–216, https://doi.org/10.5194/os-9-193-2013, 2013.
Rödenbeck, C., Bakker, D. C. E., Metzl, N., Olsen, A., Sabine, C., Cassar, N., Reum, F., Keeling,
R. F., and Heimann, M.: Interannual sea–air CO2 flux variability from an observation-driven
ocean mixed-layer scheme, 11, 4599–4613, https://doi.org/10.5194/bg-11-4599-2014,
22   2014.

Rödenbeck, C., Bakker, D. C. E., Gruber, N., Iida, Y., Jacobson, A. R., Jones, S., Landschützer,
P., Metzl, N., Nakaoka, S., Olsen, A., Park, G.-H., Peylin, P., Rodgers, K. B., Sasse, T. P.,
Schuster, U., Shutler, J. D., Valsala, V., Wanninkhof, R., and Zeng, J.: Data-based estimates of
the ocean carbon sink variability – first results of the Surface Ocean CO2 Mapping
intercomparison (SOCOM), Biogeosciences, 12, 7251–7278, https://doi.org/10.5194/bg-12-
28   7251-2015, 2015.



Rödenbeck, C., Zaehle, S., Keeling, R., and Heimann, M.: History of El Niño impacts on the
global carbon cycle 1957–2017: a quantification from atmospheric CO2 data, 373,
20170303, https://doi.org/10.1098/rstb.2017.0303, 2018.
Roobaert, A., Laruelle, G. G., Landschützer, P., and Regnier, P.: Uncertainty in the global
oceanic CO2 uptake induced by wind forcing: quantification and spatial analysis, 15, 1701–
1720, https://doi.org/10.5194/bg-15-1701-2018, 2018.
Rosan, T. M., Goldewijk, K. K., Ganzenmüller, R., O'Sullivan, M., Pongratz, J., Mercado, L. M.,
Aragao, L. E. O. C., Heinrich, V., Randow, C. V., Wiltshire, A., Tubiello, F. N., Bastos, A.,
Friedlingstein, P., and Sitch, S.: A multi-data assessment of land use and land cover
emissions from Brazil during 2000–2019, Environ. Res. Lett., 16, 074004,
https://doi.org/10.1088/1748-9326/ac08c3, 2021.
Rypdal, K., Paciornik, N., Eggleston, S., Goodwin, J., Irving, W., Penman, J., and Woodfield,
M.: Volume 1: Introduction to the 2006 Guidelines in: 2006 IPCC guidelines for national
greenhouse gas inventories., 2006.
Saatchi, S. S., Harris, N. L., Brown, S., Lefsky, M., Mitchard, E. T. A., Salas, W., Zutta, B. R.,
Buermann, W., Lewis, S. L., Hagen, S., Petrova, S., White, L., Silman, M., and Morel, A.:
Benchmark map of forest carbon stocks in tropical regions across three continents,
Proceedings of the National Academy of Sciences, 108, 9899–9904,
https://doi.org/10.1073/pnas.1019576108, 2011.
Sabine, C. L., Feely, R. A., Gruber, N., Key, R. M., Lee, K., Bullister, J. L., Wanninkhof, R.,
Wong, C. S., Wallace, D. W. R., Tilbrook, B., Millero, F. J., Peng, T.-H., Kozyr, A., Ono, T., and
Rios, A. F.: The Oceanic Sink for Anthropogenic CO2, 305, 367–371,
https://doi.org/10.1126/science.1097403, 2004.
Sarmiento, J. L., Orr, J. C., and Siegenthaler, U.: A perturbation simulation of CO2 uptake in
an ocean general circulation model, 97, 3621–3645, https://doi.org/10.1029/91JC02849,

26   1992.

Sato, M., Hansen, J. E., McCormick, M. P., and Pollack, J. B.: Stratospheric aerosol optical
depths, 1850–1990, 98, 22987–22994, https://doi.org/10.1029/93JD02553, 1993.



Saunois, M., Stavert, A. R., Poulter, B., Bousquet, P., Canadell, J. G., Jackson, R. B., Raymond,
P. A., Dlugokencky, E. J., Houweling, S., Patra, P. K., Ciais, P., Arora, V. K., Bastviken, D.,
Bergamaschi, P., Blake, D. R., Brailsford, G., Bruhwiler, L., Carlson, K. M., Carrol, M., Castaldi,
S., Chandra, N., Crevoisier, C., Crill, P. M., Covey, K., Curry, C. L., Etiope, G., Frankenberg, C.,
Gedney, N., Hegglin, M. I., Höglund-Isaksson, L., Hugelius, G., Ishizawa, M., Ito, A., Janssens-
Maenhout, G., Jensen, K. M., Joos, F., Kleinen, T., Krummel, P. B., Langenfelds, R. L., Laruelle,
G. G., Liu, L., Machida, T., Maksyutov, S., McDonald, K. C., McNorton, J., Miller, P. A.,
Melton, J. R., Morino, I., Müller, J., Murguia-Flores, F., Naik, V., Niwa, Y., Noce, S., O'Doherty,
S., Parker, R. J., Peng, C., Peng, S., Peters, G. P., Prigent, C., Prinn, R., Ramonet, M., Regnier,
P., Riley, W. J., Rosentreter, J. A., Segers, A., Simpson, I. J., Shi, H., Smith, S. J., Steele, L. P.,
Thornton, B. F., Tian, H., Tohjima, Y., Tubiello, F. N., Tsuruta, A., Viovy, N., Voulgarakis, A.,
Weber, T. S., van Weele, M., van der Werf, G. R., Weiss, R. F., Worthy, D., Wunch, D., Yin, Y.,
Yoshida, Y., Zhang, W., Zhang, Z., Zhao, Y., Zheng, B., Zhu, Q., Zhu, Q., and Zhuang, Q.: The
Global Methane Budget 2000–2017, Earth Syst. Sci. Data, 12, 1561–1623,
https://doi.org/10.5194/essd-12-1561-2020, 2020.
Schimel, D., Alves, D., Enting, I. G., Heimann, M., Joos, F., Raynaud, D., Wigley, T., Prater, M.,
Derwent, R., Ehhalt, D., Fraser, P., Sanhueza, E., Zhou, X., Jonas, P., Charlson, R., Rodhe, H.,
Sadasivan, S., Shine, K. P., Fouquart, Y., Ramaswamy, V., Solomon, S., Srinivasan, J.,
Albritton, D., Derwent, R., Isaksen, I., Lal, M., and Wuebbles, D.: Radiative Forcing of Climate
Change, in: Climate Change 1995 The Science of Climate Change, Contribution of Working
Group I to the Second Assessment Report of the Intergovernmental Panel on Climate
Change, edited by: Houghton, J. T., Meira Rilho, L. G., Callander, B. A., Harris, N., Kattenberg,
A., and Maskell, K., Cambridge University Press, Cambridge, United Kingdom and New York,
NY, USA, 1995.
Schimel, D., Stephens, B. B., and Fisher, J. B.: Effect of increasing CO 2 on the terrestrial
carbon cycle, Proc Natl Acad Sci USA, 112, 436–441,
https://doi.org/10.1073/pnas.1407302112, 2015.
Schourup-Kristensen, V., Sidorenko, D., Wolf-Gladrow, D. A., and Völker, C.: A skill
assessment of the biogeochemical model REcoM2 coupled to the Finite Element Sea Ice–
Ocean Model (FESOM 1.3), Geosci. Model Dev., 7, 2769–2802,



https://doi.org/10.5194/gmd-7-2769-2014, 2014.
Schuh, A. E., Jacobson, A. R., Basu, S., Weir, B., Baker, D., Bowman, K., Chevallier, F., Crowell,
S., Davis, K. J., Deng, F., Denning, S., Feng, L., Jones, D., Liu, J., and Palmer, P. I.: Quantifying
the Impact of Atmospheric Transport Uncertainty on CO 2 Surface Flux Estimates, Global
Biogeochem. Cycles, 33, 484–500, https://doi.org/10.1029/2018GB006086, 2019.
Schwinger, J., Goris, N., Tjiputra, J. F., Kriest, I., Bentsen, M., Bethke, I., Ilicak, M., Assmann,
K. M., and Heinze, C.: Evaluation of NorESM-OC (versions 1 and 1.2), the ocean carbon-cycle
stand-alone configuration ofthe Norwegian Earth System Model (NorESM1), Geosci. Model
Dev., 9, 2589–2622, https://doi.org/10.5194/gmd-9-2589-2016, 2016.
Séférian, R., Nabat, P., Michou, M., Saint-Martin, D., Voldoire, A., Colin, J., Decharme, B.,
Delire, C., Berthet, S., Chevallier, M., Sénési, S., Franchisteguy, L., Vial, J., Mallet, M.,
Joetzjer, E., Geoffroy, O., Guérémy, J.-F., Moine, M.-P., Msadek, R., Ribes, A., Rocher, M.,
Roehrig, R., Salas-y-Mélia, D., Sanchez, E., Terray, L., Valcke, S., Waldman, R., Aumont, O.,
Bopp, L., Deshayes, J., Éthé, C., and Madec, G.: Evaluation of CNRM Earth System Model,
CNRM-ESM2-1: Role of Earth System Processes in Present-Day and Future Climate, 11,
4182–4227, https://doi.org/10.1029/2019MS001791, 2019.
Sellar, A. A., Jones, C. G., Mulcahy, J. P., Tang, Y., Yool, A., Wiltshire, A., O'Connor, F. M.,
Stringer, M., Hill, R., Palmieri, J., Woodward, S., Mora, L., Kuhlbrodt, T., Rumbold, S. T.,
Kelley, D. I., Ellis, R., Johnson, C. E., Walton, J., Abraham, N. L., Andrews, M. B., Andrews, T.,
Archibald, A. T., Berthou, S., Burke, E., Blockley, E., Carslaw, K., Dalvi, M., Edwards, J.,
Folberth, G. A., Gedney, N., Griffiths, P. T., Harper, A. B., Hendry, M. A., Hewitt, A. J.,
Johnson, B., Jones, A., Jones, C. D., Keeble, J., Liddicoat, S., Morgenstern, O., Parker, R. J.,
Predoi, V., Robertson, E., Siahaan, A., Smith, R. S., Swaminathan, R., Woodhouse, M. T.,
Zeng, G., and Zerroukat, M.: UKESM1: Description and Evaluation of the U.K. Earth System
Model, J. Adv. Model. Earth Syst., 11, 4513–4558, https://doi.org/10.1029/2019MS001739,

26  2019.

Shu, S., Jain, A. K., Koven, C. D., and Mishra, U.: Estimation of Permafrost SOC Stock and
Turnover Time Using a Land Surface Model With Vertical Heterogeneity of Permafrost Soils,
34, e2020GB006585, https://doi.org/10.1029/2020GB006585, 2020.



Silva Junior, C. H. L., Pessôa, A. C. M., Carvalho, N. S., Reis, J. B. C., Anderson, L. O., and
Aragão, L. E. O. C.: The Brazilian Amazon deforestation rate in 2020 is the greatest of the
decade, Nat Ecol Evol, 5, 144–145, https://doi.org/10.1038/s41559-020-01368-x, 2021.
Sitch, S., Huntingford, C., Gedney, N., Levy, P. E., Lomas, M., Piao, S. L., Betts, R., Ciais, P.,
Cox, P., Friedlingstein, P., Jones, C. D., Prentice, I. C., and Woodward, F. I.: Evaluation of the
terrestrial carbon cycle, future plant geography and climate-carbon cycle feedbacks using
five Dynamic Global Vegetation Models (DGVMs): Uncertainty In Land Carbon Cycle
Feedbacks, 14, 2015–2039, https://doi.org/10.1111/j.1365-2486.2008.01626.x, 2008.
Smith, B., Wårlind, D., Arneth, A., Hickler, T., Leadley, P., Siltberg, J., and Zaehle, S.:
Implications of incorporating N cycling and N limitations on primary production in an
individual-based dynamic vegetation model, Biogeosciences, 11, 2027–2054,
https://doi.org/10.5194/bg-11-2027-2014, 2014.
Stephens, B. B., Gurney, K. R., Tans, P. P., Sweeney, C., Peters, W., Bruhwiler, L., Ciais, P.,
Ramonet, M., Bousquet, P., Nakazawa, T., Aoki, S., Machida, T., Inoue, G., Vinnichenko, N.,
Lloyd, J., Jordan, A., Heimann, M., Shibistova, O., Langenfelds, R. L., Steele, L. P., Francey, R.
J., and Denning, A. S.: Weak Northern and Strong Tropical Land Carbon Uptake from Vertical
Profiles of Atmospheric CO2, Science, 316, 1732–1735,
https://doi.org/10.1126/science.1137004, 2007.
Stocker, T., Qin, D., and Platner, G.-K.: Climate Change 2013: The Physical Science Basis.
Contribution of Working Group I to the Fifth Assessment Report of the Intergovernmental
Panel on Climate Change, edited by: Intergovernmental Panel on Climate Change,
Cambridge University Press, Cambridge, 2013.
Sweeney, C., Gloor, E., Jacobson, A. R., Key, R. M., McKinley, G., Sarmiento, J. L., and
Wanninkhof, R.: Constraining global air-sea gas exchange for CO2 with recent bomb 14C
measurements, 21, https://doi.org/10.1029/2006GB002784, 2007.
SX Coal: Monthly coal consumption estimates, available at: http://www.sxcoal.com/, last
access: 25 October 2021, 2021.
Takahashi, T., Sutherland, S. C., Wanninkhof, R., Sweeney, C., Feely, R. A., Chipman, D. W.,





Hales, B., Friederich, G., Chavez, F., Sabine, C., Watson, A., Bakker, D. C. E., Schuster, U.,
Metzl, N., Yoshikawa-Inoue, H., Ishii, M., Midorikawa, T., Nojiri, Y., Körtzinger, A., Steinhoff,
T., Hoppema, M., Olafsson, J., Arnarson, T. S., Tilbrook, B., Johannessen, T., Olsen, A.,
Bellerby, R., Wong, C. S., Delille, B., Bates, N. R., and de Baar, H. J. W.: Climatological mean
and decadal change in surface ocean $pCO_2$, and net sea–air $CO_2$ flux over the global oceans,
Deep Sea Research Part II: Topical Studies in Oceanography, 56, 554–577,
https://doi.org/10.1016/j.dsr2.2008.12.009, 2009.
Thomason, L. W., Ernest, N., Millán, L., Rieger, L., Bourassa, A., Vernier, J.-P., Manney, G.,
Luo, B., Arfeuille, F., and Peter, T.: A global space-based stratospheric aerosol climatology:
1979–2016, 10, 469–492, https://doi.org/10.5194/essd-10-469-2018, 2018.
Tian, H., Xu, X., Lu, C., Liu, M., Ren, W., Chen, G., Melillo, J., and Liu, J.: Net exchanges of
$CO_2$, $CH_4$, and $N_2O$ between China's terrestrial ecosystems and the atmosphere and their
contributions to global climate warming, 116, https://doi.org/10.1029/2010JG001393, 2011.
Tian, H., Chen, G., Lu, C., Xu, X., Hayes, D. J., Ren, W., Pan, S., Huntzinger, D. N., and Wofsy,
S. C.: North American terrestrial $CO_2$ uptake largely offset by $CH_4$ and $N_2O$ emissions:
toward a full accounting of the greenhouse gas budget, Climatic Change, 129, 413–426,
https://doi.org/10.1007/s10584-014-1072-9, 2015.
Todd-Brown, K. E. O., Randerson, J. T., Post, W. M., Hoffman, F. M., Tarnocai, C., Schuur, E.
A. G., and Allison, S. D.: Causes of variation in soil carbon simulations from CMIP5 Earth
system models and comparison with observations, Biogeosciences, 10, 1717–1736,
https://doi.org/10.5194/bg-10-1717-2013, 2013.
Tohjima, Y., Mukai, H., Machida, T., Hoshina, Y., and Nakaoka, S.-I.: Global carbon budgets
estimated from atmospheric $O_2/N_2$ and $CO_2$ observations in the western Pacific region over
a 15-year period, 19, 9269–9285, https://doi.org/10.5194/acp-19-9269-2019, 2019.
Tubiello, F. N., Conchedda, G., Wanner, N., Federici, S., Rossi, S., and Grassi, G.: Carbon
emissions and removals from forests: new estimates, 1990–2020, 13, 1681–1691,
https://doi.org/10.5194/essd-13-1681-2021, 2021.
UN: United Nations Statistics Division: National Accounts Main Aggregates Database,



available at: http://unstats.un.org/unsd/ snaama/Introduction.asp, last access: 25 October
2  2021, 2021.

UNFCCC: National Inventory Submissions, available at: https://unfccc.int/ghg-inventories-
annex-i-parties/2021, last access: 25 October 2021., 2021a.
UNFCCC: Nationally determined contributions under the Paris Agreement. Synthesis report
by the secretariat, available at: https://unfccc.int/documents/306848, last access: 25
October 2021, 2021b.
Vale, M. M., Berenguer, E., Argollo de Menezes, M., Viveiros de Castro, E. B., Pugliese de
Siqueira, L., and Portela, R. de C. Q.: The COVID-19 pandemic as an opportunity to weaken
environmental protection in Brazil, Biological Conservation, 255, 108994,
https://doi.org/10.1016/j.biocon.2021.108994, 2021.
van der Laan-Luijkx, I. T., van der Velde, I. R., van der Veen, E., Tsuruta, A., Stanislawska, K.,
Babenhauserheide, A., Zhang, H. F., Liu, Y., He, W., Chen, H., Masarie, K. A., Krol, M. C., and
Peters, W.: The CarbonTracker Data Assimilation Shell (CTDAS) v1.0: implementation and
global carbon balance 2001–2015, Geosci. Model Dev., 10, 2785–2800,
https://doi.org/10.5194/gmd-10-2785-2017, 2017.
van der Velde, I. R., Miller, J. B., Schaefer, K., van der Werf, G. R., Krol, M. C., and Peters, W.:
Terrestrial cycling of 13CO2 by photosynthesis, respiration, and biomass burning in SiBCASA,
11, 6553–6571, https://doi.org/10.5194/bg-11-6553-2014, 2014.
van der Velde, I. R., van der Werf, G. R., Houweling, S., Maasakkers, J. D., Borsdorff, T.,
Landgraf, J., Tol, P., van Kempen, T. A., van Hees, R., Hoogeveen, R., Veefkind, J. P., and
Aben, I.: Vast CO2 release from Australian fires in 2019–2020 constrained by satellite, 597,
366–369, https://doi.org/10.1038/s41586-021-03712-y, 2021.
van der Werf, G. R., Randerson, J. T., Giglio, L., Collatz, G. J., Mu, M., Kasibhatla, P. S.,
Morton, D. C., DeFries, R. S., Jin, Y., and van Leeuwen, T. T.: Global fire emissions and the
contribution of deforestation, savanna, forest, agricultural, and peat fires (1997–2009),
Atmos. Chem. Phys., 10, 11707–11735, https://doi.org/10.5194/acp-10-11707-2010, 2010.
van der Werf, G. R., Randerson, J. T., Giglio, L., van Leeuwen, T. T., Chen, Y., Rogers, B. M.,





Mu, M., van Marle, M. J. E., Morton, D. C., Collatz, G. J., Yokelson, R. J., and Kasibhatla, P. S.:
Global fire emissions estimates during 1997–2016, Earth Syst. Sci. Data, 9, 697–720,
https://doi.org/10.5194/essd-9-697-2017, 2017.
Viovy, N.: CRUNCEP data set, available at:
ftp://nacp.ornl.gov/synthesis/2009/frescati/temp/land_use_change/original/readme.htm,
last access: 25 October 2021, 2016.
Vuichard, N., Messina, P., Luyssaert, S., Guenet, B., Zaehle, S., Ghattas, J., Bastrikov, V., and
Peylin, P.: Accounting for carbon and nitrogen interactions in the global terrestrial
ecosystem model ORCHIDEE (trunk version, rev 4999): multi-scale evaluation of gross
primary production, Geosci. Model Dev., 12, 4751–4779, https://doi.org/10.5194/gmd-12-

11  4751-2019, 2019.

Walker, A. P., Quaife, T., Bodegom, P. M., De Kauwe, M. G., Keenan, T. F., Joiner, J., Lomas,
M. R., MacBean, N., Xu, C., Yang, X., and Woodward, F. I.: The impact of alternative trait-
scaling hypotheses for the maximum photosynthetic carboxylation rate ( V cmax ) on global
gross primary production, New Phytol, 215, 1370–1386,
https://doi.org/10.1111/nph.14623, 2017.
Walker, A. P., De Kauwe, M. G., Bastos, A., Belmecheri, S., Georgiou, K., Keeling, R. F.,
McMahon, S. M., Medlyn, B. E., Moore, D. J. P., Norby, R. J., Zaehle, S., Anderson-Teixeira, K.
J., Battipaglia, G., Brienen, R. J. W., Cabugao, K. G., Cailleret, M., Campbell, E., Canadell, J. G.,
Ciais, P., Craig, M. E., Ellsworth, D. S., Farquhar, G. D., Fatichi, S., Fisher, J. B., Frank, D. C.,
Graven, H., Gu, L., Haverd, V., Heilman, K., Heimann, M., Hungate, B. A., Iversen, C. M., Joos,
F., Jiang, M., Keenan, T. F., Knauer, J., Körner, C., Leshyk, V. O., Leuzinger, S., Liu, Y.,
MacBean, N., Malhi, Y., McVicar, T. R., Penuelas, J., Pongratz, J., Powell, A. S., Riutta, T.,
Sabot, M. E. B., Schleucher, J., Sitch, S., Smith, W. K., Sulman, B., Taylor, B., Terrer, C., Torn,
M. S., Treseder, K. K., Trugman, A. T., Trumbore, S. E., van Mantgem, P. J., Voelker, S. L.,
Whelan, M. E., and Zuidema, P. A.: Integrating the evidence for a terrestrial carbon sink
caused by increasing atmospheric CO2, 229, 2413–2445,
https://doi.org/10.1111/nph.16866, 2021.
Wanninkhof, R.: Relationship between wind speed and gas exchange over the ocean, 97,



7373–7382, https://doi.org/10.1029/92JC00188, 1992.
Wanninkhof, R.: Relationship between wind speed and gas exchange over the ocean
revisited, 12, 351–362, https://doi.org/10.4319/lom.2014.12.351, 2014.
Wanninkhof, R., Park, G.-H., Takahashi, T., Sweeney, C., Feely, R., Nojiri, Y., Gruber, N.,
Doney, S. C., McKinley, G. A., Lenton, A., Le Quéré, C., Heinze, C., Schwinger, J., Graven, H.,
and Khatiwala, S.: Global ocean carbon uptake: magnitude, variability and trends,
Biogeosciences, 10, 1983–2000, https://doi.org/10.5194/bg-10-1983-2013, 2013.
Watson, A. J., Schuster, U., Shutler, J. D., Holding, T., Ashton, I. G. C., Landschützer, P.,
Woolf, D. K., and Goddijn-Murphy, L.: Revised estimates of ocean-atmosphere CO2 flux are
consistent with ocean carbon inventory, Nat Commun, 11, 4422,
https://doi.org/10.1038/s41467-020-18203-3, 2020.
Watson, R. T., Rohde, H., Oeschger, H., and Siegenthaler, U.: Greenhouse Gases and
Aerosols, in: Climate Change: The IPCC Scientific Assessment. Intergovernmental Panel on
Climate Change (IPCC), edited by: Houghton, J. T., Jenkins, G. J., and Ephraums, J. J.,
Cambridge University Press, Cambridge, 140, 1990.
Weiss, R. F. and Price, B. A.: Nitrous oxide solubility in water and seawater, Marine
Chemistry, 8, 347–359, https://doi.org/10.1016/0304-4203(80)90024-9, 1980.
Wenzel, S., Cox, P. M., Eyring, V., and Friedlingstein, P.: Projected land photosynthesis
constrained by changes in the seasonal cycle of atmospheric CO2, Nature, 538, 499–501,
https://doi.org/10.1038/nature19772, 2016.
Wilkenskjeld, S., Kloster, S., Pongratz, J., Raddatz, T., and Reick, C. H.: Comparing the
influence of net and gross anthropogenic land-use and land-cover changes on the carbon
cycle in the MPI-ESM, Biogeosciences, 11, 4817–4828, https://doi.org/10.5194/bg-11-4817-

24  2014, 2014.

Wiltshire, A. J., Burke, E. J., Chadburn, S. E., Jones, C. D., Cox, P. M., Davies-Barnard, T.,
Friedlingstein, P., Harper, A. B., Liddicoat, S., Sitch, S., and Zaehle, S.: JULES-CN: a coupled
terrestrial carbon–nitrogen scheme (JULES vn5.1), 14, 2161–2186,
https://doi.org/10.5194/gmd-14-2161-2021, 2021.





Woodward, F. I. and Lomas, M. R.: Vegetation dynamics – simulating responses to climatic
change, Biol. Rev., 79, 643–670, https://doi.org/10.1017/S1464793103006419, 2004.
Wright, R. M., Le Quéré, C., Buitenhuis, E., Pitois, S., and Gibbons, M. J.: Role of jellyfish in
the plankton ecosystem revealed using a global ocean biogeochemical model, 18, 1291–
1320, https://doi.org/10.5194/bg-18-1291-2021, 2021.
Xi, F., Davis, S. J., Ciais, P., Crawford-Brown, D., Guan, D., Pade, C., Shi, T., Syddall, M., Lv, J.,
Ji, L., Bing, L., Wang, J., Wei, W., Yang, K.-H., Lagerblad, B., Galan, I., Andrade, C., Zhang, Y.,
and Liu, Z.: Substantial global carbon uptake by cement carbonation, Nature Geosci, 9, 880–
883, https://doi.org/10.1038/ngeo2840, 2016.
Xia, J., Chen, Y., Liang, S., Liu, D., and Yuan, W.: Global simulations of carbon allocation
coefficients for deciduous vegetation types, 67, 28016,
https://doi.org/10.3402/tellusb.v67.28016, 2015.
Yin, X.: Responses of leaf nitrogen concentration and specific leaf area to atmospheric $CO_2$
enrichment: a retrospective synthesis across 62 species: Leaf response to atmospheric $co_2$
enrichment, 8, 631–642, https://doi.org/10.1046/j.1365-2486.2002.00497.x, 2002.
Yuan, W., Liu, D., Dong, W., Liu, S., Zhou, G., Yu, G., Zhao, T., Feng, J., Ma, Z., Chen, J., Chen,
Y., Chen, S., Han, S., Huang, J., Li, L., Liu, H., Liu, S., Ma, M., Wang, Y., Xia, J., Xu, W., Zhang,
Q., Zhao, X., and Zhao, L.: Multiyear precipitation reduction strongly decreases carbon
uptake over northern China, 119, 881–896, https://doi.org/10.1002/2014JG002608, 2014.
Yue, C., Ciais, P., Luyssaert, S., Li, W., McGrath, M. J., Chang, J., and Peng, S.: Representing
anthropogenic gross land use change, wood harvest, and forest age dynamics in a global
vegetation model ORCHIDEE-MICT v8.4.2, 11, 409–428, https://doi.org/10.5194/gmd-11-

23  409-2018, 2018.

Yue, X. and Unger, N.: The Yale Interactive terrestrial Biosphere model version 1.0:
description, evaluation and implementation into NASA GISS ModelE2, Geosci. Model Dev., 8,
2399–2417, https://doi.org/10.5194/gmd-8-2399-2015, 2015.
Zaehle, S. and Friend, A. D.: Carbon and nitrogen cycle dynamics in the O-CN land surface
model: 1. Model description, site-scale evaluation, and sensitivity to parameter estimates:





Site-scale evaluation of a C-N model, Global Biogeochem. Cycles, 24,
https://doi.org/10.1029/2009GB003521, 2010.
Zaehle, S., Ciais, P., Friend, A. D., and Prieur, V.: Carbon benefits of anthropogenic reactive
nitrogen offset by nitrous oxide emissions, Nature Geosci, 4, 601–605,
https://doi.org/10.1038/ngeo1207, 2011.
Zeng, J., Nojiri, Y., Landschützer, P., Telszewski, M., and Nakaoka, S.: A Global Surface Ocean
fCO2 Climatology Based on a Feed-Forward Neural Network, 31, 1838–1849,
https://doi.org/10.1175/JTECH-D-13-00137.1, 2014.
Zheng, B., Chevallier, F., Yin, Y., Ciais, P., Fortems-Cheiney, A., Deeter, M. N., Parker, R. J.,
Wang, Y., Worden, H. M., and Zhao, Y.: Global atmospheric carbon monoxide budget 2000–
2017 inferred from multi-species atmospheric inversions, 26, 2019.
Zheng, B., Ciais, P., Chevallier, F., Chuvieco, E., Chen, Y., and Yang, H.: Increasing forest fire
emissions despite the decline in global burned area, 7, eabh2646,
https://doi.org/10.1126/sciadv.abh2646, 2021.
Zscheischler, J., Mahecha, M. D., Avitabile, V., Calle, L., Carvalhais, N., Ciais, P., Gans, F.,
Gruber, N., Hartmann, J., Herold, M., Ichii, K., Jung, M., Landschützer, P., Laruelle, G. G.,
Lauerwald, R., Papale, D., Peylin, P., Poulter, B., Ray, D., Regnier, P., Rödenbeck, C., Roman-
Cuesta, R. M., Schwalm, C., Tramontana, G., Tyukavina, A., Valentini, R., van der Werf, G.,
West, T. O., Wolf, J. E., and Reichstein, M.: Reviews and syntheses: An empirical
spatiotemporal description of the global surface–atmosphere carbon fluxes: opportunities
and data limitations, Biogeosciences, 14, 3685–3703, https://doi.org/10.5194/bg-14-3685-

22   2017, 2017.



# 1   **Tables**

| Table 1. Factors used to convert carbon in various units (by convention, Unit 1 = Unit 2 × conversion). | | | |
|---|---|---|---|
| Unit 1 | Unit 2 | Conversion | Source |
| GtC (gigatonnes of carbon) | ppm (parts per million) (a) | 2.124 (b) | Ballantyne et al. (2012) |
| GtC (gigatonnes of carbon) | PgC (petagrams of carbon) | 1 | SI unit conversion |
| GtCO2 (gigatonnes of carbon dioxide) | GtC (gigatonnes of carbon) | 3.664 | 44.01/12.011 in mass equivalent |
| GtC (gigatonnes of carbon) | MtC (megatonnes of carbon) | 1000 | SI unit conversion |
| (a) Measurements of atmospheric CO2 concentration have units of dry-air mole fraction. 'ppm' is an abbreviation for micromole/mol, dry air. | | | |
| (b) The use of a factor of 2.124 assumes that all the atmosphere is well mixed within one year. In reality, only the troposphere is well mixed and the growth rate of CO2 concentration in the less well-mixed stratosphere is not measured by sites from the NOAA network. Using a factor of 2.124 makes the approximation that the growth rate of CO2 concentration in the stratosphere equals that of the troposphere on a yearly basis. | | | |



| Table 2. How to cite the individual components of the global carbon budget presented here. | |
|---|---|
| **Component** | **Primary reference** |
| Global fossil CO2 emissions (EFOS), total and by fuel type | Andrew and Peters (2021) |
| National territorial fossil CO2 emissions (EFOS) | Gilfillan and Marland (2021), UNFCCC (2021a) |
| National consumption-based fossil CO2 emissions (EFOS) by country (consumption) | Peters et al. (2011b) updated as described in this paper |
| Net land-use change flux (ELUC) | This paper (see Table 4 for individual model references). |
| Growth rate in atmospheric CO2 concentration (GATM) | Dlugokencky and Tans (2021) |
| Ocean and land CO2 sinks (SOCEAN and SLAND) | This paper (see Table 4 for individual model references). |



**Table 3. Main methodological changes in the global carbon budget since 2017. Methodological changes introduced in one year are kept for the following years unless noted. Empty cells mean there were no methodological changes introduced that year. Table A7 lists methodological changes from the first global carbon budget publication up to 2016.**

| Publication year | Fossil fuel emissions | | LUC emissions | Reservoirs | | | Uncertainty & other changes |
|---|---|---|---|---|---|---|---|
| | Global | Country (territorial) | | Atmosphere | Ocean | Land | |
| 2017<br><br>Le Quéré et al. (2018a) GCB2017 | Projection includes India-specific data | | Average of two bookkeeping models; use of 12 DGVMs | | Based on eight models that match the observed sink for the 1990s; no longer normalised | Based on 15 models that meet observation-based criteria (see Sect. 2.5) | Land multi-model average now used in main carbon budget, with the carbon imbalance presented separately; new table of key uncertainties |
| 2018<br><br>Le Quéré et al. (2018b) GCB2018 | Revision in cement emissions; Projection includes EU-specific data | Aggregation of overseas territories into governing nations for total of 213 countries a | Average of two bookkeeping models; use of 16 DGVMs | Use of four atmospheric inversions | Based on seven models | Based on 16 models; revised atmospheric forcing from CRUNCEP to CRU-JRA-55 | Introduction of metrics for evaluation of individual models using observations |
| 2019<br><br>Friedlingstein et al. (2019) GCB2019 | Global emissions calculated as sum of all countries plus bunkers, rather than taken directly from CDIAC. | | Average of two bookkeeping models; use of 15 DGVMs | Use of three atmospheric inversions | Based on nine models | Based on 16 models | |
| 2020<br><br>Friedlingstein et al. (2020) GCB2020 | Cement carbonation now included in the EFOS estimate, reducing EFOS by about 0.2GtC yr-1 for the last decade | India's emissions from Andrew (2020: India); Corrections to Netherland Antilles and Aruba and Soviet emissions before 1950 as per Andrew (2020: CO2); | Average of three bookkeeping models; use of 17 DGVMs. Estimate of gross land use sources and sinks provided | Use of six atmospheric inversions | Based on nine models. River flux revised and partitioned NH, Tropics, SH | Based on 17 models | |





| | | China's coal emissions in 2019 derived from official statistics, emissions now shown for EU27 instead of EU28.Projection for 2020 based on assessment of four approaches. | | | | | |
|---|---|---|---|---|---|---|---|
| 2021 | | Official data included for a number of additional countries, new estimates for South Korea, added emissions from lime production in China. | ELUC estimate compared to the estimates adopted in national GHG inventories (NGHGI) | | Average of means of eight models and means of seven data-products. Current year prediction of SOCEAN using a feed-forward neural network method | Current year prediction of SLAND using a feed-forward neural network method | |
| Friedlingstein et al. (2021) GCB2021 (This study) | Projections are no longer an assessment of four approaches. | | | | | | |





**Table 4. References for the process models, fCO2-based ocean data products, and atmospheric inversions. All models and products are updated with new data to end of year 2020, and the atmospheric forcing for the DGVMs has been updated as described in Section 2.2.2.**

| Model/data name | Reference | Change from Global Carbon Budget 2020 (Friedlingstein et al., 2020) |
|---|---|---|
| *Bookkeeping models for land-use change emissions* | | |
| BLUE | Hansis et al. (2015) | No change to model, but simulations performed with updated LUH2 forcing. |
| updated H&N2017 | Houghton and Nassikas (2017) | Adjustment to treatment of harvested wood products. Update to FRA2020 and 2021 FAOSTAT for forest cover and land-use areas. Forest loss in excess of increases in cropland and pastures represented an increase in shifting cultivation. Extratropical peatland drainage emissions added (based on Qiu et al., 2021). |
| OSCAR | Gasser et al. (2020) | Update to OSCAR3.1.2, which provides finer resolution (96 countries/regions). LUH2-TRENDYv8 input data replaced by LUH2-TRENDYv10. FRA2015 (Houghton & Nassikas, 2017) still used as a second driving dataset, with emissions from FRA2015 extended to 2020. Constraining based on this year's budget data. |
| *Dynamic global vegetation models* | | |
| CABLE-POP | Haverd et al. (2018) | changes in parameterisation, minor bug fixes |
| CLASSIC | Melton et al. (2020) (a) | Non-structural carbohydrates are now explicitly simulated. |
| CLM5.0 | Lawrence et al. (2019) | No Change. |
| DLEM | Tian et al. (2015) (b) | Updated algorithms for land use change processes. |
| IBIS | Yuan et al. (2014) (c) | Several changes in parameterisation; Dynamic carbon allocation scheme. |
| ISAM | Meiyappan et al. (2015) (d) | ISAM now accounting for vertically-resolved soil biogeochemistry (carbon and nitrogen) module (Shu et al., 2020) |
| ISBA-CTRIP | Delire et al. (2020) (e) | Updated spinup protocol + model name updated (SURFEXv8 in GCB2017) + inclusion of crop harvesting module |
| JSBACH | Reick et al. (2021) (f) | Wood product pools per plant functional type. |
| JULES-ES | Wiltshire et al. (2021) (g) | Version 1.1 Inclusion of interactive fire Burton et al., (2019) |
| LPJ-GUESS | Smith et al. (2014) (h) | No code change. Using updated LUH2 and climate forcings. |
| LPJ | Poulter et al. (2011) (i) | Updated soil data from FAO to HWSD v2.0 |



| LPX-Bern | Lienert and Joos (2018) | No Change. |
|---|---|---|
| OCN | Zaehle and Friend (2010) (j) | No change (uses r294). |
| ORCHIDEEv3 | Vuichard et al. (2019) (k) | Updated growth respiration scheme (revision 7267) |
| SDGVM | Walker et al. (2017) (l) | No changes from version used in Friedlingstein et al. (2019), except for properly switching from grasslands to pasture in the blending of the ESA data with LUH2; this change affects mostly the semi-arid lands. |
| VISIT | Kato et al. (2013) (m) | Minor bug fix on CH4 emissions of recent few years. |
| YIBs | Yue and Unger (2015) | Inclusion of nutrient limit with down regulation approach of Arora et al. (2009) |

*Global ocean biogeochemistry models*

| NEMO-PlankTOM12 | Wright et al. (2021) (n) | Updated biochemical model to include 12 functional types. Change to spin-up, now using a looped 1990. |
|---|---|---|
| MICOM-HAMOCC (NorESM-OCv1.2) | Schwinger et al. (2016) | No change |
| MPIOM-HAMOCC6 | Lacroix et al. (2021) | Added riverine fluxes; cmip6 model version including modifications and bug-fixes in HAMOCC and MPIOM |
| NEMO3.6-PISCESv2-gas (CNRM) | Berthet et al. (2019) (o) | small bug fixes; updated model-spin-up (new forcings); atm forcing is now JRA55-Do including 2020 year and varying riverine freshwater inputs |
| FESOM-2.1-REcoM2 | Hauck et al. (2020) (p) | Updated physical model version FESOM2.1, and including 2nd zooplankton and 2nd detritus group. Used new atmospheric CO2 time series provided by GCB |
| MOM6-COBALT (Princeton) | Liao et al. (2020) | Adjustment of the piston velocity prefactor (0.337 cph/m2/s2 to 0.251 cph/m2/s2). MOM6 update from GitHub version b748b1b (2018-10-03) to version 69a096b (2021-02-24). Updated model spin-up and simulation using JRA55-do v1.5. Used new atmospheric CO2 time series provided by GCB. |
| CESM-ETHZ | Doney et al. (2009) | No change in the model. Used new atmospheric CO2 time series provided by GCB |
| NEMO-PISCES (IPSL) | Aumont et al. (2015) | No change |

*ocean fCO2-based data products*

| Landschützer (MPI-SOMFFN) | Landschützer et al. (2016) | update to SOCATv2021 measurements and time period 1982-2020; The estimate now covers the full open ocean and coastal domain as well as the Arctic Ocean extension described in Landschützer et al. (2020) |
|---|---|---|
| Rödenbeck (Jena-MLS) | Rödenbeck et al. (2014) | update to SOCATv2021 measurements, time period extended to 1957-2020, involvement of a multi-linear regression for extrapolation (combined with an explicitly interannual correction), use of OCIM (deVries, 2014) as decadal prior, carbonate chemistry parameterization now time-dependent, grid resolution increased to 2.5*2 degrees, adjustable degrees |



| | | |
|---|---|---|
| | | of freedom now also covering shallow areas and Arctic, some numerical revisions |
| CMEMS-LSCE-FFNNv2 | Chau et al. (2021) | Update to SOCATv2021 measurements and time period 1985-2020. The CMEMS-LSCE-FFNNv2 product now covers both the open ocean and coastal regions (see in Chau et al. 2021 for model description and evaluation). |
| CSIR-ML6 | Gregor et al. (2019) | Updated to SOCATv2021. Reconstruction now spans the period 1985 - 2020 and includes updates using the SeaFlux protocols (Fay et al., 2021b) |
| Watson et al | Watson et al. (2020) | Updated to SOCAT v2021. A monthly climatology of the skin temperature deviation as calculated for years 2003-2011 is now used in place of a single global average figure. SOM calculation updated to treat the Arctic as a separate biome. |
| NIES-NN | Zeng et al. (2014) | New this year |
| JMA-MLR | Iida et al. (2021) | New this year |
| OS-ETHZ-GRaCER | Gregor and Gruber (2021) | New this year |
| *Atmospheric inversions* | | |
| CAMS | Chevallier et al. (2005) (q) | No change. |
| CarbonTracker Europe (CTE) | van der Laan-Luijkx et al. (2017) | No change. |
| Jena CarboScope | Rödenbeck et al. (2018) (r) | No change. |
| UoE in-situ | Feng et al., (2016) (s) | Fossil fuels now from GCP-GridFEDv2021.2 |
| NISMON-CO2 | Niwa et al., (2017) (t) | Some inversion parameters were changed. |
| CMS-Flux | Liu et al., (2021) | New this year |

| |
|---|
| (a) see also Asaadi et al. (2018). |
| (b) see also Tian et al. (2011) |
| (c) the dynamic carbon allocation scheme was presented by Xia et al. (2015) |
| (d) see also Jain et al. (2013). Soil biogeochemistry is updated based on Shu et al. (2020) |
| (e) see also Decharme et al. (2019) and Seferian et al. (2019) |
| (f) Mauritsen et al. (2019) |
| (g) see also Sellar et al. (2019) and Burton et al., (2019). JULES-ES is the Earth System configuration of the Joint UK Land Environment Simulator as used in the UK Earth System Model (UKESM). |
| (h) to account for the differences between the derivation of shortwave radiation from CRU cloudiness and DSWRF from CRUJRA, the photosynthesis scaling parameter $\alpha a$ was modified (-15%) to yield similar results. |



| |
|---|
| (i) compared to published version, decreased LPJ wood harvest efficiency so that 50 % of biomass was removed off-site compared to 85 % used in the 2012 budget. Residue management of managed grasslands increased so that 100 % of harvested grass enters the litter pool. |
| (j) see also Zaehle et al. (2011). |
| (k) see also Zaehle and Friend (2010) and Krinner et al. (2005) |
| (l) see also Woodward and Lomas (2004) |
| (m) see also Ito and Inatomi (2012). |
| (n) see also Buitenhuis et al. (2013) |
| (o) see also Séférian et al. (2019) |
| (p) see also Schourup-Kristensen et al (2014) |
| (q) see also Remaud (2018) |
| (r) see also Rödenbeck et al. (2003) |
| (s) see also Feng et al. (2009) and Palmer et al. (2019) |
| (t) see also Niwa et al. (2020) |





**Table 5. Comparison of results from the bookkeeping method and budget residuals with results from the**
**DGVMs and inverse estimates for different periods, the last decade, and the last year available. All values**
**are in GtCyr−1. The DGVM uncertainties represent ±1σ of the decadal or annual (for 2020 only) estimates**
**from the individual DGVMs: for the inverse models the range of available results is given. All values are**
**rounded to the nearest 0.1 GtC and therefore columns do not necessarily add to zero.**

| | Mean (GtC/yr) | | | | | | |
|---|---|---|---|---|---|---|---|
| | 1960s | 1970s | 1980s | 1990s | 2000s | 2011-2020 | 2020 |
| **Land-use change emissions (ELUC)** | | | | | | | |
| Bookkeeping methods - net flux (1a) | 1.6±0.7 | 1.3±0.7 | 1.2±0.7 | 1.3±0.7 | 1.2±0.7 | 1.1±0.7 | 0.9±0.7 |
| Bookkeeping methods - source | 3.4±0.9 | 3.3±0.8 | 3.4±0.8 | 3.6±0.6 | 3.7±0.6 | 3.8±0.6 | 3.6±0.6 |
| Bookkeeping methods - sink | -1.9±0.4 | -2±0.4 | -2.1±0.3 | -2.3±0.4 | -2.5±0.4 | -2.7±0.4 | -2.8±0.4 |
| DGVMs-net flux (1b) | 1.6±0.5 | 1.3±0.4 | 1.4±0.5 | 1.4±0.5 | 1.4±0.5 | 1.5±0.5 | 1.4±0.7 |
| **Terrestrial sink (SLAND)** | | | | | | | |
| Residual sink from global budget (EFOS+ELUC-GATM-SOCEAN) (2a) | 1.8±0.8 | 1.9±0.8 | 1.6±0.9 | 2.5±0.9 | 2.7±0.9 | 2.8±0.9 | 2.1±0.9 |
| DGVMs (2b) | 1.2±0.5 | 2±0.5 | 1.8±0.5 | 2.3±0.4 | 2.6±0.5 | 3.1±0.6 | 2.9±1 |
| **Total land fluxes (SLAND-ELUC)** | | | | | | | |
| GCB2021 Budget (2b-1a) | -0.4±0.8 | 0.8±0.8 | 0.5±0.9 | 1±0.8 | 1.4±0.9 | 1.9±0.9 | 2±1.2 |
| Budget constraint (2a-1a) | 0.2±0.4 | 0.6±0.5 | 0.3±0.5 | 1.2±0.5 | 1.5±0.6 | 1.7±0.6 | 1.3±0.6 |
| DGVMs-net (2b-1b) | -0.4±0.6 | 0.7±0.4 | 0.3±0.4 | 0.9±0.4 | 1.2±0.4 | 1.6±0.6 | 1.5±0.8 |
| Inversions* | --- | --- | 0.5-0.6 (2) | 0.9-1.2 (3) | 1.3-1.8 (3) | 1.3-2 (6) | -0.1-1.3 (6) |

- Estimates are adjusted for the pre-industrial influence of river fluxes, for the cement carbonation sink, and adjusted to common EFOS (Sect. 2.6). The ranges given include varying numbers (in parentheses) of inversions in each decade (Table A4)



**Table 6. Decadal mean in the five components of the anthropogenic CO2 budget for different periods, and**
**last year available. All values are in GtC yr-1, and uncertainties are reported as ±1σ. Fossil CO$_2$ emissions**
**include cement carbonation. The table also shows the budget imbalance (B$_{IM}$), which provides a measure of**
**the discrepancies among the nearly independent estimates and has an uncertainty exceeding ± 1 GtC yr$^{-1}$. A**
**positive imbalance means the emissions are overestimated and/or the sinks are too small. All values are**
**rounded to the nearest 0.1 GtC and therefore columns do not necessarily add to zero.**

| | Mean (GtC/yr) | | | | | | | |
|---|---|---|---|---|---|---|---|---|
| | 1960s | 1970s | 1980s | 1990s | 2000s | 2011-2020 | 2020 | 2021 (Projection) |
| **Total emissions (EFOS + ELUC)** | | | | | | | | |
| Fossil CO2 emissions (EFOS)* | 3±0.2 | 4.7±0.2 | 5.5±0.3 | 6.3±0.3 | 7.7±0.4 | 9.5±0.5 | 9.3±0.5 | 9.7±0.5 |
| Land-use change emissions (ELUC) | 1.6±0.7 | 1.3±0.7 | 1.2±0.7 | 1.3±0.7 | 1.2±0.7 | 1.1±0.7 | 0.9±0.7 | 0.8±0.7 |
| Total emissions | 4.6±0.7 | 5.9±0.7 | 6.7±0.8 | 7.7±0.8 | 9±0.8 | 10.6±0.8 | 10.2±0.8 | 10.5±0.9 |
| **Partitioning** | | | | | | | | |
| Growth rate in atmos CO2 (GATM) | 1.7±0.07 | 2.8±0.07 | 3.4±0.02 | 3.1±0.02 | 4±0.02 | 5.1±0.02 | 5±0.2 | 4.2±0.4 |
| Ocean sink (SOCEAN) | 1.1±0.4 | 1.3±0.4 | 1.8±0.4 | 2±0.4 | 2.2±0.4 | 2.8±0.4 | 3±0.4 | 2.9±0.4 |
| Terrestrial sink (SLAND) | 1.2±0.5 | 2±0.5 | 1.8±0.5 | 2.3±0.4 | 2.6±0.5 | 3.1±0.6 | 2.9±1 | 3.3±1 |
| **Budget Imbalance** | | | | | | | | |
| BIM=EFOS+ELUC-(GATM+SOCEAN+SLAND) | 0.6 | -0.2 | -0.2 | 0.2 | 0.1 | -0.3 | -0.8 | 0.1 |

- Fossil emissions excluding the cement carbonation sink amount to 3.1±0.2 GtC/yr, 4.7±0.2 GtC/yr, 5.5±0.3 GtC/yr, 6.4±0.3 GtC/yr, 7.9±0.4 GtC/yr, and 9.7±0.5 GtC/yr for the decades 1960s to 2010s respectively and to 9.5±0.5 GtC/yr for 2020.





| | World | | China | | USA | | EU28 (h) | | India | | Rest of World | |
|---|---|---|---|---|---|---|---|---|---|---|---|---|
| | Projected | Actual | Projected | Actual | Projected | Actual | Projected | Actual | Projected | Actual | Projected | Actual |
| 2015 (a) | −0.6% (−1.6 to 0.5) | 0.06% | −3.9% (−4.6 to −1.1) | −0.7% | −1.5% (−5.5 to 0.3) | −2.5% | − | − | − | − | 1.2% (−0.2 to 2.6) | 1.2% |
| 2016 (b) | −0.2% (−1.0 to +1.8) | 0.20% | −0.5% (−3.8 to +1.3) | −0.3% | −1.7% (−4.0 to +0.6) | −2.1% | − | − | − | − | 1.0% (−0.4 to +2.5) | 1.3% |
| 2017 (c) | 2.0% (+0.8 to +3.0) | 1.6% | 3.5% (+0.7 to +5.4) | 1.5% | −0.4% (−2.7 to +1.0) | −0.5% | − | − | 2.00% (+0.2 to +3.8) | 3.9% | 1.6% (0.0 to +3.2) | 1.9% |
| 2018 (d) | 2.7% (+1.8 to +3.7) | 2.1% | 4.7% (+2.0 to +7.4) | 2.3% | 2.5% (+0.5 to +4.5) | 2.8% | -0.7% (-2.6 to +1.3) | -2.1% | 6.3% (+4.3 to +8.3) | 8.0% | 1.8% (+0.5 to +3.0) | 1.7% |
| 2019 (e) | 0.5% (-0.3 to +1.4) | 0.1% | 2.6% (+0.7 to +4.4) | 2.2% | -2.4% (-4.7 to -0.1) | -2.6% | -1.7% (-5.1% to +1.8%) | -4.3% | 1.8% (-0.7 to +3.7) | 1.0% | 0.5% (-0.8 to +1.8) | 0.5% |
| 2020 (f) | -6.7% | -5.4% | -1.7% | 1.4% | -12.2% | -10.6% | -11.3% (EU27) | -10.9% | -9.1% | -7.3% | -7.4% | -7.0% |
| 2021 (g) | 4.9% (4.1% to 5.7%) | | 4.0% (2.1% to 5.8%) | | 7.6% (5.3% to 10.0%) | | 7.6% (5.6% to 9.5%) | | 12.6% (10.7% to 13.6%) | | 2.9% (1.8% to 4.1%) | |

**Table 7. Comparison of the projection with realised fossil CO2 emissions (EFOS). The 'Actual' values are first the estimate available using actual data, and the 'Projected' values refers to estimates made before the end of the year for each publication. Projections based on a different method from that described here during 2008-2014 are available in Le Quéré et al., (2016). All values are adjusted for leap years.**

(a) Jackson et al. (2016) and Le Quéré et al. (2015a). (b) Le Quéré et al. (2016). (c) Le Quéré et al. (2018a). (d) Le Quéré et al. (2018b). (e) Friedlingstein et al., (2019), (f) Friedlingstein et al., (2020), (g) This study (median of four reported estimates, Section 3.4.1.2)

(h) EU28 until 2019, EU27 from 2020





**Table 8. Cumulative CO₂ for different time periods in gigatonnes of carbon (GtC). All uncertainties are reported as**
**±1σ. Fossil CO₂ emissions include cement carbonation. The budget imbalance (B_IM) provides a measure of the**
**discrepancies among the nearly independent estimates. All values are rounded to the nearest 5 GtC and therefore**
**columns do not necessarily add to zero.**

|  | 1750-2020 | 1850-2014 | 1850-2020 | 1960-2020 | 1850-2021 |
|---|---|---|---|---|---|
| Emissions |  |  |  |  |  |
| Fossil CO2 emissions (EFOS) | 460±25 | 400±20 | 455±25 | 375±20 | 465±25 |
| Land-use change emissions (ELUC) | 235±75 | 195±60 | 200±65 | 80±45 | 205±65 |
| Total emissions | 690±80 | 595±65 | 660±65 | 455±45 | 670±65 |
| Partitioning |  |  |  |  |  |
| Growth rate in atmos CO2 (GATM) | 290±5 | 235±5 | 270±5 | 205±5 | 270±5 |
| Ocean sink (SOCEAN) | 180±35 | 150±30 | 170±35 | 115±25 | 170±35 |
| Terrestrial sink (SLAND) | 215±50 | 180±40 | 195±45 | 135±25 | 200±45 |
| Budget imbalance |  |  |  |  |  |
| BIM=EFOS+ELUC-(GATM+SOCEAN+SLAND) | 10 | 30 | 25 | 0 | 25 |





**Table 9. Major known sources of uncertainties in each component of the Global Carbon Budget, defined as input data or processes that have a demonstrated effect of at least ±0.3 GtC yr-1.**

| Source of uncertainty | Time scale (years) | Location | Status | Evidence |
|---|---|---|---|---|
| Fossil CO2 emissions (EFOS; Section 2.1) | | | | |
| energy statistics | annual to decadal | global, but mainly China & major developing countries | see Sect. 2.1 | (Korsbakken et al., 2016, Guan et al., 2012) |
| carbon content of coal | annual to decadal | global, but mainly China & major developing countries | see Sect. 2.1 | (Liu et al., 2015) |
| system boundary | annual to decadal | all countries | see Sect. 2.1 | (Andrew, 2020) |
| Net land-use change flux (ELUC; section 2.2) | | | | |
| land-cover and land-use change statistics | continuous | global; in particular tropics | see Sect. 2.2 | (Houghton et al., 2012; Gasser et al., 2020) |
| sub-grid-scale transitions | annual to decadal | global | see Table A1 | (Wilkenskjeld et al., 2014) |
| vegetation biomass | annual to decadal | global; in particular tropics | see Table A1 | (Houghton et al., 2012) |
| forest degradation (fire, selective logging) | annual to decadal | tropics | | (Aragão et al., 2018; Qin et al., 2020) |
| wood and crop harvest | annual to decadal | global; SE Asia | see Table A1 | (Arneth et al., 2017, Erb et al., 2018) |
| peat burning (a) | multi-decadal trend | global | see Table A1 | (van der Werf et al., 2010, 2017) |
| loss of additional sink capacity | multi-decadal trend | global | not included; see Appendix D1.4 | (Pongratz et al, 2014, Gasser et al, 2020; Obermeier et al., 2021) |
| Atmospheric growth rate (GATM; section 2.3) no demonstrated uncertainties larger than ±0.3 GtC yr-1 (b) | | | | |
| Ocean sink (SOCEAN; section 2.4) | | | | |



| sparsity in surface fCO2 observations | mean, decadal variability and trend | global, in particular southern hemisphere | see Sect 3.5.2 | (Gloege et al., 2021, Denvil-Sommer et al., 2021, Bushinsky et al., 2019) |
|---|---|---|---|---|
| riverine carbon outgassing and its anthropogenic perturbation | annual to decadal | global, in particular partitioning between Tropics and South | see Sect. 2.4 (anthropogenic perturbations not included) | (Aumont et al., 2001, Resplandy et al., 2018, Lacroix et al., 2020) |
| interior ocean anthropogenic carbon storage | annual to decadal | global | see Sect 3.5.5 | (Gruber et al., 2019) |
| near-surface temperature and salinity gradients | mean on all time-scales | global | see Sect. 3.8.2 | (Watson et al., 2020) |
| Land sink (SLAND; section 2.5) | | | | |
| strength of CO2 fertilisation | multi-decadal trend | global | see Sect. 2.5 | (Wenzel et al., 2016; Walker et al., 2021) |
| response to variability in temperature and rainfall | annual to decadal | global; in particular tropics | see Sect. 2.5 | (Cox et al., 2013; Jung et al., 2017; Humphrey et al., 2018; 2021) |
| nutrient limitation and supply | | | | |
| tree mortality | annual | global in particular tropics | see Sect. 2.5 | (Hubau et al., 2021; Brienen et al., 2020) |
| response to diffuse radiation | annual | global | see Sect. 2.5 | (Mercado et al., 2009; O'Sullivan et al., 2021) |
| a As result of interactions between land-use and climate | | | | |
| b The uncertainties in GATM have been estimated as ±0.2 GtC yr-1, although the conversion of the growth rate into a global annual flux assuming instantaneous mixing throughout the atmosphere introduces additional errors that have not yet been quantified. | | | | |

**Figures and Captions**

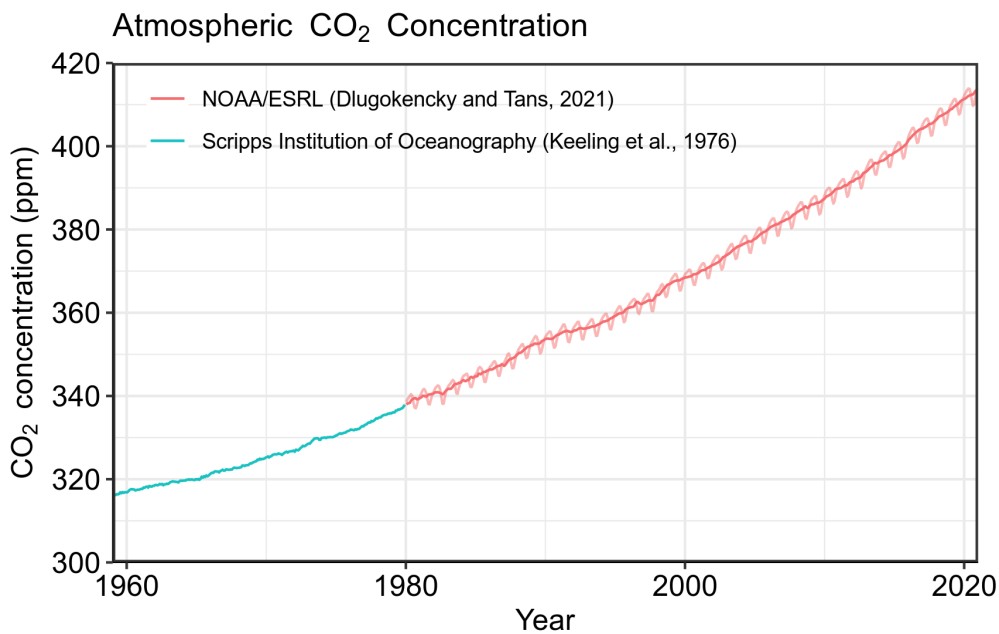

**Figure 1.** Surface average atmospheric $CO_2$ concentration (ppm). Since 1980, monthly data are
from NOAA/ESRL (Dlugokencky and Tans, 2021) and are based on an average of direct
atmospheric $CO_2$ measurements from multiple stations in the marine boundary layer (Masarie and
Tans, 1995). The 1958-1979 monthly data are from the Scripps Institution of Oceanography, based
on an average of direct atmospheric $CO_2$ measurements from the Mauna Loa and South Pole
stations (Keeling et al., 1976). To account for the difference of mean $CO_2$ and seasonality between
the NOAA/ESRL and the Scripps station networks used here, the Scripps surface average (from two
stations) was de-seasonalised and adjusted to match the NOAA/ESRL surface average (from
multiple stations) by adding the mean difference of 0.667 ppm, calculated here from overlapping
data during 1980-2012.

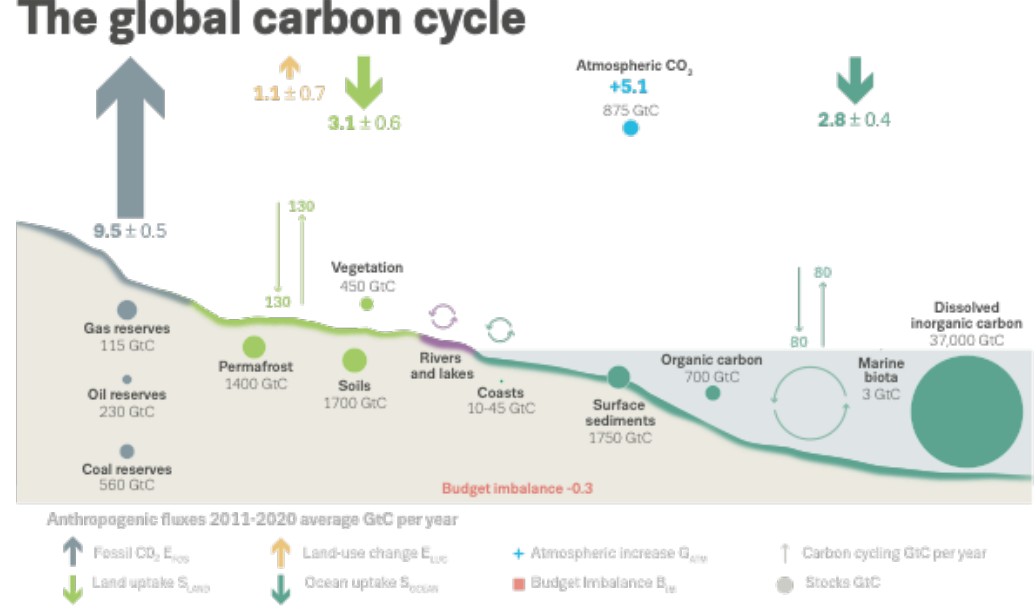

**Figure 2.** Schematic representation of the overall perturbation of the global carbon cycle caused by anthropogenic activities, averaged globally for the decade 2011-2020. See legends for the corresponding arrows and units. The uncertainty in the atmospheric $CO_2$ growth rate is very small (±0.02 GtC yr$^{-1}$) and is neglected for the figure. The anthropogenic perturbation occurs on top of an active carbon cycle, with fluxes and stocks represented in the background and taken from Canadell et al. (2021) for all numbers, except for the carbon stocks in coasts which is from a literature review of coastal marine sediments (Price and Warren, 2016).

Earth System
**Science**
**Data** Open Access Discussions

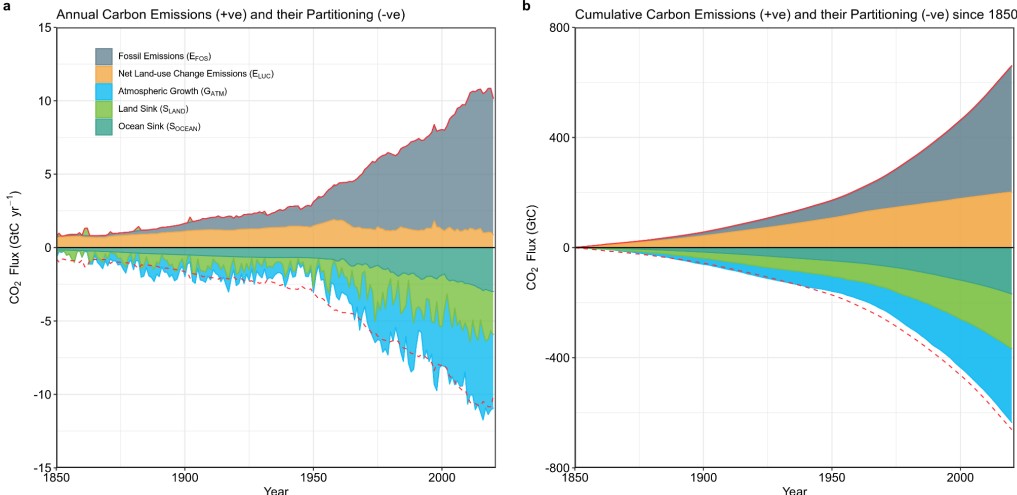

**Figure 3.** Combined components of the global carbon budget illustrated in Fig. 2 as a function of
time, for fossil $CO_2$ emissions ($E_{FOS}$, including a small sink from cement carbonation; grey) and
emissions from land-use change ($E_{LUC}$; brown), as well as their partitioning among the atmosphere
($G_{ATM}$; cyan), ocean ($S_{OCEAN}$; blue), and land ($S_{LAND}$; green). Panel (a) shows annual estimates of
each flux and panel (b) the cumulative flux (the sum of all prior annual fluxes) since the year 1850.
The partitioning is based on nearly independent estimates from observations (for $G_{ATM}$) and from
process model ensembles constrained by data (for $S_{OCEAN}$ and $S_{LAND}$) and does not exactly add up
to the sum of the emissions, resulting in a budget imbalance ($BI_M$) which is represented by the
difference between the bottom red line (mirroring total emissions) and the sum of carbon fluxes
in the ocean, land, and atmosphere reservoirs. All data are in GtC yr$^{-1}$ (panel a) and GtC (panel b).
The $E_{FOS}$ estimates are primarily from (Gilfillan and Marland, 2021), with uncertainty of about ±5%
(±1σ). The $E_{LUC}$ estimates are from three bookkeeping models (Table 4) with uncertainties of about
±0.7 GtC yr$^{-1}$. The $G_{ATM}$ estimates prior to 1959 are from Joos and Spahni (2008) with uncertainties
equivalent to about ±0.1-0.15 GtC yr$^{-1}$ and from Dlugokencky and Tans (2021) since 1959 with
uncertainties of about +-0.07 GtC yr$^{-1}$ during 1959-1979 and ±0.02 GtC yr$^{-1}$ since 1980. The $S_{OCEAN}$
estimate is the average from Khatiwala et al. (2013) and DeVries (2014) with uncertainty of about
±30% prior to 1959, and the average of an ensemble of models and an ensemble of $fCO_2$ data
products (Table 4) with uncertainties of about ±0.4 GtC yr$^{-1}$ since 1959. The $S_{LAND}$ estimate is the
average of an ensemble of models (Table 4) with uncertainties of about ±1 GtC yr$^{-1}$. See the text
for more details of each component and their uncertainties.



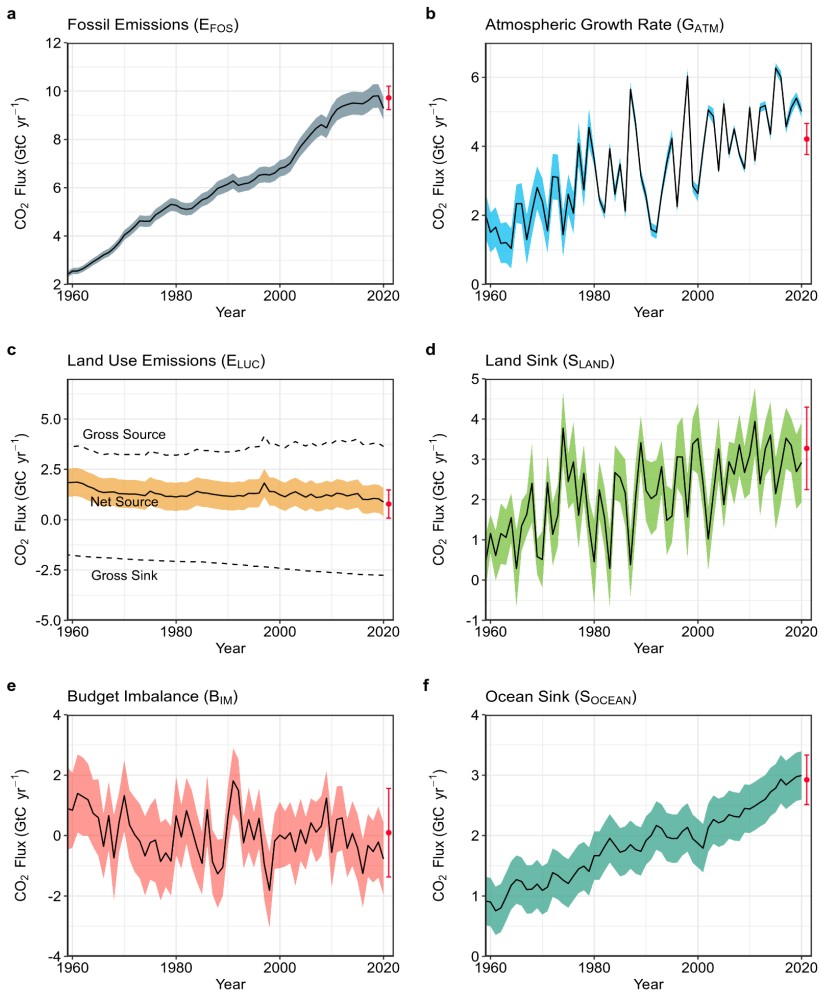

**Figure 4.** Components of the global carbon budget and their uncertainties as a function of time,
presented individually for (a) fossil $CO_2$ emissions ($E_{FOS}$), (b) growth rate in atmospheric $CO_2$
concentration ($G_{ATM}$), (c) emissions from land-use change ($E_{LUC}$), (d) the land $CO_2$ sink ($S_{LAND}$), (e)
the ocean $CO_2$ sink ($S_{OCEAN}$), (f) the budget imbalance that is not accounted for by the other terms.
Positive values of $S_{LAND}$ and $S_{OCEAN}$ represent a flux from the atmosphere to land or the ocean. All
data are in GtC yr$^{-1}$ with the uncertainty bounds representing ±1 standard deviation in shaded
colour. Data sources are as in Fig. 3. The red dots indicate our projections for the year 2021 and
the red error bars the uncertainty in the projections (see methods).



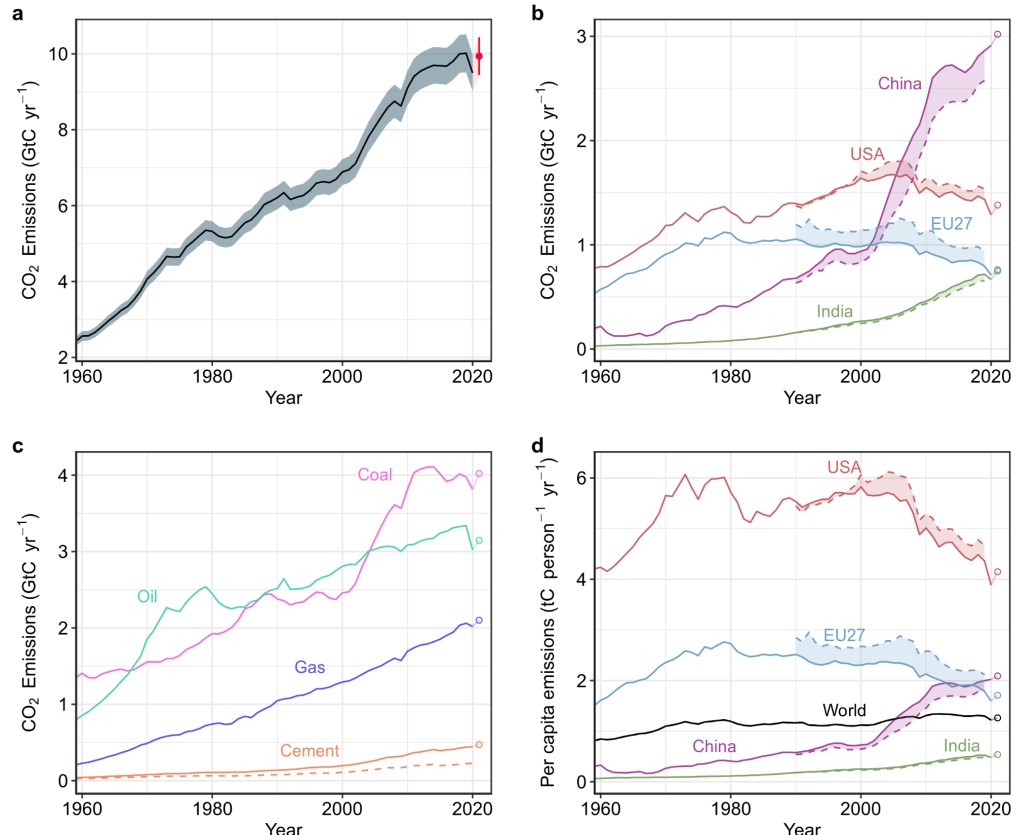

**Figure 5.** Fossil $CO_2$ emissions for (a) the globe, including an uncertainty of ± 5% (grey
shading) and a projection through the year 2021 (red dot and uncertainty range), (b)
territorial (solid lines) and consumption (dashed lines) emissions for the top three country
emitters (USA, China, India) and for the European Union (EU27), (c) global emissions by fuel
type, including coal, oil, gas, and cement, and cement minus cement carbonation (dashed),
and (d) per-capita emissions the world and for the large emitters as in panel (b).  Territorial
emissions are primarily from Gilfillan and Marland (2021) except national data for the USA
and EU27 for 1990-2018, which are reported by the countries to the UNFCCC as detailed in
the text; consumption-based emissions are updated from Peters et al. (2011b). See Section
2.1 and Appendix C.1 for details of the calculations and data sources.

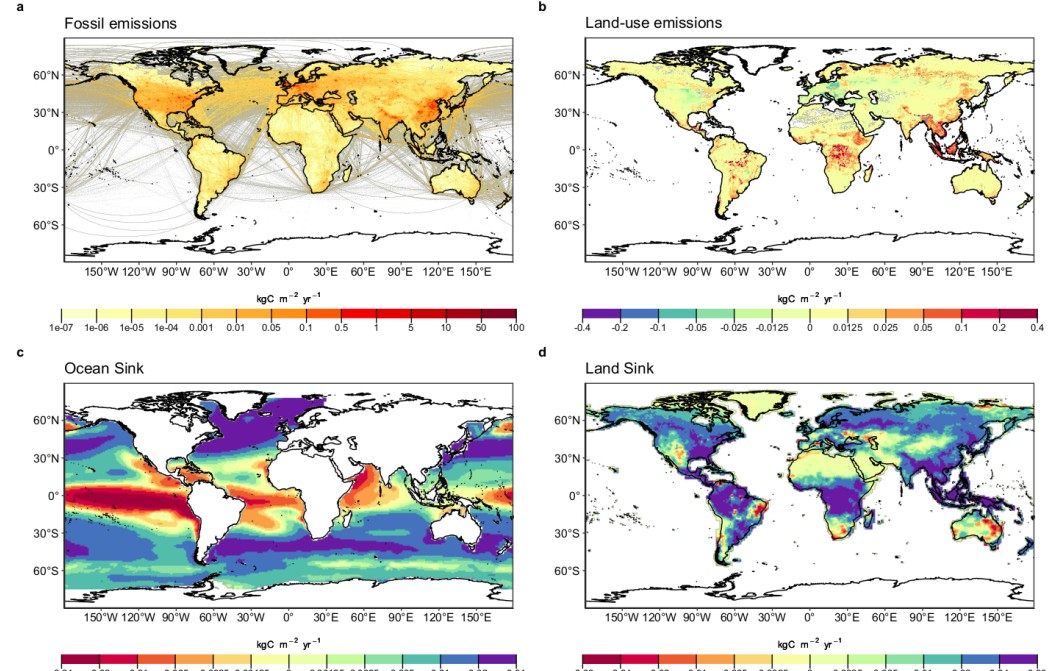

**Figure 6.** The 2011-2020 decadal mean components of the global carbon budget, presented for (a) fossil $CO_2$ emissions ($E_{FOS}$), (b) land-use change emissions ($E_{LUC}$), (c) the ocean $CO_2$ sink ($S_{OCEAN}$), and (d) the land $CO_2$ sink ($S_{LAND}$). Positive values for $E_{FOS}$ and $E_{LUC}$ represent a flux to the atmosphere, whereas positive values of $S_{OCEAN}$ and $S_{LAND}$ represent a flux from the atmosphere to the ocean or the land. In all panels, yellow/red (green/blue) colours represent a flux from (into) the land/ocean to (from) the atmosphere. All units are in kgC $m^{-2}$ $yr^{-1}$. Note the different scales in each panel. $E_{FOS}$ data shown is from GCP-GridFEDv2021.2. $E_{LUC}$ data shown is only from BLUE as the updated H&N2017 and OSCAR do not resolve gridded fluxes. $S_{OCEAN}$ data shown is the average of GOBMs and data-products means, using GOBMs simulation A, no adjustment for bias and drift applied to the gridded fields (see Sections 2.4). $S_{LAND}$ data shown is the average of DGVMs for simulation S2 (see Sections 2.5).



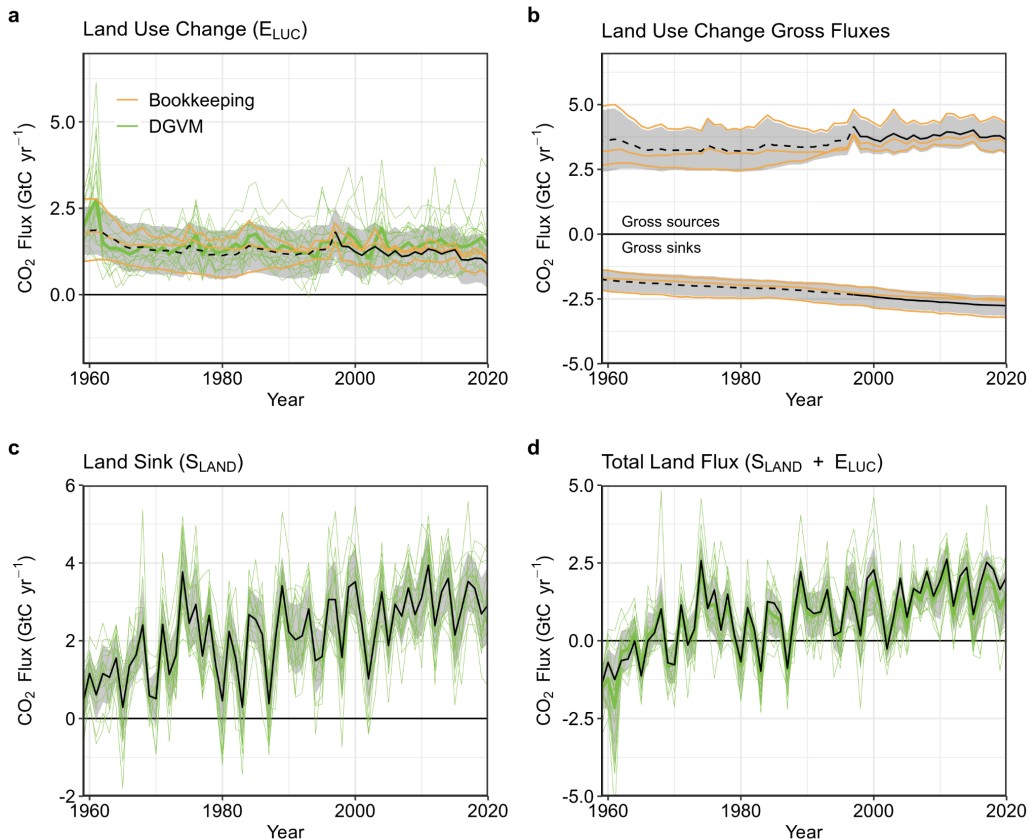

**Figure 7.** $CO_2$ exchanges between the atmosphere and the terrestrial biosphere as used in the

global carbon budget (black with ±1σ uncertainty in grey shading in all panels). (a) $CO_2$ emissions

from land-use change ($E_{LUC}$) with estimates from the three bookkeeping models (yellow lines) and

DGVMs models (green) shown individually, with DGVMs ensemble means (dark green). The

dashed line identifies the pre-satellite period before the inclusion of peatland burning. (b) $CO_2$

gross sinks (positive, from regrowth after agricultural abandonment and wood harvesting) and

gross sources (negative, from decaying material left dead on site, products after clearing of

natural vegetation for agricultural purposes, wood harvesting, and for BLUE, degradation from

primary to secondary land through usage of natural vegetation as rangeland, and also from

emissions from peat drainage and peat burning) from the three bookkeeping models (yellow

lines). The sum of the gross sinks and sources is $E_{LUC}$ shown in panel(a). (c) Land $CO_2$ sink ($S_{LAND}$)

with individual DGVMs estimates (green). (d) Total atmosphere-land $CO_2$ fluxes ($S_{LAND} - E_{LUC}$), with

individual DGVMs (green) and their multi-model mean (dark green).

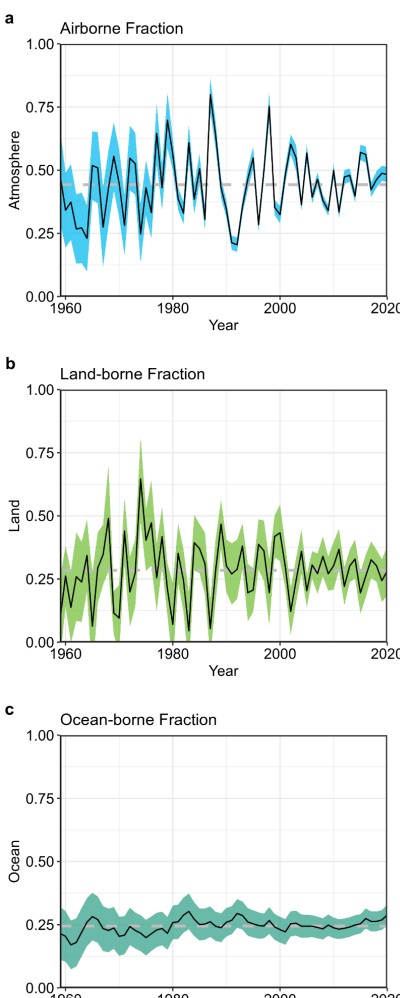

**Figure 8.** The partitioning of total anthropogenic $CO_2$ emissions ($E_{FOS}$ + $E_{LUC}$) across (a) the
atmosphere (airborne fraction), (b) land (land-borne fraction), and (c) ocean (ocean-borne
fraction). Black lines represent the central estimate, and the coloured shading represents the
uncertainty. The grey dashed lines represent the long-term average of the airborne (44%), land-
borne (28%) and ocean-borne (24%) fractions during 1959-2020.



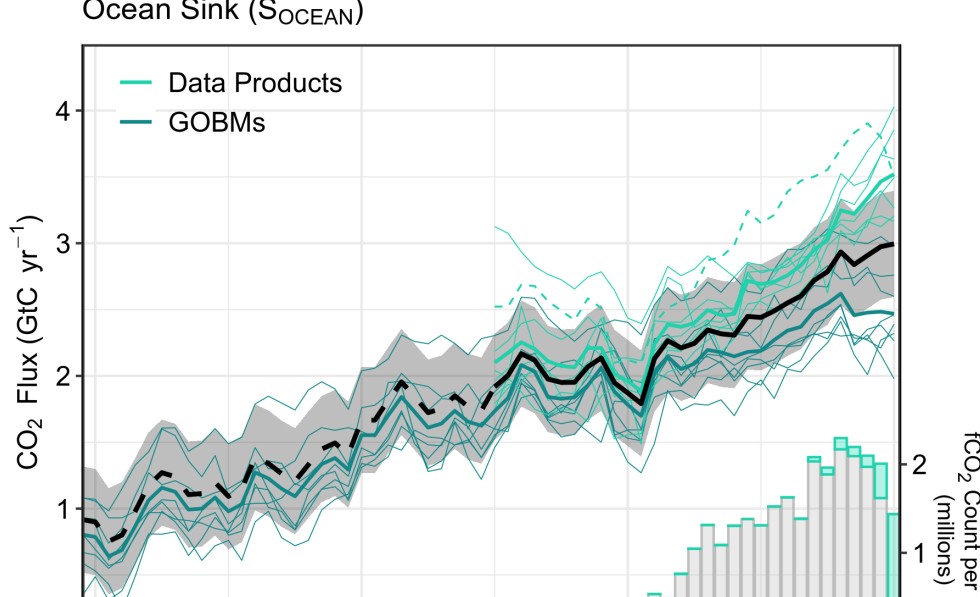

**Figure 9.** Comparison of the anthropogenic atmosphere-ocean $CO_2$ flux showing the budget values
of $S_{OCEAN}$ (black; with the uncertainty in grey shading), individual ocean models (teal), and the
ocean $fCO_2$-based data products (cyan; with Watson et al. (2020) in dashed line as not used for
ensemble mean). The $fCO_2$-based data products were adjusted for the pre-industrial ocean source
of $CO_2$ from river input to the ocean, by subtracting a source of 0.61 GtC yr$^{-1}$ to make them
comparable to $S_{OCEAN}$ (see Section 2.4). Bar-plot in the lower right illustrates the number of $fCO_2$
observations in the SOCAT v2021 database (Bakker et al., 2021). Grey bars indicate the number of
data points in SOCAT v2020, and coloured bars the newly added observations in v2021.

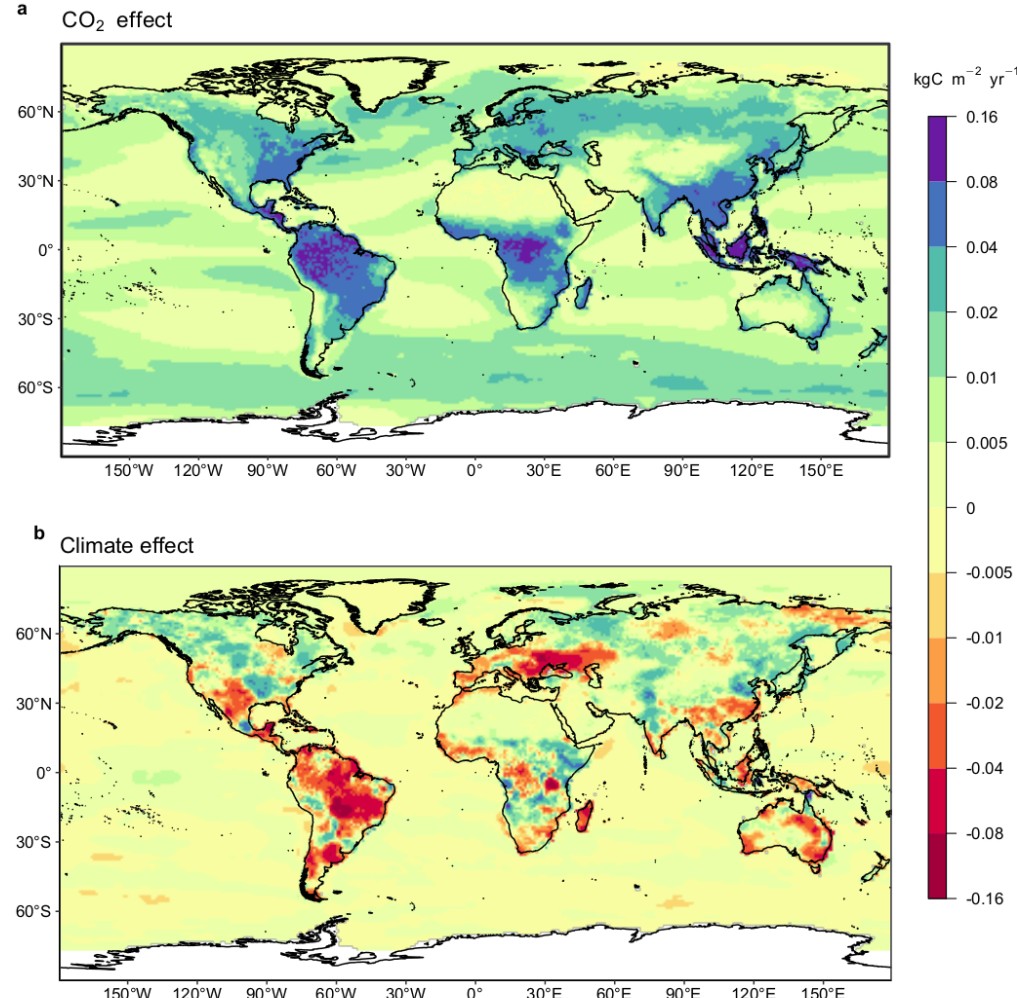

**Figure 10.** Attribution of the atmosphere-ocean ($S_{OCEAN}$) and atmosphere-land ($S_{LAND}$) $CO_2$ fluxes to

(**a**) increasing atmospheric $CO_2$ concentrations and (**b**) changes in climate, averaged over the

previous decade 2011-2020. All data shown is from the processed-based GOBMs and DGVMs. The

sum of ocean $CO_2$ and climate effects will not equal the ocean sink shown in Figure 6 which

includes the $fCO_2$-based data products. See Appendix C.3.2 and C.4.1 for attribution methodology.

Units are in kgC $m^{-2}$ $yr^{-1}$ (note the non-linear colour scale).



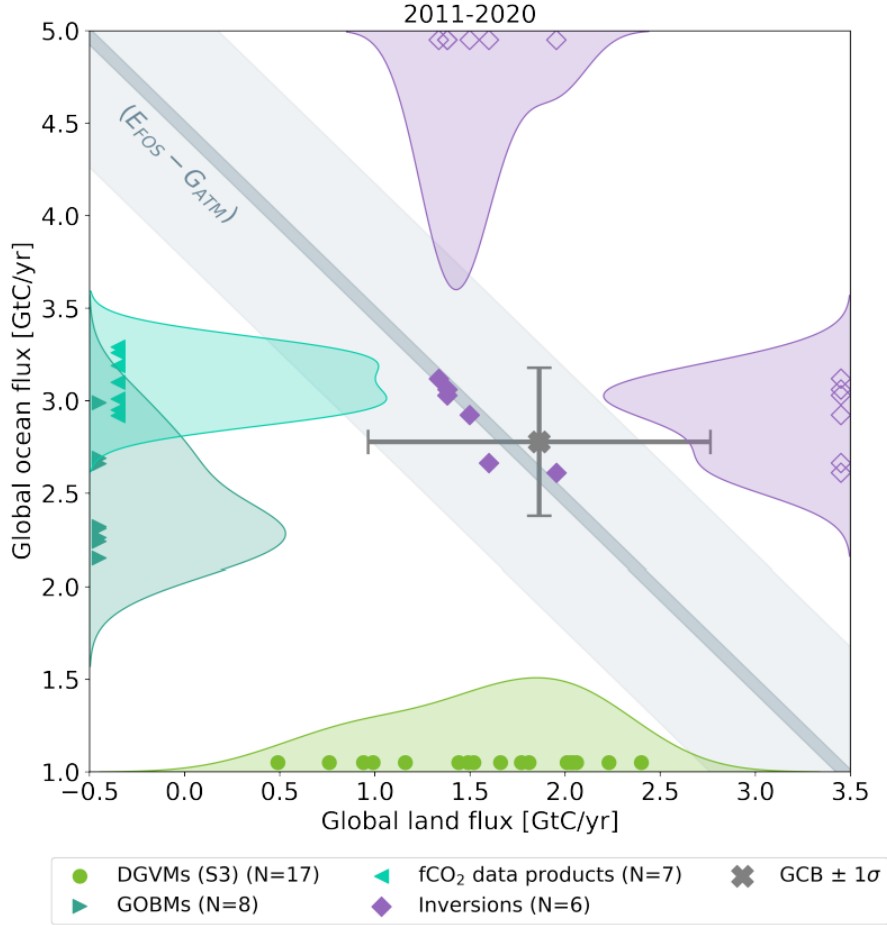

**Figure 11.** The 2011-2020 decadal mean net atmosphere-ocean and atmosphere-land fluxes

derived from the ocean models and fCO2 products (y-axis, right and left pointing blue triangles

respectively), and from the DGVMs (x-axis, green symbols), and the same fluxes estimated from

the six inversions (purple symbols on secondary x- and y-axis). The grey central point is the mean

($\pm1\sigma$) of $S_{OCEAN}$ and ($S_{LAND} - E_{LUC}$) as assessed in this budget. The shaded distributions show the

density of the ensemble of individual estimates. The grey diagonal band represents the fossil fuel

emissions minus the atmospheric growth rate from this budget ($E_{FOS} - G_{ATM}$). Note that positive

values are $CO_2$ sinks.



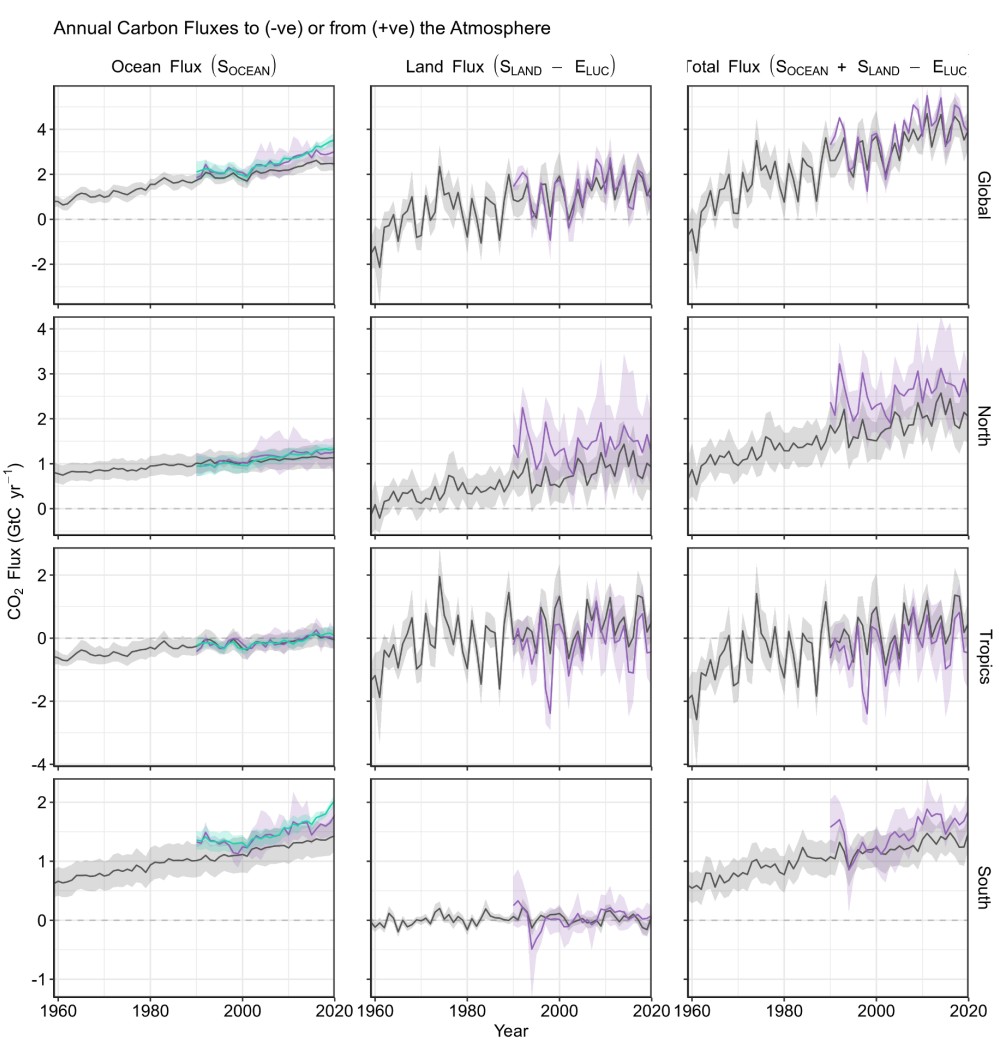

**Figure 12.** $CO_2$ fluxes between the atmosphere and the Earth's surface separated between land

and oceans, globally and in three latitude bands. The ocean flux is $S_{OCEAN}$ and the land flux is the

net atmosphere-land fluxes from the DGVMs. The latitude bands are (top row) global, (2nd row)

north (>30°N), (3rd row) tropics (30°S-30°N), and (bottom row) south (<30°S), and over ocean (left

column), land (middle column), and total (right column). Estimates are shown for: process-based

models (DGVMs for land, GOBMs for oceans); inversion models (land and ocean); and $fCO_2$-based

data products (ocean only). Positive values indicate a flux from the atmosphere to the land or the

ocean. Mean estimates from the combination of the process models for the land and oceans are





shown (black line) with ±1 standard deviation (1σ) of the model ensemble (grey shading). For the
total uncertainty in the process-based estimate of the total sink, uncertainties are summed in
quadrature. Mean estimates from the atmospheric inversions are shown (purple lines) with their
full spread (purple shading). Mean estimates from the fCO$_2$-based data products are shown for the
ocean domain (light blue lines) with their ±1σ spread (light blue shading). The global S$_{OCEAN}$ (upper
left) and the sum of S$_{OCEAN}$ in all three regions represents the anthropogenic atmosphere-to-ocean
flux based on the assumption that the preindustrial ocean sink was 0 GtC yr$^{-1}$ when riverine fluxes
are not considered. This assumption does not hold at the regional level, where preindustrial fluxes
can be significantly different from zero. Hence, the regional panels for S$_{OCEAN}$ represent a
combination of natural and anthropogenic fluxes. Bias-correction and area-weighting were only
applied to global S$_{OCEAN}$; hence the sum of the regions is slightly different from the global estimate
(<0.06 GtC yr$^{-1}$).

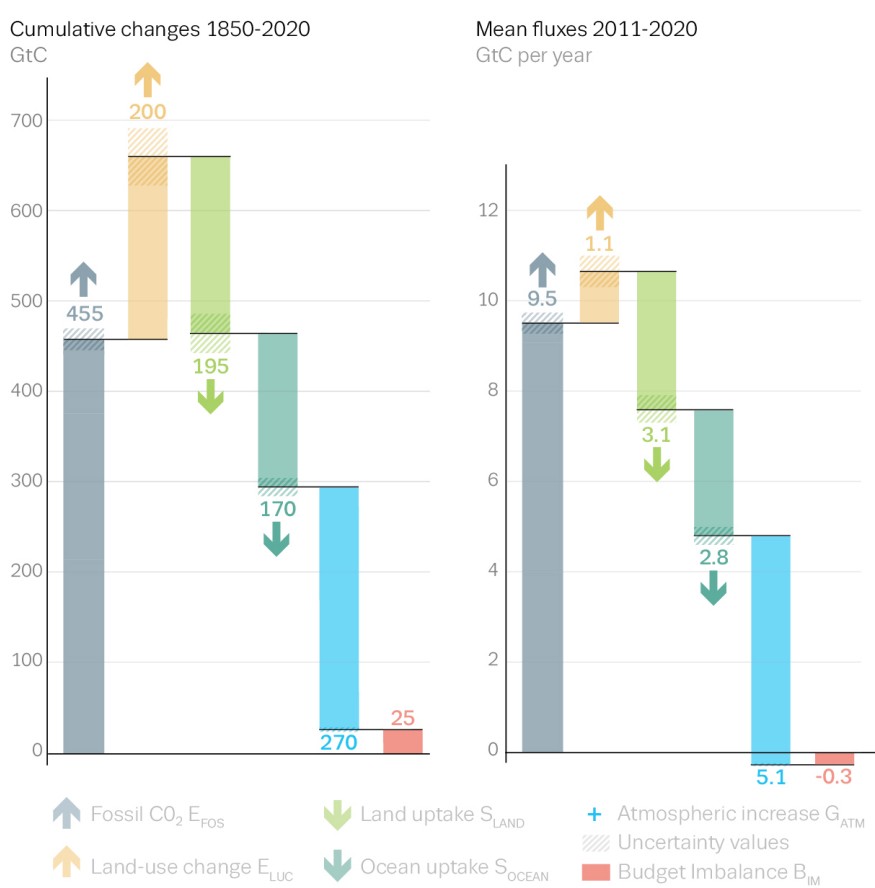

3 **Figure 13.** Cumulative changes over the 1850-2020 period (left) and average fluxes over

4 the 2011-2020 period (right) for the anthropogenic perturbation of the global carbon cycle.

5 See the caption of Figure 3 for key information and the methods in text for full details.



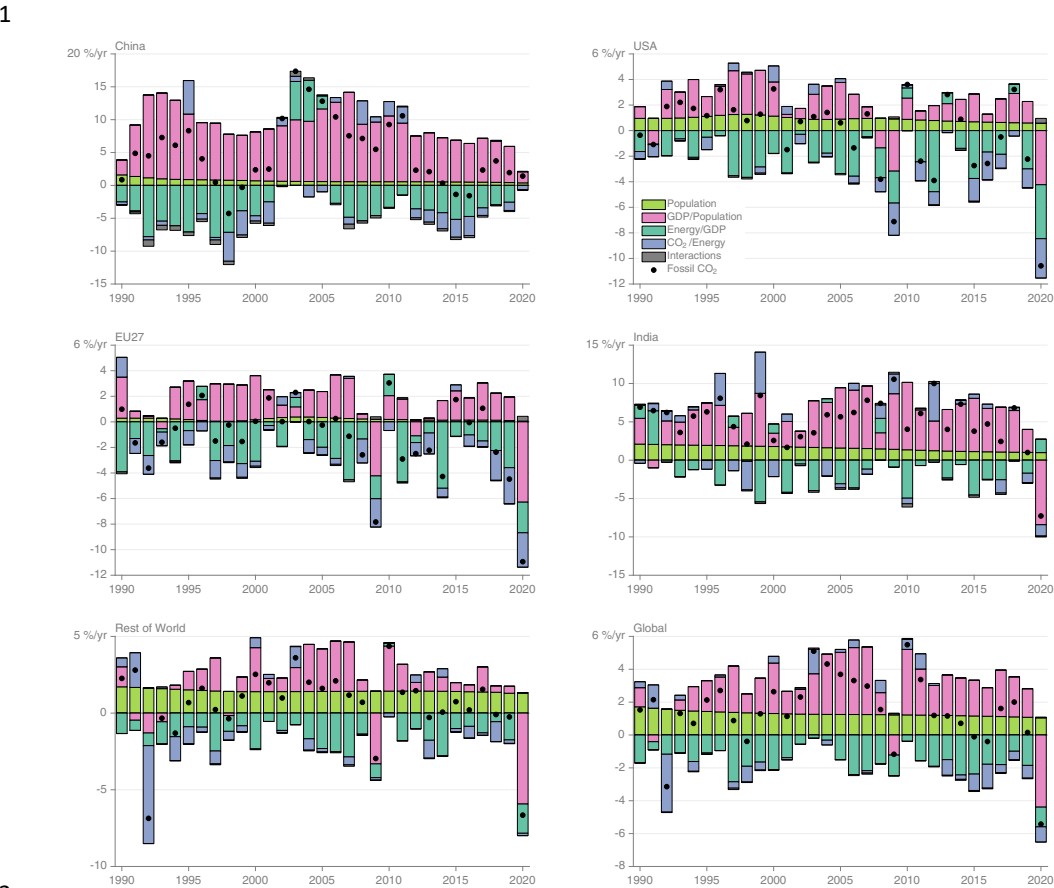

**Figure 14.** Kaya decomposition of the main drivers of fossil $CO_2$ emissions, considering population, GDP per person, Energy per GDP, and $CO_2$ emissions per energy, for China (top left), USA (top right), EU27 (middle left), India (middle right), Rest of the World (bottom left), and World (bottom right). Black dots are the annual fossil $CO_2$ emissions growth rate, coloured bars are the contributions from the different drivers. A general trend is that population and GDP growth put upward pressure on emissions, while energy per GDP and more recently $CO_2$ emissions per energy put downward pressure on emissions. The changes during 2020 led to a stark contrast to previous years, with different drivers in each region.



## 1 Appendix A. Supplementary Tables

Table A1. Comparison of the processes included in the bookkeeping method and DGVMs in their estimates of ELUC and SLAND. See Table 4 for model references. All models include deforestation and forest regrowth after abandonment of agriculture (or from afforestation activities on agricultural land). Processes relevant for ELUC are only described for the DGVMs used with land-cover change in this study.

| | Bookkeeping Models | | | DGVMs | | | | | | | | | | | | | | | | |
| | H&N | BLUE | OSCAR | CABLE-POP | CLASSIC | CLM5.0 | DLEM | IBIS | ISAM | ISBA-CTRIP(h) | JSBACH | JULES-ES | LPJ-GUESS | LPJ | LPX-Bern | OCNv2 | ORCHIDEEv3 | SDGVM | VISIT | YIBs |
|---|---|---|---|---|---|---|---|---|---|---|---|---|---|---|---|---|---|---|---|---|
| **Processes relevant for ELUC** | | | | | | | | | | | | | | | | | | | | |
| Wood harvest and forest degradation (a) | yes | yes | yes | yes | no | yes | yes | yes | yes | no | yes | no | yes | yes | no (d) | yes | yes | no | yes | no |
| Shifting cultivation / Subgrid scale transitions | no (b) | yes | yes | yes | no | yes | no | no | no | no | yes | | yes | yes | no (d) | no | no | no | yes | no |
| Cropland harvest (removed, R, or added to litter, L) | yes (R) (p) | yes (R) (p) | yes (R) | yes (R) | yes (L) | yes (R) | yes | yes (R) | yes | yes (R+L) | yes (R+L) | yes (R) | yes (R) | yes (L) | yes (R) | yes (R+L) | yes (R) | yes (R) | yse (R) | yes (L) |
| Peat fires | yes | yes | yes | no | no | yes | no | no | no | no | no | no | no | no | no | no | no | no | no | no |
| fire as a management tool | yes (p) | yes (p) | yes (j) | no | no | no | no | no | no | no | no | no | no | no | no | no | no | no | no | no |
| N fertilization | yes (p) | yes (p) | yes (j) | no | no | yes | yes | no | yes | no | no | yes(k) | yes | no | yes | yes | yes | no | no | no |
| tillage | yes (p) | yes (p) | yes (j) | no | yes (g) | no | no | no | no | no | no | no | yes | no | no | no | yes (g) | no | no | no |
| irrigation | yes (p) | yes (p) | yes (j) | no | no | yes | yes | no | yes | no | no | yes | no | no | no | no | no | no | no | no |
| wetland drainage | yes (p) | yes (p) | yes (j) | no | no | no | no | no | no | no | no | no | no | no | no | no | no | no | no | no |
| erosion | yes (p) | yes (p) | yes (j) | no | no | no | yes | no | no | no | no | no | no | no | no | no | no | no | yes | no |
| peat drainage | yes | yes | yes | no | no | no | no | no | no | no | no | no | no | no | no | no | no | no | no | no |
| Grazing and mowing Harvest (removed, r, or added to litter, l) | yes (r) (p) | yes (r) (p) | yes (r) | yes (r) | no | no | no | no | yes (l) | no | yes (l) | no | yes (r) | yes (l) | no | yes (r+l) | no | no | no | no |
| **Processes also relevant for SLAND (in addition to CO2 fertilization and climate)** | | | | | | | | | | | | | | | | | | | | |
| Fire simulation and/or suppression | N.A. | N.A. | N.A. | no | yes | yes | no | yes | no | yes | yes | yes | yes | yes | yes | no | no | yes | yes | no |
| Carbon-nitrogen interactions, including N deposition | N.A. | N.A. | N.A. | yes | no (f) | yes | yes | no | yes | no (e) | yes | yes | yes | no | yes | yes | yes | yes (c) | no | no (f) |
| Separate treatment of direct and diffuse solar radiation | N.A. | N.A | N.A | no | no | yes | no | no | no | no | no | yes | no | no | no | no | no | no | no | no |

(a) Refers to the routine harvest of established managed forests rather than pools of harvested products.
(b) No back- and forth-transitions between vegetation types at the country-level, but if forest loss based on FRA exceeded agricultural expansion based on FAO, then this amount of area was cleared for cropland and the same amount of area of old croplands abandoned.
(c) Limited. Nitrogen uptake is simulated as a function of soil C, and Vcmax is an empirical function of canopy N. Does not consider N deposition.
(d) Available but not active.
(e) Simple parameterization of nitrogen limitation based on Yin (2002; assessed on FACE experiments)
(f) Although C-N cycle interactions are not represented, the model includes a parameterization of down-regulation of photosynthesis as CO2 increases to emulate nutrient constraints (Arora et al., 2009)



(g) Tillage is represented over croplands by increased soil carbon decomposition rate and reduced humification of litter to soil carbon.

(h) ISBA-CTRIP corresponds to SURFEXv8 in GCB2018

(i) Bookkeeping models include the effect of CO2-fertilization as captured by present-day carbon densities, but not as an effect transient in time.

(j) as far as the DGVMs that OSCAR is calibrated to include it

(k) perfect fertilisation assumed, i.e. crops are not nitrogen limited and the implied fertiliser diagnosed

(m) fire intensity responds to climate and CO2, but no fire suppression

(z) Process captured implicitly by use of observed carbon densities.

2

**Table A2.** Comparison of the processes and model set up for the Global Ocean Biogeochemistry Models for their estimates of SOCEAN. See Table 4 for model references.

| | NEMO-PlankTOM12 | NEMO-PISCES (IPSL) | MICOM-HAMOCC (NorESM1-OCv1.2) | MPIOM-HAMOCC6 | FESOM-2.1-REcoM2 | NEMO3.6-PISCESv2-gas (CNRM) | MOM6-COBALT (Princeton) | CESM-ETHZ |
|---|---|---|---|---|---|---|---|---|
| **SPIN-UP procedure** | | | | | | | | |
| Initialisation of carbon chemistry | GLODAPv1 corrected for anthropogenic carbon from Sabine et al. (2004) | GLODAPv2 | GLODAP v1 (preindustrial DIC) | initialization from previous model simulations | GLODAPv2 alkalinity and preindustrial DIC | GLODAPv2 | GLODAPv2 for Alkalinity and DIC. DIC is corrected to 1959 level for simulation A and C and corrected to pre-industrial level for simulation B using Khatiwala et al. (2009, 2013) | GLODAPv2 preindustrial |
| Preindustrial spin-up prior to 1850? If yes, how long? | spin-up 1750-1947 | spin-up starting in 1836 with 3 loops of JRA55 | 1000 year spin up | yes, ~2000 years | 50 years | long spin-up (> 1000 years) | Other biogeochemical tracers are initialized from a GFDL-ESM2M spin-up (> 1000 years) | spinup 1655-1849 |
| atmospheric forcing for pre-industrial spin-up | looping NCEP year 1990 | JRA55 | CORE-I (normal year) forcing | spinup with omip climatology to reach steady state with the rivers | JRA55-do v.1.5.0 repeated year 1961 | JRA55-do | GFDL-ESM2M internal forcing | COREv2 forcing until 1835, three cycles of conditions from 1949-2009. from 1835-1850: JRA forcing |
| atmospheric forcing for historical spin-up 1850-1958 for simulation A | 1750-1947: looping NCEP year 1990; 1948-2020: NCEP | 1836-1958 : looping full JRA55 reanalysis | CORE-I (normal year) forcing; from 1948 onwards NCEP-R1 with CORE-II corrections | NCEP 6 hourly cyclic forcing (10 years starting from 1948) with co2 at 278 ppm and rivers | JRA55-do-v1.5.0 repeated year 1961 | JRA55-do cycling year 1958 | JRA55-do-v1.5 repeat year 1959 (71 years) | JRA55 version 1.3, repeat cycle between 1958-2018. |
| atmospheric CO2 for historical spin-up 1850-1958 for simulation A | provided by the GCP; converted to pCO2 temperature formulation (Sarmiento et al., 1992), monthly | xCO2 as provided by the GCB, global mean, annual resolution, converted to pCO2 with sea-level | xCO2 as provided by the GCB, converted to pCO2 with sea level pressure and water vapor correction | provided by the GCB | xCO2 as provided by the GCB, converted to pCO2 with sea-level pressure and water vapour pressure, | xCO2 as provided by the GCB, converted to pCO2 with constant sea-level pressure and water vapour | xCO2 at year 1959 level (315 ppm), converted to pCO2 with sea-level pressure and water vapour pressure, | xCO2 as provided by the GCB (new version 2021), converted to pCO2 with atmospheric pressure, |





| | resolution | pressure and water vapour pressure | | | global mean, monthly resolution | pressure, global mean, yearly resolution | global mean, yearly resolution | and locally determined water vapour pressure from SST and SSS (100% saturation) |
|---|---|---|---|---|---|---|---|---|
| atmospheric forcing for control spin-up 1850-1958 for simulation B | 1750-2020: looping NCEP 1990 | 1836-1958 : looping full JRA55 reanalysis | CORE-I (normal year) forcing | NCEP 1957 fixed forcing, co2=278 and rivers | JRA55-do-v1.5.0 repeat year 1961 | JRA55-do cycling year 1958 | JRA55-do-v1.5 repeat year 1959 (71 years) | normal year forcing created from JRA-55 version 1.3, NYF = climatology with anomalies from the year 2001 |
| atmospheric CO2 for control spin-up 1850-1958 for simulation B | constant 278ppm; converted to pCO2 temperature formulation (Sarmiento et al., 1992), monthly resolution | xCO2 of 286.46ppm, converted to pCO2 with constant sea-level pressure and water vapour pressure | xCO2 of 278 ppm, converted to pCO2 with seal level pressure and water vapor correction | 278, no conversion, assuming constant standard sea level pressure | xCO2 of 278ppm, converted to pCO2 with sea-level pressure and water vapour pressure | xCO2 of 286.46ppm, converted to pCO2 with constant sea-level pressure and water vapour pressure | xCO2 of 278ppm, converted to pCO2 with sea-level pressure and water vapour pressure | xCO2 as provided by the GCB for 1850, converted to pCO2 with atmospheric pressure, and locally determined water vapour pressure from SST and SSS (100% saturation) |
| **simulation A** | | | | | | | | |
| Atmospheric forcing for simulation A | NCEP | JRA55-v1.4 then 1.5 for 2020. | NCEP-R1 with CORE-II corrections | till1948: continue from A_spinup with cyclic NCEP forcing (1948+10) and increasing CO2 => GCBA-1777-1948 -1948-2020 : with transient NCEP forcing and transient monthly CO2 | JRA55-do-v1.5.0 | JRA55-do | JRA55-do-v1.5.0 1959-2019 and JRA55-do-v1.5.0.1b for 2020 | JRA-55 version 1.3 |
| atmospheric CO2 for simulation A | provided by the GCP; converted to pCO2 temperature formulation (Sarmiento et al., 1992), monthly resolution | xCO2 as provided by the GCB, global mean, annual resolution, converted to pCO2 with sea-level pressure and | xCO2 as provided by the GCB, converted to pCO2 with sea level pressure and water vapor correction | | xCO2 as provided by the GCB, converted to pCO2 with sea-level pressure and water vapour pressure, global mean, | xCO2 as provided by the GCB, converted to pCO2 with constant sea-level pressure and water vapour pressure, | xCO2 as provided by the GCB, converted to pCO2 with sea-level pressure and water vapour pressure, global mean, | xCO2 as provided by the GCB (new version 2021), converted to pCO2 with atmospheric pressure, and locally |





| | | water vapour pressure | | | monthly resolution | global mean, yearly resolution | yearly resolution | determined water vapour pressure from SST and SSS (100% saturation) |
|---|---|---|---|---|---|---|---|---|
| **simulation B** | | | | | | | | |
| Atmospheric forcing for simulation B | NCEP 1990 | N/A | CORE-I (normal year) forcing | 1948-2020: continue with B_spinup with fixed NCEP forcing 1957, co2=278 and rivers | JRA55-do-v1.5.0 repeat year 1961 | JRA55-do cycling year 1958 | JRA55-do-v1.5.0 repeat year 1959 | normal year forcing created from JRA-55 version 1.3, NYF = climatology with anomalies from the year 2001 |
| atmospheric CO2 for simulation B | constant 278ppm; converted to pCO2 temperature formulation (Sarmiento et al., 1992), monthly resolution | N/A | xCO2 of 278 ppm, converted to pCO2 with sea level pressure and water vapor correction | | xCO2 of 278ppm, converted to pCO2with sea-level pressure and water vapour pressure | xCO2 of 286.46ppm, converted to pCO2 with constant sea-level pressure and water vapour pressure | xCO2 of 278ppm, converted to pCO2 with sea-level pressure and water vapour pressure | xCO2 as provided by the GCB for 1850, converted to pCO2 with atmospheric pressure, and locally determined water vapour pressure from SST and SSS (100% saturation) |
| **model specifics** | | | | | | | | |
| Physical ocean model | NEMOv3.6-ORCA2 | NEMOv3.6-eORCA1L75 | MICOM (NorESM1-OCv1.2) | MPIOM | FESOM-2.1 | NEMOv3.6-GELATOv6-eORCA1L75 | MOM6-SIS2 | CESMv1.3 (ocean model based on POP2) |
| Biogeochemistry model | PlankTOM12 | PISCESv2 | HAMOCC (NorESM1-OCv1.2) | HAMOCC6 | REcoM-2-M | PISCESv2-gas | COBALTv2 | BEC (modified & extended) |
| Horizontal resolution | 2o lon, 0.3 to 1.5o lat | 1° lon, 0.3 to 1° lat | 1° lon, 0.17 to 0.25 lat (nominally 1°) | 1.5° | unstructured multi-resolution mesh. CORE-mesh, with 20-120 km resolution. Highest resolution north of 50N, intermediate in the equatorial belt and Southern Ocean, lowest in the subtropical gyres | 1° lon, 0.3 to 1° lat | 0.5° lon, 0.25 to 0.5° lat | Lon: 1.125°, Lat varying from 0.53° in the extratropics to 0.27° near the equator |



| Vertical resolution | 31 levels | 75 levels, 1m at the surface | 51 isopycnic layers + 2 layers representing a bulk mixed layer | 40 levels, layer thickness increase with depth | 46 levels, 10 m spacing in the top 100 m | 75 levels, 1m at surface | 75 levels hybrid coordinates, 2 m at surface | 60 levels (z-coordinates) |
|---|---|---|---|---|---|---|---|---|
| Total ocean area on native grid (km2) | 3.6080E+08 | 3.6270E+08 | 3.6006E+08 | 3.6598E+08 | 3.6475E+08 | 3.6270E+14 | 3.6110E+08 | 3.5926E+08 |
| Ocean area on native grid (km2) - NORTH | 6.2646E+07 | | 6.2049E+07 | 6.4440E+07 | | 6.3971E+13 | | |
| Ocean area on native grid (km2) - TROPICS | 1.1051E+08 | | 1.9037E+08 | 1.9248E+08 | | 1.9025E+14 | | |
| Ocean area on native grid (km2) - SOUTH | 1.8766E+08 | | 1.0765E+08 | 1.0986E+08 | | 1.0848E+14 | | |
| gas-exchange parameterization | Quadratic exchange formulation (function of T + 0.3*U^2)* (Sc/660)^-0.5) ; Wanninkhof (1992, Equation 8); Sweeney et al. (2007) | see Orr et al. (2017): kw parameterized from Wanninkhof (1992), with kw = a* (Sc/660)^-0.5) *u2*(1-f_ice) with a from Wanninkhof (2014) | see Orr et al. (2017): kw parameterized from Wanninkhof (1992), with kw = a* (Sc/660)^-0.5) *u2*(1-f_ice) with a=0.337 following the OCMIP2 protocols | Gas transfer velocity formulation and parameter setup of Wanninkhof (2014), including updated Schmidt number parameterizations for CO2 to comply with OMIP protocol (Orr et al., 2017) | see Orr et al. (2017): kw parameterized from Wanninkhof (1992), with kw = a* (Sc/660)^-0.5) *u2*(1-f_ice) with a from Wanninkhof (2014) | see Orr et al. (2017): kw parameterized from Wanninkhof (1992), with kw = a* (Sc/660)^-0.5) *u2*(1-f_ice) with a from Wanninkhof (2014) | see Orr et al. (2017): kw parameterized from Wanninkhof (1992), with kw = a* (Sc/660)^-0.5) *u2*(1-f_ice) with a from Wanninkhof (2014) | Gas exchange is parameterized using the Wanninkhof (1992) quadratic windspeed dependency formulation, but with the coefficient scaled down to reflect the recent 14C inventories. Concretely, we used a coefficent a of 0:31 cm hr-1 s2 m-2 to read kw = 0:31 ws^2 (1-fice) (Sc=660)^{-1/2} |
| time-step | 96 mins | 45 min | 3200 sec | 60 mins | 45 min | 15min | 30 min | 3757 sec |
| output frequency | Monthly | monthly | monthly/daily | monthly | monthly | monthly | monthly | monthly |
| CO2 chemistry routines | Following Broecker et al. (1982) | mocsy | Following Dickson et al. (2007) | as in Ilyina et al. (2013) adapted to comply with OMIP protocol (Orr et al., 2017). | mocsy | mocsy | mocsy | OCMIP2 (Orr et al., 2017) |
| river carbon input (PgC/yr) | 60.24 Tmol/yr; 0.723 PgC/yr | 0.61 PgC y-1 | 0 | 0.77 PgC/yr | 0 | ~0.611 PgC y-1 | ~0.15 PgC y-1 | 0.33 Pg C yr-1 |
| burial/net flux into the sediment (PgC/yr) | 0.723 PgC/yr | 0.59 GtC y-1 | around 0.54 | around 0.44 PgC/yr | 0 | ~0.656 GtC y-1 | ~0.18 PgC y-1 | 0.21 Pg C yr-1 |

2



| | Jena-MLS | MPI-SOMFFN | CMEMS-LSCE-FFNN | CSIR-ML6 | Watson et al | NIES-NN | JMA-MLR | OS-ETHZ-GRaCER |
|---|---|---|---|---|---|---|---|---|
| **Method** | Spatio-temporal interpolation (update of Rödenbeck et al., 2013, version oc_v2021). Specifically, the sea-air CO2 fluxes and the pCO2 field are numerically linked to each other and to the spatio-temporal field of ocean-internal carbon sources/sinks through process parametrizations, and the ocean-internal sources/sink field is then fit to the SOCATv2021 pCO2 data (Bakker et al., 2021). The fit includes a multi-linear regression against environmental drivers to bridge data gaps, and interannually explicit corrections to represent the data signals more completely. | 2-step neural network method where in a first step the global ocean is clustered into 16 biogeochemical provinces (one stand alone province for the Arctic Ocean - see Landschützer et al 2020) using a self-organizing map (SOM). In a second step, the non-linear relationship between available pCO2 measurements from the SOCAT database (Bakker et al 2016) and environmental predictor data (SST, SSS, MLD, CHL-a, atmospheric CO2 - references see Landschützer et al 2016) are established using a feed-forward neural network (FFN) for each province separately. The established relationship is then used to fill the existing data gaps (see Landschützer et al. 2013, 2016). | An ensemble of neural network models trained on 100 subsampled datasets from the Surface Ocean CO2 Atlas v2021 (SOCATv2021, Bakker et al. 2021) . Like the original data, subsamples are distributed after interpolation on 1x1 grid cells along ship tracks. Sea surface salinity, temperature, sea surface height, mixed layer depth, atmospheric CO2 mole fraction, chlorophyll-a, pCO2 climatology, latitude and longitude are used as predictors. The models are used to reconstruct sea surface pCO2 and convert to air-sea CO2 fluxes (see the proposed ensemble-based approach and analysis in Chau et al. 2020, 2021). | An ensemble average of six machine learning estimates of surface ocean pCO2 using the approach described in Gregor et al. (2019) with the updated product using SOCAT v2021 (Bakker et al., 2016). All ensemble members use a cluster-regression approach. Two different cluster configurations are used: (1) based on K-means clustering; (2) Fay and McKinley (2014) 's CO2 biomes. Three regression algorithms are used: (1) gradient boosted decision trees; (2) feed-forward neural network; (3) support vector regression. The product of the cluster configurations and the regression algorithms results in an ensemble with six members., hence the CSIR-ML6. | Derived from the SOCAT(v2021) pCO2 database, but corrected to the subskin temperature of the ocean as measured by satellite, using the methodology described by Goddijn-Murphy et al. (2015). A correction to the flux calculation is also applied for the cool and salty surface skin. In other respects the product uses interpolation of the data using the two step neural network based on MPI-SOMFFN :in the first step the ocean is divided into a monthly climatology of 16 biogeochemical provinces using a SOM, In the second step a feed-forward neural network establishes non-linear relationships between pCO2 and SST, SSS, mixed layer depth(MLD) and atmospheric xCO2 in each | A feed forward neural network model was used to reconstruct monthly global surface ocean CO2 concentrations 1x1 degree meshes and estimate air-sea CO2 fluxes. The target variable is the per cruise weighted fCO2 mean of SOCAT 2021. Feature variables include sea surface temperature (SST), salinity, chlorophyll-a, mixed layer depth, and the monthly nomaly of SST. See Zeng et al. (2014) | Fields of total alkalinity (TA) were estimated by using a multiple linear regressions (MLR) method based on GLODAPv2.2021 and satellite observation data. TA = f(SSDH, SSS) SOCATv2021 fCO2 data were converted to total dissolved inorganic carbon (DIC) concentrations in combination with the TA, and then fields of DIC were estimated by using a MLR method based on the DIC and satellite observation data. DIC = f(SSDH, SST, SSS, log(Chl), log(MLD), time) | OceanSODA-ETHZ's Geospatial Random Cluster Ensemble Regression is a two-step cluster-regression approach, where multiple clustering instances with slight variations are run to create an ensemble of estimates (n_membersd= 16). We use K-means clustering (n_clusters=21) for the clustering step and a combination of Gradient boosted trees (n_members=8) and Feed-forward neural-networks (n_members=8) to estimate SOCAT v2021 fCO2. Clustering is performed on the following variables: SOCOM_pCO2_climatology, SST_clim, MLD_clim, CHL_clim. Regression is performed on the following variables: xCO2atm, SST, SST_anomaly, SSS, CHL, MLD, u10_wind, v10_wind, sea- |

Table A3: Description of ocean data-products used for assessment of SOCEAN. See Table 4 for references.




| | | | | | of the 16 provinces. Further description in Watson et al. (2020). | | | ice changes, SSH (note that the latter two variables are an update from Gregor and Gruber, 2021). |
|---|---|---|---|---|---|---|---|---|
| **Gas-exchange parameterization** | Quadratic exchange formulation ($k*U^2*(Sc/660)^{-0.5}$) (Wanninkhof, 1992) with the transfer coefficient k scaled to match a global mean transfer rate of 16.5 cm/hr by Naegler (2009) | Quadratic exchange formulation ($k*U^2*(Sc/660)^{-0.5}$) (Wanninkhof, 1992) with the transfer coefficient k scaled to match a global mean transfer rate of 16.5 cm/hr (calculated myself over the full period 1982-2020) | Quadratic exchange formulation ($k*U^2*(Sc/660)^{-0.5}$) (Wanninkhof., 2014) with the transfer coefficient k scaled to match a global mean transfer rate of 16.5 cm/hr (Naegler, 2009). | Quadratic formulation kw = a $*U10^2 * (Sc/660)^{0.5}$ (). We use scaled kw for ERA5 reanalysis wind data, which is scaled globally to 16.5 cm/hr (after Naegler 2009) like in Fay and Gregor et al. (2021) https://doi.org/10.5194/essd-2021-16 | Nightingale et al. (2000) formulation : $K=((Sc/600)^{-0.5})*(0.333*U+0.222*U^2)$ | $Kw=0.251*Wnd*Wnd/sqrt(Sc/660.0)$ (Wanninkhof, 2014) | Quadratic exchange formulation ($k*U^2*(Sc/660)^{-0.5}$) (Wanninkhof., 2014) with the transfer coefficient k scaled to match a global mean transfer rate of 16.5 cm/hr (Naegler, 2009) under fitted to the JRA55 wind field. | Quadratic formulation of bulk air-sea CO2 flux: kw = a $* U10^2 * (Sc/660)^{0.5}$ We use individually scaled kw's for JRA55, ERA5, and NCEP-R1, which are all scaled globally to 16.5 cm/hr (after Naegler, 2009). See Fay and Gregor et al. (2021) |
| **Wind product** | JMA55-do reanalysis | ERA 5 | ERA5 | ERA5 | CCMP wind product, 0.25 x 0.25 degrees x 6-hourly, from which we calculate mean and mean square winds over 1 x 1 degree and 1 month intervals. CCMP product does not cover years 1985-1987, for which we use a monthly climatology calculated as the means of 1988-1991. | ERA5 | JRA55 | JRA55, ERA5, NCEP1 |
| **Spatial resolution** | 2.5 degrees longitude * 2 degrees latitude | 1x1 degree | 1x1 degree | 1 x 1 | 1 x 1 degree | 1x1 degree | 1x1 degree | 1x1 degree |



| Temporal resolution | daily | monthly | monthly | monthly | monthly | monthly | monthly | monthly |
|---|---|---|---|---|---|---|---|---|
| **Atmospheric CO2** | Spatially and temporally varying field based on atmospheric CO2 data from 169 stations (Jena CarboScope atmospheric inversion sEXTALL_v2021) | atmospheric pCO2_wet calculated from the NOAA ESRL marine boundary layer xCO2 and the NCEP sea level pressure with the moisture correction by Dickson et al 2007 (details and references can be obtained from Appendix A3 in Landschützer et al 2013) | Spatially and monthly varying fields of atmospheric pCO2 computed from CO2 mole fraction ( Chevallier, 2013; CO2 atmospheric inversion from the Copernicus Atmosphere Monitoring Service ), and atmospheric dry-air pressure which is derived from monthly surface pressure (ERA5) and water vapour pressure fitted by Weiss and Price (1980) | The NOAA's marine boundary layer product for the mole fraction of carbon dioxide (xCO2) is linearly interpolated onto a 1°x1° grid and resampled from weekly to monthly. Basically, xCO2 is multiplied by ERA5 mean sea level pressure (MSLP), and a water vapour pressure correction is applied to MSLP using the equation from Dickson et al. (2007). This results in monthly 1°x 1° atmospheric pCO2. | Atmospheric pCO2 (wet) calculated from NOAA marine boundary layer XCO2 and NCEP sea level pressure, with pH2O calculated from Cooper et al. (1998). (2019 XCO2 marine boundary values were not available at submission so we used preliminary values, estimated from 2018 values and increase at Mauna Loa.) | NOAA Greenhouse Gas Marine Boundary Layer Reference. https://gml.noaa.gov/ccgg/mbl/mbl.html | Atmospheric xCO2 fields of JMA-GSAM inversion model (Maki et al. 2010; Nakamura et al. 2015) were used. They were converted to pCO2 by using JRA55 sea level pressure. xCO2 fields in 2020 were not available at this stage, and we use observation data of obspack_co2_1_NRT_v6.1.1_2021-05-17 (Di Sarra et al. 2021) to estimate the increase from 2019 to 2020. | NOAA's marine boundary layer product for xCO2 is linearly interpolated onto a 1x1 degree grid and resampled from weekly to monthly. xCO2 is multiplied by ERA5 mean sea level pressure, where the latter corrected for water vapour pressure using Dickson et al. (2007). This results in monthly 1x1 degree pCO2atm. |
| **Total ocean area on native grid (km2)** | 3.63E+08 | 3.63E+08 | 3.46E+08 | 3.48E+08 | 3.51E+08 | 3.28E+08 (3.23E+08 to 3.35E+08, depending on ice cover) | 3.05E+08 (2.98E+08 to 3.15E+08, depending on ice cover) | 3.55E+08 |
| **method to extend product to full global ocean coverage** | | | Arctic and marginal seas added following Landschützer et al. (2020). previously applied coastal cut (1degree off coast) was dropped | | | | We used the same method as Fay et al. (2021a) | Method has near full coverage |
| **Ocean area on native grid (km2) - NORTH** | | | 5.4545E+07 | 5.0528E+07 | 5.0700E+07 | | 3.90E+07 (3.75E+07 to 4.09E+07, depending on ice cover) | 5.9771E+07 |
| **Ocean area on native grid (km2) - TROPICS** | | | 1.8875E+08 | 1.8933E+08 | 1.9230E+08 | | 1.74E+08 | 1.8779E+08 |





| | | | 1.0241E+08 | 1.0767E+08 | 1.0868E+08 | | 9.20E+07 (8.47E+07 to 1.02E+08, depending on ice cover) | 1.0705E+08 |
|---|---|---|---|---|---|---|---|---|
| **Ocean area on native grid (km2) - SOUTH** | | | | | | | | |

2



**Table A4. Comparison of the inversion set up and input fields for the atmospheric inversions. Atmospheric inversions see the full CO2 fluxes, including the anthropogenic and pre-industrial fluxes. Hence they need to be adjusted for the pre-industrial flux of CO2 from the land to the ocean that is part of the natural carbon cycle before they can be compared with SOCEAN and SLAND from process models. See Table 4 for references.**

| | CarbonTracker Europe (CTE) | Jena CarboScope | Copernicus Atmosphere Monitoring Service (CAMS) | UoE | CMS-Flux | NISMON-CO2 |
|---|---|---|---|---|---|---|
| **Version number** | CTE2021 | sEXTocNEET_v2021 | v20r2 | in-situ | | v2021.1 |
| **Observations** | | | | | | |
| **Atmospheric observations** | Hourly resolution (well-mixed conditions) obspack GLOBALVIEWplus v6.1 and NRT_v6.1.1 (a) | Flasks and hourly from various institutions (outliers removed by 2-sigma criterion) | Hourly resolution (well-mixed conditions) obspack GLOBALVIEWplus v6.1 and NRT_v6.1.1 (a), WDCGG, RAMCES and ICOS ATC | Hourly resolution (well-mixed conditions) obspack GLOBALVIEWplus v6.1 and NRT_v6.1.1 (a) | ACOS-GOSAT v9 (6) retrievals between July 2009 and Dec 2014 and OCO-2 b10 (7) retrievals between Jan 2015 to Dec 2015. In addition, surface flask observations from remote sites were also assimilated from GLOBALVIEWplus v6.1 and NRT_v6.1.1 . | Hourly resolution (well-mixed conditions) obspack GLOBALVIEWplus v6.1 and NRT_v6.1.1 (a) |
| **Period covered** | 2001-2020 | 1957-2020 | 1979-2021 | 2001-2020 | 2010-2020 | 1990-2020 |
| **Prior fluxes** | | | | | | |
| **Biosphere and fires** | SIBCASA biosphere (b) with 2019-2020 climatological, GFAS fires | No prior | ORCHIDEE (climatological), GFEDv4.1s | CASA v1.0, climatology after 2016 & GFED4.0 | yearly repeating CARDAMOM biosphere+fires | VISIT & GFEDv4.1s |
| **Ocean** | oc_v2020 (Rodenbeck et al., 2014), with updates, For 2020: climatology based on years 2015-2019 | oc_v2021 (Rödenbeck et al., 2014) with updates | CMEMS Copernicus ocean fluxes (Denvil-Sommer et al., 2019), with updates | Takahashi climatology | MOM6 | JMA global ocean mapping (Iida et al., 2015) |
| **Fossil fuels** | GCP-GridFEDv2021.1 (Jones et al., 2021b) for 2000-2018, GCP-GridFEDv2021.2 for 2019+2020 (c) | GCP-GridFEDv2021.2 (Jones et al., 2021b) (c) | GCP-GridFEDv2021.2 (Jones et al., 2021b) (c) | GCP-GridFEDv2021.2 (Jones et al., 2021b) (c) | GCP-GridFEDv2021.2 (Jones et al., 2021b) (c) | GCP-GridFEDv2021.2 (Jones et al., 2021b) (c) |
| **Transport and optimization** | | | | | | |
| **Transport model** | TM5 | TM3 | LMDZ v6 | GEOS-CHEM | GEOS-CHEM | NICAM-TM |
| **Weather forcing** | ECMWF | NCEP | ECMWF | MERRA2 | MERRA-2 | JRA55 |





| Horizontal Resolution | Global: 3° x 2°, Europe: 1° x 1°, North America: 1° x 1° | Global: 4° x 5° | Global: 3.75° x 1.875° | Global: 4° x 5° | Global: 4° x 5° | isocahedral grid: ~225km |
|---|---|---|---|---|---|---|
| Optimization | Ensemble Kalman filter | Conjugate gradient (re-ortho-normalization) (d) | Variational | Ensemble Kalman filter | Variational | Variational |

| | |
|---|---|
| (a) (Cox et al., 2021; Di Sarra et al., 2021) | |
| (b) (van der Velde et al., 2014) | |
| (c) GCP-GridFEDv2021.2 (Jones et al., 2021b) is an update through the year 2020 of the GCP-GridFED dataset presented by Jones et al. (2021a). | |
| (d) ocean prior not optimised | |




**Table A5 Attribution of fCO2 measurements for the year 2020 included in SOCATv2021 (Bakker et al., 2016, 2021) to inform ocean fCO2-based data products.**

| Platform name | Regions | No. of measurements | Principal Investigators | No. of data sets | Platform type |
|---|---|---|---|---|---|
| *1 degree* | North Atlantic, Coastal | 8,652 | Gutekunst, S. | 2 | Ship |
| *Allure of the Seas* | North Atlantic, Tropical Atlantic, Coastal | 19,321 | Wanninkhof, R.; Pierrot, D. | 8 | Ship |
| *Atlantic Explorer* | North Atlantic | 15,665 | Bates, N. | 11 | Ship |
| *Atlantic Sail* | North Atlantic, Coastal | 25,082 | Steinhoff, T.; Körtzinger, A. | 6 | Ship |
| *Aurora Australis* | Southern Ocean | 14,316 | Tilbrook, B. | 1 | Ship |
| *Bjarni Saemundsson* | Coastal | 3,269 | Benoit-Cattin A.; Ólafsdóttir, S. R. | 1 | Ship |
| *BlueFin* | North Pacific, Tropical Pacific, Coastal | 76,505 | Alin, S. R.; Feely, R. A. | 12 | Ship |
| *Cap San Lorenzo* | Tropical Atlantic, Coastal | 12,417 | Lefèvre, N. | 2 | Ship |
| *Celtic Explorer* | North Atlantic, Coastal | 18,617 | Cronin, M. | 6 | Ship |
| *Colibri* | North Atlantic, Tropical Atlantic, Coastal | 13,402 | Lefèvre, N. | 2 | Ship |
| *Equinox* | North Atlantic, Coastal | 25,052 | Wanninkhof, R.; Pierrot, D. | 11 | Ship |
| *F. G. Walton Smith* | Coastal | 10,460 | Rodriguez, C.; Millero, F. J.; Pierrot, D.; Wanninkhof, R. | 6 | Ship |
| *Finnmaid* | Coastal | 253,894 | Rehder, G.; Glockzin, M. | 11 | Ship |
| *Flora* | Tropical Pacific | 4,099 | Wanninkhof, R.; Pierrot, D. | 2 | Ship |
| *G.O. Sars* | Arctic, North Atlantic, Coastal | 75,833 | Skjelvan, I. | 7 | Ship |
| *GAKOA_149W_60 N* | Coastal | 68 | Cross, J. N.; Monacci, N. M. | 3 | Mooring |
| *Gulf Challenger* | Coastal | 2,717 | Salisbury, J.; Vandemark, D.; Hunt, C. | 3 | Ship |
| *Healy* | Arctic, North Pacific, Coastal | 16,943 | Sweeney, C.; Newberger, T.; Sutherland, S. C.; Munro, D. R. | 4 | Ship |
| *Henry B. Bigelow* | North Atlantic, Coastal | 14,436 | Wanninkhof, R.; Pierrot, D. | 4 | Ship |
| *Heron Island* | Coastal | 768 | Tilbrook B. | 1 | Mooring |
| *James Clark Ross* | Southern Ocean | 2,000 | Kitidis, V. | 1 | Ship |
| *James Cook* | North Atlantic, Tropical Atlantic, Coastal | 46,710 | Theetaert, H. | 1 | Ship |
| *KC_BUOY* | Coastal | 1,983 | Evans, W. | 1 | Mooring |
| *Laurence M. Gould* | Southern Ocean | 25,414 | Sweeney, C.; Newberger, T.; Sutherland, S. C.; Munro, D. R. | 4 | Ship |
| *Maria. S. Merian* | Tropical Atlantic, Coastal | 35,806 | Ritschel, M. | 1 | Ship |
| *Marion Dufresne* | Southern Ocean, Indian | 4,709 | Lo Monaco, C.; Metzl, N. | 1 | Ship |
| *Nathaniel B. Palmer* | Southern Ocean, Tropical Pacific | 34,357 | Sweeney, C.; Newberger, T.; Sutherland, S. C.; Munro, D. R. | 3 | Ship |
| *New Century 2* | North Pacific, Tropical Pacific, Tropical Atlantic, North Atlantic, Coastal | 27,793 | Nakaoka, S.-I. | 14 | Ship |
| *Nuka Arctica* | North Atlantic, Coastal | 26,576 | Becker, M.; Olsen, A. | 6 | Ship |
| *Oscar Dyson* | Arctic, North Pacific, Coastal | 28,196 | Alin, S. R.; Feely, R. A. | 6 | Ship |
| *Quadra Island Field Station* | Coastal | 78,098 | Evans, W. | 1 | Mooring |
| *Ronald H. Brown* | Southern Ocean, Tropical Atlantic, North Atlantic, Coastal | 51,611 | Wanninkhof, R.; Pierrot, D. | 6 | Ship |



| | | | | | |
|---|---|---|---|---|---|
| *Saildrone1030* | North Atlantic, Tropical Atlantic, Coastal | 4,080 | Skjelvan, I.; Fiedler, B.; Pfeil, B.; Jones, S. D. | 1 | Saildrone |
| *Sea Explorer* | Southern Ocean, Tropical Atlantic, North Atlantic, Coastal | 89,896 | Landschützer, P.; Tanhua, T. | 6 | Ship |
| *Sikuliaq* | Arctic, North Pacific, Coastal | 36,278 | Sweeney, C.; Newberger, T.; Sutherland, S. C.; Munro, D. R. | 10 | Ship |
| *Simon Stevin* | Coastal | 16,448 | Gkritzalis, T. | 4 | Ship |
| *Soyo Maru* | Coastal | 46,280 | Ono, T. | 2 | Ship |
| *Tangaroa* | Southern Ocean, Tropical Pacific | 121,135 | Currie, K. I. | 13 | Ship |
| TAO110W_0N | Tropical Pacific | 1,518 | Sutton, A. J. | 3 | Mooring |
| *Tavastland* | Coastal | 4,214 | Willstrand Wranne, A., Steinhoff, T. | 5 | Ship |
| *Thomas G. Thompson* | Southern Ocean, Tropical Atlantic | 1,317 | Alin, S. R.; Feely, R. A. | 1 | Ship |
| *Trans Carrier* | Coastal | 24,135 | Omar, A. M. | 13 | Ship |
| *Trans Future 5* | Southern Ocean, Coastal | 16,404 | Nakaoka, S.-I.; Nojiri, Y. | 15 | Ship |
| *Wakataka Maru* | North Pacific, Coastal | 101,327 | Tadokoro, K.; Ono, T. | 7 | Ship |




| Table A6. Aircraft measurement programs archived by Cooperative Global Atmospheric Data Integration Project (CGADIP; Cox et al., 2021) that contribute to the evaluation of the atmospheric inversions (Figure B4). | | | | |
|---|---|---|---|---|
| Site code | Measurement program name in Obspack | Specific doi | Data providers | used in 2021 |
| AAO | Airborne Aerosol Observatory, Bondville, Illinois | | Sweeney, C.; Dlugokencky, E.J. | yes |
| ACG | Alaska Coast Guard | | Sweeney, C.; McKain, K.; Karion, A.; Dlugokencky, E.J. | yes |
| ACT | Atmospheric Carbon and Transport - America | | Sweeney, C.; Dlugokencky, E.J.; Baier, B; Montzka, S.; Davis, K. | yes |
| ALF | Alta Floresta | | Gatti, L.V.; Gloor, E.; Miller, J.B.; | yes |
| AOA | Aircraft Observation of Atmospheric trace gases by JMA | | ghg_obs@met.kishou.go.jp | yes |
| BGI | Bradgate, Iowa | | Sweeney, C.; Dlugokencky, E.J. | yes |
| BNE | Beaver Crossing, Nebraska | | Sweeney, C.; Dlugokencky, E.J. | yes |
| BRZ | Berezorechka, Russia | | Sasakama, N.; Machida, T. | yes |
| CAR | Briggsdale, Colorado | | Sweeney, C.; Dlugokencky, E.J. | yes |
| CMA | Cape May, New Jersey | | Sweeney, C.; Dlugokencky, E.J. | yes |
| CON | CONTRAIL (Comprehensive Observation Network for TRace gases by AIrLiner) | http://dx.doi.org/10.17595/20180208.001 | Machida, T.; Matsueda, H.; Sawa, Y. Niwa, Y. | yes |
| CRV | Carbon in Arctic Reservoirs Vulnerability Experiment (CARVE) | | Sweeney, C.; Karion, A.; Miller, J.B.; Miller, C.E.; Dlugokencky, E.J. | yes |
| DND | Dahlen, North Dakota | | Sweeney, C.; Dlugokencky, E.J. | yes |
| ESP | Estevan Point, British Columbia | | Sweeney, C.; Dlugokencky, E.J. | yes |
| ETL | East Trout Lake, Saskatchewan | | Sweeney, C.; Dlugokencky, E.J. | yes |
| FWI | Fairchild, Wisconsin | | Sweeney, C.; Dlugokencky, E.J. | yes |
| GSFC | NASA Goddard Space Flight Center Aircraft Campaign | | Kawa, S.R.; Abshire, J.B.; Riris, H. | yes |
| HAA | Molokai Island, Hawaii | | Sweeney, C.; Dlugokencky, E.J. | yes |
| HFM | Harvard University Aircraft Campaign | | Wofsy, S.C. | yes |
| HIL | Homer, Illinois | | Sweeney, C.; Dlugokencky, E.J. | yes |
| HIP | HIPPO (HIAPER Pole-to-Pole Observations) | https://doi.org/10.3334/CDIAC/HIPPO_010 | Wofsy, S.C.; Stephens, B.B.; Elkins, J.W.; Hintsa, E.J.; Moore, F. | yes |
| IAGOS-CARIBIC | In-service Aircraft for a Global Observing System | | Obersteiner, F.; Boenisch., H; Gehrlein, T.; Zahn, A.; Schuck, T. | yes |
| INX | INFLUX (Indianapolis Flux Experiment) | | Sweeney, C.; Dlugokencky, E.J.; Shepson, P.B.; Turnbull, J. | yes |
| LEF | Park Falls, Wisconsin | | Sweeney, C.; Dlugokencky, E.J. | yes |
| NHA | Offshore Portsmouth, New Hampshire (Isles | | Sweeney, C.; Dlugokencky, E.J. | yes |





| | | | | |
|---|---|---|---|---|
| | of Shoals) | | | |
| OIL | Oglesby, Illinois | | Sweeney, C.; Dlugokencky, E.J. | yes |
| PFA | Poker Flat, Alaska | | Sweeney, C.; Dlugokencky, E.J. | yes |
| RBA-B | Rio Branco | | Gatti, L.V.; Gloor, E.; Miller, J.B. | yes |
| RTA | Rarotonga | | Sweeney, C.; Dlugokencky, E.J. | yes |
| SCA | Charleston, South Carolina | | Sweeney, C.; Dlugokencky, E.J. | yes |
| SGP | Southern Great Plains, Oklahoma | | Sweeney, C.; Dlugokencky, E.J.; Biraud, S. | yes |
| TAB | Tabatinga | | Gatti, L.V.; Gloor, E.; Miller, J.B. | yes |
| TGC | Offshore Corpus Christi, Texas | | Sweeney, C.; Dlugokencky, E.J. | yes |
| THD | Trinidad Head, California | | Sweeney, C.; Dlugokencky, E.J. | yes |
| WBI | West Branch, Iowa | | Sweeney, C.; Dlugokencky, E.J. | yes |





**Table A7. Main methodological changes in the global carbon budget since first publication. Methodological changes introduced in one year are kept for the following years unless noted. Empty cells mean there were no methodological changes introduced that year.**

| Publication year | Fossil fuel emissions | | | LUC emissions | Reservoirs | | | Uncertainty & other changes |
|---|---|---|---|---|---|---|---|---|
| | Global | Country (territorial) | Country (consumption) | | Atmosphere | Ocean | Land | |
| 2006 (a) | | Split in regions | | | | | | |
| 2007 (b) | | | | ELUC based on FAO-FRA 2005; constant ELUC for 2006 | 1959-1979 data from Mauna Loa; data after 1980 from global average | Based on one ocean model tuned to reproduced observed 1990s sink | | ±1σ provided for all components |
| 2008 (c) | | | | Constant ELUC for 2007 | | | | |
| 2009 (d) | | Split between Annex B and non-Annex B | Results from an independent study discussed | Fire-based emission anomalies used for 2006-2008 | | Based on four ocean models normalised to observations with constant delta | First use of five DGVMs to compare with budget residual | |
| 2010 (e) | Projection for current year based on GDP | Emissions for top emitters | | ELUC updated with FAO-FRA 2010 | | | | |
| 2011 (f) | | | Split between Annex B and non-Annex B | | | | | |
| 2012 (g) | | 129 countries from 1959 | 129 countries and regions from 1990-2010 based on GTAP8.0 | ELUC for 1997-2011 includes interannual anomalies from fire-based emissions | All years from global average | Based on 5 ocean models normalised to observations with ratio | Ten DGVMs available for SLAND; First use of four models to compare with ELUC | |
| 2013 (h) | | 250 countriesb | 134 countries and regions 1990-2011 based on GTAP8.1, with detailed estimates for years 1997, 2001, 2004, and 2007 | ELUC for 2012 estimated from 2001-2010 average | | Based on six models compared with two data-products to year 2011 | Coordinated DGVM experiments for SLAND and ELUC | Confidence levels; cumulative emissions; budget from 1750 |
| 2014 (i) | Three years of BP data | Three years of BP data | Extended to 2012 with updated GDP data | ELUC for 1997-2013 includes interannual anomalies from fire-based emissions | | Based on seven models | Based on ten models | Inclusion of breakdown of the sinks in three latitude bands and comparison with three atmospheric inversions |





| | | | | | | | |
|---|---|---|---|---|---|---|---|
| 2015 (j) | Projection for current year based Jan-Aug data | National emissions from UNFCCC extended to 2014 also provided | Detailed estimates introduced for 2011 based on GTAP9 | | | Based on eight models | Based on ten models with assessment of minimum realism | The decadal uncertainty for the DGVM ensemble mean now uses ±1σ of the decadal spread across models |
| 2016 (k) | Two years of BP data | Added three small countries; China's emissions from 1990 from BP data (this release only) | | Preliminary ELUC using FRA-2015 shown for comparison; use of five DGVMs | | Based on seven models | Based on fourteen models | Discussion of projection for full budget for current year |
| a Raupach et al. (2007) | | | | | | | |
| b Canadell et al. (2007) | | | | | | | |
| c GCP (2008) | | | | | | | |
| d Le Quéré et al. (2009) | | | | | | | |
| e Friedlingstein et al. (2010) | | | | | | | |
| f Peters et al. (2012b) | | | | | | | |
| g Le Quéré et al. (2013), Peters et al. (2013) | | | | | | | |
| h Le Quéré et al. (2014) | | | | | | | |
| i Le Quéré et al. (2015a) | | | | | | | |
| j Le Quéré et al. (2015b) | | | | | | | |
| k Le Quéré et al. (2016) | | | | | | | |





| Table A8: Mapping of scientific land flux definitions to the definition of the LULUCF net flux used in national reporting Note that estimates are based on the global carbon budget estimates from Friedlingstein et al (2020), which estimated higher emissions from the net land-use change flux (ELUC) and a larger natural terrestrial sink Non-intact lands are a proxy for "managed lands" in the country reporting | | | | |
|---|---|---|---|---|
| | | | 2000-2009 | 2010-2019 |
| ELUC from bookkeeping estimates (from Tab. 5) | | | 1.44 | 1.61 |
| SLAND | Total | from DGVMs | -2.90 | -3.40 |
| | on non-forest lands | from DGVMs | -1.05 | -1.38 |
| | on non-intact forest | from DGVMs | -1.39 | -1.54 |
| | on intact land (intact forest only for DGVMs) | from DGVMs | -0.46 | -0.49 |
| | | from cohort-based ORCHIDEE | -1.29 | -1.47 |
| | | | | |
| SLAND on non-intact lands plus ELUC | | from DGVMs and bookkeeping ELUC | 0.05 | 0.08 |
| | | from cohort-based ORCHIDEE | 1.00 | 0.61 |
| National greenhouse gas inventories (LULUCF) | | | 0.00 | -0.31 |
| FAOSTAT (LULUCF) | | | 0.39 | 0.20 |



**Table A9.** Funding supporting the production of the various components of the global carbon budget in addition to the authors' supporting institutions (see also acknowledgements).

| Funder and grant number (where relevant) | Author Initials |
|---|---|
| Australia, Integrated Marine Observing System (IMOS) | BT |
| Australian National Environment Science Program (NESP) | JGC |
| Belgium, FWO (Flanders Research Foundation, contract IRI I001019N) | TG |
| BNP Paribas Foundation through Climate & Biodiversity initiative, philanthropic grant for developments of the Global Carbon Atlas | PC |
| Canada, Tula Foundation | WE |
| China, National Natural Science Foundation (grant no. 41975155) | XY |
| Commonwealth Scientific and Industrial Organization (CSIRO) - Climate Science Centre | JGC, JK |
| EC Copernicus Atmosphere Monitoring Service implemented by ECMWF on behalf of the European Commission | FC |
| EC Copernicus Marine Environment Monitoring Service implemented by Mercator Ocean | TTTC |
| EC H2020 (4C; grant no 821003) | PF, RMA, SS, GPP, PC, JIK, TI, LB, PL, LG, SL, NG |
| EC H2020 (CHE; grant no 776186) | MWJ |
| EC H2020 (CoCO2: grant no. 958927) | RMA, GPP |
| EC H2020 (COMFORT: grant no. 820989) | DCEB, LG |
| EC H2020 (CONSTRAIN: grant no 820829) | RS, PMF, TG |
| EC H2020 (CRESCENDO: grant no. 641816) | RS, EJ AJPS, TI |
| EC H2020 (ESM2025 – Earth System Models for the Future; grant agreement No 101003536). | RS, TG, TI, LB, BD |
| EC H2020 (EuroSea: grant no. 862626) | SDJ |
| EC H2020 (JERICO-S3: grant no. 871153) | GR |
| EC H2020 (QUINCY; grant no 647204) | SZ |
| EC H2020 (RINGO: grant no. 730944) | DCEB |
| EC H2020 (VERIFY: grant no. 776810) | MWJ, RMA, GPP, PC, JIK, NV, GG |
| Efg International | TT |
| EFG International | TT |
| European Space Agency Climate Change Initiative ESA-CCI RECCAP2 project 655 (ESRIN/4000123002/18/I-NB) | PF, SS, PC |
| European Space Agency OceanSODA project (grant no. 4000112091/14/I-LG) | LG |
| France, ICOS (Integrated Carbon Observation System) France | NL |
| France, Institut de Recherche pour le Développement (IRD) | NL |
| Germany, Blue Ocean and Federal Ministry of Education (BONUS INTEGRAL; Grant No. 03F0773A) | GR |
| Germany, Deutsche Forschungsgemeinschaft (DFG) under Germany's Excellence Strategy – EXC 2037 'Climate, Climatic Change, and Society' – Project Number: 390683824 | TI |
| Germany, Federal Ministry for Education and Research (BMBF) | GR |
| Germany, GEOMAR Helmholtz Centre for Ocean Research | SKL |
| Germany, German Federal Ministry of Education and Research under project "DArgo2025" (03F0857C) | AK |
| Germany, Helmholtz Association ATMO programme | PA |
| Germany, Helmholtz Young Investigator Group Marine Carbon and Ecosystem Feedbacks in the Earth System (MarESys), grant number VH-NG-1301 | JH, OG |
| Germany, ICOS (Integrated Carbon Observation System) Germany | GR, NL |
| Hapag-Lloyd | TT |



| | |
|---|---|
| Ireland, Marine Institute | MC |
| Japan, Environment Research and Technology Development Fund of the Ministry of the Environment (JPMEERF21S20810) | YN |
| Japan, Global Environmental Research Coordination System, Ministry of the Environment (grant number E1751) | SN, TO, CW |
| Kuehne + Nagel International AG | TT |
| Mediterranean Shipping Company (MSc) | TT |
| Monaco, Fondation Prince Albert II de Monaco | TT |
| Monaco, Yacht Club de Monaco | TT |
| NASA Interdisciplinary Research in Earth Science Program. | BP |
| Netherlands Organization for Scientific Research (NWO; grant no. SH-312, 17616) | WP |
| New Zealand, NIWA MBIE Core funding | KIC |
| Norway, Norwegian Research Council (grant no. 270061) | JS |
| Norway, Research Council of Norway, ICOS (Integrated Carbon Observation System) Norway and OTC (Ocean Thematic Centre) (grant no. 245927) | SKL, MB, SDJ |
| PEAK6 Investments | SKL |
| Saildrone Inc. | SKL |
| South Africa, Department of Science and Innovation | LD |
| South Africa, National Science Foundation | LD |
| Swiss National Science Foundation (grant no. 200020_172476) | SL |
| UK Royal Society (grant no. RP\R1\191063) | CLQ |
| UK, CLASS ERC funding | TG |
| UK, National Centre for Atmospheric Science (NCAS) | PCM |
| UK, Natural Environment Research Council (SONATA: grant no. NE/P021417/1) | DW |
| UK, Natural Environmental Research Council (NE/R016518/1) | LF |
| UK, Newton Fund, Met Office Climate Science for Service Partnership Brazil (CSSP Brazil) | AJWi |
| UK, Royal Society: The European Space Agency OCEANFLUX projects | AJWa |
| UK, University of Reading Research Endowment Trust Fund | PCM |
| USA, Department of Commerce, Office of Oceanic and Atmospheric Research (OAR)'s / National Oceanic and Atmospheric Administration (NOAA)'s Global Ocean Monitoring and Observation Program (GOMO) | DRM, CS, DP, RW, SRA, RAF, AJS, NRB |
| USA, Department of Commerce, Office of Oceanic and Atmospheric Research (OAR)'s / National Oceanic and Atmospheric Administration (NOAA)'s Ocean Acidification Program | DP, RW, SRA, RAF, AJS |
| USA, Department of Energy, Office of Science and BER prg. (grant no. DE-SC000 0016323) | AKJ |
| USA, Department of Energy, SciDac (DESC0012972) | GCH, LPC |
| USA, NASA Carbon Monitoring System probram and OCO Science team program (80NM0018F0583) . | JL |
| USA, NASA Interdisciplinary Research in Earth Science (IDS) (80NSSC17K0348) | GCH, LPC |
| USA, National Science Foundation (grant number 1903722) | HT |
| USA, National Science Foundation (grant number PLR 1543457) | DRM, CS |
| USA, Princeton University Environmental Institute and the NASA OCO2 science team, grant number 80NSSC18K0893. | LR |
| **Computing resources** | |
| bwHPC, High Performance Computing Network of the State of Baden-Württemberg, Germany | PA |
| Cheyenne supercomputer, Computational and Information Systems Laboratory (CISL) at National Center for Atmospheric Research (NCAR) | DK |
| Deutsches Klimarechenzentrum (allocation bm0891) | JEMSN, JP |
| MRI (FUJITSU Server PRIMERGY CX2550M5) | YN |



| Netherlands Organization for Scientific Research (NWO; NWO-2021.010) | ITL |
|---|---|
| NIES (SX-Aurora) | YN |
| NIES supercomputer system | EK |
| supercomputer 'Gadi' of the National Computational Infrastructure (NCI), Australia | JK |
| Supercomputing time was provided by the Météo-France/DSI supercomputing center. | RS, BD |
| TGCC under allocation 2019-A0070102201 made by GENCI | FC |
| UEA High Performance Computing Cluster, UK | MWJ, CLQ, DRW |
| UNINETT Sigma2, National Infrastructure for High Performance Computing and Data Storage in Norway (NN2980K/NS2980K) | JS |

**Appendix B. Supplementary Figures**

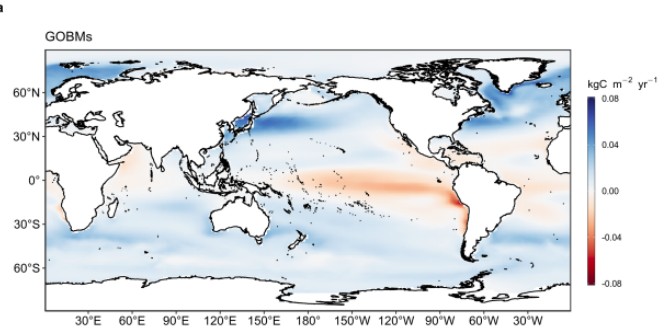

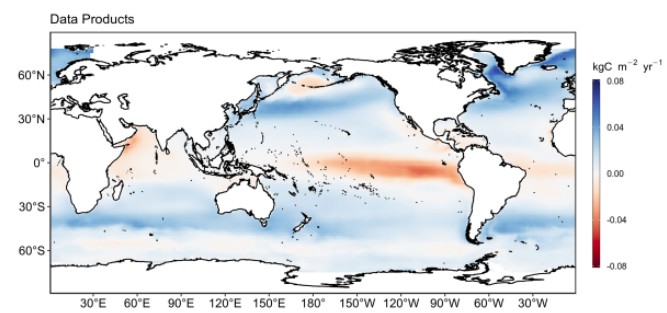

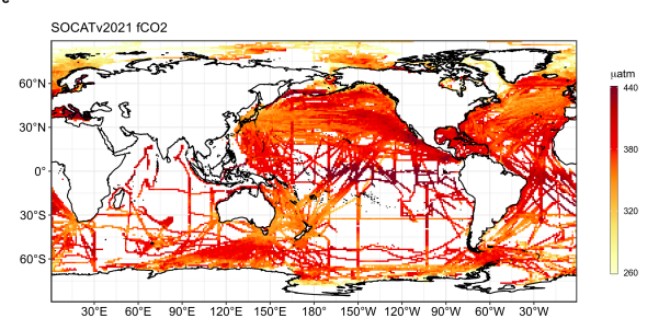

**Figure B1.** Ensemble mean air-sea $CO_2$ flux from a) global ocean biogeochemistry models and b)
$fCO_2$ based data products, averaged over 2011-2020 period (kgC $m^{-2}$ $yr^{-1}$). Positive numbers
indicate a flux into the ocean. c) gridded SOCAT v2021 $fCO_2$ measurements, averaged over the
2011-2020 period (µatm). In (a) model simulation A is shown. The data-products represent the
contemporary flux, i.e. including outgassing of riverine carbon, which is estimated to amount to
0.615 GtC $yr^{-1}$ globally.



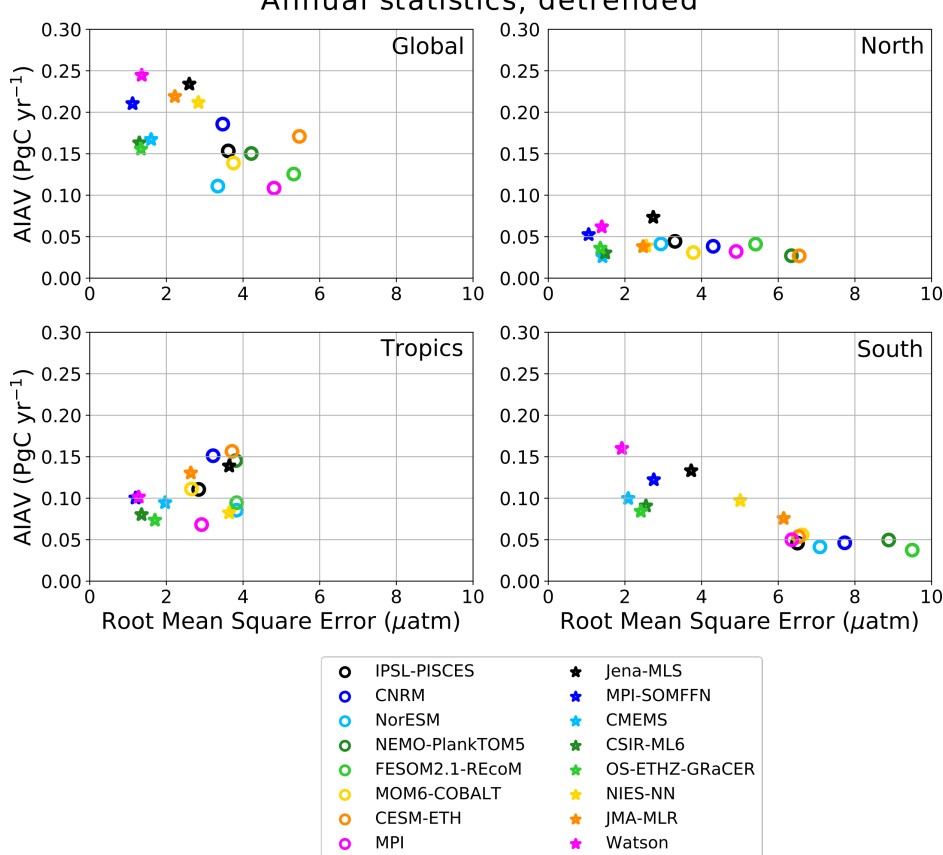

**Figure B2.** Evaluation of the GOBMs and data products using the root mean squared error (RMSE)

for the period 1990 to 2020, between the individual surface ocean $fCO_2$ mapping schemes and the

SOCAT v2021 database. The y-axis shows the amplitude of the interannual variability (A-IAV, taken

as the standard deviation of a detrended time series calculated as a 12-months running mean over

the monthly flux time series, Rödenbeck et al., 2015). Results are presented for the globe, north

(>30°N), tropics (30°S-30°N), and south (<30°S) for the GOBMs (see legend circles) and for the

$fCO_2$-based data products (star symbols). The $fCO_2$-based data products use the SOCAT database

and therefore are not independent from the data (see section 2.4.1).

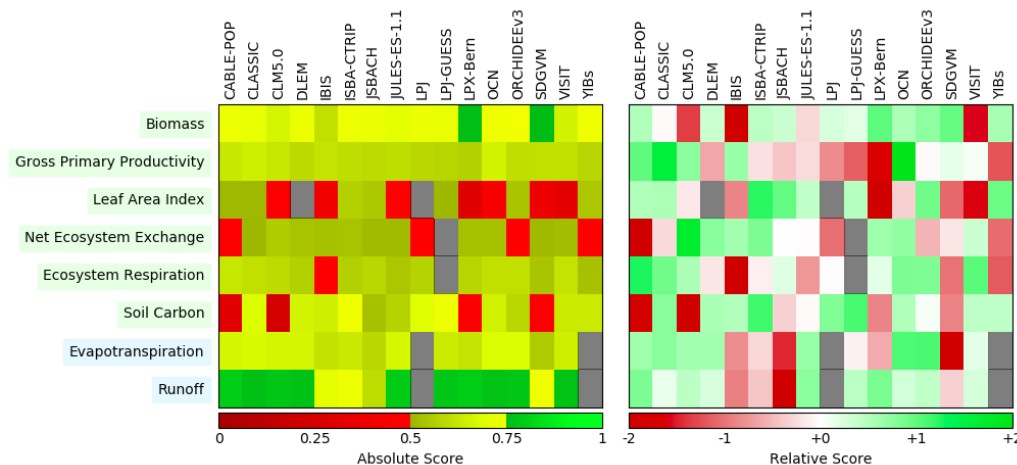

**Figure B3.** Evaluation of the DGVMs using the International Land Model Benchmarking system (ILAMB; Collier et al., 2018) (left) absolute skill scores and (right) skill scores relative to other models. The benchmarking is done with observations for vegetation biomass (Saatchi et al., 2011; and GlobalCarbon unpublished data; Avitabile et al., 2016), GPP (Jung et al., 2010; Lasslop et al., 2010), leaf area index (De Kauwe et al., 2011; Myneni et al., 1997), net ecosystem exchange (Jung et al., 2010;Lasslop et al., 2010), ecosystem respiration (Jung et al., 2010;Lasslop et al., 2010), soil carbon (Hugelius et al., 2013;Todd-Brown et al., 2013), evapotranspiration (De Kauwe et al., 2011), and runoff (Dai and Trenberth, 2002). For each model-observation comparison a series of error metrics are calculated, scores are then calculated as an exponential function of each error metric, finally for each variable the multiple scores from different metrics and observational data sets are combined to give the overall variable scores shown in the left panel. Overall variable scores increase from 0 to 1 with improvements in model performance. The set of error metrics vary with data set and can include metrics based on the period mean, bias, root mean squared error, spatial distribution, interannual variability and seasonal cycle. The relative skill score shown in the right panel is a Z-score, which indicates in units of standard deviation the model scores relative to the multi-model mean score for a given variable. Grey boxes represent missing model data.



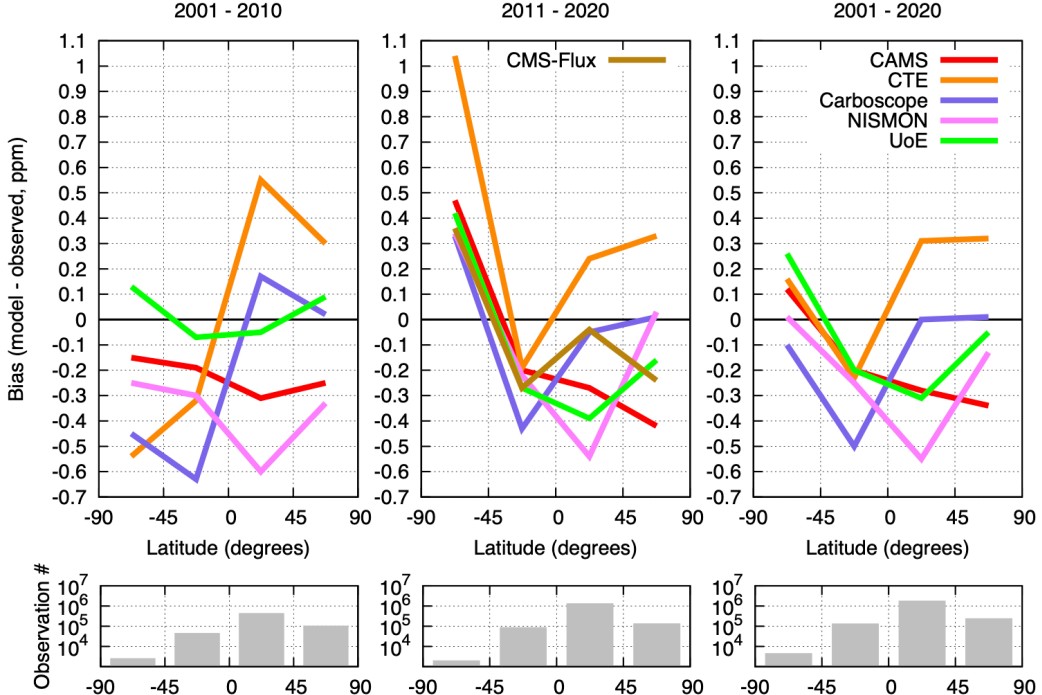

**Figure B4.** Evaluation of the atmospheric inversion products. The mean of the model minus
observations is shown for four latitude bands in three periods: (left) 2001-2010, (centre) 2011-
2020, (right) 2001-2020. The six models are compared to independent $CO_2$ measurements made
onboard aircraft over many places of the world between 2 and 7 km above sea level. Aircraft
measurements archived in the Cooperative Global Atmospheric Data Integration Project (CGADIP;
Cox et al., 2021) from sites, campaigns or programs that cover at least 9 months between 2001
and 2020 and that have not been assimilated, have been used to compute the biases of the
differences in four 45° latitude bins. Land and ocean data are used without distinction, and
observation density varies strongly with latitude and time as seen on the lower panels.

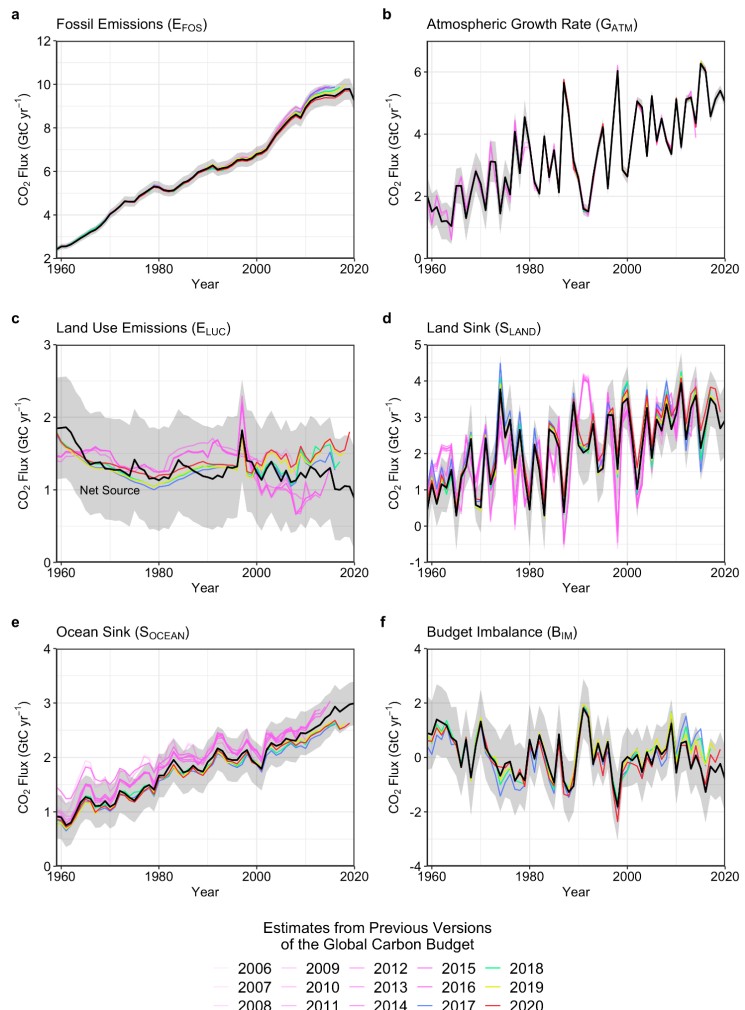

**Figure B5.** Comparison of the estimates of each component of the global carbon budget in this study (black line) with the estimates released annually by the GCP since 2006. Grey shading shows the uncertainty bounds representing ±1 standard deviation of the current global carbon budget, based on the uncertainty assessments described in Appendix C. $CO_2$ emissions from (a) fossil $CO_2$ emissions ($E_{FOS}$), and (b) land-use change ($E_{LUC}$), as well as their partitioning among (c) the atmosphere ($G_{ATM}$), (d) the land ($S_{LAND}$), and (e) the ocean ($S_{OCEAN}$). See legend for the corresponding years, and Tables 3 and A7 for references. The budget year corresponds to the year when the budget was first released. All values are in GtC yr$^{-1}$.





**Appendix C. Extended Methodology**
**Appendix C.1 Methodology Fossil Fuel CO$_2$ emissions (E$_{FOS}$)**
**C.1.1 Cement carbonation**
From the moment it is created, cement begins to absorb CO$_2$ from the atmosphere, a process
known as 'cement carbonation'. We estimate this CO$_2$ sink, as the average of two studies in the
literature (Cao et al., 2020; Guo et al., 2021). Both studies use the same model, developed by Xi et
al. (2016), with different parameterisations and input data, with the estimate of Guo and
colleagues being a revision of Xi et al (2016). The trends of the two studies are very similar.
Modelling cement carbonation requires estimation of a large number of parameters, including the
different types of cement material in different countries, the lifetime of the structures before
demolition, of cement waste after demolition, and the volumetric properties of structures, among
others (Xi et al., 2016). Lifetime is an important parameter because demolition results in the
exposure of new surfaces to the carbonation process. The main reasons for differences between
the two studies appear to be the assumed lifetimes of cement structures and the geographic
resolution, but the uncertainty bounds of the two studies overlap. In the present budget, we
include the cement carbonation carbon sink in the fossil CO$_2$ emission component (E$_{FOS}$).
**C.1.2 Emissions embodied in goods and services**
CDIAC, UNFCCC, and BP national emission statistics 'include greenhouse gas emissions and
removals taking place within national territory and offshore areas over which the country has
jurisdiction' (Rypdal et al., 2006), and are called territorial emission inventories. Consumption-
based emission inventories allocate emissions to products that are consumed within a country,
and are conceptually calculated as the territorial emissions minus the 'embodied' territorial
emissions to produce exported products plus the emissions in other countries to produce
imported products (Consumption = Territorial – Exports + Imports). Consumption-based emission
attribution results (e.g. Davis and Caldeira, 2010) provide additional information to territorial-
based emissions that can be used to understand emission drivers (Hertwich and Peters, 2009) and
quantify emission transfers by the trade of products between countries (Peters et al., 2011b). The
consumption-based emissions have the same global total, but reflect the trade-driven movement
of emissions across the Earth's surface in response to human activities. We estimate consumption-



based emissions from 1990-2018 by enumerating the global supply chain using a global model of
the economic relationships between economic sectors within and between every country (Andrew
and Peters, 2013; Peters et al., 2011a). Our analysis is based on the economic and trade data from
the Global Trade and Analysis Project (GTAP; Narayanan et al., 2015), and we make detailed
estimates for the years 1997 (GTAP version 5), 2001 (GTAP6), and 2004, 2007, and 2011
(GTAP9.2), covering 57 sectors and 141 countries and regions. The detailed results are then
extended into an annual time series from 1990 to the latest year of the Gross Domestic Product
(GDP) data (2018 in this budget), using GDP data by expenditure in current exchange rate of US
dollars (USD; from the UN National Accounts main Aggregrates database; UN, 2021) and time
series of trade data from GTAP (based on the methodology in Peters et al., 2011a). We estimate
the sector-level $CO_2$ emissions using the GTAP data and methodology, include flaring and cement
emissions from CDIAC, and then scale the national totals (excluding bunker fuels) to match the
emission estimates from the carbon budget. We do not provide a separate uncertainty estimate
for the consumption-based emissions, but based on model comparisons and sensitivity analysis,
they are unlikely to be significantly different than for the territorial emission estimates (Peters et
al., 2012a).
**C.1.3 Uncertainty assessment for $E_{FOS}$**
We estimate the uncertainty of the global fossil CO2 emissions at ±5% (scaled down from the
published ±10 % at ±2σ to the use of ±1σ bounds reported here; Andres et al., 2012). This is
consistent with a more detailed analysis of uncertainty of ±8.4% at ±2σ (Andres et al., 2014) and at
the high-end of the range of ±5-10% at ±2σ reported by (Ballantyne et al., 2015). This includes an
assessment of uncertainties in the amounts of fuel consumed, the carbon and heat contents of
fuels, and the combustion efficiency. While we consider a fixed uncertainty of ±5% for all years,
the uncertainty as a percentage of emissions is growing with time because of the larger share of
global emissions from emerging economies and developing countries (Marland et al., 2009).
Generally, emissions from mature economies with good statistical processes have an uncertainty
of only a few per cent (Marland, 2008), while emissions from strongly developing economies such
as China have uncertainties of around ±10% (for ±1σ; Gregg et al., 2008; Andres et al., 2014).
Uncertainties of emissions are likely to be mainly systematic errors related to underlying biases of
energy statistics and to the accounting method used by each country.

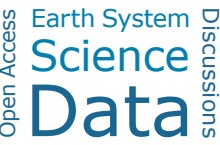

**C.1.4 Growth rate in emissions**
We report the annual growth rate in emissions for adjacent years (in percent per year) by
calculating the difference between the two years and then normalising to the emissions in the first
year: (EFOS(t0+1)-EFOS(t0))/EFOS(t0)×100%. We apply a leap-year adjustment where relevant to
ensure valid interpretations of annual growth rates. This affects the growth rate by about 0.3% yr-
1 (1/366) and causes calculated growth rates to go up approximately 0.3% if the first year is a leap
year and down 0.3% if the second year is a leap year.
The relative growth rate of $E_{FOS}$ over time periods of greater than one year can be rewritten using
its logarithm equivalent as follows:

$$\frac{1}{E_{FOS}}\frac{dE_{FOS}}{dt} = \frac{d(lnE_{FOS})}{dt} \qquad (2)$$

Here we calculate relative growth rates in emissions for multi-year periods (e.g. a decade) by
fitting a linear trend to $ln(E_{FOS})$ in Eq. (2), reported in percent per year.
**C.1.5 Emissions projection for 2021**
To gain insight on emission trends for 2021, we provide an assessment of global fossil $CO_2$
emissions, $E_{FOS}$, by combining individual assessments of emissions for China, USA, the EU, and
India (the four countries/regions with the largest emissions), and the rest of the world. We
provide full year estimates for two datasets: IEA (2021b) and our own analysis. This approach
differs from last year where we used four independent estimates including our own, because of
the unique circumstances related to the COVID-19 pandemic. This year's analysis is more in line
with earlier budgets.
Previous editions of the Global Carbon Budget (GCB) have estimated YTD emissions, and
performed projections, using sub-annual energy consumption data from a variety of sources
depending on the country or region. The YTD estimates have then been projected to the full year
using specific methods for each country or region. The methods described in detail below.
**China**: The method for the projection uses: (1) the sum of monthly domestic production of raw
coal, crude oil, natural gas and cement from the National Bureau of Statistics (NBS, 2021), (2)
monthly net imports of coal, coke, crude oil, refined petroleum products and natural gas from the
General Administration of Customs of the People's Republic of China (2021); proprietary monthly
estimates of sectoral coal consumption by the consultancy SX Coal (2021); and (3) annual energy



consumption data by fuel type and annual production data for cement from the NBS, using data
for 2000-2020 (NBS, 2021), with the last year being a preliminary estimate. We estimate the full-
year growth rate for 2021 using a Bayesian regression for the ratio between the annual energy
consumption data (3 above) from 2014 through 2019, and monthly production plus net imports
through August of each year (1+2 above) or the corresponding estimate from SX Coal for coal. The
uncertainty range uses the standard deviations of the resulting posteriors. Sources of uncertainty
and deviations between the monthly and annual growth rates include lack of or incomplete
monthly data on stock changes and energy density, variance in the trend during the last three
months of the year, and partially unexplained discrepancies between supply-side and
consumption data even in the final annual data.
Note that in recent years, the absolute value of the annual growth rate for coal energy
consumption, and hence total $CO_2$ emissions, has been consistently lower (closer to zero) than the
growth or decline suggested by the monthly, tonnage-based production and import data, and this
is reflected in the projection. This pattern is only partially explained by stock changes and changes
in energy content, and it is therefore not possible to be certain that it will continue in any given
year. For 2020 and 2021, COVID-19-related lockdown and reopening in China, similar but delayed
restrictions in major export markets, unusual amounts of flooding and extreme weather during
the summer months and extraordinarily high local and global prices of many energy products
imply that seasonal patterns and correlations between supply, stock changes and consumption
may be quite different this year than in the previous years that the regression is based on. Shocks
in the housing market and heightened perceptions of political risk among investors may also affect
consumption patterns. This adds a major but unquantified amount of uncertainty to the estimate.
**USA**: We use emissions estimated by the U.S. Energy Information Administration (EIA) in their
Short-Term Energy Outlook (STEO) for emissions from fossil fuels to get both YTD and a full year
projection (EIA, 2021). The STEO also includes a near-term forecast based on an energy
forecasting model which is updated monthly (last update with preliminary data through
September 2021), and takes into account expected temperatures, household expenditures by fuel
type, energy markets, policies, and other effects. We combine this with our estimate of emissions
from cement production using the monthly U.S. cement clinker production data from USGS for
January-June 2021, assuming changes in cement production over the first part of the year apply
throughout the year.


**India**:  We use monthly emissions estimates for India updated from Andrew (2020b) through
August 2021. These estimates are derived from many official monthly energy and other activity
data sources to produce direct estimates of national $CO_2$ emissions, without the use of proxies.
Emissions from coal are then extended to September using a regression relationship based on
power generated from coal, coal dispatches by Coal India Ltd., the composite PMI, time, and days
per month. For the last 3-4 months of the year, each series is extrapolated assuming typical
trends.
**EU**: We use a refinement to the methods presented by Andrew (2021), deriving emissions from
monthly energy data reported by Eurostat. Some data gaps are filled using data from the Joint
Organisations Data Initiative (JODI, 2021). Sub-annual cement production data are limited, but
data for Germany and Poland, the two largest producers, suggest a small decline. For fossil fuels
this provides estimates through July. We extend coal emissions through September using a
regression model built from generation of power from hard coal, power from brown coal, total
power generation, and the number of working days in Germany and Poland, the two biggest coal
consumers in the EU. These are then extended through the end of the year assuming typical
trends. We extend oil emissions by building a regression model between our monthly CO2
estimates and oil consumption reported by the EIA for Europe in its Short-Term Energy Outlook
(October edition), and then using this model with EIA's monthly forecasts. For natural gas, the
strong seasonal signal allows the use of the bias-adjusted Holt-Winters exponential smoothing
method (Chatfield, 1978).
**Rest of the world**: We use the close relationship between the growth in GDP and the growth in
emissions (Raupach et al., 2007) to project emissions for the current year. This is based on a
simplified Kaya Identity, whereby $E_{FOS}$ (GtC yr$^{-1}$) is decomposed by the product of GDP (USD yr$^{-1}$)
and the fossil fuel carbon intensity of the economy ($I_{FOS}$; GtC USD$^{-1}$) as follows:
$E_{FOS} = GDP \times I_{FOS}$         (3)
Taking a time derivative of Equation (3) and rearranging gives:
$\frac{1}{E_{FOS}}\frac{dE_{FOS}}{dt} = \frac{1}{GDP}\frac{dGDP}{dt} + \frac{1}{I_{FOS}}\frac{dI_{FOS}}{dt}$       (4)
where the left-hand term is the relative growth rate of $E_{FOS}$, and the right-hand terms are the
relative growth rates of GDP and $I_{FOS}$, respectively, which can simply be added linearly to give the
overall growth rate.



The $I_{FOS}$ is based on GDP in constant PPP (Purchasing Power Parity) from the International Energy
Agency (IEA) up to 2017 (IEA/OECD, 2019) and extended using the International Monetary Fund
(IMF) growth rates through 2020 (IMF, 2021). Interannual variability in $I_{FOS}$ is the largest source of
uncertainty in the GDP-based emissions projections. We thus use the standard deviation of the
annual $I_{FOS}$ for the period 2009-2019 as a measure of uncertainty, reflecting a $\pm 1\sigma$ as in the rest of
the carbon budget.
**World**: The global total is the sum of each of the countries and regions.
**Appendix C.2 Methodology $CO_2$ emissions from land-use, land-use change and forestry ($E_{LUC}$)**
The net $CO_2$ flux from land-use, land-use change and forestry ($E_{LUC}$, called land-use change
emissions in the rest of the text) includes $CO_2$ fluxes from deforestation, afforestation, logging and
forest degradation (including harvest activity), shifting cultivation (cycle of cutting forest for
agriculture, then abandoning), and regrowth of forests following wood harvest or abandonment
of agriculture. Emissions from peat burning and drainage are added from external datasets (see
section C.2.1 below). Only some land-management activities are included in our land-use change
emissions estimates (Table A1). Some of these activities lead to emissions of $CO_2$ to the
atmosphere, while others lead to $CO_2$ sinks. $E_{LUC}$ is the net sum of emissions and removals due to
all anthropogenic activities considered. Our annual estimate for 1960-2020 is provided as the
average of results from three bookkeeping approaches (Section C.2.1 below): an estimate using
the Bookkeeping of Land Use Emissions model (Hansis et al., 2015; hereafter BLUE) and one using
the compact Earth system model OSCAR (Gasser et al., 2020), both BLUE and OSCAR being
updated here to new land-use forcing covering the time period until 2020, and an updated version
of the estimate published by Houghton and Nassikas (2017) (hereafter updated H&N2017). All
three data sets are then extrapolated to provide a projection for 2021 (Section C.2.5 below). In
addition, we use results from Dynamic Global Vegetation Models (DGVMs; see Section 2.5 and
Table 4) to help quantify the uncertainty in $E_{LUC}$ (Section C.2.4), and thus better characterise our
understanding. Note that in this budget, we use the scientific $E_{LUC}$ definition, which counts fluxes
due to environmental changes on managed land towards $S_{LAND}$, as opposed to the national
greenhouse gas inventories under the UNFCCC, which include them in $E_{LUC}$ and thus often report



smaller land-use emissions (Grassi et al., 2018; Petrescu et al., 2020). However, we provide a
methodology of mapping of the two approaches to each other further below (Section C.2.3).
**C.2.1 Bookkeeping models**
Land-use change $CO_2$ emissions and uptake fluxes are calculated by three bookkeeping models.
These are based on the original bookkeeping approach of Houghton (2003) that keeps track of the
carbon stored in vegetation and soils before and after a land-use change (transitions between
various natural vegetation types, croplands, and pastures). Literature-based response curves
describe decay of vegetation and soil carbon, including transfer to product pools of different
lifetimes, as well as carbon uptake due to regrowth. In addition, the bookkeeping models
represent long-term degradation of primary forest as lowered standing vegetation and soil carbon
stocks in secondary forests, and include forest management practices such as wood harvests.
BLUE and the updated H&N2017 exclude land ecosystems' transient response to changes in
climate, atmospheric $CO_2$ and other environmental factors, and base the carbon densities on
contemporary data from literature and inventory data. Since carbon densities thus remain fixed
over time, the additional sink capacity that ecosystems provide in response to $CO_2$-fertilization
and some other environmental changes is not captured by these models (Pongratz et al., 2014).
On the contrary, OSCAR includes this transient response, and it follows a theoretical framework
(Gasser and Ciais, 2013) that allows separating bookkeeping land-use emissions and the loss of
additional sink capacity. Only the former is included here, while the latter is discussed in Appendix
D4. The bookkeeping models differ in (1) computational units (spatially explicit treatment of land-
use change for BLUE, regional-/ mostly country-level for the updated H&N2017 and OSCAR), (2)
processes represented (see Table A1), and (3) carbon densities assigned to vegetation and soil of
each vegetation type (literature-based for the updated H&N2017 and BLUE, calibrated to DGVMs
for OSCAR). A notable difference between models exists with respect to the treatment of shifting
cultivation. The update of H&N2017 changed the approach over the earlier H&N2017 version:
H&N2017 had assumed the "excess loss" of tropical forests (i.e., when FRA indicated a forest loss
larger than the increase in agricultural areas from FAO) resulted from converting forests to
croplands at the same time older croplands were abandoned. Those abandoned croplands began
to recover to forests after 15 years. The updated H&N2017 now assumes that forest loss in excess
of increases in cropland and pastures represented an increase in shifting cultivation. When the
excess loss of forests was negative, it was assumed that shifting cultivation was returned to forest.



Historical areas in shifting cultivation were extrapolated taking into account country-based
estimates of areas in fallow in 1980 (FAO/UNEP, 1981) and expert opinion (from Heinimann et al.,
2017). In contrast, the BLUE and OSCAR models include sub-grid-scale transitions between all
vegetation types. Furthermore, the updated H&N2017 assume conversion of natural grasslands to
pasture, while BLUE and OSCAR allocate pasture proportionally on all natural vegetation that
exists in a grid-cell. This is one reason for generally higher emissions in BLUE and OSCAR.
Bookkeeping models do not directly capture carbon emissions from peat fires, which can create
large emissions and interannual variability due to synergies of land-use and climate variability in
Southeast Asia, particularly during El-Niño events, nor emissions from the organic layers of
drained peat soils. To correct for this, the updated H&N2017 includes carbon emissions from
burning and draining of peatlands in Indonesia, Malaysia, and Papua New Guinea (based on the
Global Fire Emission Database (GFED4s; van der Werf et al., 2017) for fire and Hooijer et al. for
drainage. Further, estimates of carbon losses from peatlands in extra-tropical regions are added
from Qiu et al. (2021). We add GFED4s peat fire emissions to BLUE and OSCAR output as well as
the global FAO peat drainage emissions 1990-2018 from croplands and grasslands (Conchedda
and Tubiello, 2020), keeping post-2018 emissions constant. We linearly increase tropical drainage
emissions from 0 in 1980, consistent with H&N2017's assumption, and keep emissions from the
often old drained areas of the extra-tropics constant pre-1990. This adds 9.0 GtC for FAO
compared to 5.6 GtC for Hooijer et al. (2010). Peat fires add another 2.0 GtC over the same
period.
The three bookkeeping estimates used in this study differ with respect to the land-use change
data used to drive the models. The updated H&N2017 base their estimates directly on the Forest
Resource Assessment of the FAO which provides statistics on forest-area change and management
at intervals of five years currently updated until 2020 (FAO, 2020). The data is based on country
reporting to FAO and may include remote-sensing information in more recent assessments.
Changes in land-use other than forests are based on annual, national changes in cropland and
pasture areas reported by FAO (FAOSTAT, 2021). On the other hand, BLUE uses the harmonised
land-use change data LUH2-GCB2021 covering the entire 850-2020 period (an update to the
previously released LUH2 v2h dataset; Hurtt et al., 2017; Hurtt et al., 2020), which was also used
as input to the DGVMs (Sec. 2.2.2). It describes land-use change, also based on the FAO data as
well as the HYDE3.3 dataset (Goldewijk et al., 2017a, 2017b), but provided at a quarter-degree





spatial resolution, considering sub-grid-scale transitions between primary forest, secondary forest,
primary non-forest, secondary non-forest, cropland, pasture, rangeland, and urban land (Hurtt et
al., 2020; Chini et al., 2021). LUH2-GCB2021 provides a distinction between rangelands and
pasture, based on inputs from HYDE. To constrain the models' interpretation on whether
rangeland implies the original natural vegetation to be transformed to grassland or not (e.g.,
browsing on shrubland), a forest mask was provided with LUH2-GCB2021; forest is assumed to be
transformed to grasslands, while other natural vegetation remains (in case of secondary
vegetation) or is degraded from primary to secondary vegetation (Ma et al., 2020). This is
implemented in BLUE. OSCAR was run with both LUH2-GCB2021 and FAO/FRA (as used by
Houghton and Nassikas, 2017), where emissions from the latter were extended beyond 2015 with
constant 2011–2015 average values. The best-guess OSCAR estimate used in our study is a
combination of results for LUH2-GCB2021 and FAO/FRA land-use data and a large number of
perturbed parameter simulations weighted against an observational constraint. All three
bookkeeping estimates were extended from 2020 to provide a projection for 2021 by adding the
annual change in emissions from tropical deforestation and degradation and peat burning and
drainage to the respective model's estimate for 2020 (van der Werf et al., 2017, Conchedda &
Tubiello, 2020).
For $E_{LUC}$ from 1850 onwards we average the estimates from BLUE, the updated H&N2017 and
OSCAR. For the cumulative numbers starting 1750 an average of four earlier publications is added
(30 ± 20 PgC 1750-1850, rounded to nearest 5; Le Quéré et al., 2016).
We provide estimates of the gross land use change fluxes from which the reported net land-use
change flux, $E_{LUC}$, is derived as a sum. Gross fluxes are derived internally by the three bookkeeping
models: Gross emissions stem from decaying material left dead on site and from products after
clearing of natural vegetation for agricultural purposes, wood harvesting, emissions from peat
drainage and peat burning, and, for BLUE, additionally from degradation from primary to
secondary land through usage of natural vegetation as rangeland. Gross removals stem from
regrowth after agricultural abandonment and wood harvesting. Gross fluxes for the updated
H&N2017 2016-2020 and for the 2021 projection of all three models were based on a regression
of gross sources (including peat emissions) to net emissions for recent years.
Due to an artifact in the HYDE3.3 data causing large abrupt transitions, an unrealistic peak in
emissions occurs around 1960 in BLUE and OSCAR. To correct for this, we replace the estimates
for 1959-1961 by the average of 1958 and 1962 in each BLUE and OSCAR. Abrupt transitions will
immediately influence gross emissions, which have a larger instantaneous component. Processes
with longer timescales, such as slow legacy emissions and regrowth, are inseparable from the
carbon dynamics due to subsequent land-use change events. We therefore do not adjust gross
removals, but only gross emissions to match the corrected net flux. Since DGVMs estimates are
only used for an uncertainty range of $E_{LUC}$, which is independent of land-use changes, no
correction is applied to the DGVMs data.
**C.2.2 Dynamic Global Vegetation Models (DGVMs)**
Land-use change $CO_2$ emissions have also been estimated using an ensemble of 17 DGVMs
simulations. The DGVMs account for deforestation and regrowth, the most important components
of $E_{LUC}$, but they do not represent all processes resulting directly from human activities on land
(Table A1). All DGVMs represent processes of vegetation growth and mortality, as well as
decomposition of dead organic matter associated with natural cycles, and include the vegetation
and soil carbon response to increasing atmospheric $CO_2$ concentration and to climate variability
and change. Most models explicitly simulate the coupling of carbon and nitrogen cycles and
account for atmospheric N deposition and N fertilisers (Table A1). The DGVMs are independent
from the other budget terms except for their use of atmospheric $CO_2$ concentration to calculate
the fertilization effect of $CO_2$ on plant photosynthesis.
DGVMs that do not simulate subgrid scale transitions (i.e., net land-use emissions; see Table A1)
used the HYDE land-use change data set (Goldewijk et al., 2017a, 2017b), which provides annual
(1700-2019), half-degree, fractional data on cropland and pasture. The data are based on the
available annual FAO statistics of change in agricultural land area available until 2015. The new
HYDE3.3 cropland/grazing land dataset which now in addition to FAO country-level statistics is
constrained spatially based on multi-year satellite land cover maps from ESA CCI LC. Data from
HYDE3.3 is based on a FAO which includes yearly data from 1961 up to and including the year
2017. After the year 2017 HYDE extrapolates the cropland, pasture, and urban data, based on the
trend over the previous 5 years, to generate data until the year 2020. HYDE also uses satellite
imagery from ESA-CCI from 1992 – 2018 for more detailed yearly allocation of cropland and
grazing land. The 2018 map is also used for the 2019-2020 period. The original 300 meter
resolution data from ESA was aggregated to a 5 arc minute resolution according to the
classification scheme as described in Klein Goldewijk et al (2017a). DGVMs that simulate subgrid




scale transitions (i.e., gross land-use emissions; see Table A1) also use the LUH2-GCB2021 data set,
an update of the more comprehensive harmonised land-use data set (Hurtt et al., 2020), that
further includes fractional data on primary and secondary forest vegetation, as well as all
underlying transitions between land-use states (850-2020; Hurtt et al., 2011, 2017, 2020; Chini et
al., 2021; Table A1). This new data set is of quarter degree fractional areas of land-use states and
all transitions between those states, including a new wood harvest reconstruction, new
representation of shifting cultivation, crop rotations, management information including irrigation
and fertilizer application. The land-use states include five different crop types in addition to the
pasture-rangeland split discussed before. Wood harvest patterns are constrained with Landsat-
based tree cover loss data (Hansen et al. 2013). Updates of LUH2-GCB2021 over last year's version
(LUH2-GCB2020) are using the most recent HYDE/FAO release (covering the time period up to
2021 included). We also use the most recent FAO wood harvest data for all years from 1961 to
2019. After the year 2019 we extrapolated the wood harvest data until the year 2020. The
HYDE3.3 population data is also used to extend the wood harvest time series back in time. Other
wood harvest inputs (for years prior to 1961) remain the same in LUH2.
DGVMs implement land-use change differently (e.g., an increased cropland fraction in a grid cell
can either be at the expense of grassland or shrubs, or forest, the latter resulting in deforestation;
land cover fractions of the non-agricultural land differ between models). Similarly, model-specific
assumptions are applied to convert deforested biomass or deforested area, and other forest
product pools into carbon, and different choices are made regarding the allocation of rangelands
as natural vegetation or pastures.
The difference between two DGVMs simulations (See Section C4.1 below), one forced with
historical changes in land-use and a second with time-invariant pre-industrial land cover and pre-
industrial wood harvest rates, allows quantification of the dynamic evolution of vegetation
biomass and soil carbon pools in response to land-use change in each model ($E_{LUC}$). Using the
difference between these two DGVMs simulations to diagnose $E_{LUC}$ means the DGVMs account for
the loss of additional sink capacity (around 0.4 ± 0.3 GtC yr-1; see Section 2.7.4, Appendix D4),
while the bookkeeping models do not.
As a criterion for inclusion in this carbon budget, we only retain models that simulate a positive
$E_{LUC}$ during the 1990s, as assessed in the IPCC AR4 (Denman et al., 2007) and AR5 (Ciais et al.,
2013).  All DGVMs met this criterion, although one model was not included in the $E_{LUC}$ estimate





from DGVMs as it exhibited a spurious response to the transient land cover change forcing after
its initial spin-up.
**C.2.3 Mapping of national GHG inventory data to $E_{LUC}$**
For the first time, an approach is implemented to reconcile the large gap between ELUC from
bookkeeping models and land use, land-use change and forestry (LULUCF) from national GHG
Inventories (NGHGI) (see Tab. A8). This gap is due to different approaches to calculating
"anthropogenic" $CO_2$ fluxes related to land-use change and land management (Grassi et al. 2018).
In particular, the land sinks due to environmental change on managed lands are treated as non-
anthropogenic in the global carbon budget, while they are generally considered as anthropogenic
in NGHGIs ("indirect anthropogenic fluxes"; Eggleston et al., 2006). Building on previous studies
(Grassi et al. 2021), the approach implemented here adds the DGVMs estimates of $CO_2$ fluxes due
to environmental change from countries' managed forest area (part of the $S_{LAND}$) to the original
$E_{LUC}$ flux. This sum is expected to be conceptually more comparable to LULUCF than simply $E_{LUC}$.
ELUC data are taken from bookkeeping models, in line with the global carbon budget approach. To
determine $S_{LAND}$ on managed forest, the following steps were taken: Spatially gridded data of
"natural" forest NBP ($S_{LAND}$ i.e., due to environmental change and excluding land use change
fluxes) were obtained with S2 runs from DGVMs up to 2019 from the TRENDY v9 dataset. Results
were first masked with the Hansen forest map (Hansen et al. 2013), with a 20% tree cover and
following the FAO definition of forest (isolated pixels with maximum connectivity less than 0.5 ha
are excluded), and then further masked with the "intact" forest map for the year 2013, i.e. forest
areas characterized by no remotely detected signs of human activity (Potapov et al. 2017). This
way, we obtained the $S_{LAND}$ in "intact" and "non-intact" forest area, which previous studies (Grassi
et al. 2021) indicated to be a good proxy, respectively, for "unmanaged" and "managed" forest
area in the NGHGI. Note that only 4 models (CABLE-POP, CLASSIC, YIBs and ORCHIDEE-CNP) had
forest NBP at grid cell level. Two models (OCN and ISBA-CTRIP) provided forest NEP and simulated
disturbances at pixel level that were used as basis, in addition to forest cover fraction, to estimate
forest NBP. For the other DGVMs, when a grid cell had forest, all the NBP was allocated to forest.
LULUCF data from NGHGIs are from Grassi et al. (2021) until 2017, updated until 2019 for Annex I
countries. For non-Annex I countries, the years 2018 and 2019 were assumed equal to the average
2013-2017. This data includes all CO2 fluxes from land considered managed, which in principle



encompasses all land uses (forest land, cropland, grassland, wetlands, settlements, and other
land), changes among them, emissions from organic soils and from fires. In practice, although
almost all Annex I countries report all land uses, many non-Annex I countries report only on
deforestation and forest land, and only few countries report on other land uses. In most cases,
NGHGI include most of the natural response to recent environmental change, because they use
direct observations (e.g., national forest inventories) that do not allow separating direct and
indirect anthropogenic effects (Eggleston et al., 2006).
To provide additional, largely independent assessments of fluxes on unmanaged vs managed
lands, we include a DGVM that allows diagnosing fluxes from unmanaged vs managed lands by
tracking vegetation cohorts of different ages separately. This model, ORCHIDEE-MICT (Yue et al.,
2018), was run using the same LUH2 forcing as the DGVMs used in this budget (Section 2.5) and
the bookkeeping models BLUE and OSCAR (Section 2.2). Old-aged forest was classified as primary
forest after a certain threshold of carbon density was reached again, and the model-internal
distinction between primary and secondary forest used as proxies for unmanaged vs managed
forests; agricultural lands are added to the latter to arrive at total managed land.
Tab. A8 shows the resulting mapping of global carbon cycle models' land flux definitions to that of
the NGHGI (discussed in Sec. 3.2.2). Note that estimates in this table are based on the global
carbon budget estimates from Friedlingstein et al. (2020), which estimated higher emissions from
the net land-use change flux ($E_{LUC}$) and a larger natural terrestrial sink. ORCHIDEE-MICT estimates
for SLAND on intact forests are expected to be higher than based on DGVMs in combination with
the NGHGI managed/unmanaged forest data because the unmanaged forest area, with about 27
mio km2, is estimated to be substantially larger by ORCHIDEE-MICT than, with less than 10 mio
km2, by the NGHGI, while managed forest area is estimated to be smaller (22 compared to 32 mio
km2). Related to this, $S_{LAND}$ on non-intact lands plus $E_{LUC}$ is a larger source estimated by ORCHIDEE-
MICT compared to NGHGI. We also show as comparison FAOSTAT emissions totals (FAO, 2021),
which include emissions from net forest conversion and fluxes on forest land (Tubiello et al., 2021)
as well as $CO_2$ emissions from peat drainage and peat fires.
**C.2.4 Uncertainty assessment for $E_{LUC}$**
Differences between the bookkeeping models and DGVMs models originate from three main
sources: the different methodologies, which among others lead to inclusion of the loss of



additional sink capacity in DGVMs (see Appendix D1.4), the underlying land-use/land cover data
set, and the different processes represented (Table A1). We examine the results from the DGVMs
models and of the bookkeeping method and use the resulting variations as a way to characterise
the uncertainty in $E_{LUC}$.
Despite these differences, the $E_{LUC}$ estimate from the DGVMs multi-model mean is consistent with
the average of the emissions from the bookkeeping models (Table 5). However there are large
differences among individual DGVMs (standard deviation at around 0.5 GtC yr$^{-1}$; Table 5), between
the bookkeeping estimates (average difference 1850-2020 BLUE-updated H&N2017 of 0.8 GtC yr$^{-1}$,
BLUE-OSCAR of 0.4 GtC yr$^{-1}$, OSCAR-updated H&N2017 of 0.3 GtC yr$^{-1}$), and between the updated
estimate of H&N2017 and its previous model version (Houghton et al., 2012). A factorial analysis
of differences between BLUE and H&N2017 attributed them particularly to differences in carbon
densities between natural and managed vegetation or primary and secondary vegetation (Bastos
et al., 2021). Earlier studies additionally showed the relevance of the different land-use forcing as
applied (in updated versions) also in the current study (Gasser et al., 2020).
The uncertainty in $E_{LUC}$ of ±0.7 GtC yr$^{-1}$ reflects our best value judgment that there is at least 68%
chance (±1σ) that the true land-use change emission lies within the given range, for the range of
processes considered here. Prior to the year 1959, the uncertainty in $E_{LUC}$ was taken from the
standard deviation of the DGVMs. We assign low confidence to the annual estimates of $E_{LUC}$
because of the inconsistencies among estimates and of the difficulties to quantify some of the
processes in DGVMs.
**C.2.5 Emissions projections for $E_{LUC}$**
We project the 2021 land-use emissions for BLUE, the updated H&N2017 and OSCAR, starting
from their estimates for 2020 assuming unaltered peat drainage, which has low interannual
variability, and the highly variable emissions from peat fires, tropical deforestation and
degradation as estimated using active fire data (MCD14ML; Giglio et al., 2016). Those latter scale
almost linearly with GFED over large areas (van der Werf et al., 2017), and thus allows for tracking
fire emissions in deforestation and tropical peat zones in near-real time. During most years,
emissions during January-September cover most of the fire season in the Amazon and Southeast
Asia, where a large part of the global deforestation takes place, and our estimates capture
emissions until the end of September.



**Appendix C.3 Methodology Ocean CO$_2$ sink**

**C.3.1 Observation-based estimates**

We primarily use the observational constraints assessed by IPCC of a mean ocean CO$_2$ sink of 2.2 ±

0.7 GtC yr$^{-1}$ for the 1990s (90% confidence interval; Ciais et al., 2013) to verify that the GOBMs

provide a realistic assessment of S$_{OCEAN}$.  This is based on indirect observations with seven

different methodologies and their uncertainties, using the methods that are deemed most reliable

for the assessment of this quantity (Denman et al., 2007; Ciais et al., 2013). The observation-based

estimates use the ocean/land CO$_2$ sink partitioning from observed atmospheric CO$_2$ and O$_2$/N$_2$

concentration trends (Manning and Keeling, 2006; Keeling and Manning, 2014), an oceanic

inversion method constrained by ocean biogeochemistry data (Mikaloff Fletcher et al., 2006), and

a method based on penetration time scale for chlorofluorocarbons (McNeil et al., 2003). The IPCC

estimate of 2.2 GtC yr$^{-1}$ for the 1990s is consistent with a range of methods (Wanninkhof et al.,

2013). We refrain from using the IPCC estimates for the 2000s (2.3 ± 0.7 GtC yr$^{-1}$), and the period

2002-2011 (2.4  ± 0.7 GtC yr$^{-1}$, Ciais et al., 2013) as these are based on trends derived mainly from

models and one data-product (Ciais et al., 2013). Additional constraints summarized in AR6

(Canadell et al., 2021) are the interior ocean anthropogenic carbon change (Gruber et al., 2019)

and ocean sink estimate from atmospheric CO$_2$ and O$_2$/N$_2$ (Tohjima et al., 2019) which are used

for model evaluation and discussion, respectively.

We also use eight estimates of the ocean CO$_2$ sink and its variability based on surface ocean fCO$_2$

maps obtained by the interpolation of surface ocean fCO$_2$ measurements from 1990 onwards due

to severe restriction in data availability prior to 1990 (Figure 9).  These estimates differ in many

respects: they use different maps of surface fCO$_2$, different atmospheric CO$_2$ concentrations, wind

products and different gas-exchange formulations as specified in Table A3. We refer to them as

fCO$_2$-based flux estimates. The measurements underlying the surface fCO$_2$ maps are from the

Surface Ocean CO$_2$ Atlas version 2021 (SOCATv2021; Bakker et al., 2021), which is an update of

version 3 (Bakker et al., 2016) and contains quality-controlled data through 2020 (see data

attribution Table A5). Each of the estimates uses a different method to then map the SOCAT

v2021 data to the global ocean. The methods include a data-driven diagnostic method (Rödenbeck

et al., 2013; referred to here as Jena-MLS), three neural network models (Landschützer et al.,





2014; referred to as MPI-SOMFFN; Chau et al., 2021; Copernicus Marine Environment Monitoring
Service, referred to here as CMEMS-LSCE-FFNN; and Zeng et al., 2014; referred to as NIES-FNN),
two cluster regression approaches (Gregor et al., 2019; referred to here as CSIR-ML6; and Gregor
and Gruber, 2021, referred to as OS-ETHZ-GRaCER), and a multi-linear regression method (Iida et
al., 2021; referred to as JMA-MLR). The ensemble mean of the $fCO_2$-based flux estimates is
calculated from these seven mapping methods. Further, we show the flux estimate of Watson et
al. (2020) who also use the MPI-SOMFFN method to map the adjusted $fCO_2$ data to the globe, but
resulting in a substantially larger ocean sink estimate, owing to a number of adjustments they
applied to the surface ocean $fCO_2$ data and the gas-exchange parameterization. Concretely, these
authors adjusted the SOCAT $fCO_2$ downward to account for differences in temperature between
the depth of the ship intake and the relevant depth right near the surface, and included a further
adjustment to account for the cool surface skin temperature effect. The Watson et al. flux
estimate hence differs from the others by their choice of adjusting the flux to a cool, salty ocean
surface skin. Watson et al. (2020) showed that this temperature adjustment leads to an upward
correction of the ocean carbon sink, up to 0.9 GtC $yr^{-1}$, that, if correct, should be applied to all
$fCO_2$-based flux estimates. So far, this adjustment is based on a single line of evidence and hence
associated with low confidence until further evidence is available.  The Watson et al flux estimate
presented here is therefore not included in the ensemble mean of the $fCO_2$-based flux estimates.
This choice will be re-evaluated in upcoming budgets based on further lines of evidence.
The $CO_2$ flux from each $fCO_2$-based product is either already at or above 98% areal coverage (Jena-
MLS, OS-ETHZ-GRaCER), filled by the data-provider (using Fay et al., 2021a, method for JMA-MLR;
and Landschützer et al., 2020, methodology for MPI-SOMFFN) or scaled for the remaining
products by the ratio of the total ocean area covered by the respective product to the total ocean
area (361.9e6 $km^2$) from ETOPO1 (Amante and Eakins, 2009; Eakins and Sharman, 2010). In
products where the covered area varies with time (e.g., CMEMS-LSCE-FFNN) we use the maximum
area coverage. The lowest coverage is 93% (NIES-NN), resulting in a maximum adjustment factor
of 1.08 (Table A3, Hauck et al., 2020).
We further use results from two diagnostic ocean models, Khatiwala et al. (2013) and DeVries
(2014), to estimate the anthropogenic carbon accumulated in the ocean prior to 1959. The two
approaches assume constant ocean circulation and biological fluxes, with $S_{OCEAN}$ estimated as a
response in the change in atmospheric $CO_2$ concentration calibrated to observations. The



uncertainty in cumulative uptake of ±20 GtC (converted to ±1σ) is taken directly from the IPCC's
review of the literature (Rhein et al., 2013), or about ±30% for the annual values (Khatiwala et al.,

3    2009).

**C.3.2 Global Ocean Biogeochemistry Models (GOBMs)**
The ocean $CO_2$ sink for 1959-2019 is estimated using eight GOBMs (Table A2). The GOBMs
represent the physical, chemical, and biological processes that influence the surface ocean
concentration of $CO_2$ and thus the air-sea $CO_2$ flux. The GOBMs are forced by meteorological
reanalysis and atmospheric $CO_2$ concentration data available for the entire time period. They
mostly differ in the source of the atmospheric forcing data (meteorological reanalysis), spin up
strategies, and in their horizontal and vertical resolutions (Table A2). All GOBMs except one
(CESM-ETHZ) do not include the effects of anthropogenic changes in nutrient supply (Duce et al.,
2008). They also do not include the perturbation associated with changes in riverine organic
carbon (see Section 2.7.3).
Three sets of simulations were performed with each of the GOBMs. Simulation A applied historical
changes in climate and atmospheric $CO_2$ concentration. Simulation B is a control simulation with
constant atmospheric forcing (normal year or repeated year forcing) and constant pre-industrial
atmospheric $CO_2$ concentration. Simulation C is forced with historical changes in atmospheric $CO_2$
concentration, but repeated year or normal year atmospheric climate forcing. To derive $S_{OCEAN}$
from the model simulations, we subtracted the annual time series of the control simulation B from
the annual time series of simulation A. Assuming that drift and bias are the same in simulations A
and B, we thereby correct for any model drift. Further, this difference also removes the natural
steady state flux (assumed to be 0 GtC yr$^{-1}$ globally without rivers) which is often a major source of
biases. Simulation B of IPSL had to be treated differently as it was forced with constant
atmospheric $CO_2$ but observed historical changes in climate. For IPSL, we fitted a linear trend to
the simulation B and subtracted this linear trend from simulation A. This approach assures that
the interannual variability is not removed from IPSL simulation A.
The absolute correction for bias and drift per model in the 1990s varied between <0.01 GtC yr$^{-1}$
and 0.26 GtC yr$^{-1}$, with six models having positive biases, and one model having essentially no bias
(NorESM). The remaining model (MPI) uses riverine input and therefore simulates outgassing in
simulation B, i.e., a seemingly negative bias. By subtracting simulation B, also the ocean carbon





sink of the MPI model follows the definition of $S_{OCEAN}$. This correction reduces the model mean
ocean carbon sink by 0.03 GtC yr$^{-1}$ in the 1990s. The ocean models cover 99% to 101% of the total
ocean area, so that area-scaling is not necessary.
**C.3.3 GOBM evaluation and uncertainty assessment for $S_{OCEAN}$**
The ocean $CO_2$ sink for all GOBMs and the ensemble mean falls within 90% confidence of the
observed range, or 1.5 to 2.9 GtC yr$^{-1}$ for the 1990s (Ciais et al., 2013) after applying adjustments.
An exception is the MPI model, which simulates a low ocean carbon sink of 1.38 GtC yr$^{-1}$ for the
1990s in simulation A owing to the inclusion of riverine carbon flux. After adjusting to the GCB's
definition of $S_{OCEAN}$ by subtracting simulation B, the MPI model falls into the observed range with
an estimated sink of 1.69 GtC yr$^{-1}$.
The GOBMs and data products have been further evaluated using the fugacity of sea surface $CO_2$
(fCO$_2$) from the SOCAT v2021 database (Bakker et al., 2016, 2021). We focused this evaluation on
the root mean squared error (RMSE) between observed and modelled fCO$_2$ and on a measure of
the amplitude of the interannual variability of the flux (modified after Rödenbeck et al., 2015).
The RMSE is calculated from detrended, annually and regionally averaged time series calculated
from GOBMs and data-product fCO$_2$ subsampled to open ocean (water depth > 400 m) SOCAT
sampling points to measure the misfit between large-scale signals (Hauck et al., 2020) The
amplitude of the $S_{OCEAN}$ interannual variability (A-IAV) is calculated as the temporal standard
deviation of the detrended $CO_2$ flux time series (Rödenbeck et al., 2015, Hauck et al., 2020). These
metrics are chosen because RMSE is the most direct measure of data-model mismatch and the A-
IAV is a direct measure of the variability of $S_{OCEAN}$ on interannual timescales. We apply these
metrics globally and by latitude bands. Results are shown in Fig. B2 and discussed in Section 3.5.5.
We quantify the 1-σ uncertainty around the mean ocean sink of anthropogenic $CO_2$ by assessing
random and systematic uncertainties for the GOBMs and data-products. The random
uncertainties are taken from the ensemble standard deviation (0.3 GtC yr$^{-1}$ for GOBMs, 0.3 GtC yr$^{-}$
$^1$ for data-products). We derive the GOBMs systematic uncertainty by the deviation of the DIC
inventory change 1994-2007 from the Gruber et al (2019) estimate (0.5 GtC yr$^{-1}$) and suggest
these are related to physical transport (mixing, advection) into the ocean interior. For the data-
products, we consider systematic uncertainties stemming from uncertainty in fCO$_2$ observations
(0.2 GtC yr$^{-1}$, Takahashi et al., 2009; Wanninkhof et al., 2013), gas-transfer velocity (0.2 GtC yr$^{-1}$,





Ho et al., 2011; Wanninkhof et al., 2013; Roobaert et al., 2018), wind product (0.1 GtC yr$^{-1}$, Fay et
al., 2021a), river flux adjustment (0.2 GtC yr$^{-1}$, Jacobson et al., 2007; Resplandy et al., 2018), and
$fCO_2$ mapping (0.2 GtC yr$^{-1}$, Landschützer et al., 2014). Combining these uncertainties as their
squared sums, we assign an uncertainty of ± 0.6 GtC yr$^{-1}$ to the GOBMs ensemble mean and an
uncertainty of ± 0.5 GtC yr$^{-1}$ to the data-product ensemble mean. These uncertainties are
propagated as $\sigma(S_{OCEAN}) = (1/2^2 * 0.6^2 + 1/2^2 * 0.5^2)^{1/2}$ GtC yr$^{-1}$ and result in an ± 0.4 GtC yr$^{-1}$
uncertainty around the best estimate of $S_{OCEAN}$.
We examine the consistency between the variability of the model-based and the $fCO_2$-based data
products to assess confidence in $S_{OCEAN}$. The interannual variability of the ocean fluxes (quantified
as A-IAV, the standard deviation after detrending, Figure B2) of the seven $fCO_2$-based data
products plus the Watson et al. (2020) product for 1990-2020, ranges from 0.16 to 0.26 GtC yr$^{-1}$
with the lower estimates by the three ensemble methods (CSIR-ML6, CMEMS-LSCE-FFNN, OS-
ETHZ-GRaCER). The inter-annual variability in the GOBMs ranges between 0.10 and 0.19 GtC yr$^{-1}$,
hence there is overlap with the lower A-IAV estimates of three data-products.
Individual estimates (both GOBMs and data products) generally produce a higher ocean $CO_2$ sink
during strong El Niño events. There is emerging agreement between GOBMs and data-products on
the patterns of decadal variability of $S_{OCEAN}$ with a global stagnation in the 1990s and an extra-
tropical strengthening in the 2000s (McKinley et al., 2020, Hauck et al., 2020). The central
estimates of the annual flux from the GOBMs and the $fCO_2$-based data products have a correlation
*r* of 0.94 (1990-2020). The agreement between the models and the data products reflects some
consistency in their representation of underlying variability since there is little overlap in their
methodology or use of observations.
**Appendix C.4 Methodology Land $CO_2$ sink**
**C.4.1 DGVM simulations**
The DGVMs model runs were forced by either the merged monthly Climate Research Unit (CRU)
and 6 hourly Japanese 55-year Reanalysis (JRA-55) data set or by the monthly CRU data set, both
providing observation-based temperature, precipitation, and incoming surface radiation on a
0.5°x0.5° grid and updated to 2020 (Harris et al., 2014, 2020). The combination of CRU monthly
data with 6 hourly forcing from JRA-55 (Kobayashi et al., 2015) is performed with methodology
used in previous years (Viovy, 2016) adapted to the specifics of the JRA-55 data.



New to this budget is the revision of incoming short-wave radiation fields to take into account
aerosol impacts and the division of total radiation into direct and diffuse components as
summarised below.
The diffuse fraction dataset offers 6-hourly distributions of the diffuse fraction of surface
shortwave fluxes over the period 1901-2020. Radiative transfer calculations are based on
monthly-averaged distributions of tropospheric and stratospheric aerosol optical depth, and 6-
hourly distributions of cloud fraction. Methods follow those described in the Methods section of
Mercado et al. (2009), but with updated input datasets.
The time series of speciated tropospheric aerosol optical depth is taken from the historical and
RCP8.5 simulations by the HadGEM2-ES climate model (Bellouin et al., 2011). To correct for biases
in HadGEM2-ES, tropospheric aerosol optical depths are scaled over the whole period to match
the global and monthly averages obtained over the period 2003-2020 by the CAMS Reanalysis of
atmospheric composition (Inness et al., 2019), which assimilates satellite retrievals of aerosol
optical depth.
The time series of stratospheric aerosol optical depth is taken from the climatology by Sato et al.
(1993), which has been updated to 2012. Years 2013-2020 are assumed to be background years so
replicate the background year 2010. That assumption is supported by the Global Space-based
Stratospheric Aerosol Climatology time series (1979-2016; Thomason et al., 2018). The time series
of cloud fraction is obtained by scaling the 6-hourly distributions simulated in the Japanese
Reanalysis (Kobayashi et al., 2015) to match the monthly-averaged cloud cover in the CRU TS
v4.03 dataset (Harris et al., 2021). Surface radiative fluxes account for aerosol-radiation
interactions from both tropospheric and stratospheric aerosols, and for aerosol-cloud interactions
from tropospheric aerosols, except mineral dust. Tropospheric aerosols are also assumed to exert
interactions with clouds.
The radiative effects of those aerosol-cloud interactions are assumed to scale with the radiative
effects of aerosol-radiation interactions of tropospheric aerosols, using regional scaling factors
derived from HadGEM2-ES. Diffuse fraction is assumed to be 1 in cloudy sky. Atmospheric
constituents other than aerosols and clouds are set to a constant standard mid-latitude summer
atmosphere, but their variations do not affect the diffuse fraction of surface shortwave fluxes.





In summary, the DGVMs forcing data include time dependent gridded climate forcing, global
atmospheric $CO_2$ (Dlugokencky and Tans, 2021), gridded land cover changes (see Appendix C.2.2),
and gridded nitrogen deposition and fertilisers (see Table A1 for specific models details).
Four simulations were performed with each of the DGVMs. Simulation 0 (S0) is a control
simulation which uses fixed pre-industrial (year 1700) atmospheric CO2 concentrations, cycles
early 20th century (1901-1920) climate and applies a time-invariant pre-industrial land cover
distribution and pre-industrial wood harvest rates. Simulation 1 (S1) differs from S0 by applying
historical changes in atmospheric CO2 concentration and N inputs. Simulation 2 (S2) applies
historical changes in atmospheric $CO_2$ concentration, N inputs, and climate, while applying time-
invariant pre-industrial land cover distribution and pre-industrial wood harvest rates. Simulation 3
(S3) applies historical changes in atmospheric CO2 concentration, N inputs, climate, and land
cover distribution and wood harvest rates.
S2 is used to estimate the land sink component of the global carbon budget ($S_{LAND}$). S3 is used to
estimate the total land flux but is not used in the global carbon budget. We further separate $S_{LAND}$
into contributions from $CO_2$ (=S1-S0) and climate (=S2-S1-S0).
**C.4.2 DGVM evaluation and uncertainty assessment for $S_{LAND}$**
We apply three criteria for minimum DGVMs realism by including only those DGVMs with (1)
steady state after spin up, (2) global net land flux ($S_{LAND} - E_{LUC}$) that is an atmosphere-to-land
carbon flux over the 1990s ranging between -0.3 and 2.3 GtC yr$^{-1}$, within 90% confidence of
constraints by global atmospheric and oceanic observations (Keeling and Manning, 2014;
Wanninkhof et al., 2013), and (3) global $E_{LUC}$ that is a carbon source to the atmosphere over the
1990s, as already mentioned in section 2.2.2. All 17 DGVMs meet these three criteria.
In addition, the DGVMs results are also evaluated using the International Land Model
Benchmarking system (ILAMB; Collier et al., 2018). This evaluation is provided here to document,
encourage and support model improvements through time. ILAMB variables cover key processes
that are relevant for the quantification of $S_{LAND}$ and resulting aggregated outcomes. The selected
variables are vegetation biomass, gross primary productivity, leaf area index, net ecosystem
exchange, ecosystem respiration, evapotranspiration, soil carbon, and runoff (see Fig. B3 for the
results and for the list of observed databases). Results are shown in Fig. B3 and discussed in
Section 3.6.5.



For the uncertainty for $S_{LAND}$, we use the standard deviation of the annual $CO_2$ sink across the
DGVMs, averaging to about ± 0.6 GtC $yr^{-1}$ for the period 1959 to 2019. We attach a medium
confidence level to the annual land $CO_2$ sink and its uncertainty because the estimates from the
residual budget and averaged DGVMs match well within their respective uncertainties (Table 5).
**Appendix C.5 Methodology Atmospheric Inversions**
Six atmospheric inversions (details of each in Table A4) were used to infer the spatio-temporal
distribution of the $CO_2$ flux exchanged between the atmosphere and the land or oceans. These
inversions are based on Bayesian inversion principles with prior information on fluxes and their
uncertainties. They use very similar sets of surface measurements of $CO_2$ time series (or subsets
thereof) from various flask and in situ networks. One inversion system also used satellite xCO2
retrievals from GOSAT and OCO-2.
Each inversion system uses different methodologies and input data but is rooted in Bayesian
inversion principles. These differences mainly concern the selection of atmospheric $CO_2$ data and
prior fluxes, as well as the spatial resolution, assumed correlation structures, and mathematical
approach of the models. Each system uses a different transport model, which was demonstrated
to be a driving factor behind differences in atmospheric inversion-based flux estimates, and
specifically their distribution across latitudinal bands (Gaubert et al., 2019; Schuh et al., 2019).
The inversion systems prescribe same global fossil fuel emissions for $E_{FOS}$; specifically, the GCP's
Gridded Fossil Emissions Dataset version 2021 (GCP-GridFEDv2021.2; Jones et al., 2021b), which is
an update through 2020 of the first version of GCP-GridFED presented by Jones et al. (2021a).
GCP-GridFEDv2021.2 scales gridded estimates of $CO_2$ emissions from EDGARv4.3.2 (Janssens-
Maenhout et al., 2019) within national territories to match national emissions estimates provided
by the GCP for the years 1959-2020, which were compiled following the methodology described in
Appendix C.1 based on all information available on 31st July 2021 (R. Andrew, *pers. comm.*).
Typically, the GCP-GridFED adopts the seasonal variation in emissions (the monthly distribution of
annual emissions) from EDGAR and applies small corrections based on heating or cooling degree
days to account for the effects of inter-annual climate variability on the seasonality emissions
(Jones et al., 2021a). However, strategies taken to deal with the COVID-19 pandemic during 2020
mean that the seasonality of emissions diverged substantially in 2020 from a typical year. To
account for this change, GCP-GridFEDv2021.2 adopts the national seasonality in emissions from
Carbon Monitor (Liu et al., 2020a,b) during the years 2019-2020 (Jones et al. 2021b).



The consistent use of GCP-GridFEDv2021.2 for $E_{FOS}$ ensures a close alignment with the estimate of
$E_{FOS}$ used in this budget assessment, enhancing the comparability of the inversion-based estimate
with the flux estimates deriving from DGVMs, GOBMs and $fCO_2$-based methods. To account for
small differences in regridding, and the use of a slightly earlier file version (GCP-GridFEDv2021.1)
for 2000-2018 in CarbonTracker Europe, small fossil fuel corrections were applied to all inverse
models to make the estimated uptake of atmospheric CO2 fully consistent. Finally, we note that
GCP-GridFEDv2021.2 includes emissions from cement production, but it does not include the
cement carbonation $CO_2$ sink (Xi et al., 2016; Cao et al., 2020; Guo et al. 2021) that is applied to
the GCB estimate of $E_{FOS}$ in Table 6.
The land and ocean $CO_2$ fluxes from atmospheric inversions contain anthropogenic perturbation
and natural pre-industrial $CO_2$ fluxes. On annual time scales, natural pre-industrial fluxes are
primarily land $CO_2$ sinks and ocean $CO_2$ sources corresponding to carbon taken up on land,
transported by rivers from land to ocean, and outgassed by the ocean. These pre-industrial land
$CO_2$ sinks are thus compensated over the globe by ocean $CO_2$ sources corresponding to the
outgassing of riverine carbon inputs to the ocean, using the exact same numbers and distribution
as described for the oceans in Section 2.4. To facilitate the comparison, we adjusted the inverse
estimates of the land and ocean fluxes per latitude band with these numbers to produce historical
perturbation $CO_2$ fluxes from inversions. Finally, for the presentation of the comparison in Figure
11 we modified the FF-corrected and riverine-adjusted land sinks from the inversions further, by
removing a 0.2 $GtCyr^{-1}$ $CO_2$ sink that is ascribed to cement carbonation in the GCB, rather than to
terrestrial ecosystems. The latter is not applied in the inversion products released through GCB or
the original data portals of these products.
All participating atmospheric inversions are checked for consistency with the annual global growth
rate, as both are derived from the global surface network of atmospheric CO2 observations. In this
exercise, we use the conversion factor of 2.086 GtC/ppm to convert the inverted carbon fluxes to
mole fractions, as suggested by Prather (2012). This number is specifically suited for the
comparison to surface observations that do not respond uniformly, nor immediately, to each
year's summed sources and sinks. This factor is therefore slightly smaller than the GCB conversion
factor in Table 1 (2.142 GtC/ppm, Ballantyne et al., 2012). Overall, the inversions agree with the
growth rate with biases between 0.03-0.08 ppm (0.06-0.17 $GtCyr^{-1}$) on the decadal average.



The atmospheric inversions are also evaluated using vertical profiles of atmospheric $CO_2$
concentrations (Fig. B4). More than 30 aircraft programs over the globe, either regular programs
or repeated surveys over at least 9 months, have been used in order to draw a robust picture of
the model performance (with space-time data coverage irregular and denser in the 0-45°N
latitude band; Table A6). The six models are compared to the independent aircraft $CO_2$
measurements between 2 and 7 km above sea level between 2001 and 2020. Results are shown in
Fig. B4, where the inversions generally match the atmospheric mole fractions to within 0.6 ppm at
all latitudes, except for CT Europe in 2010-2020 over the more sparsely sampled southern
hemisphere.
**Appendix D Processes not included in the global carbon budget**
**Appendix D.1 Contribution of anthropogenic CO and $CH_4$ to the global carbon budget**
Equation (1) includes only partly the net input of $CO_2$ to the atmosphere from the chemical
oxidation of reactive carbon-containing gases from sources other than the combustion of fossil
fuels, such as: (1) cement process emissions, since these do not come from combustion of fossil
fuels, (2) the oxidation of fossil fuels, (3) the assumption of immediate oxidation of vented
methane in oil production. However, it omits any other anthropogenic carbon-containing gases
that are eventually oxidised in the atmosphere, such as anthropogenic emissions of CO and $CH_4$.
An attempt is made in this section to estimate their magnitude and identify the sources of
uncertainty. Anthropogenic CO emissions are from incomplete fossil fuel and biofuel burning and
deforestation fires. The main anthropogenic emissions of fossil $CH_4$ that matter for the global
(anthropogenic) carbon budget are the fugitive emissions of coal, oil and gas sectors (see below).
These emissions of CO and $CH_4$ contribute a net addition of fossil carbon to the atmosphere.
In our estimate of $E_{FOS}$ we assumed (Section 2.1.1) that all the fuel burned is emitted as $CO_2$, thus
CO anthropogenic emissions associated with incomplete fossil fuel combustion and its
atmospheric oxidation into $CO_2$ within a few months are already counted implicitly in $E_{FOS}$ and
should not be counted twice (same for $E_{LUC}$ and anthropogenic CO emissions by deforestation
fires). Anthropogenic emissions of fossil $CH_4$ are however not included in $E_{FOS}$, because these
fugitive emissions are not included in the fuel inventories. Yet they contribute to the annual $CO_2$
growth rate after $CH_4$ gets oxidized into $CO_2$. Emissions of fossil $CH_4$ represent 30% of total
anthropogenic $CH_4$ emissions (Saunois et al. 2020; their top-down estimate is used because it is



consistent with the observed $CH_4$ growth rate), that is 0.083 GtC $yr^{-1}$ for the decade 2008-2017.
Assuming steady state, an amount equal to this fossil $CH_4$ emission is all converted to $CO_2$ by OH
oxidation, and thus explain 0.083 GtC $yr^{-1}$ of the global $CO_2$ growth rate with an uncertainty range
of 0.061 to 0.098 GtC $yr^{-1}$ taken from the min-max of top-down estimates in Saunois et al. (2020).
If this min-max range is assumed to be 2 σ because Saunois et al. (2020) did not account for the
internal uncertainty of their min and max top-down estimates, it translates into a 1-σ uncertainty
of 0.019 GtC $yr^{-1}$.
Other anthropogenic changes in the sources of CO and $CH_4$ from wildfires, vegetation biomass,
wetlands, ruminants, or permafrost changes are similarly assumed to have a small effect on the
$CO_2$ growth rate. The $CH_4$ and CO emissions and sinks are published and analysed separately in the
Global Methane Budget and Global Carbon Monoxide Budget publications, which follow a similar
approach to that presented here (Saunois et al., 2020; Zheng et al., 2019).
**Appendix D.2 Contribution of other carbonates to $CO_2$ emissions**
Although we do account for cement carbonation (a carbon sink), the contribution of emissions of
fossil carbonates (carbon sources) other than cement production is not systematically included in
estimates of $E_{FOS}$, except at the national level where they are accounted for in the UNFCCC
national inventories. The missing processes include $CO_2$ emissions associated with the calcination
of lime and limestone outside cement production. Carbonates are also used in various industries,
including in iron and steel manufacture and in agriculture. They are found naturally in some coals.
$CO_2$ emissions from fossil carbonates other than cement are estimated to amount to about 1% of
$E_{FOS}$ (Crippa et al., 2019), though some of these carbonate emissions are included in our estimates
(e.g., via UNFCCC inventories).
**Appendix D.3 Anthropogenic carbon fluxes in the land-to-ocean aquatic continuum**
The approach used to determine the global carbon budget refers to the mean, variations, and
trends in the perturbation of $CO_2$ in the atmosphere, referenced to the pre-industrial era. Carbon
is continuously displaced from the land to the ocean through the land-ocean aquatic continuum
(LOAC) comprising freshwaters, estuaries, and coastal areas (Bauer et al., 2013; Regnier et al.,
2013). A substantial fraction of this lateral carbon flux is entirely 'natural' and is thus a steady
state component of the pre-industrial carbon cycle. We account for this pre-industrial flux where
appropriate in our study (see Appendix C.3). However, changes in environmental conditions and



land-use change have caused an increase in the lateral transport of carbon into the LOAC – a
perturbation that is relevant for the global carbon budget presented here.
The results of the analysis of Regnier et al. (2013) can be summarized in two points of relevance
for the anthropogenic $CO_2$ budget. First, the anthropogenic perturbation of the LOAC has
increased the organic carbon export from terrestrial ecosystems to the hydrosphere by as much as
$1.0 \pm 0.5$ GtC yr$^{-1}$ since pre-industrial, mainly owing to enhanced carbon export from soils. Second,
this exported anthropogenic carbon is partly respired through the LOAC, partly sequestered in
sediments along the LOAC and to a lesser extent, transferred to the open ocean where it may
accumulate or be outgassed. The increase in storage of land-derived organic carbon in the LOAC
carbon reservoirs (burial) and in the open ocean combined is estimated by Regnier et al. (2013) at
$0.65 \pm 0.35$ GtC yr$^{-1}$. The inclusion of LOAC related anthropogenic $CO_2$ fluxes should affect
estimates of $S_{LAND}$ and $S_{OCEAN}$ in Eq. (1) but does not affect the other terms. Representation of the
anthropogenic perturbation of LOAC $CO_2$ fluxes is however not included in the GOBMs and
DGVMs used in our global carbon budget analysis presented here.
**Appendix D.4 Loss of additional land sink capacity**
Historical land-cover change was dominated by transitions from vegetation types that can provide
a large carbon sink per area unit (typically, forests) to others less efficient in removing $CO_2$ from
the atmosphere (typically, croplands). The resultant decrease in land sink, called the 'loss of
additional sink capacity', can be calculated as the difference between the actual land sink under
changing land-cover and the counterfactual land sink under pre-industrial land-cover. This term is
not accounted for in our global carbon budget estimate. Here, we provide a quantitative estimate
of this term to be used in the discussion. Seven of the DGVMs used in Friedlingstein et al. (2019)
performed additional simulations with and without land-use change under cycled pre-industrial
environmental conditions. The resulting loss of additional sink capacity amounts to $0.9 \pm 0.3$ GtC
yr$^{-1}$ on average over 2009-2018 and $42 \pm 16$ GtC accumulated between 1850 and 2018 (Obermeier
et al., 2021). OSCAR, emulating the behaviour of 11 DGVMs finds values of the loss of additional
sink capacity of $0.7 \pm 0.6$ GtC yr$^{-1}$ and $31 \pm 23$ GtC for the same time period (Gasser et al., 2020).
Since the DGVM-based ELUC estimates are only used to quantify the uncertainty around the
bookkeeping models' ELUC we do not add the loss of additional sink capacity to the bookkeeping
estimate.