# Peer review of "Global Carbon Budget 2021"

_Earth System Science Data, 2021_

## Referee Comment (RC1)

Review 2021-386, global carbon budget

Thank you opportunity to review global carbon budget. I conclude authors have submitted (another) excellent product. Comprehensive and careful compilations and analysis, great graphics, excellent presentation (other ESSD papers should emulate Tables 1, 2 and 3), easy-to-access and easy-to-use spreadsheet. Thanks to authors and thanks to ESSD.

I make a few comments about confusing organization and also (mindful of admonition from editor to find ways to shorten) a few suggestions on ways to shorten. Overall, I enthusiastically recommend publication.

1) Abstract, exec summary, introduction, etc. I suppose authors might intend abstract and introduction for other scientific users, as for many ESSD papers, with executive summary for non-experts? Reader now confronts abstract of three paragraphs, with middle paragraph devoted entirely to numbers, then executive summary organized as highlight sentences followed by explanatory numbers. To this reader, all this text seems highly (carefully?) technical, backed by best numbers. If so, then I do not understand purpose for exec summary. If authors want exec summary included in a reviewed document, they should eliminate as much redundancy as possible? Summary highlight sentences should go in conclusions? Or into abstract, in place of quantitative paragraph? Or, they could make exec summary as a separate document, used on GCP website? In present formats, I do not understand purpose of exec summary.

2) Page 5 Lines 15, 16 - "patterns reflect the stringency of the COVID-19 confinement levels": Which patterns? 5% drop in 2020? 5% rebound in 2021. Combined drop-then-rebound pattern? Country-to-country comparisons? Authors raise confusion with this statement.

3) Page 5 Lines 17 to 19 - 23 countries achieved significant reduced emissions over past decade but these countries do not represent the big emitters? But sources remain national reports which can be 'cooked' according to country preference? Here the authors seem to intend to write for a non-expert reader but they somewhat fail to make their point?

4) Page 5 lines 20 to 30 - more confusion here? All LULUC refers to managed land? Emissions only changed due to changes in management? Decrease (significant?, not significant?) over prior decades not well documented here?

5) Page 5 line 29 - "the (rising) importance of degradation": Degradation here means land previously used as cropland or grazing land now abandoned? Or, burned? Not clear what the authors intend here?

6) Page 6 lines 6 to 10: important statement, has gotten or will get much attention, but seems somewhat of a diversion for GCB? Also, line 13 refers to "net zero". Do authors adopt political conventions of assuming unproven unrealistic $CO_2$ removals, or do they mean quantitatively-rigorous zero emissions? Need more careful language here, to avoid misinterpretations?

7) Page 6 line 21; global reduction of 0.18 ppm? Really? Hard to find any evidence for such reduction on e.g. CO2 Earth (NOAA)? Ironically, 0.18 change here equals exactly the calibration change (likewise 0.18 ppm) outlined on page 16. Circumstance?

8) Page 6 lines 22 to 29: A statement about real change in ocean uptake or about current deficiencies in ocean observations? Both? Can one conclude the former while acknowledging the latter?

9) Page 7 lines 5 to 7: Important statement here but redundant with text above?

At this point, before conventional Introduction, reader has encountered exec summary containing / highlighting seven important points. Authors have conveyed mixed message. If abstract and introduction, followed by methods, results, and discussion/conclusion, proceed in well-written sequence as expected for ESSD product, with exec summary added to provide a short-cut for readers who do not want, or lack time, to read entire manuscript, good. But language of exec summary remains too similar to normal technical text, so that it does not

appear friendly to non-experts? In attempting to extract, authors have too many times added confusion. Get someone else, outside of this group, to write an exec summary? Make it a separate product from this complete description manuscript? I think I understand intent, and I applaud effort for this product, but outcome not as clear or distinct as authors might have liked. Making it a separate product would save approx 3 pages here?

Page 8 line 4: reader sees 412 ppm here but just a few lines (page 6 line 17) earlier saw 414.7, almost 415 ppm. Need to make clearer distinction between final QC'd value for 2020 (here) and first projections (earlier)?

Page 8 line 25, 26 - "to quantify the permissible emissions for a given climate stabilization target": But, 'permissible' emissions involve a morass of social assumptions and choices, as evidenced by massive literature around SSPs. Can this group really quantify future emissions as a function of a biogeochemical carbon cycle separate from vague social choices? I understand desire to see this product take a greater impact on social change but I worry slightly that authors 'bend' their otherwise-excellent quantitatively-rigorous product toward highly-uncertain future estimates.

Page 13 line 15 - "all other countries combined"- a bit vague, somewhat awkward, allows reader confusion?

Page 16 lines 16, 17: these represent impacts of calibration corrections, not actual emissions reductions! Authors know exactly what they intend but they here open the door slightly to reader misinterpretation.

Page 17 line 9 - "0.17 GtC yr$_{-1}$ for 1980-2020": This number is essentially identical to the decrease authors have reported due to Covid-19?

Page 23 line 10: same 30+ aircraft CO2 measurement projects as listed in prior verions?  If so (I have only done a perfunctory comparison), these could be cited rather than re-reported? Save another formatted page or two?

Page 24 line 10, Fossil fuel emissions: In all these results sections, authors report based on historical (1850-2020), recent (1960-2020), final (2020) and projection (2021). Not until partitioning discussion at section 3.7 (top of page 40) does reader again encounter decadal results. But, at least through exec summary and introduction, authors have presented decadal outcomes and trends as of primary interest. Reader needs to know beforehand that decadal trends relate only (primarily) to component exchanges and regional or country-specific assessments?

Page 43 line 23: extra or misplaced comma here?

Page 44 line 16: FACE experiments?

BIM discussion starting from page 45 line 22 = excellent!

Page 47 line 24 'Tracking progress': Again a small voice of caution. After pages, table and charts of highest quality with itemized uncertainties and extensive validation, authors next turn to a purely political agreement from Paris. If the information here seems or proves relevant to that agreement, good. But readers can get the sense that Paris agreement contains hard QC'd targets, which it emphatically does not. Authors use their hard (and, hard-won) numbers to show progress toward a soft target?

All these discussion rely entirely on national reports which - unfortunately - we must regard as estimates at best or as manipulated at worst. Uncertainty discussion (e.g on page 50) focuses on sectors, definition boundaries, activity factors etc. but seems nowhere to acknowledge deliberate mis-reporting? Here, however, authors take readers through a logical, orderly, quantitative assessment (e.g. through pages 48, 49 etc.) Meanwhile, as reported earlier in manuscript and as reported hourly, daily, monthly, annually etc. by NOAA, atmospheric $CO_2$ concentrations rise consistently and relentlessly. Nothing I write will seem unfamiliar to these authors, but their products seems to divide here: the reality of atmospheric concentrations on the one hand vs these hopeful projections about remaining emissions on the other. Annual GCB must acknowledge and report latest data relevant to both views, but in a way that leaves readers/users clear about what represents hard data and what represents political and politically-motivated targets? I don't know what to suggest as clear separation or resolution but present format seems to give equal credence to both aspects. Perhaps make clearer somehow (as in your uncertainty sections) where you rely on measured concentrations and where on national reports? If national reports referenced to actual concentrations? Great! But national reports with the same apparent credibility as measurements? Cautious.

Page 49, lines 15 to 17: Confused again. a) Net zero here means quantitive zero net emissions, not politically expedient net zero based on future imagined reductions? b) If apparent 0.5 GtC reductions due to Covid-19 in 2020 were NOT matched by equivalent reductions in atmospheric $CO_2$, then how can we advocate similar or continued emissions reductions of that magnitude? I do not dispute goals or policy, but data shown here suggest that 2020 Efos changes had minor to zero quantifiable impact on global atmospheric $CO_2$?

Page 51 line 8 - NGHGI?: One suspects 'National GreenHouse Gas Inventories' but the acronym needs accurate definition.

For the most part, consider this list of comments as suggestions only. One reader encountering this product from a personal viewpoint. A few typos you will want to fix but otherwise not much that requires definite change.

---

## Referee Comment (RC2)

**ACP review essd-2021-386, global carbon budget 2021**

Thank you very much for the opportunity I was offered to be reviewer of this excellent study. This new global carbon budget study, is a useful and comprehensive study for the carbon cycle scientific community. I would also like to thank the authors for this exceptional effort.

Please find below a few comments.

1) Line 10, page 12 : 'Short-cycle carbon emissions - for example from combustion of biomass - are not included.' Which short-cycle emissions from combustion of biomass are you referencing exactly and why not including it?

2) Are the fraction of wildfire emissions included in your ELUC emissions?

2021 is projected in this study to be a La Niña year (with a reduction to the ocean sink, page 36), which has been linked to increase fires severity in 2019/2020 in the Northern Hemisphere due to severe drought. Was La Niña event and its possible impact on ELUC emissions also considered in the 2021 projection? The dry conditions for a non-El Niño year in 2019 are mentioned for "Final year 2020" (3.2.3) but not for 2021 projection (3.2.4).

3) What are the uncertainties estimates of the fossil fuel emission inventories you considered in this study?

4) Page 22, "Multiple inversions […] were previously tested with satellite $XCO_2$ retrievals from GOSAT or OCO-2 measurements, but their results at the larger scales did not deviate substantially from their in-situ counterparts and are therefore not separately included". Which results/studies are you referring to? What are the differences and what are the results at latitudinal scales?

The differences between these two sets of observations are particularly large at latitudinal/regional scale, but even if satellite measurements do not deviate significantly from in-situ data at the global scale, some differences are still present. With the MIP (Model Inter-comparison Project) ensemble, Peiro et al. (2022) found when using OCO-2 v9 that a small difference could be observed at the global terrestrial scale (largest sink from ~1PgC/yr to ~2PgC/yr for in-situ fluxes relative to posterior OCO-2 LNLG fluxes) and at the global ocean scale (largest sink for OCO-2 LNLG fluxes of about 1.5PgC/yr relative to in-situ). This was also observed, even if the difference was smaller, with OCO-2 v7 retrievals (Crowell et al., 2019).

Could this small difference, between these two sets of observations, impact your results if you did not include them separately? Do you think the results could have been different by not separating them, particularly when you looked at latitudinal scales (such as the tropics)?

You mentioned not including separately the three inversions tested with GOSAT and OCO-2 data (and only using in-situ data here, table A4) but in your discussion you mentioned that additional information could have been obtained with inversions assimilating satellite observations, is this not contradictory?

5) Which OCO-2 b10 retrievals (LNLG, LNLGOG...) did the CMS-Flux inversion use?

Did all inversions optimize biosphere and fires, Ocean, and fossil fuels fluxes?

6) The use of satellite observations from GOSAT and OCO-2 with CMS-Flux is new compared to Friedlingstein et al. (2020), where MIROC inversion was used instead. I saw no discussion (even in appendix) of the possible disagreement or agreement between the satellite and in-situ analysis with bottom-up fluxes used here; and how the results (accuracy, uncertainty, …) could have changed here with a simulation assimilating satellite observations compared to the previous study of Friedlingstein et al. (2020) where no satellite observations were used?

This could explain the high uncertainty and fluxes ranges in the tropics observed with the inversions, for example (page 42), where the previous studies of Crowell et al. (2019) and Peiro et al., (2022), observed more net sources with OCO-2 inversions than with in-situ inversions.

On page 50, you mentioned "Additional information could also be obtained through […] the introduction of inferred fluxes such as those based on satellite $CO_2$ retrievals", but do not go further knowing you used an inversion with satellite $CO_2$ retrievals.

7) Page 25: 2020 has a global fossil $CO_2$ emissions 5.4% lower than in 2019. This was probably related to COVID, but it is not mentioned here, for some reason? China has not observed a decline in growth rate compared to other countries, do you have any assumptions/explanations why? In Friedlingstein et al, (2020), the projection of 2020 for China was a decrease in emissions which appeared to be less pronounced than other countries. However, here we don't see a decrease but an increase. How do you explain this difference for China between the two studies?

8) On page 28, the gross emissions are influenced by the temporary decrease in deforestation, which is one of the changes that could explain the decrease in net ELUC emissions over the last few years. However, have not forest wildfires been more intense in recent years? Also, in term of prevision,

studies show that fires will increase in intensity and frequency, so do we expect fires to have a larger contribution in the projection? If not, why?

9) On page 30, You mentioned the consequence of dry conditions from La Niña leading to fire emissions in Equatorial Asia. What about the large and severe fires in Australia which ceased in early March 2020? Additionally, you mention fires severity in the tropics, but the northern hemisphere like California experienced in 2020 the largest fires in Californian history. Why not mention it in this ELUC section (3.2)? I was only able to find this information by accessing the Land sink section (3.6, page 38).

*Technical comments:*

Figure 2, could it be possible to have a better quality figure?

Page 12, line 9, the meaning of UNFCCC is needed for those who do not know what it is (Like reviewer #1 mentioned, a lot of acronyms definition are missing).

Page 12, BP is mentioned without information on the abbreviation meaning.

Lin 9 page 12, 'UNFCCC Annex 1', could not find Annex1 in the manuscript, so if this is from the UNFCCC report, the reference is missing here.

Page 14, line 16: DGVMs is not defined.

Page 15, line 1, FAO is only defined page 178 but not in page 15.

Page 18, line 15: In table 4 and table A4, it seems there is 8 ocean based data-products and not 7.

Page 23, line 22, CH4 should be $CH_4$

page 36, line 9, La Niña need an accent.

---

## Author Comment (AC1)

Response to reviewer 1

Thank you opportunity to review global carbon budget. I conclude authors have submitted (another) excellent product. Comprehensive and careful compilations and analysis, great graphics, excellent presentation (other ESSD papers should emulate Tables 1, 2 and 3), easy- to-access and easy-to-use spreadsheet. Thanks to authors and thanks to ESSD.

I make a few comments about confusing organization and also (mindful of admonition from editor to find ways to shorten) a few suggestions on ways to shorten. Overall, I enthusiastically recommend publication.

**Thank you.**

1)        Abstract, exec summary, introduction, etc. I suppose authors might intend abstract and introduction for other scientific users, as for many ESSD papers, with executive summary for non-experts? Reader now confronts abstract of three paragraphs, with middle paragraph devoted entirely to numbers, then executive summary organized as highlight sentences followed by explanatory numbers. To this reader, all this text seems highly (carefully?) technical, backed by best numbers. If so, then I do not understand purpose for exec summary. If authors want exec summary included in a reviewed document, they should eliminate as much redundancy as possible? Summary highlight sentences should go in conclusions? Or into abstract, in place of quantitative paragraph? Or, they could make exec summary as a separate document, used on GCP website? In present formats, I do not understand purpose of exec summary.

**We felt, in agreement with the ESSD editors, that an executive summary was appropriate given the length and amount of information in the main text. We tried to improve the language in the executive summary to make it less technical.**

2)  Page 5 Lines 15, 16 - "patterns reflect the stringency of the COVID-19 confinement levels": Which patterns? 5% drop in 2020? 5% rebound in 2021. Combined drop-then-rebound pattern? Country-to-country comparisons? Authors raise confusion with this statement.

**Thank you. We rephrased to clarify: These changes in 2021 emissions patterns reflect the stringency of the COVID-19 confinement levels in 2020 and the pre-covid background trends in emissions in these countries.**

3)  Page 5 Lines 17 to 19 - 23 countries achieved significant reduced emissions over past decade but these countries do not represent the big emitters? But sources remain national reports which can be 'cooked' according to country preference? Here the authors seem to intend to write for a non-expert reader but they somewhat fail to make their point?

**Indeed, as we wrote, "these 23 countries contribute to only about one quarter of world CO2 fossil emissions". Not clear what the rest of the comment is exactly about, our point is quite simple and seem clear: .**

4)  Page 5 lines 20 to 30 - more confusion here? All LULUC refers to managed land? Emissions only changed due to changes in management? Decrease (significant?, not significant?) over prior decades not well documented here?

**These are global CO2 emissions from land-use, land-use change, and forestry as written in the headline sentence. This refers indeed to managed land (i.e. land used). The decrease is over the past two decades (2000-2019) as written in the headline sentence.**

5)  Page 5 line 29 - "the (rising) importance of degradation": Degradation here means land previously used as cropland or grazing land now abandoned? Or, burned? Not clear what the authors intend here?

**Forest degradation here refers to the partial loss of forest function and structural integrity from disturbance, but does not result in a change of land cover**

6)  Page 6 lines 6 to 10: important statement, has gotten or will get much attention, but seems somewhat of a diversion for GCB? Also, line 13 refers to "net zero". Do authors adopt political conventions of assuming unproven unrealistic CO2 removals, or do they mean quantitatively-rigorous zero emissions? Need more careful language here, to avoid misinterpretations?

**We follow the same approach as in IPCC AR6 WG1. What we present here is the remaining carbon budget to keep warming below 1.5, 1.7 or 2°C. It is update from IPCC as IPCC AR6 estimated the budget since 2019. Here we account for the emissions that occurred in 2020 and 2021, hence reducing the remaining carbon budget. To avoid confusion and policy interpretation, we changed "net-zero" by "zero" in this statement as this estimate does not make any assumption on carbon dioxide removal.**

7)  Page 6 line 21; global reduction of 0.18 ppm? Really? Hard to find any evidence for such reduction on e.g. CO2 Earth (NOAA)? Ironically, 0.18 change here equals exactly the calibration change (likewise 0.18 ppm) outlined on page 16. Circumstance?

**There is a misunderstanding here. We don't say atmospheric $CO_2$ went down by 0.18ppm, we only said that the reduction in emissions of 0.7 GtC in 2020 implies a reduction in the $CO_2$ growth rate of about 0.18 ppm. However, we decided to remove this sentence as potentially confusing and not critical that paragraph.**

8)  Page 6 lines 22 to 29: A statement about real change in ocean uptake or about current deficiencies in ocean observations? Both? Can one conclude the former while acknowledging the latter?

**These lines describe consistent changes across methods ('resumed a more rapid growth') and discrepancies between methods. Discrepancies between methods are clearly reported ('the growth of the ocean CO2 sink in the past decade has an uncertainty of a factor of three, with estimates based on data products and estimates based on models showing an ocean sink increase of 0.9 GtC yr-1 and 0.3 GtC yr-1 since 2010, respectively'). This source of discrepancy (model bias or lack of observations or other methodological issue) cannot be clearly identified. We refer the reader to the main text for more information.**

9)  Page 7 lines 5 to 7: Important statement here but redundant with text above?

**Sorry, we don't see where this text is redundant with previous text.**

At this point, before conventional Introduction, reader has encountered exec summary containing / highlighting seven important points. Authors have conveyed mixed message. If abstract and introduction, followed by methods, results, and discussion/conclusion, proceed in well-written

sequence as expected for ESSD product, with exec summary added to provide a short-cut for readers who do not want, or lack time, to read entire manuscript, good. But language of exec summary remains too similar to normal technical text, so that it does not appear friendly to non-experts? In attempting to extract, authors have too many times added confusion. Get someone else, outside of this group, to write an exec summary? Make it a separate product from this complete description manuscript? I think I understand intent, and I applaud effort for this product, but outcome not as clear or distinct as authors might have liked. Making it a separate product would save approx 3 pages here?

**See earlier comment. We felt, in agreement with the ESSD editors that an executive summary was appropriate, given the length and amount of information in the main text. We tried to improve to the language in the executive summary to make it less technical.**

Page 8 line 4: reader sees 412 ppm here but just a few lines (page 6 line 17) earlier saw 414.7, almost 415 ppm. Need to make clearer distinction between final QC'd value for 2020 (here) and first projections (earlier)?

**Indeed, as described in the text, the numbers here are the observed atmospheric CO2 data for 2020 while the numbers in the executive summary are the projection for 2021.**

Page 8 line 25, 26 - "to quantify the permissible emissions for a given climate stabilization target": But, 'permissible' emissions involve a morass of social assumptions and choices, as evidenced by massive literature around SSPs. Can this group really quantify future emissions as a function of a biogeochemical carbon cycle separate from vague social choices? I understand desire to see this product take a greater impact on social change but I worry slightly that authors 'bend' their otherwise-excellent quantitatively-rigorous product toward highly-uncertain future estimates.

**There is a misunderstanding here. We are only saying that an understanding of the global carbon budget is necessary to quantify the link between emissions and climate target. Every climate target is associated with a given amount of $CO_2$ emission (that is the remaining carbon budget). Improving our understanding on the carbon cycle can only help quantifying this remaining carbon budget. We slightly revised the sentence to make it clearer.**

Page 13 line 15 - "all other countries combined"- a bit vague, somewhat awkward, allows reader confusion?

**Not sure why this is awkward/confusing. We literally mean to say: all other countries, aggregated as one group.**

Page 16 lines 16, 17: these represent impacts of calibration corrections, not actual emissions reductions! Authors know exactly what they intend but they here open the door slightly to reader misinterpretation.

**Indeed. Clarified now.**

Page 17 line 9 - "0.17 GtC yr-1 for 1980-2020": This number is essentially identical to the decrease authors have reported due to Covid-19?

**This is pure coincidence.**

Page 23 line 10: same 30+ aircraft CO2 measurement projects as listed in prior verions? If so (I have only done a perfunctory comparison), these could be cited rather than re-reported? Save another formatted page or two?

**Thank you for the suggestion. This information does indeed include only minor updates from version to version and we could save space by backward citing and listing only updates. However, we feel that if we only list the updates per year/release it would become nearly impossible to piece together the full suite of datasets used per GCP release. It would require the reader to access previous versions as well as the current one to get the information complete. We therefore prefer to keep the table complete as is, at the expense of 1,5 pages of formatted space as indicated by the reviewer.**

Page 24 line 10, Fossil fuel emissions: In all these results sections, authors report based on historical (1850-2020), recent (1960-2020), final (2020) and projection (2021). Not until partitioning discussion at section 3.7 (top of page 40) does reader again encounter decadal results. But, at least through exec summary and introduction, authors have presented decadal outcomes and trends as of primary interest. Reader needs to know beforehand that decadal trends relate only (primarily) to component exchanges and regional or country-specific assessments?

**We do provide decadal information in the sections about the recent period 1960-2020 (sections 3.1.2, 3.4.2, 3.5.2, 3.6.2). In these sections, we give the decadal values for the first (1960s) and last (2010s) decades and we also refer to Table 6 that gives data for each individual decade. The 2010s decadal estimate is provided in each component section, the only exception was for the land use emission 3.2.2 where we omitted to give it. This has been corrected now.**

Page 43 line 23: extra or misplaced comma here?

**Done**

Page 44 line 16: FACE experiments?

**Defined now**

BIM discussion starting from page 45 line 22 = excellent!

**Thank you**

Page 47 line 24 'Tracking progress': Again a small voice of caution. After pages, table and charts of highest quality with itemized uncertainties and extensive validation, authors next turn to a purely political agreement from Paris. If the information here seems or proves relevant to that agreement, good. But readers can get the sense that Paris agreement contains hard QC'd targets, which it emphatically does not. Authors use their hard (and, hard-won) numbers to show progress toward a soft target?

**Misunderstanding. This section is not a "purely political agreement from Paris". The information we provide here are: 1) an brief analysis of the countries where emissions have been declining over the last decade; 2) a Kaya analysis of drivers of growth/decline in emissions for the major economies (Fig 14) and 3) an update of the IPCC AR6 estimate remaining carbon budget consistent with climate targets. These are factual analysis, nothing "purely political".**

All these discussion rely entirely on national reports which - unfortunately - we must regard as estimates at best or as manipulated at worst. Uncertainty discussion (e.g on page 50) focuses on sectors, definition boundaries, activity factors etc. but seems nowhere to acknowledge deliberate mis-reporting? Here, however, authors take readers through a logical, orderly, quantitative assessment (e.g. through pages 48, 49 etc.) Meanwhile, as reported earlier in manuscript and as reported hourly, daily, monthly, annually etc. by NOAA, atmospheric CO2 concentrations rise consistently and relentlessly. Nothing I write will seem unfamiliar to these authors, but their products seems to divide here: the reality of atmospheric concentrations on the one hand vs these hopeful projections about remaining emissions on the other. Annual GCB must acknowledge and report latest data relevant to both views, but in a way that leaves readers/users clear about what represents hard data and what represents political and politically-motivated targets? I don't know what to suggest as clear separation or resolution but present format seems to give equal credence to both aspects. Perhaps make clearer somehow (as in your uncertainty sections) where you rely on measured concentrations and where on national reports? If national reports referenced to actual concentrations? Great! But national reports with the same apparent credibility as measurements? Cautious.

**Not sure how to respond to this long comment with several unrelated threads. 1) Reporting of fossil fuel emissions is the best we have. They could be subject to mis-reporting manipulation but that's virtually impossible to assess. We note that such misreporting would ned to be consistent in time as we do not see any suspicious change in annual estimates from individual countries.2) We don't understand the second comment about "our products seeming to divide" atmospheric CO₂ concentration is going up because of continuous anthropogenic CO₂ emissions, hence the remaining carbon budget is reducing. We do not see where in the manuscript we would have written anything about "hopeful projections ". Likewise, we don't understand what the reviewer means by "data relevant to both views". We don't know what these both views are supposed to be and why the reviewer refers to politically mitigated targets. 3) the last comments seem to confuse emissions and concentrations. Fossil fuel emissions are relying on country level reports, global atmospheric concentrations are directly inferred from measurements from the NOOA/ESRL network. This is clearly described in section 2.**

Page 49, lines 15 to 17: Confused again. a) Net zero here means quantitive zero net emissions, not politically expedient net zero based on future imagined reductions? b) If apparent 0.5 GtC reductions due to Covid-19 in 2020 were NOT matched by equivalent reductions in atmospheric CO2, then how can we advocate similar or continued emissions reductions of that magnitude? I do not dispute goals or policy, but data shown here suggest that 2020 Efos changes had minor to zero quantifiable impact on global atmospheric CO2?

**Again, the reviewer seems to be confusing emissions and concentrations and/or concentration and concentration growth. Fossil fuel emissions declined by 0.5GtC (5%) due to the COVID-19 pandemic. Given than the world still emitted more than 10GtC in 2020, atmospheric concentration kept increasing. As we wrote, such reduction should be sustained every year until we reach zero emissions in order to stop atmospheric CO2 increase. One single year is not enough.**

Page 51 line 8 - NGHGI?: One suspects 'National GreenHouse Gas Inventories' but the acronym needs accurate definition.

**Already defined in section 2.2.**

For the most part, consider this list of comments as suggestions only. One reader encountering this product from a personal viewpoint. A few typos you will want to fix but otherwise not much that requires definite change.

**Thank you.**

---

## Author Comment (AC2)

Response to reviewer 2

Thank you very much for the opportunity I was offered to be reviewer of this excellent study. This new global carbon budget study, is a useful and comprehensive study for the carbon cycle scientific community. I would also like to thank the authors for this exceptional effort.
Please find below a few comments.

**Thank you**

Line 10, page 12 : 'Short-cycle carbon emissions - for example from combustion of biomass - are not included.' Which short-cycle emissions from combustion of biomass are you referencing exactly and why not including it?

**CO2 emissions from combustion of biomass are included in the land use change (ELUC) part of the carbon budget (see section 2.2).**

Are the fraction of wildfire emissions included in your ELUC emissions?
2021 is projected in this study to be a La Niña year (with a reduction to the ocean sink, page 36), which has been linked to increase fires severity in 2019/2020 in the Northern Hemisphere due to severe drought. Was La Niña event and its possible impact on ELUC emissions also considered in the 2021 projection? The dry conditions for a non-El Niño year in 2019 are mentioned for "Final year 2020" (3.2.3) but not for 2021 projection (3.2.4).

**Wildfire are not included in ELUC as ELUC only treats with human induced land use and land cover changes. Natural fluxes such as wildfires are part of the land sink (SLAND). However, deforestation and/or degradation fires are inlcudd in ELUC and those are also subject to year to year variability induced by climatic conditions (more human fires during drier years).**

What are the uncertainties estimates of the fossil fuel emission inventories you considered in this study?

**Unclear what this means, since we consider one inventory with its uncertainty reported in section 2.1.**

Page 22, "Multiple inversions [...] were previously tested with satellite XCO2 retrievals from GOSAT or OCO-2 measurements, but their results at the larger scales did not deviate substantially from their in-situ counterparts and are therefore not separately included". Which results/studies are you referring to? What are the differences and what are the results at latitudinal scales? The differences between these two sets of observations are particularly large at latitudinal/regional scale, but even if satellite measurements do not deviate significantly from in-situ data at the global scale, some differences are still present. With the MIP (Model Inter-comparison Project) ensemble, Peiro et al. (2022) found when using OCO-2 v9 that a small difference could be observed at the global terrestrial scale (largest sink from ~1PgC/yr to ~2PgC/yr for in-situ fluxes relative to posterior OCO-2 LNLG fluxes) and at the global ocean scale (largest sink for OCO-2 LNLG fluxes of about 1.5PgC/yr relative to in-situ). This was also observed, even if the difference was smaller, with OCO-2 v7 retrievals (Crowell et al., 2019).  Could this small difference, between these two sets of observations, impact your results if you did not include them separately? Do you think the results could have been different by not separating them, particularly when you looked at latitudinal scales (such as the tropics)? You mentioned not including separately the three inversions tested with GOSAT and OCO-2 data (and only using in-situ data here, table A4) but in your discussion you mentioned that additional

information could have been obtained with inversions assimilating satellite observations, is this not contradictory?

**We thank the reviewer for the elaborate question, and for bringing the Peiro et al paper to our attention. It is an impressive analysis that we can impossibly match, or repeat, with our set of model outcomes. Nevertheless, we would like to reply to some of the points made. The sentence on XCO2 results not deviating from their in-situ counterparts on the larger-scales of GCP is based on our own analysis of the resulting flux ensemble for GCP. This analysis is part of the annual written synthesis by the modelling team, which is not published by itself. Instead, it serves as the basis of our presentation of the inverse GCP flux ensemble in the paper. In the GCP paper we typically only discuss the ensemble and refrain from highlighting specific models, or specific model-model differences. In the mentioned elaborate synthesis, we found the fluxes on tropical-NH-SH scale for the models with more than one solution to be similar enough to not warrant introducing them as a separate member in the GCP ensemble. From that perspective, we agree with the contradiction in the discussion that the reviewer points out. To make our point less ambiguous we suggest changing the last statement to: "... Additional information specifically on smaller scales not currently analysed in GCP (but see Peiro et al., 2022) could be extracted from inversions using satellite data." Note that the reverse request: to analyse the differences between inversions with- and without satellite data in more depth, is beyond the scope of the GCP inverse exercise. This research is much better addressed in specific studies that systematically investigate XCO2 constraints on carbon fluxes across more models. In GCP, the time period analysed exceeds the XCO2-era by far and also only a few models explicitly include/exclude these data for the past years. Also, GCP does not have protocols for how to use XCO2 nor which retrieval versions to assimilate, in contrast to OCO-MIP. To try to come to other conclusions here based on our set would not be feasible, nor credible.**

Which OCO-2 b10 retrievals (LNLG, LNLGOG...) did the CMS-Flux inversion use? Did all inversions optimize biosphere and fires, Ocean, and fossil fuels fluxes?

**The information requested here is provided in Table A4, where we list the specific data constraints and inversion details of each model. In short, the models do not all optimize the same variables and in fact none of them have estimated fossil fuel fluxes.**

The use of satellite observations from GOSAT and OCO-2 with CMS-Flux is new compared to Friedlingstein et al. (2020), where MIROC inversion was used instead. I saw no discussion (even in appendix) of the possible disagreement or agreement between the satellite and in-situ analysis with bottom-up fluxes used here; and how the results (accuracy, uncertainty, ...) could have changed here with a simulation assimilating satellite observations compared to the previous study of Friedlingstein et al. (2020) where no satellite observations were used? This could explain the high uncertainty and fluxes ranges in the tropics observed with the inversions, for example (page 42), where the previous studies of Crowell et al. (2019) and Peiro et al., (2022), observed more net sources with OCO-2 inversions than with in-situ inversions.
On page 50, you mentioned "Additional information could also be obtained through [...] the introduction of inferred fluxes such as those based on satellite $CO_2$ retrievals", but do not go further knowing you used an inversion with satellite $CO_2$ retrievals.

**Indeed, we do not separately discuss the introduction of CMS flux and its impact on the ensemble, and we do not analyze any changes in the flux distribution that CMS-flux might have caused through the use of OCO-2 data. The reason for this is twofold: (1) that in all the analyses we performed before our synthesis, the CMS-fluxes are never an outlier or remarkable change from previous fluxes we presented in 2020, or before. It's land/ocean distribution, NH-Tropics-SH**

distribution, its agreement with the global growth rate, and agreement with aircraft data all fall within the current (and previous year's) range of fluxes in GCP. (2) is that we typically do not discuss individual models, or model-model differences in the paper, but instead focus on the budget constraint suggested by the ensemble. The remark on page 50, of the possible extra information to gain from inversions with satellite CO2 retrievals will be changed as indicated in our reply to the previous remark. Thank you for pointing this out.

Page 25: 2020 has a global fossil $CO_2$ emissions 5.4% lower than in 2019. This was probably related to COVID, but it is not mentioned here, for some reason? China has not observed a decline in growth rate compared to other countries, do you have any assumptions/explanations why? In Friedlingstein et al, (2020), the projection of 2020 for China was a decrease in emissions which appeared to be less pronounced than other countries. However, here we don't see a decrease but an increase. How do you explain this difference for China between the two studies?

**Indeed, the decline is due to the COVID-19 pandemic, we clarified this again here. The difference between Friedlingstein et al 2020 and the estimates here is that the former were our projection based on 4 different approaches. Here we report the actual 2020 estimate based entirely on reported emissions from countries. Our projection for 2020 emissions from China was too low compared to their actual emissions.**

On page 28, the gross emissions are influenced by the temporary decrease in deforestation, which is one of the changes that could explain the decrease in net ELUC emissions over the last few years. However, have not forest wildfires been more intense in recent years? Also, in term of prevision, studies show that fires will increase in intensity and frequency, so do we expect fires to have a larger contribution in the projection? If not, why?

**In general, wildfires are considered in the natural land sink term unless they are associated with a land cover change (e.g. deforestation), or clearly attributable to human intervention (tropical forest degradation/peat fires). Andela et al., Science, 356, 1356-1362 shows a deadline in the global area burnt over recent decades, albeit with large regional variation. This was attributed to agricultural expansion and intensification. Therefore it is not a priori clear the direction of the near-future trend in biomass burning. Although Zeng et al. suggest an increase in emissions despite a reduction in burnt area, DOI: 10.1126/sciadv.abh2646**

On page 30, You mentioned the consequence of dry conditions from La Niña leading to fire emissions in Equatorial Asia. What about the large and severe fires in Australia which ceased in early March 2020? Additionally, you mention fires severity in the tropics, but the northern hemisphere like California experienced in 2020 the largest fires in Californian history. Why not mention it in this ELUC section (3.2)? I was only able to find this information by accessing the Land sink section (3.6, page 38).

**The severe fires between end 2019-early 2020 are indeed mentioned in the manuscript on P39 line7. Wildfires, ie those that do not result in a change in land cover (e.g. deforestation fires) or are clearly attributable to human intervention (tropical forest degradation / peat fires) are considered in the natural land sink and simulated by several of the ensemble of DGVMs.**

*Technical comments:*

Figure 2, could it be possible to have a better quality figure?

**Done, hope it's better now.**

Page 12, line 9, the meaning of UNFCCC is needed for those who do not know what it is (Like reviewer #1 mentioned, a lot of acronyms definition are missing).

**Done, thank you**

Page 12, BP is mentioned without information on the abbreviation meaning.

**BP used to be an acronym, but it hasn't been for many years. But we now write " BP energy company" to make it clear.**

Lin 9 page 12, 'UNFCCC Annex 1', could not find Annex1 in the manuscript, so if this is from the UNFCCC report, the reference is missing here.

**UNFCCC Annex 1 countries is a list of 43 countries, classified by the UN as "industrialized countries and economies in transition". We clarified the sentence.**

Page 14, line 16: DGVMs is not defined.

**Done, thank you**

Page 15, line 1, FAO is only defined page 178 but not in page 15.

**Done, thank you**
Page 18, line 15: In table 4 and table A4, it seems there is 8 ocean based data-products and not 7.

**Indeed, there are 8 ocean based data products but only 7 are used to calculated the ocean carbon sink. This is clarified now.**

Page 23, line 22, $CH4$ should be $CH_4$

**Done, thank you**
page 36, line 9, La Niña need an accent.

**Done, thank you**

---

## Author Comment (AC3)

Response to community comment 1 (Robert Gieseke)

Thanks to all authors and contributors for this valuable and important resource. Thanks also for opening up the peer review discussion process to the community. Given this opportunity I have the following comments you might consider for the final version.

Page 6, lines 6-16: This paragraph was a bit confusing to me trying to understand where numbers came from. I probably wouldn't say that the remaining carbon budget has shrunk but rather only say that emissions from 2020 and 2021 used up 77 GtCO2 of the remaining carbon budget as assessed in IPCC AR6. The 2021 cumulative CO2 emissions from 1850 - 2019 (2393 Gt CO2) are closer than the 2020 version (2411 Gt CO2) to the historic emissions shown in Table SPM.2 (2390 ± 240 Gt CO2), so if a newer assessment (like this study) of historical emissions had an effect on the remaining carbon budget calculation, the 2021 version would actually be in better agreement with the IPCC AR6 assessment than before. As noted by anonymous reviewer 1 in their comment for page 47, line 24 with regards to 'hard' targets of the Paris Agreement it might be clearer to simply state the remaining budget in relation to the temperature of 1.5 degrees.

**We slightly rephrased this paragraph to clarify that the remaining carbon budget is now reduced to 120GtC, etc. We also removed the reference to the Paris Agreement. Note that the estimate of the remaining carbon budget is independent of the estimate of the historical carbon budget. The remaining carbon budget only depends on the warming to date, the given climate target (ex 1.5°C), the transient climate response to cumulative CO$_2$ emissions (TCRE) and the contribution from non-CO2 agents.**

The Excel sheet proposes a conversion factor for carbon to CO2 of 3.664. The numbers shown here appear to use different factors, probably due to rounding?

**We use a conversion factor of 3.664 indeed. If there are apparent discrepancies, it is because of rounding.**

Page 12, line 21: Is it planned to publish Andrew and Peters (2021) as a separate publication? It is a very valuable resource containing interesting and important points. The information could also be part of this peer reviewed publication, maybe as supplementary material.

**Andrew and Peters (2021) dataset is already available online: (https://doi.org/10.5281/ZENODO.5569235) and is referred in this paper.**

Page 24, line 21, line 27: PRIMAP-hist 2.3.1 does not seem to include bunker emissions either, see https://www.pik-potsdam.de/paris-reality-check/primap-hist/PRIMAP-hist_v2.3.1_data-description.pdf
"Emissions from international aviation and shipping are not included in the dataset."

**Thank you. We have added the following clause to the sentence finishing on line 29 of that page: "and omits emissions from international transport entirely".**

Page 53, lines 13-22: The data availability section and the header information in the Excel files should probably be updated to include a reference to the data being released under a CC-BY 4.0 license. The ICOS page and file metadata include it but it would be clearer to write this in the manuscript and Excel files as well.

**As for all ESSD papers, it is clearly indicated on paper's home page: This work is distributed under the Creative Commons Attribution 4.0 License." We are not aware of references to CC-BY licenses being mentioned in manuscripts.**

Figures: Some figures have very light gray text which is hard to read.

**Done, thank you**